# Can Neural Networks Achieve Optimal Computational-statistical Tradeoff? An Analysis on Single-Index Model

**Siyu Chen**[*1]**, Beining Wu**[*2]**, Miao Lu**[3]**, Zhuoran Yang**[1]**, Tianhao Wang**[4]

[1]Yale University, [2]University of Chicago, [3]Stanford University, [4]Toyota Technological Institute at Chicago
{siyu.chen.sc3226, zhuoran.yang}@yale.edu, beiningw@uchicago.edu
miaolu@stanford.edu, tianhao.wang@ttic.edu

## Abstract

In this work, we tackle the following question: Can neural networks trained with gradient-based methods achieve the optimal statistical-computational tradeoff in learning Gaussian single-index models? Prior research has shown that any polynomial-time algorithm under the statistical query (SQ) framework requires $\Omega(d^{s^\star/2} \vee d)$ samples, where $s^\star$ is the generative exponent representing the intrinsic difficulty of learning the underlying model. However, it remains unknown whether neural networks can achieve this sample complexity. Inspired by prior techniques such as label transformation and landscape smoothing for learning single-index models, we propose a unified gradient-based algorithm for training a two-layer neural network in polynomial time. Our method is adaptable to a variety of loss and activation functions, covering a broad class of existing approaches. We show that our algorithm learns a feature representation that strongly aligns with the unknown signal $\theta^\star$, with sample complexity $\widetilde{O}(d^{s^\star/2} \vee d)$, matching the SQ lower bound up to a polylogarithmic factor for all generative exponents $s^\star \geq 1$. Furthermore, we extend our approach to the setting where $\theta^\star$ is $k$-sparse for $k = o(\sqrt{d})$ by introducing a novel weight perturbation technique that leverages the sparsity structure. We derive a corresponding SQ lower bound of order $\widetilde{\Omega}(k^{s^\star})$, matched by our method up to a polylogarithmic factor. Our framework, especially the weight perturbation technique, is of independent interest, and suggests potential gradient-based solutions to other problems such as sparse tensor PCA.

## 1 Introduction

The success of neural networks is largely attributed to their remarkable ability to learn rich and useful features from data during gradient-based training (Girshick et al., 2014). This feature-learning capability allows them to outperform traditional methods like kernel-based approaches, which rely on predefined features (Allen-Zhu & Li, 2019; Ghorbani et al., 2019; Refinetti et al., 2021). However, when trained using (stochastic) gradient descent, neural networks can sometimes fall into a "kernel regime", where their behavior resembles that of a fixed kernel method, constrained by their random initialization (Jacot et al., 2018; Chizat et al., 2019). In this regime, the ability of the network to learn complex representations is severely limited, undermining the primary advantage of deep learning. Therefore, it is crucial to understand when and how neural networks trained with gradient-based method can perform effective feature learning to unlock their full potential, particularly in scenarios where a balance between computational efficiency and statistical performance is essential.

In this work, we approach this question in the context of Gaussian single-index models, a canonical class of problems in statistics and learning (MacCullagh & Nelder, 1989; Ichimura, 1993; Hristache et al., 2001; Härdle et al., 2004). The model is defined as follows: for covariates $z \sim \mathcal{N}(0, I_d)$, the output $y$ depends on the inner product $\langle \theta^\star, z \rangle$ with an unknown signal $\theta^\star \in \mathbb{R}^d$ through a link distribution $p$, i.e., $y \sim p(\cdot \mid \langle \theta^\star, z \rangle)$. The goal is to recover $\theta^\star$ using i.i.d. samples $(z_1, y_1), \ldots, (z_n, y_n)$ generated by the underlying model. While $n = \Omega(d)$ samples suffice to recover $\theta^\star$ information-theoretically

---
[*]Equal Contribution

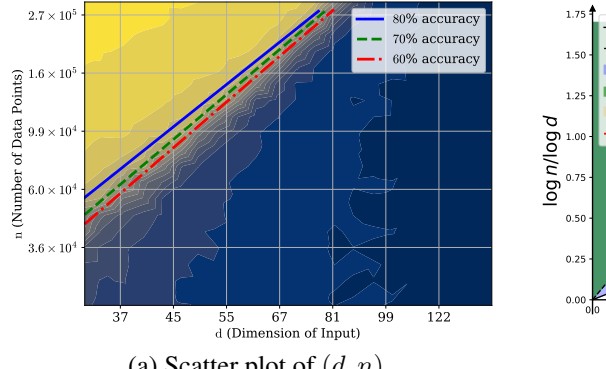
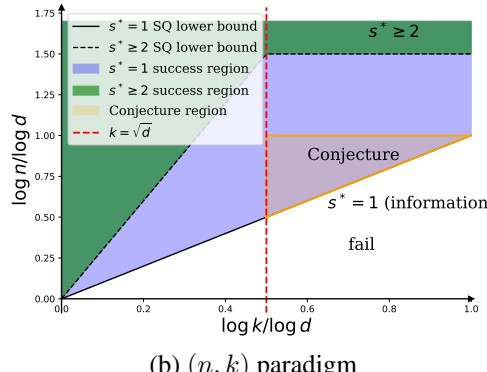

(a) Scatter plot of $(d, n)$          (b) $(n, k)$ paradigm

Figure 1: (a) The contour plots of $(\log d, \log n, \texttt{acc}(d, n))$ for Algorithm 1 under model $y = \langle z, \theta^\star \rangle^2 \exp(-\langle z, \theta^\star \rangle^2)$, which has generative exponent $s^\star = 4$ (Example 2.2). Here $\texttt{acc}(d, n)$ is the average of the largest 8 values of the alignment between the neuron weights and the unknown signal $\theta^\star$. The slopes of these contour lines are all close to 2, indicating a sample complexity $n \approx d^2$ for $s^\star = 4$. (b) The paradigm of sample complexity achieved by our algorithm for different generative exponent $s^\star$ and sparsity level $k$, illustrating the success of achiving computational-statistical tradeoff.

(Bach, 2017; Damian et al., 2024), achieving this efficiently is difficult for polynomial-time algorithms, where the required sample size also depends on properties of the link distribution $p$, creating a computational-statistical gap. For example, when $y$ is a polynomial of $\langle \theta^\star, z \rangle$, it has been shown that two-layer neural networks with square loss need $d^{\Theta(q^\star)}$ samples (Arous et al., 2021; Bietti et al., 2022; Damian et al., 2023), where $q^\star$ is the information exponent of the polynomial link function (Arous et al., 2021; Dudeja & Hsu, 2018). Such sample complexity is indeed inevitable under the correlational statistical query (CSQ) framework, leading to a computational-statistical gap for $q^\star \geq 2$.

However, the CSQ framework does not capture the fundamental limits of all gradient-based algorithms. Recent works have shown that by leveraging higher-order terms in the gradient, neural networks can learn polynomials with as few as $\widetilde{O}(d)$ samples (Lee et al., 2024; Arnaboldi et al., 2024). It turns out that the intrinsic learning difficulty is captured by another quantity called the *generative exponent* $s^\star$, which is at most 2 for polynomial link functions, and the corresponding SQ lower bound on the sample complexity is $n = \Omega(d^{s^\star/2})$[1] (Damian et al., 2024). Thus, there is no computational-statistical gap up to poly-log factors for learning polynomial single-index models. However, for general single-index models with $s^\star \geq 3$, no gradient-based algorithm for neural networks has been shown to match the SQ lower bound, leaving it an open problem (Arnaboldi et al., 2024; Lee et al., 2024).

Furthermore, learning the Gaussian single-index model can benefit from additional structures in the signal $\theta^\star$, such as sparsity, which can significantly reduce the sample complexity compared to those depending on the ambient dimension $d$ (Candès et al., 2006; Donoho et al., 2009; Raskutti et al., 2012). Recent work by Vural & Erdogdu (2024) examines the effectiveness of pruning in learning sparse features, demonstrating that it matches the correlated statistical query (CSQ) lower bound $d^{q^\star}$ for $k \ll \sqrt{d}$. However, the method fails to achieve the CSQ lower bound in non-sparse settings. For sparse single-index models with information exponent $q^\star = 1$, gradient descent on diagonal linear networks nearly achieves the information-theoretic lower bound thanks to its implicit regularization effect (Fan et al., 2023). Nonetheless, how to achieve the optimal sample complexity for general $s^\star \geq 1$ is also unknown under the sparse setting.

**Contributions.** Towards characterizing the fundamental feature learning capability of neural networks in the Gaussian single-index model, our main contributions are as follows:

1. We propose a unified recipe of gradient-based algorithms for training a two-layer neural network to learn the Gaussian single-index model. Our method integrates a general gradient oracle with a weight perturbation technique, carefully designed to exploit the underlying structure of the

---

[1]This $\Omega(d^{s^\star/2})$ sample complexity lower bound is essentially for the detection problem. Dudeja & Hsu (2021) shows that there is an estimation-detection gap for tensor PCA under the SQ framework, though it is unclear whether such gap exists universally. Throughout the paper, we always refer to the SQ lower bound as the detection lower bound, since detection in general is assumed to be easier than estimation.

Gaussian single-index model. This allows the neural network to perform feature learning of the unknown signal $\theta^\star$ in a computationally efficient manner. Our framework encompasses many existing approaches as special cases, such as batch reusing (Dandi et al., 2024; Lee et al., 2024), label transformation (Chen & Meka, 2020), and landscape smoothing (Damian et al., 2023).

2. We show that for an *unknown link distribution* $p$ with *any* generative exponent $s^\star \geq 1$, the weights of the neural network achieve strong recovery of the true signal $\theta^\star$ after training by our algorithm using $\widetilde{O}(d^{s^\star/2} \vee d)$ samples and polynomial running time. Our method achieves the SQ lower bound up to a polylogarithmic factor, and is the first gradient-based algorithm for training two-layer neural networks that attains the nearly optimal computational-statistical tradeoff for Gaussian single-index models with any $s^\star \geq 1$. Figure 1 (a) illustrates an example for $s^\star = 4$.

3. Furthermore, our method is able to take advantage of additional structural information of the true signal $\theta^\star$. Specifically, we consider the case where $\theta^\star$ is $k$-sparse for $k = o(\sqrt{d})$, and develop a *novel weight perturbation procedure* tailored to the sparsity of $\theta^\star$. Equipped with this, we show that the weights of the neural network can achieve strong recovery of the sparse signal $\theta^\star$ after training with $\widetilde{O}(k^{s^\star})$ samples in polynomial time for any generative exponent $s^\star \geq 1$. This sample complexity is also nearly optimal according to the sample complexity lower bound we establish for SQ algorithms, which might be of independent interest. Also, our method suggests a new approach to achieve the computational-statistical tradeoff for sparse tensor PCA.

In summary, our work provides a unified framework for training neural networks that can achieve the nearly optimal computational-statistical tradeoff for the Gaussian single-index model with any generative exponent $s^\star \geq 1$. Our method not only tackles the intrinsic difficulty of learning the underlying model posed by the link distribution $p$, but also leverages the additional structural information of the true singal $\theta^\star$ that benefits the learning process. Integrating these results, our method attains nearly optimal balance between computational efficiency and statistical performance across almost all regimes of sparsity levels and generative exponent $s^\star \geq 1$, as illustrated in Figure 1 (b).

## 2  PROBLEM SETUP

We begin by introducing the notation used in the paper, and then describe the problem setup. For a probability distribution $\mathbb{P}$, we denote by $L^2(\mathbb{P})$ the space of square-integrable functions with respect to $\mathbb{P}$, and $\overset{L^2(\mathbb{P})}{=}$ means equality in $L^2(\mathbb{P})$. We denote the normalized probabilist's Hermite polynomials by $\{h_s(\cdot)\}_{s \geq 0}$, where each $h_s(x) := \frac{(-1)^s}{\sqrt{s!}} \cdot e^{x^2/2} \cdot \frac{\mathrm{d}^s}{\mathrm{d}x^s} e^{-x^2/2}$. These polynomials form an orthonormal basis for $L^2(\mathcal{N}(0,1))$, i.e., the space of square-integrable functions under the Gaussian measure.

**Gaussian single-index model.** We study the following Gaussian single-index model: The environment first samples an unobservable signal $\theta^\star \sim \pi$ from some known prior $\pi \in \mathscr{P}(\mathbb{S}^{d-1})$. Then i.i.d. data $(z_1, y_1), \ldots, (z_n, y_n) \in \mathbb{R}^d \times \mathbb{R}$ are generated according to the following distribution $\mathbb{P}_{\theta^\star}$ given $\theta^\star$:

$$\mathbb{P}_{\theta^\star}: \quad z \sim \mathcal{N}(0, I_d), \quad y \sim p(\cdot \,|\, \langle \theta^\star, z \rangle). \tag{2.1}$$

Here $p(\cdot \,|\, \cdot) : \mathbb{R} \mapsto \mathscr{P}(\mathbb{R})$ is referred to as the *link distribution*. A canonical example is the additive model where $y = \phi(\langle \theta^\star, z \rangle) + \epsilon$ for some deterministic link function $\phi : \mathbb{R} \to \mathbb{R}$ and random noise $\epsilon$. See Damian et al. (2024) for more complicated examples.

**Generative exponent.** The following discussion on the generative exponent is based on the work of Damian et al. (2024). We aim to learn (2.1) where the link distribution $p$ has *generative exponent* $s^\star \geq 1$, a measure of the computational-statistical gap for learning single-index models. We let $x = \langle \theta^\star, z \rangle$. Notice that $\mathbb{P}_{\theta^\star}(y, z) = \mathbb{P}(y, x) \cdot \mathcal{N}(z^\perp; 0, I_{d-1})$ where we use $\mathbb{P}$ to denote the joint distribution of $(x, y)$ as this joint distribution is independent of $\theta^\star$. As the marginal distribution of $y$ is also independent of $\theta^\star$, we define the *null distribution* $\mathbb{Q}(y, z) := \mathcal{N}(z; 0, I_d) \otimes \mathbb{Q}(y)$ and denote $\mathbb{Q}(y, x) := \mathcal{N}(x; 0, 1) \otimes \mathbb{Q}(y)$ where $\mathbb{Q}(y) = \int_{\mathbb{R}} \mathbb{P}(y, x) \mathrm{d}x$. Under a square-integrable condition under $\mathbb{Q}$, the likelihood ratio admits a Hermite expansion with coefficient functions $\{\zeta_s(y)\}_{s \geq 1}$, i.e.,

$$\frac{\mathbb{P}_{\theta^\star}(y,z)}{\mathbb{Q}(y,z)} = \frac{\mathbb{P}(y,x)}{\mathbb{Q}(y,x)} \overset{L^2(\mathbb{Q})}{=} \sum_{s=0}^\infty \zeta_s(y) \cdot h_s(x), \quad \text{where } \zeta_s(y) = \mathbb{E}_{\mathbb{P}}[h_s(x)|y], \tag{2.2}$$

and $\mathbb{E}_{\mathbb{Q}}[\zeta_s(y)^2] \leq 1$ for all $s \geq 1$. Note that (2.2) makes sense only when we are working with the inner product of $\mathbb{P}/\mathbb{Q}$ and a square-integrable function under the null distribution $\mathbb{Q}$.

**Definition 2.1** (Generative exponent). *For the Gaussian single-index model defined in* (2.1)*, the generative exponent* $s^\star$ *of the link distribution* $p$ *is defined as* $s^\star(p) := \min\{s \geq 1 : \mathbb{E}_{\mathbb{Q}}[\zeta_s(y)^2] > 0\}$.

**Example 2.2** (Example 2.7, Damian et al. (2024)). *Consider the special case of the Gaussian single-index model* (2.1) *where* $y = \phi(\langle \theta^\star, z \rangle)$ *for a deterministic link function* $\phi : \mathbb{R} \to \mathbb{R}$. *When* $\phi$ *is a polynomial function, it holds that* $s^\star(\phi) \leq 2$, *and the equality holds if and only if* $f$ *is an even polynomial. In particular,* $s^\star(h_s) = 1$ *for odd* $s$ *and* $s^\star(h_s) = 2$ *for even* $s$. *While for the example of* $\phi(x) = x^2 \exp(-x^2)$, *which is not a polynomial, it has generative exponent* $s^\star(\phi) = 4$.

**Two-layer neural networks.** We consider using a two-layer neural network with $M$ hidden neurons to learn the single-index model (2.1). The weight vector for each neuron $m \in [M]$ is $\theta_m \in \mathbb{R}^d$, and the weights of the second layer are $a_1, \ldots, a_M \in \mathbb{R}$. We collect all the weights and denote $\boldsymbol{\theta} = (\theta_1, \ldots, \theta_M) \in \mathbb{R}^{d \times M}$, $\boldsymbol{a} = (a_1, \ldots, a_M)^\top \in \mathbb{R}^M$. Now for any input $z \in \mathbb{R}^d$, the output of the network is given by $f(z; \boldsymbol{\theta}, \boldsymbol{a}) := \sum_{m=1}^M a_m \cdot \sigma(\langle z, \theta_m \rangle)$, where $\sigma : \mathbb{R} \to \mathbb{R}$ is the activation.

## 3 OVERVIEW OF TECHNIQUES

In this work, we apply gradient-based methods to learn Gaussian single-index models, with a focus on feature learning in neural networks and the corresponding computational-statistical tradeoff. To motivate the techniques involved, we begin by discussing an illustrative example that highlights such tradeoffs. For this overview, we focus on $s^\star > 2$ and the uniform prior $\pi = \mathrm{Unif}(\mathbb{S}^{d-1})$. It has been shown that a gap exists between the information-theoretic lower bound $\Omega(d)$ and the SQ lower bound $\Omega(d^{s^\star/2})$ under this setting when $s^\star > 2$ (Bach, 2017; Damian et al., 2024).

For illustration, let us consider training a two-layer network with a single neuron under the population square loss. When the weight of the second layer is small, the rescaled negative gradient $g$ satisfies

$$g = -(2a)^{-1} \nabla_\theta \big( f(z; \theta, a) - y \big)^2 = -\big( a \cdot \sigma(\langle z, \theta \rangle) - y \big) \cdot \sigma'(\langle z, \theta \rangle) \cdot z = (y \sigma'(\langle z, \theta \rangle) + \mathrm{err}) \cdot z,$$

where we take $y\sigma'(\langle z, \theta \rangle)$ as the signal term and treat $-a\sigma(\langle z, \theta \rangle)\sigma'(\langle z, \theta \rangle)$ as the error term since it scales with $a$.[2] We ignore the error term in the following discussion. Taking expectation over $(z, y) \sim \mathbb{P}_{\theta^\star}$ and using the likelihood ratio decomposition in (2.2), we have

$$\mathbb{E}_{\mathbb{P}_{\theta^\star}}[g] \approx \underbrace{\mathbb{E}_{\mathbb{Q}}[y] \cdot \mathbb{E}_{\mathbb{Q}}[\sigma'(\langle z, \theta \rangle) \cdot z]}_{\text{bias}} + \sum_{s \geq s^\star} \underbrace{\mathbb{E}_{\mathbb{Q}}[y\zeta_s(y)] \cdot \mathbb{E}_{\mathbb{Q}}[h_s(\langle \theta^\star, z \rangle) \cdot \sigma'(\langle z, \theta \rangle) \cdot z]}_{\text{informative queries}}, \quad (3.1)$$

where we use the fact that $y$ and $z$ are independent under the null distribution $\mathbb{Q}$. Note that the *bias* term does not contain any information about $\theta^\star$, and it can be easily removed by a debiasing procedure, so we assume for simplicity that $\mathbb{E}[y] = 0$.

**Failure of vanilla online minibatch SGD.** We first consider the vanilla online minibatch SGD, which updates the weight vector $\theta$ by $\theta \leftarrow \theta - \eta \sum_{i=1}^n g_i$ for a minibatch of size $n$. The sample complexity of gradient-based methods is determined by the signal-to-noise ratio (SNR) of the one-sample gradient, which in our case is defined as $\mathrm{SNR} := \mathbb{E}[\langle g, \theta^\star \rangle]^2 / \mathbb{E}[\|g\|_2^2]$. This is the square of the alignment between $g$ and $\theta^\star$, governed primarily by the informative query corresponding to the lowest degree $s^\star$ in (3.1) assuming that $\mathbb{E}_{\mathbb{Q}}[y\zeta_{s^\star}(y)] \neq 0$. It can be shown that the inner product between the lowest-degree informative query in (3.1) and the signal $\theta^\star$ satisfies (see Lemma H.1)

$$\mathbb{E}_{\mathbb{Q}}[h_{s^\star}(\langle \theta^\star, z \rangle) \cdot \sigma'(\langle z, \theta \rangle) \cdot \langle z, \theta^\star \rangle] \approx s^\star \cdot \widehat{\sigma}_{s^\star} \cdot \langle \theta^\star, \theta \rangle^{s^\star - 1} = \widehat{\sigma}_{s^\star} \cdot O(d^{-(s^\star - 1)/2}), \quad (3.2)$$

where $\widehat{\sigma}_{s^\star}$ is the $s^\star$-th coefficient in the Hermite expansion of $\sigma$. While for $\|g\|_2$, we have

$$\mathbb{E}_{\mathbb{P}_{\theta^\star}}\big[\|g\|_2^2\big] \approx d \cdot \mathbb{E}_{\mathbb{Q}}\big[y^2 \sigma'(\langle z, \theta \rangle)^2\big] = \Omega(d),$$

where the high-order terms in the likelihood ratio decomposition are ignored and we come back to this point later. Now we can argue why vanilla online minibatch SGD has difficulty achieving the SQ lower bound for generative exponent $s^\star > 2$: Suppose $\mathbb{E}_{\mathbb{Q}}[y\zeta_{s^\star}(y)]$ and $\widehat{\sigma}_{s^\star}$ are both nonzero constants. Then the one-sample SNR is $O(d^{-s^\star})$. For a minibatch with $n$ samples, the SNR of the gradient averaged over the minibatch is roughly $n$ times the one-sample SNR[3], i.e., $nd^{-s^\star}$. To ensure

---

[2]A rigorous derivation of the error term with multiple neurons and general loss function $\ell$ can be found in the proof of Example 4.6 in Appendix C.2.2.

[3]This argument is not fully rigorous because $\mathbb{E}_{\mathbb{P}_{\theta^\star}}[\|g\|_2^2]$ also includes "bias" $\|\mathbb{E}_{\mathbb{P}_{\theta^\star}}[g]\|_2^2$ besides the fluctuations, but it remains valid as long as $\|g\|_2^2$ is dominated by fluctuations from all $d$ directions at initialization.

one update step achieves alignment, i.e., the square root of the $n$-sample SNR, $\sqrt{nd^{-s^\star}}$, exceeding the trivial $d^{-1/2}$ threshold attained by a random vector, it requires at least $d^{s^\star-1}$ samples. Note that the sample complexity would become even worse if $s^\star < \arg\min_{s \geq s^\star}\{s : \mathbb{E}_{\mathbb{Q}}[y\zeta_s(y)] \neq 0\}$. This contrasts with the sample complexity $O(d^{s^\star/2})$ suggested by the SQ lower bound.

The above failure of vanilla online minibatch SGD exposes three key challenges:

   (i) (**Non-polynomial**) How to handle the infinite sum of high-order terms in the likelihood ratio?
  (ii) (**Low SNR**) How to enhance the SNR to achieve the SQ lower bound?
 (iii) (**Zero correlation**) How to ensure that the algorithm still works if $\mathbb{E}_{\mathbb{Q}}[y\zeta_{s^\star}(y)] = 0$?

Below we discuss our techniques for addressing these challenges.

**Label transformation via general gradient oracle.** The idea to fix the zero correlation problem is to apply a nonlinear transformation $\mathcal{T} : \mathbb{R} \to \mathbb{R}$ to $y$ such that $\mathcal{T}(y)$ has nonzero correlation with $\zeta_{s^\star}(y)$. This label transformation technique has been widely used in the literature (Lu & Li, 2020; Mondelli & Montanari, 2018; Dudeja & Hsu, 2018; Chen & Meka, 2020; Damian et al., 2024). In particular, Lee et al. (2024) show that the label transformation can be automatically realized by running two gradient steps on the same batch, a technique termed as *batch-reusing* (Dandi et al., 2024; Arnaboldi et al., 2024). In this work, we study a more *general class of gradient-based methods* with gradient of form $g = \psi(y, \langle\theta, z\rangle)z$, which is an abstract form of the transformed gradient $\mathcal{T}(y)\sigma'(\langle z, \theta\rangle)z$. The desired condition becomes $\mathbb{E}_{\mathbb{Q}}[\widehat{\psi}_{s^\star-1}(y)\zeta_{s^\star}(y)] \neq 0$, where $\widehat{\psi}_s(y)$ is the $s$-th Hermite coefficient function of $\psi(y, x)$ in the Hermite basis of $x$. One particular way to obtain such a gradient is to use a modified loss function, similar to the approach in Joshi et al. (2024), while in our case the specific choice of $\psi$ is also related to the other two challenges addressed as follows.

**Exploration by weight perturbation with high-pass activation.** The low-SNR challenge corresponds to the fact that points on the equator of $\mathbb{S}^{d-1}$ orthogonal to $\theta^\star$ are all saddle points in terms of $|\langle\theta, \theta^\star\rangle|$, and random initialization typically lies near this equator. To efficiently escape from such saddle points, we perform random weight perturbation, akin to the approach in Jin et al. (2017) for non-convex optimization. To understand the effectiveness of weight perturbation, we still stick to the squared loss and the two-layer neural network for the following second-moment calculation.[4] Specifically, suppose the activation $\sigma$ is high-pass and has the lowest degree $s^\star$, i.e., $\sigma(x) = \sum_{s \geq s^\star} \widehat{\sigma}_s h_s(x)$, and consider for simplicity the case of odd $s^\star$. In the extreme case where $\theta$ is perturbed into i.i.d. pure noise $\theta_1, \ldots, \theta_L \sim \mathrm{Unif}(\mathbb{S}^{d-1})$, we compute the gradient for each $\theta_l$ and aggregate them into $g = L^{-1}(g_1 + \cdots + g_L)$. Using the properties of the Gaussian noise operator (see Appendix B for details), the second moment of this aggregated gradient satisfies

$$\mathbb{E}\left[\|g\|_2^2\right] \approx \frac{d}{L^2} \sum_{l,l'=1}^{L} \mathbb{E}_{\mathbb{Q}}[y^2] \cdot \mathbb{E}_{\mathbb{Q}}[\sigma'(\langle z, \theta_l\rangle)\sigma'(\langle z, \theta_{l'}\rangle)] \approx d \sum_{s \geq s^\star} s \cdot \widehat{\sigma}_s^2 \cdot \mathbb{E}_{\theta, \theta'}[\langle\theta, \theta'\rangle^{s-1}],$$

where $\theta, \theta'$ are drown independently from $\mathrm{Unif}(\mathbb{S}^{d-1})$. Since $\langle\theta, \theta'\rangle \approx d^{-1/2}$, we have $\mathbb{E}[\|g\|_2^2] \approx O(d^{-(s^\star-3)/2})$, yielding a higher one-sample SNR as the first moment remains unchanged and pushing the sample complexity towards the SQ lower bound. Moreover, we also see from the above calculation that the weight perturbation resolves the non-polynomial issue thanks to the near-orthogonality of the perturbed weights. The above heuristics can be made rigorous for polynomially large $L$, thereby handling non-polynomial link and activation functions.

Our approach also draws inspiration from the landscape smoothing method in Damian et al. (2024), but in constrast to their problem setup, we do not require full knowledge of the link distribution in advance. Instead, it suffices to know the generative exponent $s^\star$ to construct a high-pass activation function as well as the gradient oracle $\psi$. See Example 4.6 for a detailed discussion on this.

## 4    GRADIENT-BASED ALGORITHM FOR UNIFORM PRIOR

We first present our method and results for the case of $\theta^\star \sim \mathrm{Unif}(\mathbb{S}^{d-1})$, or equivalently, when there is no structural information on $\theta^\star$. Motivated by the discussion in Section 3, we propose a gradient-based meta algorithm (Algorithm 1) that can train a two-layer neural network to learn the unknown signal $\theta^\star$ with $\widetilde{O}(d^{s^\star/2} \vee d)$ sample complexity, nearly matching the corresponding SQ lower bound. This

---

[4]The label transformation only affects the second moment by a constant factor as we show in Appendix C.1.

---

**Algorithm 1** Meta-Algorithm for Gradient-based Feature Learning for Uniform Signal Prior

---

1: **Input**: Initialization $\boldsymbol{\theta}^{(0)} = (\theta_1^{(0)}, \ldots, \theta_M^{(0)}) \in \mathbb{R}^{d \times M}$, where $\theta_m^{(0)} \overset{\text{i.i.d.}}{\sim} \text{Unif}(\mathbb{S}^{d-1})$, $\boldsymbol{a} = a \cdot \mathbf{1} \in \mathbb{R}^M$, number of iterations $T \in \mathbb{N}$, learning rate $\eta > 0$, batch size $n \in \mathbb{N}$, polarization level $\gamma \in (0, 1)$, number of perturbation $L \in \mathbb{N}$.
2: For $t = 0, 1, \ldots, T - 1$, in the $t$-th iteration:
3:     Sample a fresh mini-batch of data $\{(z_i^{(t)}, y_i^{(t)})\}_{i=1}^n$.
4:     Perturb weights $w_{m,l}^{(t)} = (\gamma \theta_m^{(t)} + \xi_{m,l}^{(t)})/\|\gamma \theta_m^{(t)} + \xi_{m,l}^{(t)}\|_2$, $\xi_{m,l}^{(t)} \overset{\text{i.i.d.}}{\sim} \text{Unif}(\mathbb{S}^{d-1})$ for all $m, l$.
5:     Compute the gradients $g_{m,l,i}^{(t)} = (\psi(y_i^{(t)}, \langle w_{m,l}^{(t)}, z_i^{(t)} \rangle) + \text{err}_{m,l,i}^{(t)}) \cdot z_i^{(t)}$ for all $m, l, i$.
6:     Aggregate the gradients: $g_m^{(t)} = (nL)^{-1} \sum_{i=1}^n \sum_{l=1}^L (g_{m,l,i}^{(t)} - \widehat{\psi}_1(y_i^{(t)}) w_{m,l}^{(t)})$ for all $m$.
7:     Normalize the update step: $\bar{g}_m^{(t)} = g_m^{(t)}/\|g_m^{(t)}\|_2$ for all $m$.
8:     Update the weights $\theta_m^{(t+1)} = (\theta^{(t)} + \eta \bar{g}_m^{(t)})/\|\theta^{(t)} + \eta \bar{g}_m^{(t)}\|_2$ for all $m$.
9: **Output**: Final model weights $\boldsymbol{\theta}^{(T)}$.

---

meta-algorithm (Algorithm 1) covers an extensive family of gradient-based methods including the batch-reusing and gradient descent with label transformation or modified loss.

## 4.1 GRADIENT-BASED TRAINING ALGORITHM (ALGORITHM 1)

We initialize each neuron $m$ with $\theta_m^{(0)} \sim \text{Unif}(\mathbb{S}^{d-1})$, and we set $a_m^{(t)} \equiv a$ for some sufficiently small $a > 0$ throughout the training. In each iteration $t \in [T]$, we sample a new data batch of size $n$.

**Weight perturbation.** Before calculating the gradients, we first perturb the weights of each neuron to get $L$ noisy replica, by injecting uniform noise from the sphere $\mathbb{S}^{d-1}$ as in Line 4. There is a simple rule for choosing the polarization level $\gamma$. In the previous section, we discussed how $\mathbb{E}_{\mathbb{P}_{\theta^\star}}[\|g\|_2^2]$ in the one-sample SNR depends on the following quantity:

$$\mathbb{E}_{\xi_{m,l}^{(t)}, \xi_{m,l'}^{(t)}} \langle w_{m,l}^{(t)}, w_{m,l'}^{(t)} \rangle^{s^\star - 1} \lesssim (\gamma^2 \|\theta_m^{(t)}\|_2^2)^{s^\star - 1} + \mathbb{E}_{\xi_{m,l}^{(t)}, \xi_{m,l'}^{(t)}} \langle \xi_{m,l}^{(t)}, \xi_{m,l'}^{(t)} \rangle^{s^\star - 1} \approx (\gamma^2 \vee d^{-1/2})^{s^\star - 1}.$$

In this context, $\gamma^2$ represents the bias from the *exploitation* of the learned search direction, and $d^{-1/2}$ accounts for the variance from the *exploration* for the unknown signal. In fact, $\gamma$ should be set as large as possible to maximize exploitation while still ensuring that the exploration noise dominates. This balance is necessary to fully gain the SNR enhancement from weight perturbation. This gives rise to the choice $\gamma = \widetilde{\Theta}(d^{-1/4})$. Moreover, it suffices to set $L = \widetilde{\Omega}(n\sqrt{d})$ as stated in Theorem 4.2.

**Gradient aggregation and debiasing.** Then for each neuron $m$ and its perturbed weights $w_{m,l}^{(t)}$, we calculate the gradient $g_{m,l,i}^{(t)}$ for every sample (Line 5). Here, we express the gradient as a sum of an oracle term $\psi(y_i^{(t)}, \langle z_i^{(t)}, w_{m,l}^{(t)} \rangle)$ and an error term $\text{err}_{m,l,i}^{(t)}$, which we will discuss in more detail in the next paragraph. Next, we aggregate the gradients for each neuron $m$ by averaging over the $n$ samples and $L$ perturbations to get $g_m^{(t)}$ as shown in Line 6. Here we additionally subtract a term $\widehat{\psi}_1(y_i^{(t)}) w_{m,l}^{(t)}$ to debias the gradient, where $\widehat{\psi}_1(y)$ is the first coefficient function in the Hermite expansion of the oracle function $\psi(y, x)$ with respect to $x$. Finally, we update $\theta_m^{(t)}$ according to Line 7 and Line 8.

**Implementation for a general loss function.** The gradient step in Line 5 is an implicit decomposition of the actual implementation, and we present the meta-algorithm in this way for the purpose of theoretical analysis. Specifically, for a general loss function $\ell(y, f(z; \boldsymbol{\theta}, \boldsymbol{a}))$ evaluated on the two-layer neural network, the actual $g_{m,l,i}^{(t)}$ is the negative gradient of the loss rescaled by $a_m^{-1}$, i.e.,

$$g_{m,l,i}^{(t)} = -a_m^{-1} \nabla_{\theta_m} \ell(y_i^{(t)}, f(z_i^{(t)}; \{w_{m,l}^{(t)}\}, \boldsymbol{a})) = -\ell'(y_i^{(t)}, f(z_i^{(t)}; \{w_{m,l}^{(t)}\}, \boldsymbol{a})) \cdot \sigma'(\langle w_{m,l}^{(t)}, z_i^{(t)} \rangle) \cdot z_i^{(t)},$$

where we denote by $\ell'(y, f)$ the partial derivative of the loss function with respect to the output $f$. Correspondingly, the oracle term and the error term are given by

$$\psi(y_i^{(t)}, \langle w_{m,l}^{(t)}, z_i^{(t)} \rangle) = -\ell'(y_i^{(t)}, 0) \cdot \sigma'(\langle w_{m,l}^{(t)}, z_i^{(t)} \rangle),$$

$$\text{err}_{m,l,i}^{(t)} = \left( \ell'(y_i^{(t)}, 0) - \ell'\big(y_i^{(t)}, f(z_i^{(t)}; \{w_{m,l}^{(t)}\}_{m=1}^M, \boldsymbol{a})\big) \right) \cdot \sigma'(\langle w_{m,l}^{(t)}, z_i^{(t)} \rangle).$$

Note that $\text{err}_{m,l,i}^{(t)}$ can be made small by setting $\boldsymbol{a}$ to be small. Again, we emphasize that these two terms are not calculated explicitly, but rather from an implicit decomposition of the actual gradient.

Beyond the above example, the meta-algorithm encompasses more general gradient-based methods, and the theoretical results presented later are valid as long as the actual gradient can be decomposed in the same way such that the assumptions of our main theorem are satisfied.

## 4.2 FEATURE ALIGNMENT AND STATISTICAL COMPLEXITY

For any $\psi \in L^2(\mathbb{Q})$, we write its Hermite expansion as $\psi(y, x) = \sum_{s=0}^{\infty} \widehat{\psi}_s(y) \cdot h_s(y)$ in the $L^2(\mathbb{Q})$ sense, where $\widehat{\psi}_s(y)$ is the $s$-th coefficient function and $\sum_{s=0}^{\infty} \mathbb{E}_{\mathbb{Q}}[\widehat{\psi}_s(y)^2] < \infty$.

**Assumption 4.1.** *For the Gaussian single-index model in (2.1) with generative exponent $s^\star \geq 1$, the oracle $\psi : \mathbb{R} \times \mathbb{R} \to \mathbb{R}$ satisfies the following conditions:*

(a) *(Quadruple-integrable under $\mathbb{Q}$). Both $\psi(y, x) \cdot x$ and $\psi(y, x)$ belong to $L^4(\mathbb{Q})$.*

(b) *(High-pass under $\mathbb{Q}$). For all $s = 1, \ldots, s^\star - 2$, the $s$-th coefficient function is zero, i.e., $\widehat{\psi}_s(y) \equiv 0$. In addition, there exists a constant $C > 0$ such that $|\mathbb{E}_{\mathbb{Q}}[\zeta_{s^\star}(y) \cdot \widehat{\psi}_{s^\star-1}(y)]| \geq C$.*

(c) *(Polynomial-like tail under $\mathbb{P}$ and $\mathbb{Q}$). There exists a constant $C > 0$ such that for all $r \geq 1$, $\max\{\mathbb{E}_{\mathbb{P}}[|\psi(y, x)|^r], \mathbb{E}_{\mathbb{Q}}[|\psi(y, x)|^r]\} \leq C \cdot r^{Cr}.$*

The quadruple-integrability condition (Assumption 4.1(a)) ensures that the decomposition of the likelihood ratio in (2.2) is well defined for calculations involving the second moment, i.e.,

$$\mathbb{E}_{\mathbb{P}}\left[ \psi(y, x)^2 x^2 \right] = \mathbb{E}_{\mathbb{Q}}\left[ \psi(y, x)^2 x^2 \cdot \frac{\mathbb{P}(y, x)}{\mathbb{Q}(y, x)} \right] = \mathbb{E}_{\mathbb{Q}}\left[ \psi(y, x)^2 x^2 \cdot \sum_{s=0}^{\infty} \zeta_s(y) h_s(x) \right].$$

The high-pass condition (Assumption 4.1(b)) has been motivated in Section 3 and guarantees noise reduction for the second moment. The polynomial-like tail condition (Assumption 4.1(c)) is used for concentration arguments in the proof. Note that this condition is analogue to the Gaussian hypercontractivity property, where $\mathbb{E}_{x \sim \mathcal{N}(0,1)}[|f(x)|^r] \lesssim r^{Dr/2}$ if $f(x)$ is a polynomial of degree at most $D$. In particular, $\psi$ can be constructed as $\psi(y, x) = \ell'(y)\sigma'(x)$, where $\ell$ is the loss function and $\sigma$ is the activation in the two-layer network. It suffices to use a loss $\ell$ with bounded derivative and a polynomial activation $\sigma$ for the polynomial-like tail condition (see Section 4.2.1 and 4.2.2).

Now we are ready to state our first main result on the sample complexity of Algorithm 1 for uniform prior. See Appendix D for a proof sketch for even $s^\star \geq 2$ and Appendix E for a detailed proof.

**Theorem 4.2** (Sample complexity for uniform prior). *Under Assumption 4.1, set the initialization of the weights as $\theta_1^{(0)}, \ldots, \theta_M^{(0)} \overset{\text{i.i.d.}}{\sim} \text{Unif}(\mathbb{S}^{d-1})$. Suppose that the event $\mathcal{E} = \{|\text{err}_{m,l,i}^{(t)}| \leq d^{-10s^\star}, \forall (m, l, i, t) \in [M] \times [L] \times [n] \times [T]\}$ holds with probability at least $1 - O(d^{-c_0})$ for some constant $c_0 > 0$ with $(M, L, n, T)$ specified as follows. Set the learning rate $\eta > 2$, the polarization level $\gamma = d^{-1/4}(\log d)^{1/4}$, and the number of neurons $M = \Theta(1)$. Suppose*

$$n = \Theta\big((d^{s^\star/2}(\log d)^{1+s^\star/2}) \vee d(\log d)^2\big), \quad L = \Theta\big(d^{(s^\star+1)/2} \log d\big).$$

*Define $\tau = \eta^{-2}/(1 - \eta^{-1})^2$, and let $\Delta = (\log d)^{-1/2}$ if $s^\star \leq 2$ and $\Delta = d^{-1/4}(\log d)^{1/4}$ if $s^\star \geq 3$. Then with probability at least $1 - O(d^{-c})$ for some constant $c > 0$, after running Algorithm 1 for $T = O(\log d + \log(\Delta^{-1})/\log(\tau^{-1}))$ steps, there are at least $\Omega(M)$ neurons having alignment $|\langle \theta_m^{(T)}, \theta^\star \rangle| \geq 1 - O(\sqrt{\Delta})$.*

Theorem 4.2 shows that the sample complexity of Algorithm 1 is $nT = \widetilde{\Theta}(d^{s^\star/2} \vee d)$, matching the SQ sample complexity lower bound for all $s^\star \geq 1$ established by Damian et al. (2024). Compared to the partial trace method in Damian et al. (2024), our algorithm does not require special warm-start initialization. Meanwhile, the computational complexity of Algorithm 1 is $MLnT = \widetilde{\Theta}(d^{s^\star+1/2} \vee d^2)$. Indeed, one is allowed to choose $\eta = +\infty$, which resembles tensor power iteration. So far the gradient

oracle $\psi$ is still an abstract object, and next we will instantiate the above general theorem with concrete examples of $\psi$ that yield implementable algorithms. We consider the special case of polynomial link functions with $s^\star \leq 2$ in Section 4.2.1, and then the general case for any $s^\star \geq 1$ in Section 4.2.2.

**Remark 4.3** (Benefit of overparametrization). *Algorithm 1 trains a two-layer neural network with constant width $M$, involving $L$ times of perturbation for every neuron in each step. Indeed, this is equivalent to train a two-layer neural network with width $LM = \Theta\big(d^{(s^\star-1)/2} \vee (d \log d)^{1/2}\big)$, where we divide the neurons into $L$ groups, each having $M$ neurons. In each iteration we perturb the weights and compute the gradients, and then aggregate the gradients within each group of $M$ neurons. This combination of weight perturbation and* gradient sharing *exploits the* benefit of overparametrization.

### 4.2.1 ONLINE SGD WITH BATCH REUSING

The oracle function $\psi$ can be specialized to two consecutive gradient descent steps on the same batch under square loss to handle polynomial link functions corresponding to $s^\star \leq 2$ (Lee et al., 2024).

**Example 4.4** (Batch-reusing: $\psi$ for polynomial link functions). *Suppose that the link distribution is a polynomial of degree $q$. We consider $\psi$ induced by batch-reusing on single neuron, i.e., $\psi(y, x) = y\sigma'(x) + y\sigma'(x + y\sigma'(x))$ (see Section 4.2 of Lee et al. (2024) for deduction) and choose $\sigma'(x) = \sum_{i=0}^{C_q} c_i h_i(x)$ where $C_q \in \mathbb{N}$ is a constant depending only on $q$ and each $c_i \sim \text{Unif}([0,1])$.*

**Corollary 4.5** (Batch-reusing for polynomial link function). *Suppose that the link distribution is given by a polynomial link function. Under the same setups in Theorem 4.2 with the oracle $\psi$ given by Example 4.4, the sample complexity of Algorithm 1 is $\widetilde{\Theta}(d)$, recovering the result of Lee et al. (2024).*

The proof is deferred to Appendix C.2.1. However, batch-reusing may not be optimal for $s^\star \geq 3$ due to violation of the high-pass condition, necessitating a more general approach to construct $\psi$.

### 4.2.2 LABEL TRANSFORMATION VIA MODIFIED LOSS

We discuss another approach to construct $\psi$ by modifying the loss function, a universal method for arbitrary generative exponent $s^\star \geq 1$. Additional details and proofs are postponed to Appendix C.2.2.

**Example 4.6** ($\psi$ based on modified loss). *Let $\psi(y, x) = \ell'(y)\sigma'(x)$ with $\ell(y)$ being certain loss function and $\sigma(x)$ being some activation function. Such a form corresponds to the gradient of the loss $\ell(y - f(z; \boldsymbol{\theta}))$ (assuming that $\boldsymbol{a}$ is fixed and has small entries), since*

$$a_m^{-1} \nabla_{\theta_m} \ell\big(y - f(z; \boldsymbol{\theta})\big) = -\ell'\big(y - f(z; \boldsymbol{\theta})\big) \cdot \sigma'(\langle \theta_m, z \rangle) \cdot z = -\underbrace{\ell'(y) \cdot \sigma'(\langle \theta_m, z \rangle)}_{:= \psi(y, \langle \theta_m, z \rangle)} \cdot z + \text{err}_m \cdot z,$$

*where $\text{err}_m := [\ell'(y) - \ell'(y - f(z; \boldsymbol{\theta}, \boldsymbol{a}))] \cdot \sigma'(\langle \theta_m, z \rangle) = O(f(z; \boldsymbol{\theta}, \boldsymbol{a})) \cdot \sigma'(\langle \theta_m, z \rangle)$ denotes small error for sufficiently small $\boldsymbol{a}$. In Appendix C.2.2, we provide specific choice of the activation function $\sigma(x)$ (order-$s^\star$ Hermite polynomial) and the loss function $\ell(y)$ (a carefully designed random loss function), satisfying all the conditions in Assumption 4.1.*

**Corollary 4.7** (Modified loss for general $s^\star$). *The oracle $\psi$ given by Example 4.6 satisfies all the assumptions of Theorem 4.2, thus the results of Theorem 4.2 hold for Algorithm 1 using this $\psi$.*

To satisfy the high-pass condition with high probability, we construct a random loss function based on the spectral properties of $\zeta_{s^\star}$ in the likelihood ratio decomposition, as detailed in Appendix C.2.2. Furthermore, we show an equivalence between direct label transformation and modified loss functions (see Appendix C.1.1 for details). Therefore, using a random label transformation with the standard squared loss can also achieve a similar effect. (See Appendix C.4 for a discussion on the limitations.)

## 5 EXPLOITING THE STRUCTURE: ALGORITHM FOR SPARSE PRIOR

We have seen that when the prior on $\theta^\star$ is uninformative, our method nearly achieves the SQ lower bound that scales with the ambient dimension $d$. It is natural to ask whether our method can benefit from extra structural information on $\theta^\star$, one classic example being sparsity. To this end, we consider an extension of the framework in the previous section to the setting where $\theta^\star$ is a $k$-sparse vector. We first introduce the algorithm for extreme sparsity $k = o(\sqrt{d})$ and then discuss the general sparse case.

**Gaussian single-index model with sparse signal.** Given sparsity level $k = o(\sqrt{d})$, we consider the Gaussian single-index model in (2.1) with $\theta^\star$ drawn from a $k$-sparse prior:

$$\pi_k: \quad \theta^\star \mid \phi^\star \sim \text{Unif}\big(\mathbb{S}^{k-1}(\phi^\star)\big), \quad \phi^\star \sim \text{Unif}(\mathcal{S}_k), \tag{5.1}$$

where $\mathcal{S}_k := \{\phi \subset [d] : |\phi| = k\}$ is the collection of all $k$-sparse support sets, and $\mathbb{S}^{k-1}(\phi) := \{x \in \mathbb{R}^d : \sum_{i \in \phi} x_i^2 = 1, x_j = 0, \forall j \notin \phi\}$ is the associated $k$-dimensional unit sphere for any $\phi \in \mathcal{S}_k$.

We show that Algorithm 1 for the uniform prior can be easily adapted to leverage the sparsity of $\theta^\star$, striking a nearly optimal computational-statistical balance with $\widetilde{O}(k^{s^\star})$ sample complexity.

### 5.1 ALGORITHM DESIGN: HOW TO LEVERAGE SPARSITY?

Note that Algorithm 1 can also learn the $k$-sparse Gaussian single-index model, albeit with $\widetilde{O}(d^{s^\star/2} \vee d)$ samples, which is apparently suboptimal in light of the classic example of sparse linear regression. Here the key challenge is *support identification* of $\phi^\star$, and the issue of Algorithm 1 lies in the weight perturbation using noise $\xi \sim \text{Unif}(\mathbb{S}^{d-1})$, thus unaware of the sparsity of $\theta^\star$. Below we discuss how to calibrate the weight perturbation with the sparse prior.

**Perturbation by replicating the prior is not enough.** An intuitive first-cut attempt is to use perturbation noise from the same distribution as the sparse prior $\pi_k$ in (5.1), which turns out to be suboptimal as well. To illustrate this, we assume for simplicity a balanced $\theta^\star$ where every nonzero entry of $\theta^\star$ is equal to $k^{-1/2}$, and consider i.i.d. $\xi_1, \ldots, \xi_L \sim \pi_k$. For each $j \in \phi^\star$, consider the $j$-th entry of the lowest-degree informative query (analogous to (3.2)), whose first moment satisfies

$$\mathbb{E}_{\mathbb{P}_{\theta^\star}}[g_j] \approx \mathbb{E}_{\mathbb{Q}}[y\zeta_s(y)] \cdot s^\star \hat{\sigma}_{s^\star} \cdot \frac{1}{L} \sum_{l=1}^L \langle \theta^\star, \theta_l \rangle^{s^\star-1} \theta_j \approx \frac{\mathbb{E}_{\theta \sim \pi_k}[\langle \theta^\star, \theta \rangle^{s^\star-1}]}{\sqrt{k}} \simeq \frac{k^2}{d} \cdot \frac{k^{-(s^\star-1)}}{\sqrt{k}},$$

where the last step follows from direct calculation for $\theta \sim \pi_k$. Similarly, the second moment satisfies

$$\mathbb{E}_{\mathbb{P}_{\theta^\star}}[g_j^2] \approx \sum_{s \geq s^\star} s \cdot (\hat{\sigma}_s)^2 \cdot \mathbb{E}_{\theta, \theta' \sim \pi_k}[\langle \theta, \theta' \rangle^{s-1}] \simeq \frac{k^2}{d} \cdot k^{-(s^\star-1)},$$

where $\theta$ and $\theta'$ are drawn independently from the prior $\pi_k$. This calculation implies that the fluctuation of each entry of the aggregated gradient is of order $\sqrt{k^2/d} \cdot k^{-(s^\star-1)/2} \cdot n^{-1/2}$ for batch size $n$. To successfully identify the true support $\phi^\star$, the signal must be larger than the fluctuation for entries in $\phi^\star$, resulting in a sample complexity of $n = \widetilde{O}(k^{s^\star} \cdot d/k^2)$. In comparison to the SQ lower bound in Theorem 5.4, this is suboptimal by a factor $d/k^2$. However, for $s^\star = 1$, we observe that $\mathbb{E}_{\mathbb{P}_{\theta^\star}}[g_j] = \Omega(k^{-1})$ for $j \in \phi^\star$ and $\mathbb{E}_{\mathbb{P}_{\theta^\star}}[g_j^2] = O(1)$, indicating that the support $\phi^\star$ can still be identified using $\widetilde{O}(k)$ samples. The form of the perturbation does not matter here since both $\langle \theta^\star, \theta \rangle$ and $\langle \theta, \theta' \rangle$ are degree zero in terms of $k^{-1}$. Therefore, we conjecture that our algorithm succeeds for $s^\star = 1$ even without weight perturbation as outlined in Conjecture 5.3.

**Perturbation by groups that cover the prior.** The suboptimality of the previous stragety originates from the fact that $\phi^\star$ is sampled from a uniform distribution over $\binom{d}{k}$ different $k$-sparse support sets, making it unlikely for two independent sets to overlap (only with $k^2/d$ probability). Then how to perturb the weights in a way that guarantees a significant overlap with $\phi^\star$? The solution is to construct *a polynomial-size cover* for the prior $\pi_k$. Specifically, we divide $\mathcal{S}_k$ into $d$ subsets, where the $j$-th subset is define as $\mathcal{S}_{k,j} := \{\phi \in \mathcal{S}_k \mid j \in \phi\}$, which contains all $k$-sparse support sets that include the $j$-th coordinate. Now suppose $\theta^\star \in \mathcal{S}_{k,j}$, then for any perturbed weight $\theta_l$ with support from the same subset $\mathcal{S}_{k,j}$, its support overlaps with $\phi^\star$ almost surely, thereby eliminating the $d/k^2$ factor.

In particular, considering a two-layer neural network with width $d$, the above strategy can be carried out by perturbing each neuron $m$ using $\theta_{m,1}, \ldots, \theta_{m,L}$ whose support sets are sampled from the same group $\mathcal{S}_{k,m}$. As a result, at least $k$ neurons will have one or more overlapping coordinates with the true signal $\theta^\star$. For these neurons, the signal in the aggregated gradient would be strong enough for simple thresholding methods to correctly identify the true support $\phi^\star$ with $\widetilde{O}(k^\star)$ samples. We further refine this by first projecting the aggregated gradient for each neuron onto its top-$k$ support, and then selecting the *strongest* projected gradient to update the weights.

Combining these yields Algorithm 2 for the sparse case, where we define the support projection matrix $P_\phi := \sum_{i \in \phi} e_i e_i^\top$ and the top-$k$ operator $\text{Top}_k(v) := \text{argmax}_{\phi \subset \mathcal{S}_k} \|P_\phi(v)\|_1$, which extracts the $k$-sparse support $\phi$ corresponding to the largest (in absolute value) $k$ entries of $v$. We set the polarization level $\gamma = k^{-1/2}$, following the same balance between exploitation and exploration as in the uniform case, since the exploration noise is now of order $k^{-1}$.

---

**Algorithm 2** Meta Algorithm for Gradient-based Feature Learning for Sparse Signal Prior

---

1: **Input**: Initialization $\boldsymbol{\theta}^{(0)} = (\theta_1^{(0)}, \ldots, \theta_M^{(0)}) \in \mathbb{R}^{d \times M}$, where $\theta_m^{(0)} = e_m$, $\boldsymbol{a} = a \cdot \mathbf{1} \in \mathbb{R}^M$ with number of neurons $M = d$, number of iterations $T \in \mathbb{N}$, batch size $n \in \mathbb{N}$, polarization level $\gamma \in (0, 1)$, number of perturbation $L \in \mathbb{N}$.
2: **for** iteration $t = 0, 1, \ldots, T-1$ **do**
3:      Sample a fresh mini-batch $\{(z_i^{(t)}, y_i^{(t)})\}_{i=1}^n$.
4:      Perturb as Line 4 in Algorithm 1 with $\xi_{m,l}^{(t)} \overset{\text{i.i.d.}}{\sim} \text{Unif}(\mathbb{S}^{k-1}(\phi_{m,l}))$ and $\phi_{m,l} \overset{\text{i.i.d.}}{\sim} \text{Unif}(\mathcal{S}_{k,m})$.
5:      Compute and aggregate the gradients to get $g_m^{(t)}$, same as Line 5&6 in Algorithm 1.
6:      Find the top-$k$ support of $g_m^{(t)}$ and project: $\phi_m^{(t)} = \text{Top}_k(g_m^{(t)})$, $\widetilde{g}_m^{(t)} = P_{\phi_m^{(t)}}(g_m^{(t)})$ for all $m$.
7:      Locate the neuron with the largest $\|\widetilde{g}_m^{(t)}\|_2$: $\widehat{m} = \text{argmax}_m \|\widetilde{g}_m^{(t)}\|_2$.
8:      Update weights by gradient sharing: $\theta_m^{(t+1)} = \widetilde{g}_{\widehat{m}}^{(t)} / \|\widetilde{g}_{\widehat{m}}^{(t)}\|_2$ for all $m$.
9: **end for**
10: **Output**: Model weights $\boldsymbol{\theta}^{(T)}$.

---

## 5.2 SAMPLE COMPLEXITY ANALYSIS FOR SPARSE PRIOR

**Theorem 5.1** (Sample complexity for sparse prior). *Under Assumption 4.1, consider the sparse prior in (5.1) with sparsity level $k$ satisfying $\omega(d^\iota) < k < o(\sqrt{d})$ for a small $\iota > 0$. Suppose that the event $\mathcal{E} = \{|\text{err}_{m,l,i}^{(t)}| \leq d^{-10s^\star}, \forall (m, l, i, t) \in [M] \times [L] \times [n] \times [T]\}$ holds with probability at least $1 - O(d^{-c_0})$ for some constant $c_0 > 0$ with $(M, L, n, T)$ specified as follows. Let $\gamma = k^{-1/2}$, $n = \Omega((k \log^3 k)^{s^\star} \cdot \log d)$, and $L = \Omega(k^{(s^\star+3)/2} \cdot \log(k)^{s^\star-1})$. Then with probability at least $1 - O(k^{-c})$ for some $c > 0$, after running Algorithm 2 with $T = 2$ iterations, there are at least $\Omega(M)$ neurons that have alignment $|\langle \theta_m^{(T)}, \theta^\star \rangle| \geq 1 - O(\Delta)$, where $\Delta = k^{-1} \vee (k^{-(s^\star-1)/2} \cdot \log(k)^{-3/2}) = o(1)$.*

Theorem 5.1 shows that the sample complexity of Algorithm 2 is $n = \widetilde{O}(k^{s^\star})$, matching the SQ lower bound established in Theorem 5.4 which will be presented below. Here for simplicity, we essentially use an infinitely large learning rate when updating the weights in Line 8 of Algorithm 2, so it takes only two iterations to achieve strong alignment. This is equivalent to running the same algorithm with a finite learning rate but with a larger number of iterations, which is omitted for brevity.

**Remark 5.2** (Implication for sparse tensor PCA). *The connection between the Gaussian single-index model and tensor PCA has been discussed in Damian et al. (2023), by showing that estimating $\theta^\star$ corresponds to a tensor PCA problem defined over the empirical Hermite tensors. Our weight perturbation technique can be potentially applied to iteratively solve sparse tensor PCA problems.*

**Conjecture 5.3.** *If $d^\iota < k < o(d)$ for a small $\iota > 0$ and $s^\star = 1$, Algorithm 2 succeeds with sample complexity $n = \widetilde{O}(k)$. Furthermore, the same guarantee applies even without perturbing the weights.*

Finally, we present the following SQ lower bound for the sparse prior, complementing Theorem 5.1.

**Theorem 5.4** (SQ lower bound). *Consider the Gaussian single-index model in (2.1) with generative exponent $s^\star \geq 1$. Suppose $\theta^\star$ is $k$-sparse for $\omega((\log d)^2) \leq k \leq d/2$. Take $c > 2$ as a constant. For any (stochastic) algorithm using $\exp(\Omega((\log d)^c))$ calls to the $\text{VSTAT}(\mathbb{P}_{\theta^\star}, n)$ oracle with sample size $n$, in order to achieve nontrivial alignment $|\langle \widehat{\theta}, \theta^\star \rangle| > \rho$ with probability at least $2/3$, it requires*

$$n \gtrsim \frac{k^{s^\star}}{(\log d)^{cs^\star}}, \quad \text{where} \quad \rho = \widetilde{\omega}(k^{-1}) \quad \text{if } (\log d)^2 < k < \sqrt{d(\log d)^c}, \tag{5.2}$$

$$n \gtrsim \frac{d^{s^\star/2}}{(\log d)^{cs^\star/2}}, \quad \text{where} \quad \rho = \widetilde{\omega}(d^{-1/2}) \quad \text{if } \sqrt{d(\log d)^c} \leq k \leq d/2. \tag{5.3}$$

More discussion on Theorem 5.4 is available in Remark G.8. For $k = o(\sqrt{d})$, running Algorithm 2 will succeed with $\widetilde{O}(k^{s^\star})$ samples, matching the lower bound in (5.2) for every $s^\star \geq 1$. For $k = \Omega(\sqrt{d})$, running Algorithm 1 will succeed with $\widetilde{O}(d^{s^\star/2})$ samples, matching the lower bound in (5.3) for $s^\star \geq 2$. In addition, for $k = \Omega(\sqrt{d})$ with $s^\star = 1$, we conjecture $n = \widetilde{O}(k)$ samples to be sufficient, where the information-theoretic lower bound is $\Omega(k \log(d/k))$ (Neykov et al., 2016). This gives rise to the paradigm in Figure 1 (b).

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

CONTENTS

## A   NUMERICAL EXPERIMENTS

We conduct extensive simulation experiments to validate the sample complexity result established in
Theorem 4.2. In specific, for a fixed Gaussian single-index model, we run Algorithm 1 extensively
over a variety of problem instances with diverse scales and report the average accuracy in terms of
the alignment. We lay out the details of the experiment setting as follows.

• **Gaussian single-index model.** We focus on the Gaussian single-index model introduced in (2.1)
with a deterministic link function $p$ and Gaussian additive noise. Here we set $p(x) = x^2 \cdot \exp(-x^2) + \epsilon$
where $\epsilon \sim \mathcal{N}(0, \sigma^2)$ with $\sigma^2 = 0.5$. As shown in Example 2.2, the generative exponent of the function
$p$ is $s^\star(p) = 4$. In addition, the signal parameter $\theta^\star$ is uniformly sampled from the unit sphere in $\mathbb{R}^d$.

• **Neural network architecture.** We adopt the two-layer neural network introduced in Section 2 with
$M$ set to 15 and $a_m = 1$ for all $m \in [M]$ in all experiments. Since $s^\star(p) = 4$, we set the activation
function as $\sigma(x) = h_4(x)$, i.e., the fourth-order Hermite polynomial.

• **Training using Algorithm 1.** In Algorithm 1, we set $\psi(x, y) = y \cdot \sigma'(x)$, as stated in Example 4.6.
Such a $\psi$ is justified by considering the following alignment loss:

$$L(\boldsymbol{\theta}) = 1 - y \cdot f(z; \boldsymbol{\theta}, \boldsymbol{a}) = 1 - \sum_{m=1}^{M} a \cdot \sigma(\langle z, \theta_m \rangle) \cdot y, \tag{A.1}$$

where recall that each entry of $\boldsymbol{a}$ is equal to $a$. As a result, by (A.1) we have

$$a^{-1} \cdot \nabla_{\theta_m} L(\boldsymbol{\theta}) = y \cdot \sigma'(\langle z, \theta_m \rangle) \cdot z = \psi(\langle z, \theta_m \rangle, y) \cdot z.$$

As a result, we can alternatively interpret the gradients in Algorithm 1 as those with respect to the
alignment loss $L(\boldsymbol{\theta})$. Thus, the choice of $a$ does not matter in this case, and we set $a = 1$ for simplicity.
Furthermore, other details of Algorithm 1 are specified as follows:

- The parameters $\{\theta_n\}_{m \in [M]}$ are initialized as i.i.d. random vectors in $\mathbb{R}^d$ uniformly sampled
  from the unit sphere.
- We fix $M = 15$, $a = 1$, $\eta = 3$, $T = 24$, and $L = 500$ throughout all experiments with
  different values of $n$ and $d$.

- We enumerate $n$ and $d$ over a grid with $d \in [32, 499]$ and $n \in [5 \times 10^3, 3 \times 10^6]$. Note that $\log d \in (3, 7)$, our choice of $T$ satisfies the requirement in Theorem 4.2.

• **Choices of** $(d, n)$. We select 40 different values of $d$ and 30 different values of $n$ within the ranges $d \in [32, 499]$ and $n \in [5 \times 10^3, 3 \times 10^6]$, respectively. These values form an evenly spaced grid in terms of $\log n$ and $\log d$. See Figure 2 for an illustration.

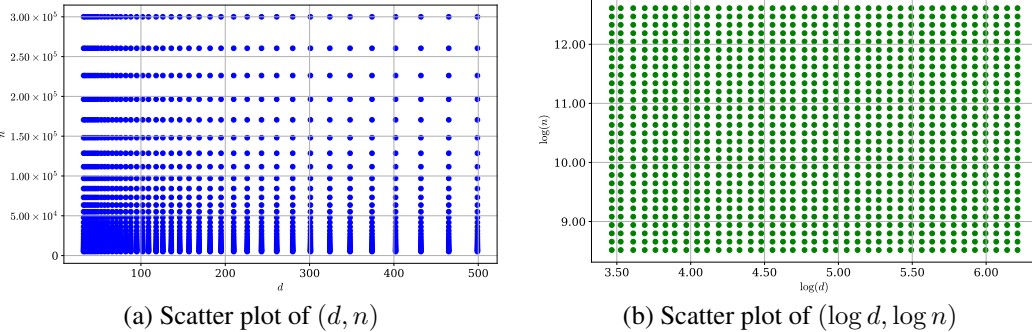

(a) Scatter plot of $(d, n)$          (b) Scatter plot of $(\log d, \log n)$

Figure 2: Scatter plots of $(d, n)$ and $(\log d, \log n)$. In (a) we plot $n$ against $d$ and in (b) we plot $\log n$ agains $\log d$. As shown in (b), we choose $n$ and $d$ such that they form an evenly-spaced grid after logarithm.

• **Evaluation.** We report the accuracy of Algorithm 1 based on 25 repeated experiments for every choice of $(n, d)$. We report two types of accuracy metrics:

(i) Average accuracy: We report $M^{-1} \sum_{m=1}^{M} |\langle \theta_m, \theta^\star \rangle|$ in each experiment and then average over the 25 experiments.

(ii) Top-8 accuracy: Given $\{\theta_m\}_{m \in [M]}$ returned by the algorithm, we sort the alignment values $\{|\langle \theta_m, \theta^\star \rangle|\}_{m \in [M]}$. Then we report the average of the largest 8 numbers. The rationale is that if the top-8 accuracy is close to one, at least half of the neurons correctly find $\theta^\star$.

**Contour plots.** After calculating these two versions of accuracy for every $(d, n)$ pair, we generate the contour plots based on $(\log d, \log n, \mathrm{acc}(d, n))$, where $\mathrm{acc}(d, n)$ is one of the two versions of average accuracy introduced above. We report these two contour plots in Figure 3 and Figure 4, where in Figure 3 we zoom in to a smaller range of $d$ for better visualization. In these plots, points with the same color indicate $(\log d, \log n)$ with the same level of accuracy.

**Validate** $\widetilde{\Theta}(d^{s^\star/2})$ **sample complexity.** As shown in these figures, the average accuracy and the top-8 accuracy clearly exhibit **a linear relationship**. That is, for a fixed accuracy level $\delta$, $(d, n)$ satisfying $\mathrm{acc}(d, n) = \delta$ is a line segment. That is, $\log n = c_1 \cdot \log d + c_2$. To determine $c_1$ and $c_2$, we further fit linear models for $(\log d, \log n)$ with the same accuracy level $\delta$, where $\delta \in \{0.6, 0.7, 0.8\}$. For both the average accuracy and the top-8 accuracy, the coefficient $c_1$ in the linear models is close to 2. We report the linear models corresponding to different accuracy levels in Table 1. This finding indicates that $n \propto d^2$. Note that $s^\star = 4$. Moreover, since we compute the accuracy for all $(d, n)$ on the grid. The fact that $c_1 \approx 2$ indicates that the $\widetilde{\Theta}(d^{s^\star/2})$ sample complexity is sharp.

Table 1: Fitted linear equations of the form $\log n = c_1 \cdot \log d + c_2$ for $n, d$ with the desired accuracy level. Notably, the slopes of these equations are all close to $s^\star/2 = 2$, which shows that $n \propto d^{s^\star/2}$.

| Accuracy level | Average accuracy | Top-8 accuracy |
|---|---|---|
| 0.8 | $\log(n) = 1.9058 \cdot \log(d) + 4.4516$ | $\log(n) = 1.8201 \cdot \log(d) + 4.6218$ |
| 0.7 | $\log(n) = 1.9103 \cdot \log(d) + 4.3273$ | $\log(n) = 1.9343 \cdot \log(d) + 4.0790$ |
| 0.6 | $\log(n) = 1.9640 \cdot \log(d) + 4.0361$ | $\log(n) = 1.9653 \cdot \log(d) + 3.8901$ |

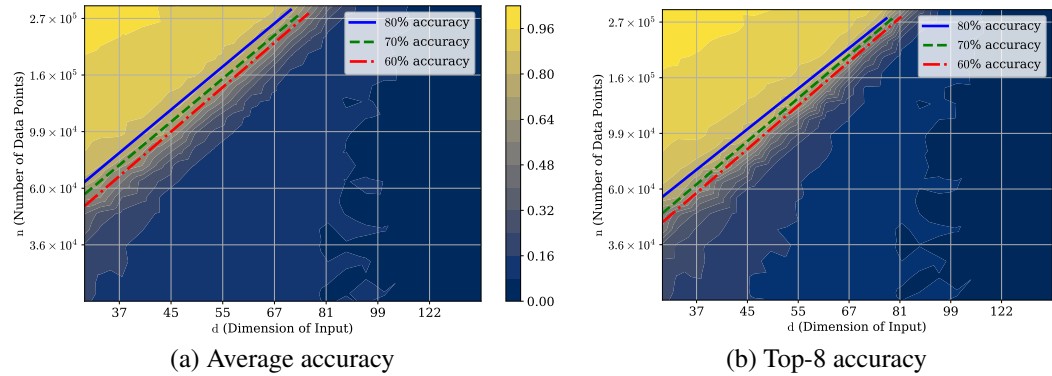

(a) Average accuracy                    (b) Top-8 accuracy

Figure 3: The contour plots of $(\log d, \log n, \mathtt{acc}(d, n))$, where $\mathtt{acc}(d, n)$ is either the average accuracy and top-8 accuracy. Here we zoom in to a smaller subset of $d$'s for better visualization. We also plot the lines containing $(\log d, \log n)$ with the same accuracy level among $\{0.6, 0.7, 0.8\}$. The slopes of these lines are all close to 2. This indicates that $n \approx d^2$ samples are sufficient and necessary for accurate estimation.

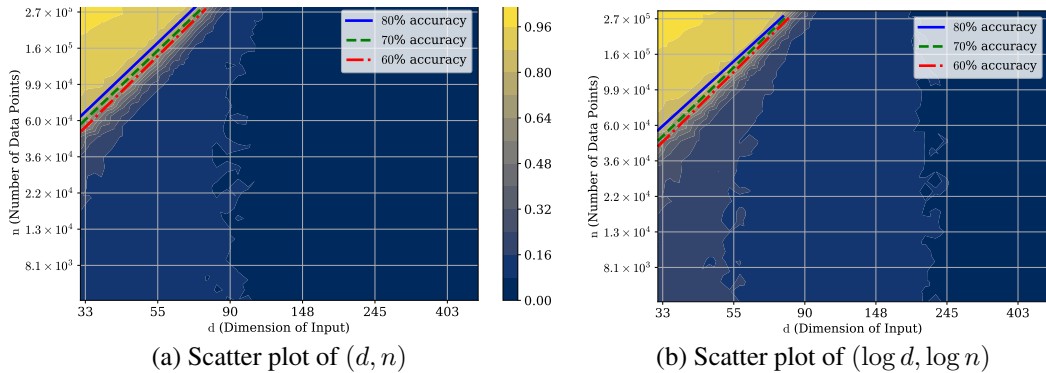

(a) Scatter plot of $(d, n)$              (b) Scatter plot of $(\log d, \log n)$

Figure 4: The contour plots of $(\log d, \log n, \mathtt{acc}(d, n))$, where $\mathtt{acc}(d, n)$ is either the average accuracy and top-8 accuracy. We also plot the lines containing $(\log d, \log n)$ with the same accuracy level among $\{0.6, 0.7, 0.8\}$. The slopes of these lines are all close to 2. This indicates that $n \approx d^2$ samples are sufficient and necessary for accurate estimation.

## B NOTATION, PRELIMINARIES AND RELATED WORKS

**Notations.** We use $\mathbb{N}$ to denote the set of positive integers and $\mathbb{N}_0$ to denote the set of nonnegative integers. For vector $z \in \mathbb{R}^d$, we denote by $\mathbb{R}_n[z]$ the set of polynomials of degree at most $n$ in $z$ with real coefficients. For $s \in \mathbb{N}$, we denote by $\Pi_s$ the symmetric group of all permutations of $[s]$. We denote by $\mathcal{N}_d(\cdot)$ and $\mathcal{N}(\cdot)$ the standard normal distribution in $\mathbb{R}^d$ and $\mathbb{R}$, respectively.

For two tensors $S \in (\mathbb{R}^d)^{\otimes s}$ and $T \in (\mathbb{R}^d)^{\otimes t}$ where $s \geq t$,

$$(S[T])_{j_1, \dots, j_{s-t}} := \sum_{i_1, \dots, i_t = 1}^{d} S_{j_1, \dots, j_{s-t}, i_1, \dots, i_t} T_{i_1, \dots, i_t}.$$

Here, $S[T]$ produces a tensor of order $s - t$ and dimension $d$. We also define the symmetrization operation for a tensor $T \in (\mathbb{R}^d)^{\otimes t}$ as

$$\mathrm{Sym}(T)_{i_1, \dots, i_t} := \frac{1}{t!} \sum_{\pi \in \Pi_t} T_{i_{\pi(1)}, \dots, i_{\pi(t)}}.$$

The followings are some notations for the relationship between two quantities (or matrices):

$a \simeq b$: There exists a constant $C = O(1)$ such that $a \leq Cb$ and $b \leq Ca$. Note that $a$ and $b$ should have the same sign. $a = \Theta(b)$ also has the same meaning.

$a \cong b$: $a \leq \mathrm{polylog}(d) \cdot b$ and $b \leq \mathrm{polylog}(d) \cdot a$, and the same for $a = \widetilde{\Theta}(b)$.

$a \lesssim b$: There exists a constant $C = O(1)$ such that $a \leq Cb$, and the same for $a = \Omega(b)$. The use of $a \gtrsim b$ is similar.

$a \lessapprox b$: $a \leq \mathrm{polylog}(d) \cdot b$, and the same for $a = \widetilde{O}(b)$. The use of $a \gtrapprox b$ and $a = \widetilde{\Omega}(b)$ is similar.

$a \ll b$: $a \leq (\mathrm{polylog}(d))^{-1} \cdot b$. The use of $a \gg b$ is similar.

In addition, we denote by $a = b \pm \varepsilon$, $a \simeq b \pm \varepsilon$, $a \cong b \pm \varepsilon$ that $b - \varepsilon \leq a \leq b + \varepsilon$, $a - \varepsilon \lesssim b \lesssim a + \varepsilon$, $a - \varepsilon \lessapprox b \lessapprox a + \varepsilon$, respectively.

For square matrices $A$ and $B$, $A \preceq B$ means that $B - A$ is positive semi-definite, and $A \precsim B$ means that there exists a constant $C = O(1)$ such that $C \cdot B - A$ is positive semidefinite.

## B.1 BACKGROUND ON HERMITE POLYNOMIALS

The probabilist's Hermite polynomials satisfy the following recurrence relations

$$h_s(x)' = \sqrt{s} \cdot h_{s-1}(x), \quad x \cdot h_s(x) = \sqrt{s+1} \cdot h_{s+1}(x) + \sqrt{s} \cdot h_{s-1}(x), \tag{B.1}$$

where we adopt the convention that $h_{-1}(x) \equiv 0$.

For any function $f \in L^2(\mathcal{N}(0,1))$, its Hermite expansion is given by

$$f(x) = \sum_{s=0}^{\infty} \widehat{f}_s \cdot h_s(x),$$

where we denote by $\widehat{f}_s$ the $s$-th coefficient of the Hermite expansion of $f$.

**Gaussian noise operator.** For $\rho \in [-1, 1]$, define the Gaussian noise operator as

$$\mathrm{U}_\rho f(x) = \mathbb{E}_{x' \sim \mathcal{N}(0,1)}\big[f(\rho x + \sqrt{1 - \rho^2} \cdot x')\big].$$

Proposition 11.37 of O'Donnell (2014) shows that the Hermite expansion of $\mathrm{U}_\rho f$ is given by

$$\mathrm{U}_\rho f(x) \overset{L^2(\mathcal{N}(0,1))}{=} \sum_{s=0}^{\infty} \rho^s \cdot \widehat{f}_s \cdot h_s(x).$$

A direct implication of this identity is

$$\mathbb{E}_{x \sim \mathcal{N}(0,1)}[\mathrm{U}_\rho f(x) g(x)] = \mathbb{E}_{x \sim \mathcal{N}(0,1)}[f(x) \mathrm{U}_\rho g(x)] = \sum_{s=0}^{\infty} \rho^s \widehat{f}_s \widehat{g}_s. \tag{B.2}$$

As a result, for any fixed $w, \theta \in \mathbb{S}^{d-1}$, it holds that

$$\mathbb{E}_{z \sim \mathcal{N}(0,I_d)}\left[f(\langle w, z \rangle) h_s(\langle \theta, z \rangle)\right] = \mathbb{E}_{x \sim \mathcal{N}(0,1)}\left[\mathrm{U}_\rho f \cdot h_s(x)\right] = \langle w, \theta \rangle^s \cdot \widehat{f}_s. \tag{B.3}$$

**Hermite tensor.** Corresponding to the Hermite polynomials defined for scalar variables, we define the Hermite tensors over $z \in \mathbb{R}^d$:

$$\boldsymbol{h}_s(z) := \frac{(-1)^s}{\sqrt{s!}} \cdot e^{\|z\|_2^2/2} \cdot \nabla^s e^{-\|z\|_2^2/2} \in (\mathbb{R}^d)^{\otimes s}, \text{ for } s \geq 0.$$

The scalar-valued Hermite polynomials and the tensor-valued Hermite tensors are related as follows:

$$h_s(\langle \theta, z \rangle) = \boldsymbol{h}_s(z)[\theta^{\otimes s}], \quad \forall \theta \in \mathbb{S}^{d-1}. \tag{B.4}$$

Now let $f : \mathbb{R}^d \to \mathbb{R}$ be a $s$-times differentiable function such that for all $k \leq s$, every component of $\nabla^k f$ belongs to $L^2(\mathcal{N}(0, I_d))$. Then it follows from integration by parts that

$$\mathbb{E}_{z \sim \mathcal{N}(0,I_d)}\left[f(z) \boldsymbol{h}_s(z)\right] = \frac{1}{\sqrt{s!}} \cdot \mathbb{E}_{z \sim \mathcal{N}(0,I_d)}\left[\nabla^s f(z)\right]. \tag{B.5}$$

This is a version of Stein's lemma for tensor-valued functions.

### B.2 Additional Related Works

**Related Works.** Our work contributes to the recent research on the computation-statistical tradeoff in learning single-index models. The information-theoretic limit for estimating the latent signal is $n = \Omega(d)$ (Bach, 2017; Damian et al., 2024), but the sample complexity lower bound varies across computational models, potentially revealing a computational-statistical gap.

The information exponent $q^\star$ (Dudeja & Hsu, 2018; Arous et al., 2021) governs the sample complexity for learning Gaussian single-index models in the CSQ framework (Chen et al., 2020; Bietti et al., 2022; Damian et al., 2022; Dandi et al., 2023; Abbe et al., 2023; Ba et al., 2023). Notably, Arous et al. (2021) show that online SGD has a sample complexity of $n = \widetilde{O}(d^{q^\star - 1})$, which is worse than the CSQ lower bound $n = \Omega(d^{q^\star/2})$ (Abbe et al., 2023; Damian et al., 2022). This gap can be closed by a loss landscape smoothing technique (Damian et al., 2023) originally developed for tensor PCA (Anandkumar et al., 2017; Biroli et al., 2020).

Our work extends beyond the CSQ framework, aligning with more general SQ algorithms (Feldman et al., 2017; Feldman, 2017), where the sample complexity lower bound is $\Omega(d^{s^\star/2})$, with $s^\star$ as the generative exponent (Damian et al., 2024). In this context, online SGD with batch reusing suffices for learning polynomial link functions (Dandi et al., 2024; Lee et al., 2024), while for $s^\star \geq 3$, only the partial trace estimator proposed by Damian et al. (2024) can match the SQ lower bound.

In the sparse setting, including sparse linear models (Vaskevicius et al., 2019; Zhao et al., 2022; Gamarnik & Zadik, 2017), sparse PCA (Arous et al., 2020), and planted models (Bandeira et al., 2022), computational-statistical gaps also exist. For example, in matrix PCA, the best rank-1 estimator achieves a near-optimal sample complexity of $\widetilde{O}(d)$ due to the BBP transition (Baik et al., 2005; Choo & d'Orsi, 2021). Under extreme sparsity, however, sparse estimators require $\widetilde{O}(k^2)$ samples, utilizing methods such as diagonal thresholding (Johnstone & Lu, 2009) or semidefinite relaxation (d'Aspremont et al., 2004). These approaches improve upon the $\widetilde{O}(d)$ sample complexity but reveal a notable computational-statistical gap relative to the information-theoretic lower bound of $\Omega(k \log d)$ (Wang et al., 2016).

Related to our work, Fan et al. (2023) provide a $\widetilde{O}(k)$ sample complexity for learning single index models with $q^\star = 1$ using diagonal linear networks, and Neykov et al. (2016) report a $\widetilde{O}(k^2)$ result for phase retrieval where $q^\star = 2$. However, as previously noted, the information exponent does not fully characterize the intrinsic computational-statistical tradeoff. Our work completes the picture by providing a gradient-based framework that simultaneously handles all sparsity levels and any generative exponent $s^\star \geq 1$.

On the other hand, Ba et al. (2023) and Mousavi-Hosseini et al. (2023) demonstrate that additional structures in the data, such as spiked covariance in the input, can break the $d^{p^\star}$ sample complexity lower bound compared to the standard Gaussian single-index model. A promising direction for future research is to explore whether similar structural properties could be utilized to surpass the $d^{s^\star/2}$ sample complexity lower bound in the SQ setting.

Other works show manifold method can break CSQ or SQ lower bounds. Mousavi-Hosseini et al. (2024) investigate learning multi-index models using two-layer neural networks trained with the Mean-Field Langevin Algorithm (MFLA). In a Riemannian setting, constraining weights to a compact manifold allows MFLA to achieve polynomial-time convergence while surpassing the CSQ sample complexity lower bound.

## C Supplementary Proofs for the Main Context

### C.1 Proofs for Section 3

In this section, we first argue why $\mathbb{E}_{x \sim \mathcal{N}(0,1)}[y\zeta_{s^\star}(y)] = 0$ is the major difficulty for vanilla (stochastic) gradient descent to achieve the information-theoretical lower bound $O(d)$ (the same for SQ lower bound) when the information exponent $q^\star$ is larger than 2. It has been shown by Damian et al. (2024) that the generative exponent $s^\star$ for polynomial model is either 1 or 2. Consider the information exponent $q^\star > 2$. We have the following lemma saying that the correlation $\mathbb{E}_{x \sim \mathcal{N}(0,1)}[y\zeta_s(y)] = 0$ for any $s < q^\star$.

**Lemma C.1.** *Recall that $\zeta_s$ is the coefficient function for degree $s$ in the decomposition of the likelihood ratio $\mathbb{P}(x, y)/\mathbb{Q}(x, y)$ in (2.2). For any $q^\star \geq 2$, consider the Gaussian single-index model given by $y = \beta_0 + \sum_{p \geq p^\star} \beta_p h_p(x)$ with $x \sim \mathcal{N}(0, 1)$. Then for any $1 \leq s < p^\star$, $\mathbb{E}_{\mathbb{Q}}[y\zeta_s(y)] = 0$.*

*Proof.* The proof can be done by noting that $\zeta_s(y) = \mathbb{E}_{\mathbb{Q}}[\mathbb{P}(x, y)/\mathbb{Q}(x, y) \cdot h_s(x) \,|\, y]$, and

$$\mathbb{E}_{\mathbb{Q}}[y\zeta_s(y)] = \mathbb{E}_{\mathbb{Q}}\Big[y \cdot \mathbb{E}_{\mathbb{Q}}\Big[\frac{\mathbb{P}(x, y)}{\mathbb{Q}(x, y)} \cdot h_s(x)\,\Big|\, y\Big]\Big] = \mathbb{E}_{\mathbb{P}}[y \cdot h_s(x)]$$
$$= \beta_0 \cdot \mathbb{E}_{x \sim \mathcal{N}(0,1)}[h_s(x)] + \sum_{i \geq p^\star} \beta_i \cdot \mathbb{E}_{x \sim \mathcal{N}(0,1)}[h_i(x)h_s(x)] = 0,$$

where the second equality follows from the independence between $x$ and $y$ under $\mathbb{Q}$. $\square$

Therefore, the first nonzero term in the informative queries of (3.1) is of order at least $p^\star$. This gives rise to sample complexity $d^{p^\star-1}$ for vanilla online SGD (Arous et al., 2021) and $d^{p^\star/2}$ for SGD after smoothing the landscape (Damian et al., 2023). This sample complexity $d^{p^\star/2}$ matches the correlated statistical query (CSQ) lower bound with gradient of the form $y\phi(z)$ (Abbe et al., 2023; Damian et al., 2022).

### C.1.1 EQUIVALENCE BETWEEN DIRECT LABEL TRANSFORMATION AND MODIFIED LOSS FUNCTION

In this derivation, we focus on a single neuron $f(z; \theta, a) = a\sigma(\langle\theta, z\rangle)$ with $\theta \in \mathbb{S}^{d-1}$ and sufficiently small $a \in \mathbb{R}$. A more general case with multiple neurons is handled in the proof of Example 4.6 in Appendix C.2.2. We consider the following cases:

(i) $L_2$ loss with direct label transformation $y \to \mathcal{T}(y)$.

(ii) Label transformation with modified loss function $\ell(f(x), y)$.

Let $\ell'(f, y)$ denote the partial derivative of $\ell(f, y)$ with respect to $f$. We have the following observations:

**Lemma C.2.** *Take $\mathcal{T}(\cdot) = \ell'(0, \cdot)$. Assume $\ell'(f, \cdot)$ is $B$-Lipschitz with respect to $f$. The gradients obtained from both cases (i) and (ii) are the same up to a small error term that scales with $a$.*

*Proof.* For case (i), the rescaled negative gradient is obtained as

$$g = -a^{-1} \cdot \nabla_\theta(\mathcal{T}(y) - f(z; \theta, a))^2 = (\mathcal{T}(y) - f(z; \theta, a)) \cdot \sigma'(\langle\theta, z\rangle) \cdot z$$
$$= \mathcal{T}(y) \cdot \sigma'(\langle\theta, z\rangle) \cdot z - a \cdot \sigma(\langle\theta, z\rangle) \cdot \sigma'(\langle\theta, z\rangle) \cdot z$$

For case (ii), the gradient is given by

$$g = -a^{-1} \cdot \nabla_\theta \ell(f(z; \theta, a), y) = -\ell'(f(z; \theta, a), y) \cdot \sigma'(\langle\theta, z\rangle) \cdot z$$
$$= -\ell'(0, y) \cdot \sigma'(\langle\theta, z\rangle) \cdot z + a \cdot (\ell'(0, y) - \ell'(f(z; \theta, a), y)) \cdot \sigma'(\langle\theta, z\rangle) \cdot z,$$

where $|\ell'(0, y) - \ell'(f(z; \theta, a), y)| \leq |a \cdot B \cdot \sigma(\langle\theta, z\rangle)|$. Therefore, in the following we just focus on case (i) and omit the error term that scales with $a$ for simplicity. $\square$

The assumption on the Lipschitz constant $B$ is just for the sake of simplicity, which is not required in our main results.

### C.1.2 MOMENT CALCULATION AROUND INITIALIZATION

In the following, we provide a detailed calculation of the first and second moments of the gradient $g = \mathcal{T}(y)\sigma'(\langle\theta, z\rangle) \cdot z$.

**Without Weight Perturbation.** Consider gradient $g = \mathcal{T}(y)\sigma'(\langle \theta, z \rangle) \cdot z$ for a single neuron with weight $\theta \in \mathbb{S}^{d-1}$ and input $z \in \mathbb{R}^d$, where we omit the error term for simplicity. The first moment of the gradient is given by

$$\mathbb{E}_{\mathbb{P}_{\theta^\star}}\left[\mathcal{T}(y) \cdot \sigma'(\langle \theta, z \rangle) \cdot z\right]$$

$$= \mathbb{E}_{\mathbb{Q}}\left[\mathcal{T}(y) \cdot \sigma'(\langle \theta, z \rangle) \cdot z \cdot \left(1 + \sum_{s \geq s^\star} \zeta_s(y) h_s(\langle \theta^\star, z \rangle)\right)\right]$$

$$= \underbrace{\mathbb{E}_{\mathbb{Q}}\left[\mathcal{T}(y) \cdot \sigma'(\langle \theta, z \rangle) \cdot z\right]}_{\text{bias}} + \sum_{s \geq s^\star} \underbrace{\mathbb{E}_{\mathbb{Q}}\left[\mathcal{T}(y)\zeta_s(y)\right] \cdot \mathbb{E}_{\mathbb{Q}}\left[\sigma'(\langle \theta, z \rangle)h_s(\langle \theta^\star, z \rangle)z\right]}_{\text{informative queries}}.$$

The right-hand side of the above equation gives rise to (3.1) in the main text. From the above derivation, we can see clearly that using a direct label transformation or a modified loss function can handle the zero correlation issue $\mathbb{E}_{\mathbb{Q}}[y\zeta_s(y)] = 0$ for the informative queries. Invoking Lemma H.1 with $\psi(y, \langle w, z \rangle) = \sigma'(\langle \theta, z \rangle)$ where we replace $w \to \theta$ and $\theta \to \theta^\star$ in the lemma statement, we obtain

$$\mathbb{E}_{\mathbb{Q}}[\sigma'(\langle \theta, z \rangle) \cdot h_s(\langle \theta^\star, z \rangle) \cdot z] = \sqrt{s+1} \cdot \widehat{(\sigma')}_{s+1} \cdot \langle \theta, \theta^\star \rangle^s \cdot \theta + \sqrt{s} \cdot \widehat{(\sigma')}_{s-1} \cdot \langle \theta, \theta^\star \rangle^{s-1} \cdot \theta^\star,$$

where $\widehat{(\sigma')}_s$ is the $s$-th coefficient of the Hermite expansion of $\sigma'$, and we have by a simple calculation that $\widehat{(\sigma')}_s = \sqrt{s+1} \cdot \widehat{\sigma}_{s+1}$. Since we are only interested in the alignment between the informative queries and the unknown signal $\theta^\star$, we only need to focus on the second term for $s = s^\star$ as the remaining terms are of higher order in $\langle \theta, \theta^\star \rangle = d^{-1/2}$ at initialization. With a debiasing procedure (Line 6 in Algorithm 1), we can also cancel the bias term in the first moment of the gradient. For simplicity, we just assume $\mathbb{E}_{\mathbb{Q}}[\mathcal{T}(y)] = 0$ in the following calculations while the more general version is handled in the proofs of both Theorem 4.2 and Theorem 5.1. This gives rise to the first moment

$$\mathbb{E}_{\mathbb{P}_{\theta^\star}}[\langle g, \theta^\star \rangle] = \mathbb{E}_{\mathbb{Q}}[\mathcal{T}(y)\zeta_s(y)] \cdot s^\star \cdot \widehat{\sigma}_{s^\star} \cdot \langle \theta, \theta^\star \rangle^{s^\star - 1} + O(d^{-s^\star/2}) = O(d^{-(s^\star-1)/2}). \quad \text{(C.1)}$$

For the second moment, we have

$$\mathbb{E}_{\mathbb{P}_{\theta^\star}}[\|g\|_2^2] = \mathbb{E}_{\mathbb{P}_{\theta^\star}}\left[\mathcal{T}(y)^2 \cdot \sigma'(\langle \theta, z \rangle)^2 \cdot \|z\|_2^2\right] = O(d),$$

where $\|z\|_2^2$ contributes to the $O(d)$ term.

**With Weight Perturbation.** Let $w_l = (\gamma\theta + \xi_l)/\|\gamma\theta + \xi_l\|_2$ for $l \in [M]$ with $\xi_l \sim \text{Unif}(\mathbb{S}^{d-1})$ and $\gamma = d^{-1/4}$. As the norm $\|\gamma\theta + \xi_l\|_2 \in [1 - \gamma, 1 + \gamma]$ thanks to the triangle inequality, we can just focus on numerator, i.e., $w_l \approx \gamma\theta + \xi_l$. For given $w_l$, we denote the previous calculated gradient as $g_l$, i.e., $g_l = \mathcal{T}(y)\sigma'(\langle w_l, z \rangle) \cdot z$. Then, the aggregated gradient over all $L$ perturbed weights is given by $g = \frac{1}{L}\sum_{l=1}^L g_l$. For the following calculation, **we focus on $s^\star$ being an odd number and larger than** 1 for simplicity. In the formal proof of Theorem 4.2, we handle the remaining cases separately. Following from the same calculation for (C.1), the first moment now becomes

$$\mathbb{E}_{\mathbb{P}_{\theta^\star}}[\langle g, \theta^\star \rangle] = \frac{1}{L}\sum_{l=1}^L \underbrace{\widehat{\sigma}_{s^\star}\mathbb{E}_{\mathbb{Q}}[\mathcal{T}(y)\zeta_s(y)]}_{\neq 0} \cdot s^\star \langle w_l, \theta^\star \rangle^{s^\star - 1} + O(d^{-s^\star/2}) = O(d^{-(s^\star-1)/2}), \quad \text{(C.2)}$$

which gives the same order as (C.1). The things are quite different for the second moment. Note that in the first moment, only the $s^\star$-th coefficient of the Hermite expansion of $\sigma$ contributes to the alignment with $\theta^\star$, while the lower order terms are canceled out due to the fact that the likelihood ratio $\mathbb{P}(x, y)/\mathbb{Q}(x, y)$ has the lowest nonzero order $s^\star$. The second moment is now given by

$$\mathbb{E}_{\mathbb{P}_{\theta^\star}}[\|g\|_2^2] = \frac{1}{L^2}\sum_{l,l'=1}^L \mathbb{E}_{\mathbb{P}_{\theta^\star}}\left[\mathcal{T}(y)^2 \cdot \sigma'(\langle w_l, z \rangle) \cdot \sigma'(\langle w_{l'}, z \rangle) \cdot \|z\|_2^2\right]$$

$$\simeq \frac{d}{L^2}\sum_{l,l'=1}^L \mathbb{E}_{\mathbb{P}_{\theta^\star}}\left[\mathcal{T}(y)^2 \cdot \sigma'(\langle w_l, z \rangle) \cdot \sigma'(\langle w_{l'}, z \rangle)\right].$$

The approximation in the second line is a heuristic calculation for the second moment, and the rigorous version is available in the proof of Proposition E.4 in Appendix E.3. Now, we plug in the likelihood ratio $\mathbb{P}(x, y)/\mathbb{Q}(x, y)$, and under certain regularity assumption (See Item Assumption 4.1(c) for details) we obtain

$$
\mathbb{E}_{\mathbb{P}_{\theta^\star}}[\|g\|_2^2] \simeq \frac{d}{L^2} \sum_{l,l'=1}^{L} \mathbb{E}_{\mathbb{Q}}\left[ \mathcal{T}(y)^2 \cdot \sigma'(\langle w_l, z \rangle) \cdot \sigma'(\langle w_{l'}, z \rangle) \cdot \left( 1 + \sum_{s \geq s^\star} \zeta_s(y) h_s(\langle \theta^\star, z \rangle) \right) \right]
$$

$$
= \frac{d}{L^2} \sum_{l,l'=1}^{L} \mathbb{E}_{\mathbb{Q}}\left[ \mathcal{T}(y)^2 \right] \cdot \mathbb{E}_{\mathbb{Q}}\left[ \sigma'(\langle w_l, z \rangle) \cdot \sigma'(\langle w_{l'}, z \rangle) \right] \tag{C.3}
$$

$$
+ \frac{d}{L^2} \sum_{l,l'=1}^{L} \sum_{s \geq s^\star} \mathbb{E}_{\mathbb{Q}}\left[ \mathcal{T}(y)^2 \zeta_s(y) \right] \cdot \underbrace{\mathbb{E}_{\mathbb{Q}}\left[ \sigma'(\langle w_l, z \rangle) \cdot \sigma'(\langle w_{l'}, z \rangle) \cdot h_s(\langle \theta^\star, z \rangle) \right]}_{\text{higher order term}}.
$$

Here, we remark that when comparing the two terms on the right-hand side, the higher-order terms $s \geq s^\star$ are of order at least $d \cdot d^{-s^\star/2}$ due to the fact that both $w_l$ and $w_{l'}$ have a correlation with $\theta^\star$ of order $d^{-1/2}$ at initialization. Therefore, when considering the first term in (C.3) only, we have

$$
\mathbb{E}_{\mathbb{P}_{\theta^\star}}[\|g\|_2^2] \simeq \mathbb{E}_{\mathbb{Q}}\left[ \mathcal{T}(y)^2 \right] \cdot \left( \frac{d}{L^2} \cdot \sum_{l=1}^{L} \mathbb{E}_{\mathbb{Q}}\left[ \sigma'(\langle w_l, z \rangle)^2 \right] + \frac{d}{L^2} \cdot \sum_{l \neq l'} \mathbb{E}_{\mathbb{Q}}\left[ \sigma'(\langle w_l, z \rangle) \cdot \sigma'(\langle w_{l'}, z \rangle) \right] \right).
$$

The first term in the bracket is of order $d/L$ since $\mathbb{E}_{\mathbb{Q}}[\sigma'(\langle w_l, z \rangle)^2] = O(1)$. With large $L$, the main contribution comes from the second term in the bracket. Suppose **the lowest nonzero Hermite coefficient of $\sigma$ is $r$**, then we have by noting that $\mathbb{E}_{\xi_l, \xi_{l'}}[\langle w_l, w_{l'} \rangle^r] \leq O(\gamma^{2r} + \mathbb{E}_{\xi_l, \xi_{l'}}[|\langle \xi_l, \xi_{l'} \rangle|^r]) = O\left( d^{-r/2} \right)$ for constant $r$ and hence

$$
\frac{d}{L^2} \cdot \sum_{l \neq l'} \mathbb{E}_{\mathbb{Q}}\left[ \sigma'(\langle w_l, z \rangle) \cdot \sigma'(\langle w_{l'}, z \rangle) \right] \simeq \frac{d}{L^2} \cdot \sum_{l \neq l'} \langle w_l, w_{l'} \rangle^{r-1} \leq O(d^{-(r-3)/2}), \tag{C.4}
$$

where in the first approximation, we ignore the higher order terms. One would like to pick the largest possible $r$, which will be $s^\star$ as (C.2) requires $\widehat{\sigma}_{s^\star} \neq 0$. This leads to the second moment

$$
\mathbb{E}_{\mathbb{P}_{\theta^\star}}[\|g\|_2^2] = O(d^{-(s^\star - 3)/2}).
$$

The case for $r < s^\star$ will be discussed ilater. Here, we remark that the label transformation $\mathcal{T}(\cdot)$ is only changing the second moment up to some constant factor as the only difference that occurs for not using label transformation would be changing $\mathbb{E}_{\mathbb{Q}}[\mathcal{T}(y)^2]$ to $\mathbb{E}_{\mathbb{Q}}[y^2]$ in the above calculation. This is also the reason why we can stick to the standard $L_2$ loss in the discussion of the weight perturbation in Section 3 for showing how the weight perturbation can help boost the signal-to-noise ratio (SNR) in the second moment.

### C.1.3 HEURISTIC DERIVATION OF SAMPLE COMPLEXITY

In the following, we first discuss what is the thresholding SNR for developing a $O(1)$ alignment between the neuron weight $\theta$ and the unknown signal $\theta^\star$. Then, we provide a heuristic derivation of the sample complexity $O(d^{s^\star/2})$ for the case with weight perturbation. We still focus on the two-layer neural network with a single neuron $f(z; \theta, a) = a\sigma(\langle \theta, z \rangle)$ as in the previous discussion. We asssume the following to hold in the following discussion:

(i) $s^\star$ is an odd number greater than 1.
(ii) The activation function $\sigma$ satisfies the high-pass assumption, i.e., the lowest nonzero Hermite coefficient of $\sigma$ is $r$.
(iii) The alignment $|\langle \theta, \theta^\star \rangle| = o(1)$ (to ensure the previous simplified results are valid).
(iv) The label transformation $\mathcal{T}(\cdot)$ is such that $\mathbb{E}_{\mathbb{Q}}[\mathcal{T}(y)] = 0$ and $\mathbb{E}_{\mathbb{Q}}[\mathcal{T}(y)\zeta_{s^\star}(y)] \neq 0$.
(v) The number of perturbation $L$ is sufficiently large.

We will follow the update in Algorithm 1 and take learning rate $\eta = \infty$, which means directly updating the weight by the gradient at each step.

**Sample Complexity from the View of SNR Recursion.** Let us first consider $r = s^\star$, i.e., Assumption 4.1(b) holds. At step $t$, let the weight be $g^{(t-1)}/\|g^{(t-1)}\|_2$ and the perturbed weight be

$$w_l^{(t)} = \frac{\gamma \cdot g^{(t-1)}/\|g^{(t-1)}\|_2 + \xi_l^{(t)}}{\|\gamma \cdot g^{(t-1)}/\|g^{(t-1)}\|_2 + \xi_l^{(t)}\|_2}$$

with $\xi_l^{(t)} \sim \text{Unif}(\mathbb{S}^{d-1})$. Following the same procedure as in (C.2), we have the first moment of the gradient as

$$\mathbb{E}_{\mathbb{P}_{\theta^\star}}[\langle g^{(t)}, \theta^\star \rangle] \simeq \frac{1}{L}\sum_{l=1}^{L}\langle w_l^{(t)}, \theta^\star \rangle^{s^\star-1} \simeq (\gamma \cdot |\langle g^{(t-1)}/\|g^{(t-1)}\|_2, \theta^\star \rangle| + d^{-1/2})^{s^\star-1},$$

where the last equality follows from the fact that $s^\star - 1$ is an even number, and $\simeq$ hides some constant factors. A proof of this calculation can be found in Lemma H.4. Following the similar derivation in (C.3)-(C.4), the second moment is given by

$$\mathbb{E}_{\mathbb{P}_{\theta^\star}}[\|g^{(t)}\|_2^2] \simeq \frac{d}{L^2}\cdot\sum_{l\neq l'}\langle w_l, w_{l'}\rangle^{r-1}\bigg|_{r=s^\star} \simeq d\cdot(\gamma^2 + d^{-1/2})^{s^\star-1},$$

where $\simeq$ hides some constant factors that depends on $s^\star$. The last equality follows from the observation that $\langle w_l, w_{l'}\rangle^{s^\star-1} = \sqrt{\langle w_l, w_{l'}\rangle^2}^{s^\star-1} \simeq \sqrt{\gamma^4 + \langle \xi_l, \xi_{l'}\rangle^2}^{s^\star-1} \simeq (\gamma^2 + |\langle \xi_l, \xi_{l'}\rangle|)^{s^\star-1}$, and a concentration of $|\langle \xi_l, \xi_{l'}\rangle|^s$ for $s \leq s^\star - 1$. Hence, the one-sample SNR is given by

$$\text{SNR}^{(t)} = \frac{\mathbb{E}_{\mathbb{P}_{\theta^\star}}[\langle g^{(t)}, \theta^\star \rangle]^2}{\mathbb{E}_{\mathbb{P}_{\theta^\star}}[\|g^{(t)}\|_2^2]} = \frac{C}{d}\cdot\left(\frac{\gamma^2\cdot|\langle g^{(t-1)}/\|g^{(t-1)}\|_2, \theta^\star \rangle|^2 + d^{-1}}{\gamma^2 + d^{-1/2}}\right)^{s^\star-1},$$

where $C$ is some constant factor that depends on $s^\star$. One can replace $|\langle g^{(t-1)}/\|g^{(t-1)}\|_2, \theta^\star \rangle|^2$ by $n\text{SNR}^{(t-1)}$ as the square root of the $n$ sample SNR is roughly the alignment between the gradient and the unknown signal. Under this replacement, we always have $n\text{SNR}^{(t-1)} = o(1)$ by our assumption (iii). We also obtain the recursion

$$nd^{1/2}\cdot\text{SNR}^{(t)} = \frac{Cn}{d^{s^\star/2}}\cdot\left(\frac{\gamma^2 d^{1/2}\cdot nd^{1/2}\cdot\text{SNR}^{(t-1)} + 1}{\gamma^2 d^{1/2} + 1}\right)^{s^\star-1}.$$

Viewing the term inside the bracket as a function of $x = \gamma^2 d^{1/2}$, and taking $\alpha^{(t)} = nd^{1/2}\cdot\text{SNR}^{(t)}$, we have

$$f(x;\alpha) = \frac{\alpha x + 1}{x + 1}.$$

For $\alpha < 1$, $f$ is a decreasing function of $x$, and for $\alpha > 1$, $f$ is an increasing function of $x$. At initialization, the alignment is of order $d^{-1/2}$, which means $\alpha^{(0)} = nd^{1/2}\text{SNR}^{(0)} = O(d^{-1/2}) \ll 1$. The best choice is $x = 0$ and $f(x;\alpha^{(0)}) = 1$, and the corresponding next-step SNR is given by

$$\alpha^{(1)} = \frac{Cn}{d^{s^\star/2}}.$$

If $n \ll d^{s^\star/2}$, then $\alpha^{(1)} \ll 1$, and the same argument applies for $\alpha^{(2)} \ll 1$ and so on. This means that the alignment cannot break the barrier $O(d^{-1/4})$. On the other hand, when $n \gg d^{s^\star/2}$, we have $\alpha^{(1)} \gg 1$. Suppose we can choose $x$ as large as possible such that $f(x;\alpha^{(1)}) = \alpha^{(1)}$, then we have another recursion:

$$\alpha^{(t)} = \frac{Cn}{d^{s^\star/2}}\cdot(\alpha^{(t-1)})^{s^\star-1},$$

which means that the alignment can grow exponentially with $t$.

**Polarization Level as a Trade-off.** From the above discussion, we can see that when $\alpha \ll 1$, we want to choose $x$ as small as possible such that $f(x;\alpha) = 1$ and when $\alpha \gg 1$, we want to choose $x$ as large as possible such that $f(x;\alpha) = \alpha$. In fact, setting $x = 1$, i.e., $\gamma = d^{-1/4}$, is a good choice to balace the two cases, as it is easy to check that

$$f(1;\alpha) = \begin{cases} \frac{1}{2}, & \text{if } \alpha \ll 1, \\ \frac{\alpha}{2}, & \text{if } \alpha \gg 1, \end{cases}$$

which is optimal in the sense that it only introduces a $1/2$ factor when compared to the best choice.

**Heuristic Derivation of Sample Complexity from Alignment.** Previously, we derive from the SNR recursion that why $O(d^{s^\star/2})$ samples are necessary to achieve nontrivial alignment with weight perturbation and why $\gamma = d^{-1/4}$ is a good choice for the polarization level. Now, we provide an alternative view to understand the sample complexity $d^{s^\star/2}$ from the perspective of the alignment between the perturbed weight and the unknown signal $\theta^\star$. **The following discussion is more heuristic in nature when compared to the SNR recursion.** We still follow the assumptions (i)-(v) as in the previous discussion. We set $\gamma = d^{-1/4}$.

Consider the case of an infinitely large learning rate $\eta$, where at each step, the weight is approximately given by the gradient from the previous step, i.e., $w \approx \gamma g/\|g\|_2 + \xi$, where $g$ is the gradient from the previous step, and $\xi$ is a perturbed noise. Here, our goal is for the *perturbed weight* to exhibit nontrivial alignment with $\theta^\star$. Specifically, we require

$$\langle w, \theta^\star \rangle \approx \gamma \cdot \langle g/\|g\|_2, \theta^\star \rangle + \langle \xi, \theta^\star \rangle \gg O(d^{-1/2}).$$

Since $\langle \xi, \theta^\star \rangle$ is of order $d^{-1/2}$, it follows that we need $\gamma \cdot \langle g/\|g\|_2, \theta^\star \rangle \gg O(d^{-1/2})$. This translates into the condition $\sqrt{n\mathrm{SNR}} \gg d^{-1/4}$. From our previous moment calculations, we know that

$$\mathrm{SNR} = \frac{\mathbb{E}_{\mathbb{P}_{\theta^\star}}[\langle g, \theta^\star \rangle]^2}{\mathbb{E}_{\mathbb{P}_{\theta^\star}}[\|g\|_2^2]} = O\left(\frac{d^{-(s^\star-1)}}{d^{-(s^\star-3)/2}}\right) = O(d^{-(s^\star+1)/2}),$$

which leads to the condition $n = \widetilde{O}(d^{s^\star/2})$ when combined with $\sqrt{n\mathrm{SNR}} \gg d^{-1/4}$.

The rationale behind this criterion is that if $\gamma \cdot \langle g, \theta^\star \rangle$ does not sufficiently exceed $\langle \xi, \theta^\star \rangle$, then the perturbed noise will dominate. In such a case, the successive gradient updates would not provide more information than those derived from random initialization.

### C.1.4 FAILURE OF HIGH-PASS ASSUMPTION

Previously, we heuristically derive the sample complexity $O(d^{s^\star/2})$ for the case with weight perturbation and the activation function $\sigma$ satisfying the high-pass assumption, i.e., $r = s^\star$. Here, we discuss what happens when the high-pass assumption is violated, i.e., $1 \le r < s^\star$. We still focus on $s^\star$ being an odd number greater than 1. Here, we have $r \ge 1$ as by definition, $r$ is the lowest nonzero Hermite coefficient of $\sigma$. This is akin to the case where Assumption 4.1(b) is not satisfied. We still assume that $\widehat{\sigma}_{s^\star} \ne 0$ as this is needed for the signal term to survive in the first moment calculation. Using the second moment calculation in (C.4), we have

$$\mathrm{SNR} = \frac{\mathbb{E}_{\mathbb{P}_{\theta^\star}}[\langle g, \theta^\star \rangle]^2}{\mathbb{E}_{\mathbb{P}_{\theta^\star}}[\|g\|_2^2]} = O\left(\frac{d^{-(s^\star-1)}}{d^{-(r-3)/2}}\right) = O(d^{-(2s^\star-r+1)/2}).$$

Combined with the same condition $\sqrt{n\mathrm{SNR}} \gg d^{-1/4}$, we obtain $n = \widetilde{O}(d^{(2s^\star-r)/2})$. Hence, for general $1 \le r \le s^\star$, the sample complexity interpolates between $O(d^{s^\star/2})$ and $O(d^{s^\star-1/2})$. We would like to point out that the above calculation is heuristic, and a formal proof is left for future work. However, we believe that the above calculation provides sufficient insight on the sample complexity when the high-pass assumption is violated.

On the other hand, if $r \ge s^\star$, this is equivalent to the case where the generative exponent is $r$ (as the terms in the decomposition of the likelihood ratio with degree less than $r$ don't have any effect on both the first and second moment). Hence, we obtain a sample complexity $n = \widetilde{O}(d^{r/2})$. In summary, the sample complexity is

$$n \simeq d^{((2s^\star-r)\vee r)/2}.$$

Notably, if the activation has the lowest nonzero degree $r$, the oracle $\psi$, defined as $\psi(y, x) = \mathcal{T}(y)\sigma'(x)$, has the lowest degree $r - 1$.

### C.2 PROOFS FOR EXAMPLES OF ORACLE FUNCTION

Here we complete the discussions of the specific examples of $\psi$ in Example 4.4 and Example 4.6.

### C.2.1 Batch-reusing for polynomial link function

We consider a polynomial link function $y = p(x) = \sum_{q^\star \le q' \le q} \beta_{q'} h_{q'}(x)$ for general $q^\star \in \mathbb{N}$ and $\beta_{q'} \in \mathbb{R}$, where $q^\star$ is the information exponent of the link function, and we also denote it by $q^\star(p)$ in the sequel. For batch-reusing, we take $\psi(y, x) = y\sigma'(x) + y\sigma'(x + y\sigma'(x))$, where activation function $\sigma(x)$ satisfies that

$$\sigma(x) = \sum_{j=0}^{C_q} \alpha_j \cdot h_j(x), \quad \sigma'(x) = \sum_{j=1}^{C_q} \sqrt{j} \cdot \alpha_j \cdot h_{j-1}(x). \tag{C.5}$$

Here the degree $C_q \in \mathbb{N}_+$ only depends on the degree $q$ of the link function and is specified later, and each coefficient $\alpha_j \sim \mathrm{Unif}([0, 1])$. The second equality in (C.5) follows from the property of Hermite polynomials in (B.1). The error term $\mathrm{err}_{m,l,i}^{(t)}$ now comes from the difference between $\psi(y_i^{(t)}, \langle w_{m,l}^{(t)}, z_i^{(t)} \rangle) \cdot z_i^{(t)}$ and the exact form of the update step obtained from two consecutive gradient descent steps on the same data under the square loss. More specifically, let us consider a single neuron whose weight is $w_{m,l}$ and a single data point $(z_i, y_i)$. Here we omit the time index $t$ for convenience. Then two gradient descent step on $(z_i, y_i)$ gives

$$\begin{aligned} -g_{m,l}^{\mathrm{Re}}(z_i, y_i) &= \big(y_i - f(z_i; \{w_{m,l}\}_{m\in[M]})\big) \cdot \sigma'(\langle w_{m,l}, z_i \rangle) \cdot z_i \\ &\quad + \big(y_i - f(z_i; \{w_{m,l}^+\}_{m\in[M]})\big) \cdot \sigma'(\langle w_{m,l}^+, z_i \rangle) \cdot z_i, \end{aligned} \tag{C.6}$$

where $w_{m,l}^+ = w_{m,l} + \eta_i^{\mathrm{Re}} \cdot (y_i - f(z_i; \{w_{m,l}\}_{m\in[M]})) \cdot \sigma'(\langle w_{m,l}, z_i \rangle) \cdot z_i$. Here $\eta_i^{\mathrm{Re}}$ is the learning rate for batch reusing, different from the learning rate $\eta$ in our algorithm. More specifically, to fit the gradient form (C.6) into our general framework with oracle function $\psi(y, x)$, we take the batch-reusing learning rate $\eta_i^{\mathrm{Re}} = 1/\|z_i\|_2^2$. Then the error term is given by

$$\mathrm{err}_{m,l,i} = -g_{m,l}^{\mathrm{Re}}(z_i, y_i) - \psi(y_i, \langle w_{m,l}, z_i \rangle) = \mathrm{err}_{m,l,i,1} + \mathrm{err}_{m,l,i,2} + \mathrm{err}_{m,l,i,3}, \tag{C.7}$$

where $\mathrm{err}_{m,l,i,1}$, $\mathrm{err}_{m,l,i,2}$, and $\mathrm{err}_{m,l,i,3}$ are given by

$$\begin{aligned} \mathrm{err}_{m,l,i,1} &= -f(z_i; \{w_{m,l}\}_{m\in[M]}) \cdot \sigma'(\langle w_{m,l}, z_i \rangle), \\ \mathrm{err}_{m,l,i,2} &= -f(z_i; \{w_{m,l}^+\}_{m\in[M]}) \cdot \sigma'(\langle w_{m,l}^+, z_i \rangle), \\ \mathrm{err}_{m,l,i,3} &= y_i\sigma'\big(\langle w_{m,l}, z_i \rangle + y_i\sigma'(\langle w_{m,l}, z_i \rangle) + \mathrm{err}_{m,l,i,1}\big) \\ &\quad - y_i\sigma'\big(\langle w_{m,l}, z_i \rangle + y_i\sigma'(\langle w_{m,l}, z_i \rangle)\big). \end{aligned} \tag{C.8}$$

*Proof of Corollary 4.5.* To prove Corollary 4.5, it suffices to show that (i) Assumption 4.1 holds, and (ii) the event $\mathcal{E}$ holds with the desired high probability. In the following, we first verify Assumption 4.1, and then check the event $\mathcal{E}$.

**Verifying Assumption 4.1.** Note that the fact of $y = p(x)$ being a polynomial immediately implies that both the square-integrable condition (Assumption 4.1(a)) and the polynomial-like tail condition (Assumption 4.1(c)) are satisfied. It remains to check the high-pass condition (Assumption 4.1(b)). Since now $s^\star \le 2$, we only need to check the condition that $|\mathbb{E}_{\mathbb{Q}}[\zeta_{s^\star}(y) \cdot \widehat{\psi}_{s^\star - 1}(y)]| > 0$.

**Case 1: $s^\star = 2$.** In this case, we have that

$$\begin{aligned} \widehat{\psi}_1(y) &= \mathbb{E}_{x\sim\mathcal{N}}\Big[x \cdot \big(y\sigma'(x) + y\sigma'(x + y\sigma'(x))\big)\Big] \\ &= y \cdot \mathbb{E}_{x\sim\mathcal{N}}\Big[\sum_{j=1}^{C_q} \sqrt{j} \cdot \alpha_j \cdot x \cdot h_{j-1}(x) + \sum_{j=1}^{C_q} \sqrt{j} \cdot \alpha_j \cdot x \cdot h_{j-1}\big(x + y\sigma'(x)\big)\Big] \end{aligned} \tag{C.9}$$

For the first summation in (C.9), only the first summand is nonzero, so we obtain

$$y \cdot \mathbb{E}_{x\sim\mathcal{N}}\Big[\sum_{j=1}^{C_q} \sqrt{j} \cdot \alpha_j \cdot x \cdot h_{j-1}(x)\Big] = \sqrt{2}\alpha_2 \cdot y. \tag{C.10}$$

For the second summation in (C.9), we have the following expansion,

$$y \cdot \mathbb{E}_{x\sim\mathcal{N}}\Big[\sum_{j=1}^{C_q} j \cdot \alpha_j \cdot x \cdot h_{j-1}\big(x + y\sigma'(x)\big)\Big]$$

$$= y \cdot \mathbb{E}_{x \sim \mathcal{N}} \left[ \sum_{j=1}^{C_q} j \cdot \alpha_j \cdot x \cdot \sum_{k=0}^{j-1} r_{j-1,k} \cdot h_{j-k-1}(x) \cdot \left( y \sigma'(x) \right)^k \right]$$

$$= \sum_{k=0}^{C_q-1} \underbrace{\left\{ \sum_{j=k+1}^{C_q} j \cdot \alpha_j \cdot r_{j-1,k} \cdot \mathbb{E}_{x \sim \mathcal{N}} \left[ x \cdot h_{j-k-1}(x) \cdot \left( \sigma'(x) \right)^k \right] \right\}}_{:= \varsigma_k(\alpha)} \cdot y^{k+1}. \qquad (C.11)$$

where $\varsigma_0(\alpha), \ldots, \varsigma_{C_q-1}(\alpha)$ are just polynomials of $\alpha = (\alpha_1, \cdots, \alpha_{C_q})$ (recall the definition of $\sigma'(x)$ in (C.5)) and each $r_{j-1,k}$ is a positive number. Combining (C.10) and (C.11), we get the following decomposition of $\widehat{\psi}_1(y)$:

$$\widehat{\psi}_1(y) = \sqrt{2}\alpha_2 \cdot y + \sum_{k=0}^{C_q-1} \varsigma_k(\alpha) \cdot y^{k+1}.$$

Further using $y = p(x)$ and the definition of $\zeta_2(y)$, we get

$$\mathbb{E}_{\mathbb{Q}}[\zeta_2(y) \cdot \widehat{\psi}_1(y)] = \mathbb{E}_{\mathbb{P}}[h_2(x) \cdot \widehat{\psi}_1(y)]$$

$$= \sqrt{2}\alpha_2 \cdot \mathbb{E}_{x \sim \mathcal{N}} [h_2(x) \cdot p(x)] + \sum_{k=0}^{C_q-1} \varsigma_k(\alpha) \cdot \mathbb{E}_{x \sim \mathcal{N}} \left[ h_2(x) \cdot p(x)^{k+1} \right].$$

According to Proposition 5 of Lee et al. (2024), we can set $C_q \in \mathbb{N}^+$ (only depending on $q$) such that there exists a smallest $I \leq C_q$ such that the information exponent $q^\star(p^I) \leq 2$. We notice that in this case $s^\star(p) = 2$, where we abuse the notation and let $s^\star(p)$ be the generative exponent of the polynomial $p$. In fact, $s^\star(p) = 1$ means $\mathbb{E}_{\mathbb{P}}[\mathcal{T}(y) \cdot h_1(x)] \equiv 0$ for all label transformation $\mathcal{T}$. Hence, the only possibility is that $q^\star(p^I) = 2$ since $p^I$ is just a special case of label transformation and we cannot get any first-order term from $p^I$. Therefore, we further simplify the target quantity as

$$\mathbb{E}_{\mathbb{Q}}[\zeta_{s^\star}(y) \cdot \widehat{\psi}_{s^\star-1}(y)] = \sqrt{2}\alpha_2 \cdot \underbrace{\mathbb{E}_{x \sim \mathcal{N}} [h_2(x) \cdot p(x)]}_{:= b_1} + \sum_{k=I}^{C_q} \varsigma_{k-1}(\alpha) \cdot \underbrace{\mathbb{E}_{x \sim \mathcal{N}} \left[ h_2(x) \cdot p(x)^k \right]}_{:= b_k}$$

$$= b_1 \cdot \sqrt{2}\alpha_2 + \sum_{k=I}^{C_q} b_k \cdot \varsigma_{k-1}(\alpha), \qquad (C.12)$$

where $b_I \neq 0$ according to the definition of $I$. Now we take a closer look at the polynomials $\{\varsigma_k(\alpha_1, \cdots, \alpha_{C_q})\}_{k=I-1}^{C_q-1}$. We claim that: (i) they are different polynomials and are linearly independent, and (ii) especially, they are all nonzero. To see these, recall that

$$\varsigma_k(\alpha_1, \cdots, \alpha_{C_q}) = \sum_{j=k+1}^{C_q} \sqrt{j} \cdot \alpha_j \cdot r_{j-1,k} \cdot \mathbb{E}_{x \sim \mathcal{N}} \left[ x \cdot h_{j-k-1}(x) \cdot \left( \sigma'(x) \right)^k \right],$$

where $\sigma'(x) = \sum_{j=1}^{C_q} \sqrt{j} \cdot \alpha_j \cdot h_{j-1}(x)$. We can calculate that for $k = 0$, $\varsigma_0(\alpha_1, \cdots, \alpha_{C_q}) = \sqrt{2}\alpha_2$ which is a non-zero polynomial. For $k = 1$, we have that

$$\varsigma_1(\alpha_1, \cdots, \alpha_{C_q}) = \sum_{j=2}^{C_q} \sqrt{j} \cdot \alpha_j \cdot r_{j-1,1} \cdot \mathbb{E}_{x \sim \mathcal{N}} [x \cdot h_{j-2}(x) \cdot \sigma'(x)]$$

$$= 2r_{1,1} \cdot \alpha_2^2 + \sum_{j=3}^{C_q} \sqrt{j} \cdot \alpha_j \cdot r_{j-1,1} \cdot \mathbb{E}_{x \sim \mathcal{N}} [x \cdot h_{j-2}(x) \cdot \sigma'(x)]$$

which is non-zero (since in the summation from $j = 3$ to $C_q$ there would be no term in the form of $c_2^2$) and is linearly independent of of $\varsigma_0$ because each terms in the summation here has degree exactly 2. Now consider for $k \geq 2$,

$$\varsigma_k(\alpha_1, \cdots, \alpha_{C_q}) = \sqrt{2(k+1)k} \cdot r_{k,k} \cdot \alpha_1^{k-1} \alpha_2 \alpha_{k+1}$$

$$+ \sum_{j=k+2}^{C_q} \sqrt{j}\alpha_j r_{j-1,k} \cdot \mathbb{E}_{x\sim\mathcal{N}}\left[x \cdot h_{j-k-1}(x) \cdot \left(\sigma'(x)\right)^k\right].$$

Again, this polynomial is non-zero (since in the summation from $j = k + 2$ to $C_q$ there would be no term in the form of $\alpha_1^{k-1}\alpha_2\alpha_{k+1}$) and is linearly independent of $\varsigma_0, \cdots, \varsigma_{k-1}$ due to the fact that the highest degree of these polynomials is no larger than $k$, and the order for each term in $\varsigma_k$ is exactly $k+1$. Thus we have proved the two claims by induction. Now recall that we are aiming at proving the RHS of (C.12) is non-zero. By our two claims just proved, the RHS of (C.12) is a linear combination of $C_q - I + 2$ linearly independent and non-zero polynomials where at least one of the combination coefficient is non-zero (which is $b_I$). Thus we obtain that the RHS of (C.9) is a non-zero polynomial of $(\alpha_1, \cdots, \alpha_{C_q})$ and its zeros form a zero-measure set. This proves that with probability 1 over the randomness of $(\alpha_1, \cdots, \alpha_{C_q})$, the high-pass condition holds.

**Case 2:** $s^\star = 1$. For this case of $s^\star = 1$, the proof is almost the same as that for $s^\star = 2$, where we additionally utilize the fact that polynomial link function with generative exponent $s^\star = 1$ can not be an even polynomial (Example 2.2) and thus there always exists some $I \leq C_q \in \mathbb{N}_+$ such that the information exponent $q^\star(p^I) = 1$ (see Proposition 5 of Lee et al. (2024)). With this fact, repeating the above argument can give the desired high-pass property.

**Verifying the event $\mathcal{E}$.** Now we verify that the desired event

$$\mathcal{E} = \left\{|\mathrm{err}_{m,l,i}^{(t)}| \leq d^{-10s^\star}, \forall (m, l, i, t) \in [M] \times [L] \times [n] \times [T]\right\}$$

holds with probability at least $1 - O(d^{-c_0})$ for some constant $c_0 > 0$ that we specify later. With (C.7) and (C.8), it suffices to look at each of the error terms $\mathrm{err}_{m,l,i,1}^{(t)}$, $\mathrm{err}_{m,l,i,2}^{(t)}$, and $\mathrm{err}_{m,l,i,3}^{(t)}$ respectively. For $\mathrm{err}_{m,l,i,1}^{(t)}$,

$$\left|\mathrm{err}_{m,l,i,1}^{(t)}\right| \leq \sum_{m'=1}^{M} |a_{m'}| \cdot \left|\sigma(\langle w_{m',l}^{(t)}, z_i\rangle)\right| \cdot \left|\sigma'(\langle w_{m,l}^{(t)}, z_i\rangle)\right| \tag{C.13}$$

Note that $\{\langle w_{m,l}^{(t)}, z_i\rangle\}_{m\in[M]}$ are standard Gaussians since $\{w_{m,l}^{(t)}\}_{m\in[M]} \subset \mathbb{S}^{d-1}$. Therefore, with probability at least $1 - d^{-c_0}$ for some constant $c_0 > 0$, we have that $|\langle w_{m,l}, z_i\rangle| = \widetilde{O}(1)$. Meanwhile, since that $\sigma$ and $\sigma'$ are both polynomials with constant order and bounded coefficients, and that $M = O(d)$, then by taking $a_m = d^{-11s^\star}$, we conclude from (C.13) that

$$\left|\mathrm{err}_{m,l,i,1}^{(t)}\right| \leq \widetilde{O}(d^{-10s^\star}). \tag{C.14}$$

For the second error term $\mathrm{err}_{m,l,i,2}^{(t)}$, similarly we have that

$$\left|\mathrm{err}_{m,l,i,2}^{(t)}\right| \leq \sum_{m'\in[M]} |a_{m'}| \cdot \left|\sigma(\langle (w_{m',l}^{(t)})^+, z_i\rangle)\right| \cdot \left|\sigma'(\langle (w_{m,l}^{(t)})^+, z_i\rangle)\right|. \tag{C.15}$$

Note that the one-step updated weights satisfy that

$$\left|\langle (w_{m,l}^{(t)})^+, z_i\rangle\right| = \left|\langle w_{m,l}^{(t)}, z_i\rangle + y_i \cdot \sigma'(\langle w_{m,l}^{(t)}, z_i\rangle) + \mathrm{err}_{m,l,i,1}^{(t)}\right| = \widetilde{O}(1), \tag{C.16}$$

with probability at least $1 - d^{-c_0}$ since $y_i = p(\langle \theta^\star, z_i\rangle)$ and $p$ is also a polynomial of constant degree and coefficients. Therefore, with the choice of $a_m$'s, by (C.15), we conclude that

$$\left|\mathrm{err}_{m,l,i,2}^{(t)}\right| \leq \widetilde{O}(d^{-10s^\star}). \tag{C.17}$$

Finally, regarding $\mathrm{err}_{m,l,i,3}^{(t)}$, note that with the same argument as (C.16), we know that with probability at least $1 - d^{-c_0}$, both $\langle w_{m,l}^{(t)}, z_i\rangle + y_i \cdot \sigma'(\langle w_{m,l}^{(t)}, z_i\rangle) + \mathrm{err}_{m,l,i,1}^{(t)}$ and $\langle w_{m,l}^{(t)}, z_i\rangle + y_i \cdot \sigma'(\langle w_{m,l}^{(t)}, z_i\rangle)$ are $\widetilde{O}(1)$. Since $\sigma'$ is a polynomial, it is $\widetilde{O}(1)$-Lipschitz continuous for inputs that are $\widetilde{O}(1)$. Therefore, combined with (C.14) that we have proved, we can obtain that

$$\left|\mathrm{err}_{m,l,i,3}^{(t)}\right| \leq |y_i| \cdot \widetilde{O}\left(|\mathrm{err}_{m,l,i,1}^{(t)}|\right) = \widetilde{O}(d^{-10s^\star}). \tag{C.18}$$

Finally, combining (C.14), (C.17), and (C.18), we obtain that for given $(m, l, i, t)$, with probability at least $1 - O(d^{-c_0})$, it holds that

$$\left|\mathrm{err}_{m,l,i}^{(t)}\right| = \widetilde{O}(d^{-10s^\star}).$$

Finally, taking $c_0$ as a constant that is larger than 2 and applying a union bound argument, we can obtain that

$$\Pr(\mathcal{E}) \geq 1 - MLnT \cdot \widetilde{O}(d^{-c_0}) \geq 1 - \widetilde{\Theta}(d^2) \cdot \widetilde{O}(d^{-c_0}) \geq 1 - \widetilde{O}(d^{-c_0'})$$

for some other constant $c_0' > 0$. Here we have applied our choice of $(M, L, n, T)$ in our algorithm (see Algorithm 1, Algorithm 2). Thus we verify the property of the event $\mathcal{E}$, proving Corollary 4.5.  $\square$

## C.2.2   MODIFIED LOSS FOR GENERAL $s^\star \geq 1$

Here we give a specific choice of the activation function $\sigma$ and the loss function $\ell$. We mainly focus on the situation where $\mathbb{Q}_y$ has a continuous cumulative distribution function $F_{\mathbb{Q}_y}$ with bounded density $f_{\mathbb{Q}_y}$. For the situation where $\mathbb{Q}_y$ is a discrete distribution (e.g., classification task), we discuss them in the end of this section. For the activation function $\sigma$, we let $\sigma(x) := (1/\sqrt{s^\star}) \cdot h_{s^\star}(x)$. Since then,

$$\widehat{\psi}_s(y) = \mathbb{E}_{\mathbb{Q}}[\psi(x, y) \cdot h_s(x) \,|\, y] = \mathbb{E}_{\mathbb{Q}}[\sigma'(x) \cdot h_s(x)] \cdot \ell'(y) = 0, \quad \forall s < s^\star - 1. \tag{C.19}$$

Regarding the choice of the loss function $\ell$, we remark that if one chooses a fixed loss function, there always exist instances such that the second assumption in the high-pass condition fails. To address this issue, we propose to construct a random loss function $\ell$. To rule out pathological examples of the underlying distribution $\mathbb{P}$, we make the following assumption on the coefficient function $\zeta_{s^\star}$.

**Assumption C.3.** *We assume that the expansion of $\widetilde{\zeta}_{s^\star} := \zeta_{s^\star} \circ F_{\mathbb{Q}_y}^{-1} : [0, 1] \mapsto \mathbb{R}$ on the Fourier basis $\{\varphi_i(x)\}_{i \geq 0}$ of $[0, 1]$ has a non-zero coefficient of order at most $D = O(1)$.*

We then choose the loss function $\ell$ as the following,

$$\ell'(y) = \sum_{i=0}^{D} \alpha_i \cdot \varphi_i \circ F_{\mathbb{Q}_y}(y), \quad \alpha_i \sim \mathrm{Unif}([0, 1]), \quad \forall 0 \leq i \leq D.$$

Notice that $F_{\mathbb{Q}_y}$ can be estimated from data using a one dimensional density estimator. Thus here we directly assume the accessibility of the function $F_{\mathbb{Q}_y}$. This further gives that

$$\mathbb{E}_{\mathbb{Q}}[\zeta_{s^\star}(y) \cdot \widehat{\psi}_{s^\star-1}(y)] = \mathbb{E}_{\mathbb{Q}}[\zeta_{s^\star}(y) \cdot \ell'(y)] = \sum_{i=0}^{D} \alpha_i \cdot \mathbb{E}_{\mathrm{Unif}([0,1])}\left[\widetilde{\zeta}_{s^\star}(\widetilde{y}) \cdot \varphi_i(\widetilde{y})\right], \tag{C.20}$$

which is a non-zero polynomial of the coefficients $\{\alpha_i\}_{i \leq D}$ due to Assumption C.3.

*Proof of Corollary 4.7.* To prove Corollary 4.7, it suffices to show that (i) Assumption 4.1 holds, and (ii) the event $\mathcal{E}$ holds with the desired high probability. In the following, we first verify Assumption 4.1, and then check the event $\mathcal{E}$.

**Verifying Assumption 4.1.** First, since $\sigma'$ is a polynomial and $\ell'$ is bounded (the Fourier basis is bounded and $D = O(1)$), we know that both Assumption 4.1(a) and Assumption 4.1(c) are satisfied. Then, by the discussions before the proof, we know from (C.19) that the first condition in the high-pass assumption (Assumption 4.1(b)) is satisfied. Furthermore, according to (C.20) and Assumption C.3, $\mathbb{E}_{\mathbb{Q}}[\zeta_{s^\star}(y) \cdot \widehat{\psi}_{s^\star-1}(y)]$ is a non-zero polynomial of the coefficients $\{\alpha_i\}_{i \leq D}$ and thus its zeros form a measure-zero set. This means that with probability 1 over the randomness of $(\alpha_1, \cdots, \alpha_D)$, the second condition in the high-pass assumption is also satisfied. This verifies Assumption 4.1.

**Verifying the event $\mathcal{E}$.** Recall our definition in Example 4.6, the error term is defined as

$$\mathrm{err}_{m,l,i}^{(t)} = \left(\ell'(y_i) - \ell'\left(y_i - f(z_i; \{w_{m,l}^{(t)}\}_{m \in [M]})\right)\right) \cdot \sigma'(\langle w_{m,l}^{(t)}, z_i\rangle).$$

First, since $w_{m,l}^{(t)} \in \mathbb{S}^{d-1}$, $\langle w_{m,l}^{(t)}, z_i\rangle$ is a standard Gaussian and therefore $|\langle w_{m,l}^{(t)}, z_i\rangle| = \widetilde{O}(1)$ with probability at least $1 - d^{-c_0}$ for some constant $c_0 > 0$. Since $\sigma'$ is a polynomial of constant degree,

we then obtain that $|\sigma'(\langle w_{m,l}^{(t)}, z_i \rangle)| = \widetilde{O}(1)$ with probability at least $1 - d^{-c_0}$. Second, consider that the second derivative of the loss function $\ell''(y)$ is given by

$$\ell''(y) = \sum_{i=0}^{D} \alpha_i \cdot \varphi_i' \left( F_{\mathbb{Q}_y}(y) \right) \cdot f_{\mathbb{Q}_y}(y),$$

which satisfies $|\ell''(y)| = O(1)$ since the derivative of the Fourier basis is still bounded and that the density of $\mathbb{Q}_y$ is assumed to be bounded. Therefore, we have

$$\ell'(y_i) - \ell' \left( y_i - f(z_i; \{w_{m,l}^{(t)}\}_{m \in [M]}) \right) = O \left( \left| f(z_i; \{w_{m,l}^{(t)}\}_{m \in [M]}) \right| \right)$$

$$= O \left( \sum_{m=1}^{M} |a_m| \cdot \left| \sigma(\langle w_{m,l}^{(t)}, z_i \rangle) \right| \right).$$

Since $\sigma$ is a polynomial of constant degree and $\langle w_{m,l}^{(t)}, z_i \rangle$ are all standard Gaussians, we have that $|\sigma(\langle w_{m,l}^{(t)}, z_i \rangle)| = \widetilde{O}(1)$ with probability at least $1 - O(d^{-c_0})$. Now given that $M = O(d)$ and taking $a_m = d^{-11s^\star}$, we have that

$$\ell'(y_i) - \ell' \left( y_i - f(z_i; \{w_{m,l}^{(t)}\}_{m \in [M]}) \right) = \widetilde{O}(d^{-10s^\star}),$$

with probability at least $1 - O(d^{-c_0})$. Therefore, for any given $(m, l, i, t)$, with probability at least $1 - O(d^{-c_0})$, it holds that

$$\left| \mathrm{err}_{m,l,i}^{(t)} \right| = \widetilde{O}(d^{-10s^\star}) \cdot \widetilde{O}(1) = \widetilde{O}(d^{-10s^\star}).$$

Finally, as in the proof of Corollary 4.5, we take $c_0$ as a constant that is larger than $s^\star + 1$, and apply a union bound argument, by which we can obtain that

$$\Pr(\mathcal{E}) \geq 1 - MLnT \cdot \widetilde{O}(d^{-c_0}) \geq 1 - \widetilde{\Theta}(d^{s^\star + 1/2}) \cdot \widetilde{O}(d^{-c_0}) \geq 1 - \widetilde{O}(d^{-c_0'})$$

for some other constant $c_0' > 0$. Thus we verify the property of the event $\mathcal{E}$, proving Corollary 4.7. $\quad\square$

**Remark C.4** (Discrete labels). *For the case of discrete label $y$ that supports on a finite set $\mathcal{Y}$ (e.g., classification tasks), the construction of $\psi$ is somehow more direct. In this case, we can still consider an oracle function in the form of $\psi(y, x) = \sigma'(x) \cdot \varphi(y)$ for some function $\varphi(y)$. The activation function $\sigma(x) = (1/\sqrt{s^\star}) \cdot h_{s^\star}(x)$ and the function $\varphi(y)$ is a random function given by*

$$\varphi(y) \sim \mathrm{Unif}([0, 1]), \quad \forall y \in \mathcal{Y}. \tag{C.21}$$

*On the one hand, we can directly conclude as in (C.19) that $\widehat{\psi}_s(y) = 0$ for all $s < s^\star - 1$. On the other hand, we have that*

$$\mathbb{E}_{\mathbb{Q}}[\zeta_{s^\star}(y) \cdot \widehat{\psi}_{s^\star - 1}(y)] = \mathbb{E}_{\mathbb{Q}}[\zeta_{s^\star}(y) \cdot \varphi(y)] = \sum_{y \in \mathcal{Y}} \mathbb{Q}_y(y) \cdot \zeta_{s^\star}(y) \cdot \varphi(y).$$

*By the definition of generative exponent (Definition 2.1), $\mathbb{E}_{\mathbb{Q}_y}[\zeta_{s^\star}(y)^2] > 0$ and thus at least one of $\{\mathbb{Q}_y(y) \cdot \zeta_{s^\star}(y)\}_{y \in \mathcal{Y}}$ is non-zero. Thus under (C.21), $\mathbb{E}_{\mathbb{Q}}[\zeta_{s^\star}(y) \cdot \widehat{\psi}_{s^\star - 1}(y)]$ is non-zero with probability 1 over the randomness in $\varphi$. Thus we have verified Assumption 4.1(b). Assumption 4.1(a) and Assumption 4.1(c) can be verified in the same way as for the continuous case, and thus Assumption 4.1 is checked. Finally, we remark that in the discrete case we do not attempt to reduce the oracle function from certain loss derivative and thus we simply set the error terms $\mathrm{err}$ as zero. Thus all the conditions in Theorem 4.2 hold and Corollary 4.7 is proved.*

**Remark C.5.** *(Non-monotonicity of the loss function) Note that this loss function constructed above is not monotonically decreasing as $f$ approaches $y$. However, this non-monotonicity does not impact our analysis since our primary objective is to learn the feature direction.*

## C.3 Debiasing the gradient estimator for Unknown Oracle

In Algorithm 1, we propose a meta-algorithm that incorporates weight perturbation at each gradient step. Specifically, as outlined in Line 6, after computing the gradient estimator

$$g_{m,l,i}^{(t)} = \left(\psi(y_i^{(t)}, \langle w_{m,l}^{(t)}, z_i^{(t)} \rangle) + \mathrm{err}_{m,l,i}^{(t)}\right) \cdot z_i^{(t)},$$

we aggregate the gradients and remove bias through the following operation:

$$g_m^{(t)} = \frac{1}{nL} \sum_{i=1}^{n} \sum_{l=1}^{L} \left(g_{m,l,i}^{(t)} - \widehat{\psi}_1(y_i^{(t)}) \cdot w_{m,l}^{(t)}\right).$$

We would like to demonstrate that the debiasing operation is only needed for $s^\star \leq 2$, as the high-pass assumption already guarantees that the bias term is zero for $s^\star > 2$. Therefore, we only need to consider the case where $s^\star \leq 2$. In the two examples presented in Example 4.4 and Example 4.6 (where the modified loss $\ell$ is known), we demonstrated that the oracle $\psi(y, x)$ can be derived in closed form. Consequently, calculating $\widehat{\psi}_1(y)$ to debias the gradient poses no significant challenges in such scenarios.

However, in cases where the gradient oracle is not readily available *in closed form*, additional complexity arises. Examples include scenarios where mini-batch SGD involves multiple passes over the same batch, or when label transformations are applied using a neural network. To handle such cases, we propose to use a debiasing technique based on the U statistics.

To simplify the notation, we omit the time index $t$. Additionally, we drop the dependence on the neuron index $m$ and the perturbation index $l$. For a given mini-batch $\{(z_i, y_i)\}_{i=1}^n$, we retain $\{y_i\}_{i=1}^n$ and resample $\{\widetilde{z}_j\}_{j=1}^N$ from an isotropic Gaussian distribution. Taking the product of these two sets, we obtain a dataset of size $nN$:

$$\mathcal{D} = \{(\widetilde{z}_j, y_i)\}_{i \in [n], j \in [N]}.$$

The gradient calculated for each sample $(\widetilde{z}_j, y_i)$ is given by:

$$g_{i,j} = (\psi(y_i, \langle w, \widetilde{z}_j \rangle) + \mathrm{err}_{i,j}) \cdot \widetilde{z}_j.$$

Note that here the error term is uniformly bounded by $\widetilde{O}(d^{-10s^\star})$ under the assumptions of both Theorem 4.2 and Theorem 5.1.

Leveraging the independence between $\{\widetilde{z}_j\}_{j=1}^N$ and $\{y_i\}_{i=1}^n$, and noting that $\psi(y, x)x$ has a polynomial tail (as established in Assumption 4.1(c)), we apply Lemma J.3 to obtain the following result with probability at least $1 - \delta$:

$$\left| \frac{1}{nN} \sum_{i \in [n], j = \in [N]} \langle g_{i,j}, w \rangle - \mathbb{E}_{x \sim \mathcal{N}(0,1)} \left[ \sum_{i=1}^{n} \psi(y_i, x)x \right] \right| \leq \sqrt{\frac{\log \delta^{-1}}{N}} + \widetilde{O}(N^{-1}) + \frac{1}{nN} \sum_{i,j} |\mathrm{err}_{i,j}|$$

By choosing $N$ large enough (e.g., $N = d^C$ for some large constant $C$), the estimation error becomes negligible and can be safely absorbed into the error term for our analysis. Note that this debiasing procedure is only needed once at each iteration for a single perturbed weight, and we *don't need to repeat it for each neuron or perturbation* as the distribution of $\langle w, \widetilde{z}_j \rangle$ is the same for all neurons and perturbations. It is also worth noting that $N$ only affects computational complexity, not sample complexity. Furthermore, the computational complexity remains polynomial in $d$.

In summary, suppose the gradient oracle is not available in closed form, we can instead implement the following procedure for $s^\star \leq 2$ at each iteration of Algorithm 1:

1. Sample isotropic Gaussian vectors $\{\widetilde{z}_j\}_{j=1}^N$.
2. On synthetic sample $\{(\widetilde{z}_j, y_i)\}_{i \in [n], j \in [N]}$, perform a gradient backward to obtain each $g_{i,j}$ for weight $w_{1,1}$, which is the first perturbation of the first neuron.
3. Compute the debiasing term $\nu = \frac{1}{nN} \sum_{i \in [n], j = \in [N]} \langle g_{i,j}, w_{1,1} \rangle$.

After this procedure, in Line 6 of Algorithm 1, we compute $g_m^{(t)} = (nL)^{-1} \sum_{i=1}^{n} \sum_{l=1}^{L} \left(g_{m,l,i}^{(t)} - \nu \cdot w_{m,l}^{(t)}\right)$ instead. The other steps of the meta algorithm remain the same.

## C.4  SUMMARY OF LIMITATIONS

**Failure of standard methods.** Although being a gradient-based method, our approach is extensively modified to adapt to the structure of the problem. Although we have shown in 4.2.2 that some fatal issues prevent the vanilla mini-batch SGD from efficiently learning $\theta^\star$, it remains unknown whether there is any standard gradient-based methods (for example noisy-SGD) that can succeed in this problem.

**Minimizing the expected loss.** Current work focuses on estimating $\theta^\star$ using the two-layer neural networks. However, training the second layer with learned feature goes beyond the scope of this work. We leave it as a future direction to investigate whether a learned feature enables minimizing the expected error efficiently.

**Misspecification of the gradient oracle.** The validity of Theorem 4.2 is based on that the order of the gradient oracle is correctly specified as the generative exponent $s^\star$, i.e., Assumption 4.1(b). In general estimating tasks where the link distribution is unknown, the generative exponent is beyond the access and requires additional identification. In Appendix C.1.4, we heuristically derive that size $n = \Omega(d^{(2s^\star - r)\vee r)/2})$ samples are required for a order-$r$ high-pass oracle to obtain a non-trivial SNR. This indicates that both the optimistic or pessimistic specification of the gradient oracle deteriorates the sample efficiency of our algorithm. To address the issue of unknown generative exponent, Damian et al. (2024) propose to run their partial-trace algorithm with different order and compare the quality of the estimated feature on the hold-out validation set. We expect that a similar strategy for our algorithm can also tackle this issue. However, this brute-force approach is computationally inefficient. It is an interesting future direction to construct an efficient and adaptive algorithm that works for unknonw $s^\star$.

**Constructing modified loss in general cases.** In Appendix C.2.2, we provide an example of loss function which provably satisfies Assumption 4.1. However, the construction of this loss function involves a random linear combination of Fourier basis, which is unnatural and is rarely adopted in practice. Additionally, this construction requires access to the marginal distribution of $y$ under $\mathbb{P}$ that needs an ad-hoc estimation. It is unclear whether using an estimated version of $\mathbb{P}_y$ would cause the failure of previous argument. Besides, the minimal order of non-zero Fourier coefficient in Assumption C.3 is involved in the construction, which is typically inaccessible in practice. It would be an interesting future direction to investigate if there is any natural choice of loss function that generally follows Assumption 4.1.

## D  PROOF SKETCH OF THE MAIN THEOREM FOR UNIFORM PRIOR

In the following proof sketch, we only focus on the case $s^\star$ being an even integer and $s^\star \geq 2$. The other cases only differs technically and we defer readers to Appendix E.2 for a detailded proof. For simplicity, denote $\rho_m^{(t)} := \langle \theta_m^{(t)}, \theta^\star \rangle$, the alignment between the weights of neuron $m$ and the signal $\theta^\star$ at time $t$. Recall from Line 8 in Algorithm 1 that the update for neuron $m$ at time step $t$ is

$$\theta_m^{(t+1)} = \frac{\theta_m^{(t)} + \eta \bar{g}_m^{(t)}}{\|\theta_m^{(t)} + \eta \bar{g}_m^{(t)}\|_2}. \tag{D.1}$$

This implies that the alignment of the next iteration, $\rho_m^{(t+1)}$, is a convex combination of the previous alignment $\rho_m^{(t)}$ and the alignment of the update step $\langle \bar{g}_m^{(t)}, \theta^\star \rangle = \langle g_m^{(t)}, \theta^\star \rangle / \|g_m^{(t)}\|_2$. Therefore, to show that the alignment improves after one iteration, we need to first analyze the scale of $\langle g_m^{(t)}, \theta^\star \rangle$ and $\|g_m^{(t)}\|_2$; then we will be able to characterize the improvement of $\rho_m^{(t)}$ across iterations.

**Alignment of the update step $\langle \bar{g}_m^{(t)}, \theta^\star \rangle$.** To this end, we calculate the first moment and and second moment of $g_m^{(t)}$ over the randomness of the data $\{(z_i^{(t)}, y_i^{(t)})\}_{i=1}^n$, and combining these leads to the concentration of $\langle g_m^{(t)}, \theta^\star \rangle / \|g_m^{(t)}\|_2$. More specifically, for the first moment of $g_m^{(t)}$, we have

$$\langle \mathbb{E}_{\mathbb{P}_{\theta^\star}}[g_m^{(t)}], \theta^\star \rangle \approx \rho_m^{(t)} \gamma \cdot (|\rho_m^{(t)}|\gamma + d^{-1/2})^{s^\star - 2},$$

while the magnitude of $\mathbb{E}_{\mathbb{P}_{\theta^\star}}[g_m^{(t)}]$ in any other direction orthogonal to $\theta^\star$ is of strictly higher order. For the second moment of $g_m^{(t)}$, setting $\gamma = \widetilde{\Theta}(d^{-1/4})$, it can be shown that for any direction $v \in \mathbb{S}^{d-1}$, $\mathbb{E}_{\mathbb{P}_{\theta^\star}}[\langle g_m^{(t)}, v \rangle^2] = \widetilde{O}(d^{-(s^\star-1)/2})$. Now chooing $n = \widetilde{\Omega}(d^{s^\star/2})$, it follows from a Bernstein-type concentration inequality that the fluctuation of $\langle g_m^{(t)}, v \rangle$ is of the same order $\widetilde{O}(d^{-s^\star/2+1/4})$ for any direction $v \in \mathbb{S}^{d-1}$. Therefore, with high probability,

$$
\begin{aligned}
\langle g_m^{(t)}, \theta^\star \rangle &\geq \rho_m^{(t)} \gamma \cdot (|\rho_m^{(t)}|\gamma + d^{-1/2})^{s^\star-2} - \widetilde{O}(d^{-s^\star/2+1/4}), \\
\|g_m^{(t)}\|_2 &\leq \rho_m^{(t)} \gamma \cdot (|\rho_m^{(t)}|\gamma + d^{-1/2})^{s^\star-2} + \widetilde{O}(d^{-s^\star/2+3/4}).
\end{aligned}
\tag{D.2}
$$

**Phase 1: from $d^{-1/2}$ to weak alignment.** Due to random initialization, it holds with high probability that $\rho_m^{(0)} = O(d^{-1/2})$ for $\Omega(M)$ many neurons. Therefore, it suffices to consider a neuron with $\rho_m^{(t)} = \Omega(d^{-1/2})$. When $\Omega(d^{-1/2}) \leq \rho_m^{(t)} \leq O(1)$, by choosing $\gamma = \widetilde{\Theta}(d^{-1/4})$, we can ensure that the first term in the lower bound for $\langle g_m^{(t)}, \theta^\star \rangle$ in (D.2) dominates the $\widetilde{O}(d^{-s^\star/2+1/4})$ fluctuation. Based on this, we can leverage (D.2) to further show that $\langle \bar{g}_m^{(t)}, \theta^\star \rangle = \langle g_m^{(t)}, \theta^\star \rangle / \|g_m^{(t)}\|_2 \geq (1+c)\rho_m^{(t)}$ for a constant $c > 0$. Consequently, it follows from (D.1) that

$$
\rho_m^{(t+1)} \geq (1+c)\rho_m^{(t)} \quad \text{for some constant } c > 0.
$$

Therefore, it takes $O(\log d)$ many steps for $\rho_m^{(t)}$ to increase from $d^{-1/2}$ to $O(1)$. During this period, the dynamics will go through two phases separated by a critical alignment level $\rho^\star$ such that

$$
\rho_m^{(t)} \gamma \cdot (|\rho_m^{(t)}|\gamma + d^{-1/2})^{s^\star-2} \approx \widetilde{O}(d^{-s^\star/2+3/4}),
$$

which gives that $\rho^\star = \widetilde{\Theta}(d^{-1/4})$. After the alignment $\rho_m^{(t)}$ reaches $\rho^\star$, there is a short period where $\rho_m^{(t)}$ grows rapidly as $\rho_m^{(t+1)}/\rho^\star \geq (\rho_m^{(t)}/\rho^\star)^{s^\star-1}$, until the alignment reaches $d^{-1/4+1/4(s^\star-1)}$. The length of this period is very short compared to the other periods on the road to weak alignment.

**Phase 2: from weak to strong alignment.** Finally, after $\rho_m^{(t)}$ grows to a constant scale, we need to track the value of $1 - \rho_m^{(t)}$. Again using (D.2), we can show that $1 - \rho_m^{(t+1)} \leq (1+c)(1 - \rho_m^{(t)})$ for some constant $c > 0$. Hence, it takes another $O(\log d)$ steps to eventually achieve strong alignment.

# E  PROOF OF THE MAIN THEOREM FOR THE UNIFORM PRIOR

Now we present the proof for Theorem 4.2. We introduce a shorthand $\rho = \langle \theta, \theta^\star \rangle$ for the alignment between $\theta$ and $\theta^\star$. This shorthand inherits the subscript and superscript of $\theta$ as well, i.e., $\rho_m^{(t)} = \langle \theta_m^{(t)}, \theta^\star \rangle$.

Recall from Algorithm 1 that at the $t$-th step, given the normalized gradient step $\bar{g}_m^{(t)} = g_m^{(t)}/\|g_m^{(t)}\|_2$, the updated weight parameter is given by

$$
\theta_m^{(t+1)} = \frac{\theta_m^{(t)} + \eta g_m^{(t)}/\|g_m^{(t)}\|_2}{\left\|\theta_m^{(t)} + \eta g_m^{(t)}/\|g_m^{(t)}\|_2\right\|_2}.
$$

Note that the alignment $\langle \theta_m^{(t+1)}, \theta^\star \rangle$ depends on the alignment of the previous iterate $\langle \theta_m^{(t)}, \theta^\star \rangle$ and the alignment of the current update step $\langle g_m^{(t)}/\|g_m^{(t)}\|, \theta^\star \rangle$, so we first need to analyze the latter.

Here, we stop to introduce an immediate result that is crucial in characterizing the alignment.

**Almost orthogonality of smoothing noise.** Recall that the perturbated weights are $w_{m,l}^{(t)} = (\gamma\theta_m^{(t)} + \xi_{m,l})/\|\gamma\theta_m^{(t)} + \xi_{m,l}\|_2$, where $\xi_{m,l} \overset{\text{i.i.d.}}{\sim} \text{Unif}(\mathbb{S}^{d-1})$. Due to the high dimensionality, $\xi_{m,l}$ is almost orthogonal to any designated direction with high probability. In the context, we are primarily interested in the alignment with $\theta^\star$. Correspondingly, we decompose $g_m^{(t)}$ with respect to the following orthonormal basis

$$
\{v_{m,1}^{(t)} = \theta^\star, v_{m,2}^{(t)} = (1-\rho^2)^{-1/2} \cdot (\theta_m^{(t)} - \langle \theta_m^{(t)}, \theta^\star \rangle \cdot \theta^\star), v_{m,3}^{(t)}, \ldots, v_{m,d}^{(t)}\}
\tag{E.1}
$$

We justify this property by defining the following nice event for $\epsilon > 0$:

$$\mathcal{E}_m^{(t)}(\epsilon) = \Big\{ \max_{1 \le i \le d} |\langle \xi_{m,l}, v_{m,i}^{(t)} \rangle| < \epsilon, \ \forall l \in [L] \Big\}.$$

This event helps in characterizing the alignment between the expected gradient $\mathbb{E}_{\mathbb{P}_{\theta^\star}}[g_m^{(t)}]$ and the signal $\theta^\star$. Additionally, we define another event for $\widetilde{\epsilon} > 0$:

$$\widetilde{\mathcal{E}}_m^{(t)}(\widetilde{\epsilon}) = \big\{ |\langle \xi_{m,l}, \xi_{m,l'} \rangle| < \widetilde{\epsilon}, \ \forall l, l' \in [L] \text{ s.t. } l \ne l' \big\}.$$

This event controls the correlation between the noise vectors, which helps in controlling the fluctuation of the gradient. The following lemma provides some direct benefits of these events.

**Lemma E.1** (Polarized weight on the nice event). *For the orthonormal directions* $\{\theta^\star, v_{m,2}^{(t)}, \dots, v_{m,d}^{(t)}\}$ *defined in* (E.1), *suppose that the corresponding nice event* $\mathcal{E}_m^{(t)}(\epsilon) \cap \widetilde{\mathcal{E}}_m^{(t)}(\widetilde{\epsilon})$ *holds and the polarization level* $\gamma \in (0, 1/2)$. *Then we have for any* $l \in [L]$ *that*

$$|\langle w_{m,l}^{(t)}, \theta^\star \rangle| \le 2(\gamma |\langle \theta_m^{(t)}, \theta^\star \rangle| + \epsilon), \quad |\langle w_{m,l}^{(t)}, v_{m,1} \rangle| \le 2\Big(\gamma \sqrt{1 - \langle \theta_m^{(t)}, \theta^\star \rangle^2} + \epsilon\Big),$$

$$|\langle w_{m,l}^{(t)}, v_{m,i}^{(t)} \rangle| \le 2 \cdot \epsilon, \quad 2 \le i \le d.$$

*Additionally, for any* $l \ne l'$, *we have that*

$$\langle w_{m,l}^{(t)}, w_{m,l'}^{(t)} \rangle \le 4(\gamma^2 + 2\gamma\epsilon + \widetilde{\epsilon}).$$

*Proof of Lemma E.1.* See Appendix E.3. $\qquad\qquad\square$

**Characterizing** $\langle g_m^{(t)}, \theta^\star \rangle$. Note that $g_m^{(t)} = n^{-1} \sum_i g_{m,i}^{(t)}$, where

$$g_{m,i}^{(t)} = \frac{1}{L} \sum_{l=1}^{L} \big( \psi(y_i^{(t)}, \langle w_{m,l}^{(t)}, z_i^{(t)} \rangle) \cdot z_i^{(t)} - \widehat{\psi}_1(y_i^{(t)}) \cdot w_{m,l} \big).$$

We characterize the alignment $\langle g_m^{(t)}, \theta^\star \rangle / \|g_m^{(t)}\|_2$ in Appendix E.1 with two steps, stated in two key propositions as follows:

1. In Proposition E.3, we analyze the magnitude of $\mathbb{E}_{\mathbb{P}_{\theta^\star}}[\langle g_m^{(t)}, \theta^\star \rangle]$ and $\|\mathbb{E}_{\mathbb{P}_{\theta^\star}}[P_{\theta^\star}^\perp \langle g_m^{(t)}, \theta^\star \rangle]\|$.

2. In Proposition E.4, we control the fluctuation of $\langle g_m^{(t)}, \theta^\star \rangle$ around its expectation using the polynomial-tail like property in Assumption 4.1(c).

Both propositions are established under the nice event $\mathcal{E}_m^{(t)}(\epsilon) \cap \widetilde{\mathcal{E}}_m^{(t)}(\widetilde{\epsilon})$. The proof of these propositions is deferred to Appendix E.3. Finally, with these two propositions, we prove Theorem 4.2 in Appendix E.2.

### E.1 PROPERTIES OF THE GRADIENT STEP

In this part, we characterize the alignment of normalized update $g_m^{(t)} / \|g_m^{(t)}\|$ with the signal $\theta^\star$, given $\theta_m^{(t)}$. Since we are focusing on the one step behavior for a fixed neuron $m \in [M]$, we omit the neuron index $m$ and time index $t$ in the sequel. To facilitate the presentation, we propose the following simplified setup that extract all the essential elements to describe the one-step behavior.

**Definition E.2.** *Fix* $\theta$ *and* $\theta^\star$ *and let* $\rho = \langle \theta, \theta^\star \rangle$. *Suppose the data points* $(z_1, y_1), \dots, (z_n, y_n)$ *are i.i.d. generated from* $\mathbb{P}_{\theta^\star}$. *Define* $w_l = (\gamma\theta + \xi_l) / \|\gamma\theta + \xi_l\|_2$ *for* $l = 1, \dots, L$, *where* $\xi_1, \dots \xi_L \overset{\text{i.i.d.}}{\sim} \text{Unif}(\mathbb{S}^{d-1})$ *are independent of* $\{(z_i, y_i)\}_{i=1}^n$. *Given the oracle* $\psi : \mathbb{R} \times \mathbb{R} \to \mathbb{R}$, *we define*

$$g = \frac{1}{nL} \sum_{i=1}^{n} \sum_{l=1}^{L} \big( \psi(y_i, \langle w_l, z_i \rangle) \cdot z_i - \widehat{\psi}_1(y_i) \cdot w_l \big).$$

*To describe the associated good event, we fix an orthonormal basis:*

$$v_1 = \theta^\star, v_2 = \frac{\theta - \rho\theta^\star}{\sqrt{1-\rho^2}}, v_3, \ldots, v_d,$$

*and define*

$$\mathcal{E}(\epsilon) = \left\{ |\langle \xi_l, \theta^\star \rangle| < \epsilon, \quad \max_{2 \leq i \leq d} |\langle \xi_l, v_i \rangle| < \epsilon, \quad \forall l \in [L] \right\};$$

$$\widetilde{\mathcal{E}}(\widetilde{\epsilon}) = \left\{ |\langle \xi_l, \xi_{l'} \rangle| < \widetilde{\epsilon}, \quad \forall l, l' \in [L] \quad \text{s.t.} \quad l \neq l' \right\}.$$

As mentioned in [Appendix D](#), we can reduce this problem to first characterizing $\mathbb{E}_{\mathbb{P}_{\theta^\star}}[g]$, and then control the fluctuation of $g$ around its expectation. To this end, we first introduce a lemma that characterizes the first moment of the gradient step. This lemma is valid for both the non-sparse and sparse setting and is helpful in understanding the structure of the expected gradient.

**Lemma H.2** (Decomposition of the first moment). *Suppose that we are working with the setting in [Definition E.2](#), where the oracle function $\psi$ follows [Assumption 4.1](#). Then it holds that*

$$\mathbb{E}_{\mathbb{P}_{\theta^\star}}[g] = \sum_{s \geq s^\star} \mathbb{E}_{\mathbb{Q}}[\zeta_s(y) \cdot \widehat{\psi}_{s-1}(y)] \cdot \frac{\sqrt{s}}{L} \sum_{l=1}^{L} \langle w_l, \theta^\star \rangle^{s-1} \cdot \theta^\star$$

$$+ \sum_{s \geq s^\star} \mathbb{E}_{\mathbb{Q}}[\zeta_s(y) \cdot \widehat{\psi}_{s+1}(y)] \cdot \frac{\sqrt{s+1}}{L} \sum_{l=1}^{L} \langle w_l, \theta^\star \rangle^{s} \cdot w_l.$$

*Proof of [Lemma H.2](#).* See [Appendix H](#). $\qquad\square$

One can easily see that the first summation term corresponds to the signal, while the second corresponds to the resilience of current weight. The structure of $w_l$ guarantees $\langle w_l, \theta^\star \rangle$ is small, therefore only the leading term in the geometric series above is dominant. Also we note that the leading term from the signal is larger than the leading term from the resilience, which indicates that the expected gradient is highly aligned with the true signal. This is justified in the next proposition.

Before stating it, we fix $M$ to be a sufficiently large constant that does not scale with $d$ and $T = O(\log d)$. The involvement of $M, T$ here is merely for the union bound argument in [Appendix E.2](#).

**Proposition E.3** (Alignment of expected gradient). *Suppose that we are working with the setting in [Definition E.2](#), where the oracle function $\psi$ follows [Assumption 4.1](#). Additionally, we set $\gamma = o(1)$, $L = \Omega\big((\epsilon \vee \gamma)^{s^\star - 1} \cdot d^{s^\star/2} \vee (d \log d)\big)$ and $\epsilon = o(1)$ is chosen such that $\Pr(\mathcal{E}(\epsilon)) = 1 - O(d^{-s^\star/2})$. Then there exists a $\{\xi_l\}_{l \in [L]}$-measurable event $\mathcal{E}_1$ with $\Pr(\mathcal{E}_1) \geq 1 - d^{-c}(MT)^{-1}$, such that on the event $\mathcal{E}_1 \cap \mathcal{E}(\epsilon)$, it holds that*

$$\langle \mathbb{E}_{\mathbb{P}_{\theta^\star}}[g], \theta^\star \rangle \simeq \begin{cases} \gamma\rho \cdot (\gamma|\rho| + d^{-1/2})^{s^\star - 2} & \text{if } s^\star \text{ is even;} \\ (\gamma|\rho| + d^{-1/2})^{s^\star - 1} & \text{if } s^\star \text{ is odd,} \end{cases}$$

*and that*

$$\left| \|\mathbb{E}_{\mathbb{P}_{\theta^\star}}[g]\|_2 - |\langle \mathbb{E}_{\mathbb{P}_{\theta^\star}}g, \theta^\star \rangle| \right| \lesssim (\gamma|\rho| + d^{-1/2})^{s^\star},$$

*as long as $\gamma|\rho| = \omega(d^{-1})$.*

*Proof of [Proposition E.3](#).* See [Appendix E.3](#). $\qquad\square$

From this proposition, it is already clear that the expected gradient is highly aligned with the signal $\theta^\star$ in the sense that $\|P_{\theta^\star}^\perp \mathbb{E}_{\mathbb{P}_{\theta^\star}}[g]\|_2 < |\langle \mathbb{E}_{\mathbb{P}_{\theta^\star}}[g], \theta^\star \rangle|$ whenever $\gamma|\rho| = \omega(d^{-1})$. Later we will see that this is indeed the case during the trajectory of [Algorithm 1](#).

**Proposition E.4** (Fluctuation of mini-batch gradient). *Under the simplified setting introduced in [Definition E.2](#) where $\psi : \mathbb{R} \times \mathbb{R} \to \mathbb{R}$ follows [Assumption 4.1](#). Suppose that we choose $\epsilon$ and $\widetilde{\epsilon}$ such that*

$$\epsilon^2 \leq \widetilde{\epsilon} \ll 1; \qquad 2\gamma\epsilon \leq \widetilde{\epsilon}.$$

*Also, suppose that sample size*

$$n = \Omega\Big(\big((\gamma^2 + \widetilde{\epsilon})^{s^\star - 1} + L^{-1}\big)^{-1} \cdot \log(d)^{2C_p + 2}\Big)$$

*where $C_p$ is defined in [Assumption 4.1(c)]. Then there exists a $\{(y_i, z_i)\}_{i \in [n]}$-measurable event $\mathcal{E}_2$ with $\Pr(\mathcal{E}_2{}^c) \leq d^{-c} \cdot (MT)^{-1}$. And it holds on $\mathcal{E}_2 \cap \mathcal{E}(\epsilon) \cap \widetilde{\mathcal{E}}(\widetilde{\epsilon})$ that*

$$\big|\langle g, \theta^\star \rangle - \langle \mathbb{E}_{\mathbb{P}_{\theta^\star}}[g], \theta^\star \rangle\big| \lesssim \sqrt{\frac{\big((\gamma^2 + \widetilde{\epsilon})^{s^\star - 1} + L^{-1}\big) \cdot \log(d)}{n}},$$

*and that*

$$\big\|g - \mathbb{E}_{\mathbb{P}_{\theta^\star}}[g]\big\|_2 \lesssim \sqrt{\frac{\big((\gamma^2 + \widetilde{\epsilon})^{s^\star - 1} + L^{-1}\big) \cdot d \log(d)}{n}}.$$

*Proof of [Proposition E.4].* See [Appendix E.3]. □

### E.2 Proof of the Main Theorem for Uniform Prior

Now we are ready to present the proof of [Theorem 4.2].

*Proof of [Theorem 4.2].* We first establish the good events required for the proof, characterize the properties of the update step on these events, and then put things together to establish the alignment of the model weights with the signal.

**Preparations.** To start with, we clarify the event we will work with by verifying that our configuration is compatible with the conditions in [Proposition E.3] and [Proposition E.4]. Recall that we set $L = \Omega\big(d^{(s^\star - 1)/2} \vee (d \log d)^{1/2}\big)$ and is at most polynomial in $d$, the scale of $L$ clearly satisfy that $L = \Omega\big((d^{1/2}\epsilon)^{s^\star} \vee (d \log d)\big)$. During our algorithm, $\gamma = (d^{-1} \cdot \log d)^{1/4} = o(1)$ is fixed. Choosing $\epsilon = d^{-1/2} \log d$, we have by [Lemma J.6] that for any $t$ and $m$, it holds that

$$1 - \Pr(\mathcal{E}_m^{(t)}(\epsilon)) \leq Ld \cdot \big(\exp(-d/16) + d^{-\log d/4}\big),$$

which decays faster than any constant-degree polynomial in $d$. Therefore, for sufficiently large $d$, it holds that $\Pr\Big(\mathcal{E}_m^{(t)}(\epsilon)^c\Big) = O(d^{-s^\star/2})$. So far, we see that all the conditions in [Proposition E.3] are satisfied and we denote the associated event as $\mathcal{E}_{m,1}^{(t)}$.

Next, we verify the conditions in [Proposition E.4]. We choose $\widetilde{\epsilon} = \sqrt{4\big(c + \log_d(MTL^2)\big) \cdot d^{-1} \log d}$, then it holds by [Lemma J.6] that

$$1 - \Pr\big(\widetilde{\mathcal{E}}_m^{(t)}(\widetilde{\epsilon})\big) \leq L^2 \cdot \big(\exp(-d/16) + d^{-\widetilde{\epsilon}^2 \cdot \log d/4}\big) \lesssim d^{-c}/MT.$$

Additionally, we see that both $\epsilon^2 \leq \widetilde{\epsilon} \ll 1$ and $2\gamma\epsilon \leq \widetilde{\epsilon}$ is satisfied for sufficiently large $d$. It is easily verified that our choice of $L = \Omega\big(d^{(s^\star + 1)/2} \vee (d \log d)\big)$ clearly meets the condition that

$$L \gtrsim (\epsilon \vee \gamma)^{s^\star - 1} \cdot d^{s^\star/2} \vee d \log d$$

we have that the sample size threshold is now

$$\frac{\log(d)^{2C_p + 2}}{\big((\gamma^2 + \widetilde{\epsilon})^{s^\star - 1} + L^{-1}\big)} \lesssim \log(d)^{2C_p + 2} \cdot d^{(s^\star - 1)/2},$$

which is satisfied by our choice $n = \Theta\Big(\big((d \log d)^{s^\star/2} \vee d \log d\big) \log d\Big)$. Hence, all the conditions in [Proposition E.4] are satisfied and we denote the associated event as $\mathcal{E}_{m,2}^{(t)}$.

Recalling that the gradient in Definition E.2 does not include the error term $\mathrm{err}_{m,l,i}^{(t)}$, we additionally need an event that controls the norm of the inputs $z_i^{(t)}{}_2^{(t)}$, which helps to control $\|\mathrm{err}_{m,l,i}^{(t)} \cdot z_i\|_2$. For this purpose, we define

$$\mathcal{E}_{m,3}^{(t)} = \left\{ \max_{i \in [n]} \|z_i^{(t)}\|_2 \leq \sqrt{d} \right\}.$$

By standard Bernstein's inequality, we have that $\Pr\left(\mathcal{E}_{m,3}^c{}^{(t)}\right) \leq Ld \cdot \exp\{-d/8\} = O(\exp\{-C'd\})$ for some $C' > 0$. To put things together, we work on the following event:

$$\mathcal{E} = \bigcap_{m=1}^{M} \bigcap_{t=1}^{T} \left( \mathcal{E}_m^{(t)}(\epsilon) \cap \widetilde{\mathcal{E}}_m^{(t)}(\widetilde{\epsilon}) \cap \mathcal{E}_{m,1}^{(t)} \cap \mathcal{E}_{m,2}^{(t)} \cap \mathcal{E}_{m,3}^{(t)} \right),$$

which is of $\Pr(\mathcal{E}) \geq 1 - O(d^{-c})$ for some $c > 0$ by the union bound argument. Denote

$$\bar{g}_m^{(t)} = \frac{1}{nL} \sum_{i=1}^{n} \sum_{l=1}^{L} \left( \psi(y_i^{(t)}, \langle w_{m,l}^{(t)}, z_i^{(t)} \rangle) \cdot z_i^{(t)} - \widehat{\psi}_1(y_i^{(t)}) \cdot w_{m,l} \right),$$

then $\bar{g}_m^{(t)}$ and the mini-batch data $\{(y_i^{(t)}, z_i^{(t)})\}_{i \in [n]}$ match the definition in Definition E.2, which allows us to apply Proposition E.3 and Proposition E.4. Thanks to the event $\mathcal{E}_{m,3}^{(t)}$, we always have for any $v \in \mathbb{S}^{d-1}$ that

$$\begin{aligned}
\left| \langle g_m^{(t)}, v \rangle - \langle \bar{g}_m^{(t)}, v \rangle \right| &\leq \left| \|g_m^{(t)}\|_2 - \|\bar{g}_m^{(t)}\|_2 \right| \\
&\leq d^{1/2} \cdot \max_{l,i} |\mathrm{err}_{m,l,i}^{(t)}| \\
&\leq d^{-9s^\star}.
\end{aligned} \tag{E.2}$$

In the sequel, we restrict our attention to neurons that have $d^{-1/2}/2$ alignment, i.e., the index $m$ such that $|\langle \theta_m^{(0)}, \theta^\star \rangle| \geq d^{-1/2}/2$. From now on, we will drop the neuron index $m$ and the iteration index $(t)$ in the following analysis for simplicity. The updated weight parameter is denoted as $\theta'$, and the alignment after the update is denoted as $\rho' = \langle \theta', \theta^\star \rangle$. Note that for large $M \gg 1$, the number of neurons with initial alignment $|\rho| \geq d^{-1/2}/2$ is at least $\Omega(M)$. For our convenience, in the following we will denote by

$$\kappa := \frac{n}{(d \log d)^{s^\star/2} \vee d \log d} \cdot (\log d)^{-1} = \Omega(1).$$

Under the preceding configuration, Proposition E.4 and Eq. (E.2) together imply that the fluctuations of $\langle g, \theta^\star \rangle$ can be further bounded by

$$\begin{aligned}
\left| \langle g, \theta^\star \rangle - \mathbb{E}_{\mathbb{P}_{\theta^\star}}[\langle \bar{g}, \theta^\star \rangle] \right| &\lesssim \sqrt{\frac{((\gamma^2 + \widetilde{\epsilon})^{s^\star - 1} + L^{-1}) \cdot \log(d)}{n}} + \left| \langle \bar{g}, \theta^\star \rangle - \langle g, \theta^\star \rangle \right| \\
&\lesssim \sqrt{\frac{(d^{-1} \log d)^{(s^\star - 1)/2} \cdot \log(d)}{n}} + d^{-9s^\star} \\
&= \begin{cases} d^{-(2s^\star - 1)/4} \cdot (\log d)^{-1/4} \cdot \kappa^{-1/2} & \text{if } s^\star \geq 2, \\ d^{-1/2} \cdot (\log d)^{-1/2} \cdot \kappa^{-1/2} & \text{if } s^\star = 1. \end{cases}
\end{aligned} \tag{E.3}$$

On the other hand, we have by Proposition E.3 and Eq. (E.2) that

$$\begin{aligned}
\mathbb{E}_{\mathbb{P}_{\theta^\star}}[\langle g, \theta^\star \rangle] &\gtrsim \mathbb{E}_{\mathbb{P}_{\theta^\star}}[\langle \bar{g}, \theta^\star \rangle] - \left| \langle g, \theta^\star \rangle - \mathbb{E}_{\mathbb{P}_{\theta^\star}}[\langle \bar{g}, \theta^\star \rangle] \right| \\
&\geq |\rho|\gamma(|\rho|\gamma + d^{-1/2})^{s^\star - 2} - d^{-9s^\star} \\
&\geq d^{-(2s^\star - 1)/4} \cdot (\log d)^{1/4},
\end{aligned}$$

whenever $|\rho|\gamma = \Omega(d^{-3/4})$. Therefore, when $\kappa$ is sufficiently large, we have the fluctuations to be strictly bounded by half of the signal strength. Thus, we have

$$|\langle g, \theta^\star \rangle| \geq \frac{1}{2} \cdot |\langle \mathbb{E}_{\mathbb{P}_{\theta^\star}}[\bar{g}], \theta^\star \rangle|.$$

For the norm of $g$, we have that

$$
\begin{aligned}
\|g\|_2 &\leq \left|\|g\|_2 - \|\bar{g}\|_2\right| + \left|\|\bar{g}\|_2 - \|\mathbb{E}_{\mathbb{P}_{\theta^\star}}[\bar{g}]\|_2\right| \\
&\quad + \left|\|\mathbb{E}_{\mathbb{P}_{\theta^\star}}[\bar{g}]\|_2 - \mathbb{E}_{\mathbb{P}_{\theta^\star}}[\langle\bar{g},\theta^\star\rangle]\right| + \left|\mathbb{E}_{\mathbb{P}_{\theta^\star}}[\langle\bar{g},\theta^\star\rangle]\right| \\
&\leq d^{-9s^\star} + \|\bar{g} - \mathbb{E}_{\mathbb{P}_{\theta^\star}}[\bar{g}]\|_2 + \left|\|\mathbb{E}_{\mathbb{P}_{\theta^\star}}[\bar{g}]\|_2 - \langle\mathbb{E}_{\mathbb{P}_{\theta^\star}}[\bar{g}],\theta^\star\rangle\right| + |\langle\mathbb{E}_{\mathbb{P}_{\theta^\star}}[\bar{g}],\theta^\star\rangle| \\
&\lesssim \left|\mathbb{E}_{\mathbb{P}_{\theta^\star}}[\langle\bar{g},\theta^\star\rangle]\right| + (\gamma|\rho| + d^{-1/2})^{s^\star} + \sqrt{\frac{(d^{-1}\log d)^{(s^\star-1)/2}\cdot d\log d}{n}},
\end{aligned} \tag{E.4}
$$

where in the second inequality, we apply Eq. (E.2) and the triangular inequality that $\left|\|\bar{g}\| - \|\mathbb{E}_{\mathbb{P}_{\theta^\star}}[\bar{g}]\|\right| \leq \|\bar{g} - \mathbb{E}_{\mathbb{P}_{\theta^\star}}[\bar{g}]\|$. And the last inequality is deduced by combining the result Proposition E.3 and the fact that $d^{-9s^\star} \ll d^{-s^\star/2}$. For the leading term, it holds by Proposition E.3 that

$$
|\langle\mathbb{E}_{\mathbb{P}_{\theta^\star}}[\bar{g}],\theta^\star\rangle| \simeq \begin{cases} (|\rho|\gamma + d^{-1/2})^{s^\star-1} & \text{if } s^\star \text{ is odd,} \\ |\rho|\gamma(|\rho|\gamma + d^{-1/2})^{s^\star-2} & \text{if } s^\star \text{ is even.} \end{cases} \tag{E.5}
$$

Recall that the alignment admits the following iterative update rule:

$$
|\langle\theta',\theta^\star\rangle| = \left|\left\langle\frac{\theta + \eta G}{\|\theta + \eta G\|_2},\theta^\star\right\rangle\right| \geq \frac{|\langle G,\theta^\star\rangle| - \eta^{-1}|\langle\theta,\theta^\star\rangle|}{1 + \eta^{-1}},
$$

where $G = g/\|g\|_2$. In the following, we will define $\rho^\star = d^{-1/4}(\log d)^{1/4}$ as a critical threshold before the weak alignment. Specifically, in the phase I of weak alignment, we assume that $|\rho| \leq \rho^\star$. When the training process goes across this critical threshold, the dominant term in $\mathbb{E}_{\mathbb{P}_{\theta^\star}}[\langle\bar{g},\theta^\star\rangle] \asymp (\gamma\rho)^{\mathbb{1}\{s^\star \text{ is even}\}}(\gamma\rho + d^{-1/2})^{\star-1-\mathbb{1}\{s^\star \text{ is even}\}}$ becomes $\gamma\rho$ instead of $d^{-1/2}$.

**Stage I of weak alignment. Case I: $s^\star \neq 1$, $s^\star$ is odd.** Combining the results in Eq. (E.3), (E.4) and (E.5), we conclude that for $s^\star \geq 3$ being odd,

$$
\begin{aligned}
|\langle g,\theta^\star\rangle| &\gtrsim (|\rho|\cdot(d\log d)^{-1/4} + d^{-1/2})^{s^\star-1}, \\
\|g\|_2 &\lesssim (|\rho|\cdot(d\log d)^{-1/4} + d^{-1/2})^{s^\star-1} + d^{-(2s^\star-3)/4}\cdot(\log d)^{-1/4}\cdot\kappa^{-1/2}.
\end{aligned}
$$

It is important to note that here "$\gtrsim$" and "$\lesssim$" only hides constants that are independent of $d$ and $n$. Combining these two inequalities, we have that

$$
\begin{aligned}
\frac{|\langle g,\theta^\star\rangle|}{\|g\|_2} &\gtrsim \frac{(|\rho|d^{1/4}(\log d)^{-1/4} + 1)^{s^\star-1}}{(|\rho|d^{1/4}(\log d)^{-1/4} + 1)^{s^\star-1} + d^{1/4}(\log d)^{-1/4}\cdot\kappa^{-1/2}} \\
&= \frac{(|\rho|/\rho^\star + 1)^{s^\star-1}}{(|\rho|/\rho^\star + 1)^{s^\star-1} + \kappa^{-1/2}/\rho^\star}.
\end{aligned}
$$

Thus, if $|\rho| \leq \rho^\star$ and take $\kappa$ to be a sufficiently large constant, after the first gradient update, the alignment will grow to at least $\rho^\star$ by noting that $|\langle g,\theta^\star\rangle|/\|g\|_2 \gtrsim \rho^\star\kappa^{1/2}$ and that

$$
\begin{aligned}
|\rho'| &\gtrsim \frac{\rho^\star\kappa^{1/2} - \eta^{-1}|\rho|}{1 + \eta^{-1}} \\
&\geq \rho^\star\cdot\frac{\sqrt{\kappa} - \eta^{-1}}{1 + \eta^{-1}} \geq \rho^\star.
\end{aligned}
$$

As a summary of Case I(a), with one step of gradient update, the alignment will grow to at least $\rho^\star$ if $|\rho| \leq \rho^\star$.

**Stage I of weak alignment. Case II: $s^\star$ is even.** In the case where $s^\star$ is even, we have by the previous arguments that

$$
\begin{aligned}
|\langle g,\theta^\star\rangle| &\gtrsim |\rho|\cdot(d\log d)^{-1/4}\cdot(|\rho|\cdot(d\log d)^{-1/4} + d^{-1/2})^{s^\star-2}, \\
\|g\|_2 &\lesssim |\rho|\cdot(d\log d)^{-1/4}\cdot(|\rho|\cdot(d\log d)^{-1/4} + d^{-1/2})^{s^\star-2} \\
&\quad + d^{-(2s^\star-3)/4}\cdot(\log d)^{-1/4}\cdot\kappa^{-1/2}
\end{aligned}
$$

Here, we use the following fact that

$$(|\rho|\gamma + d^{-1/2})^{s^\star} \lesssim (|\rho|\gamma + d^{-1/2})^{s^\star-2} \cdot (\rho^2\gamma^2 + d^{-1}) \lesssim (|\rho|\gamma + d^{-1/2})^{s^\star-2}|\rho|\gamma,$$

where the last inequality holds since $|\rho| \geq d^{-1/2}/2$. Thus, we conclude that

$$
\begin{aligned}
\frac{|\langle g, \theta^\star \rangle|}{\|g\|_2} &\gtrsim \frac{|\rho| \cdot (|\rho|d^{1/4}(\log d)^{-1/4} + 1)^{s^\star-2}}{|\rho| \cdot (|\rho|d^{1/4}(\log d)^{-1/4} + 1)^{s^\star-2} + \kappa^{-1/2}} \\
&= \frac{|\rho| \cdot (|\rho|/\rho^\star + 1)^{s^\star-2}}{|\rho| \cdot (|\rho|/\rho^\star + 1)^{s^\star-2} + \kappa^{-1/2}}.
\end{aligned}
\tag{E.6}
$$

Note that $\kappa = \Omega(1)$. Hence before the alignment reaches $\rho^\star$, $\kappa^{-1/2}$ will dominate the denominator in Eq. (E.6), which gives us that $|\langle g, \theta^\star \rangle|/\|g\|_2 \gtrsim |\rho| \cdot \sqrt{\kappa}$. Importantly, the "$\gtrsim$" hides constants that are independent of $d$ and $\kappa$. Thus, by taking $\kappa$ to be a sufficiently large constant, we can conclude that

$$|\rho'| \geq |\rho| \cdot \frac{\sqrt{\kappa} - \eta^{-1}}{1 + \eta^{-1}} \geq 2|\rho|.$$

As a summary of Case II(a), before the alignment reaches $\rho^\star$, the alignment will grow exponentially fast, and this phase takes at most $O(\log(d))$ steps. In the following, we consider the case when $|\rho| \geq \rho^\star$ for both cases I and II.

**Stage II of weak alignment. Case I&II combined.**   Now we consider the case when $|\rho| \geq \rho^\star$ for both cases I and II, i.e., $s^\star \geq 2$. For this case, we have $|\rho|/\rho^\star + 1 \simeq |\rho|/\rho^\star$. Let us define $r = |\rho|/\rho^\star \geq 1$ and $r' = |\rho'|/\rho^\star$, and it follows that

$$\frac{|\langle g, \theta^\star \rangle|}{\|g\|_2} \gtrsim \frac{r^{s^\star-1}}{r^{s^\star-1} + \kappa^{-1/2}/\rho^\star},$$

and consequently:

$$
\begin{aligned}
r' &\gtrsim \frac{r^{s^\star-1} \cdot (r^{s^\star-1}\rho^\star + \kappa^{-1/2})^{-1} - \eta^{-1}r}{1 + \eta^{-1}} \\
&\geq \frac{(\sqrt{\kappa} \cdot r^{s^\star-1}) \wedge \rho^{\star-1} - \eta^{-1}r}{1 + \eta^{-1}}
\end{aligned}
$$

It can be noted that the maximal ratio $r \leq (\rho^\star)^{-1}$, and also $\sqrt{\kappa} \cdot r^{s^\star-1} \geq 2\eta^{-1}r$ given that $\kappa$ is sufficiently large and $r \geq 1$. Thus, we conclude that in this case

$$r' \gtrsim (\sqrt{\kappa} \cdot r^{s^\star-1}) \wedge \rho^{\star-1}.$$

For this case, the growth of the alignment will be also at least exponentially fast, until it reaches $\Omega((\rho^\star)^{-1})$, i.e., $|\rho| = C$ for some constant $C$. This phase takes at most $O(\log(d))$ steps.

**Strong alignment. Case I&II combined.**   We need a more careful analysis for this case in order to achieve strong alignment. When the alignment is on a constant level, we can deduce from its original form that

$$|\langle \mathbb{E}_{\mathbb{P}_{\theta^\star}}[\bar{g}], \theta^\star \rangle| = B(\rho, \{\xi_l\}_{l \in [L]}) \cdot (|\rho|\gamma)^{s^\star-1} + E_1,$$

where $B(\rho, \{\xi_l\}_{l \in [L]}) = \Omega(1)$ is a constant that depends on $\rho$ and the random perturbations $\{\xi_l\}_{l \in [L]}$ and the error term follows that $|E| \leq O(d^{-s^\star})$. . In the following, we will drop the dependency on $\rho$ and $\{\xi_l\}_{l \in [L]}$ and use $B$ for simplicity. We have by Eq. (E.3) that

$$
\begin{aligned}
|\langle g, \theta^\star \rangle| &\geq B(|\rho|\gamma)^{s^\star-1} - O\left(d^{-(2s^\star-1)/4} \cdot (\log d)^{-1/4} \cdot \kappa^{-1/2}\right) \\
&= B(|\rho|\gamma)^{s^\star-1} - O\left((d\log d)^{-s^\star/4} \cdot \gamma^{s^\star-1} \cdot \kappa^{-1/2}\right) \\
&= \gamma^{s^\star-1} \cdot \left(B|\rho|^{s^\star-1} - (d\log d)^{-s^\star/4} \cdot O(\kappa^{-1/2})\right),
\end{aligned}
$$

and also

$$\|g\|_2 \leq B(|\rho|\gamma)^{s^\star - 1} + O\left((|\rho|\gamma)^{s^\star} + d^{-(2s^\star - 3)/4} \cdot (\log d)^{-1/4} \cdot \kappa^{-1/2}\right)$$
$$\leq B(|\rho|\gamma)^{s^\star - 1} + O\left((|\rho|\gamma)^{s^\star} + d^{-(s^\star - 2)/4} \cdot (\log d)^{-s^\star/4} \cdot \gamma^{s^\star - 1} \cdot \kappa^{-1/2}\right)$$
$$\leq \gamma^{s^\star - 1} \cdot (B|\rho|^{s^\star - 1} + O(d^{-(s^\star - 2)/4} \cdot (\log d)^{-s^\star/4} \cdot \kappa^{-1/2})).$$

Therefore, once the alignment reaches a constant level $|\rho| \geq O(1)$, we have

$$|\langle G, \theta^\star \rangle| = \frac{|\langle g, \theta^\star \rangle|}{\|g\|_2} \geq \frac{B|\rho|^{s^\star - 1} - O(d^{-s^\star/4})}{B|\rho|^{s^\star - 1} + O(d^{-1/4}(\log d)^{1/4})}$$
$$\geq 1 - O(d^{-1/4}(\log d)^{1/4}) =: 1 - \Delta.$$

Here, $\Delta \simeq d^{-1/4}(\log d)^{1/4}$. Thus, as long as $\eta > 2$, after one step gradient,

$$|\rho'|^2 = \frac{\langle G + \eta^{-1}\theta, \theta^\star \rangle^2}{\langle G + \eta^{-1}\theta, \theta^\star \rangle^2 + \|P_{\theta^\star}^\perp(G + \eta^{-1}\theta)\|_2^2}$$
$$\geq \frac{(1 - \Delta - \eta^{-1}|\rho|)^2}{(1 - \Delta - \eta^{-1}|\rho|)^2 + (\sqrt{1 - (1 - \Delta)^2} + \eta^{-1}\sqrt{1 - \rho^2})^2}$$
$$\geq \frac{(1 - \eta^{-1} - \Delta)^2}{(1 - \eta^{-1} - \Delta)^2 + \eta^{-2}(1 - \rho^2) + 2\sqrt{2\Delta} + 2\Delta^2}$$
$$= \frac{(1 - \eta^{-1})^2}{(1 - \eta^{-1})^2 + \eta^{-2}(1 - \rho^2)} - O(\sqrt{\Delta}).$$

Here, the first equality holds by the Pythagorean theorem, the first inequality holds by the triangle inequality, and in the last line, we separate the major term and the error term that scales with $\sqrt{\Delta}$, where we use the fact that $1 - \eta^{-1} > 1/2$ with $\eta > 2$. In addition, by letting $\tau = \eta^{-2}/(1 - \eta^{-1})^2$, we have

$$1 - (\rho')^2 = 1 - \frac{(1 - \eta^{-1})^2}{(1 - \eta^{-1})^2 + \eta^{-2}(1 - \rho^2)} + O(\sqrt{\Delta})$$
$$= \frac{\tau(1 - \rho^2)}{1 + \tau(1 - \rho^2)} + O(\sqrt{\Delta})$$
$$\leq \tau(1 - \rho^2) + O(\sqrt{\Delta}).$$

Therefore, we conclude that as long as $\tau < 1$, i.e., $\eta > 2$, $1 - \rho^2$ will exponentially decrease to $O((1 - \tau)^{-1} \cdot \sqrt{\Delta})$, and achieves strong alignment in $O((\log \Delta^{-1})/(\log \tau^{-1}))$ steps.

**Weak & strong alignment. Case III: $s^\star = 1$.** In this case, we conclude from the previous arguments that regardless of the alignment level, it always holds that

$$|\langle \mathbb{E}_{\mathbb{P}_{\theta^\star}}[\bar{g}], \theta^\star \rangle| = B = O(1),$$

which gives us

$$|\langle g, \theta^\star \rangle| \geq B - O(d^{-1/2} \cdot (\log d)^{-1/2} \cdot \kappa^{-1/2}),$$

and

$$\|g\|_2 \leq B + O(d^{-1/4}(\log d)^{1/4} + (\log d)^{-1/2} \cdot \kappa^{-1/2}),$$

where we use the fact that $n = \kappa \cdot d(\log d)^2$. Therefore, we also have

$$|\langle G, \theta^\star \rangle| = \frac{|\langle g, \theta^\star \rangle|}{\|g\|_2} \geq 1 - O((\log d)^{-1/2} \cdot \kappa^{-1/2}) = 1 - \Delta,$$

where in this case, we also have $\Delta \simeq (\log d)^{-1/2}$ just like $s^\star = 2$ in the previous case, and the rest of the proof follows the same arguments as in the previous case for the strong alignment. $\qquad\square$

### E.3 Proof of Key Results

*Proof of Lemma E.1.* We begin with proving the first part of the lemma. For conciseness, we drop the superscript $(t)$ and simply denote $\theta_m$ as the present weight. We have the projection of $w_{m,l}$ onto $\theta^\star$ as

$$|\langle w_{m,l}, \theta^\star \rangle| = \left| \frac{\gamma \langle \theta_m, \theta^\star \rangle + \langle \xi_{m,l}, \theta^\star \rangle}{\|\gamma \theta_m + \xi_{m,l}\|_2} \right| \leq 2(\gamma |\rho_m| + \epsilon).$$

For direction $v_{m,2}$, by definition we have

$$\begin{aligned}
|\langle w_{m,l}, v_{m,2} \rangle| &= \left| \frac{\gamma \langle \theta_m, v_{m,2} \rangle + \langle \xi_{m,l}, v_{m,2} \rangle}{\|\gamma \theta_m + \xi_{m,l}\|_2} \right| \\
&\leq 2 \cdot \left( \gamma \left| \left\langle \theta_m, \frac{\theta_m - \rho_m \theta^\star}{\|\theta_m - \rho_m \theta^\star\|_2} \right\rangle \right| + \epsilon \right) \\
&= 2 \left( \gamma \sqrt{1 - \rho_m^2} + \epsilon \right).
\end{aligned}$$

For the remaining directions, we always have

$$|\langle w_{m,l}, v_{m,i} \rangle| = \left| \frac{\gamma \langle \theta_m, v_{m,i} \rangle + \langle \xi_{m,l}, v_{m,i} \rangle}{\|\gamma \theta_m + \xi_{m,l}\|_2} \right| \leq 2 \cdot \epsilon,$$

where we use the fact that $\langle \theta_m, v_{m,i} \rangle = 0$ for $i \geq 2$. This completes the proof for the first part.

On the joint nice event $\widetilde{\mathcal{E}}_m(\widetilde{\epsilon})$, we have that

$$\begin{aligned}
|\langle w_{m,l}, w_{m,l'} \rangle| &\leq 4 \cdot \langle \gamma \theta_m + \xi_{m,l}, \gamma \theta_m + \xi_{m,l'} \rangle \\
&\leq 4 (\gamma^2 + \widetilde{\epsilon} + \gamma \langle \theta_m, \xi_{m}, l \rangle + \gamma \langle \theta_m, \xi_{m,l'} \rangle).
\end{aligned}$$

On the other hand, it holds on the event $\mathcal{E}_m(\epsilon)$ that

$$|\langle \theta_m, \xi_{m,l} \rangle| = |\langle \sqrt{1 - \rho^2} \cdot v_{m,2} + \rho v_{m,1}, \xi_{m,l} \rangle| \leq (\sqrt{1 - \rho^2} + |\rho|) \cdot \epsilon \leq 2\epsilon.$$

Therefore, we have for any $l \neq l'$ that

$$|\langle w_{m,l}, w_{m,l'} \rangle| \leq 4(\gamma^2 + 4\epsilon\gamma + \widetilde{\epsilon}).$$

$\square$

*Proof of Proposition E.3.* Invoking Lemma H.2 with the fact that $\|\theta^\star\|_2 = 1$, we can decompose $\langle \mathbb{E}_{\mathbb{P}_{\theta^\star}}[g], \theta^\star \rangle$ as

$$\begin{aligned}
\langle \mathbb{E}_{\mathbb{P}_{\theta^\star}}[g], \theta^\star \rangle &= \sum_{s \geq s^\star} \frac{\sqrt{s+1}}{L} \sum_{l=1}^{L} \mathbb{E}_{\mathbb{Q}}[\zeta_s(y) \cdot \widehat{\psi}_{s+1}(y)] \cdot \langle w_l, \theta^\star \rangle^{s+1} \\
&\quad + \sum_{s \geq s^\star} \frac{\sqrt{s}}{L} \sum_{l=1}^{L} \mathbb{E}_{\mathbb{Q}}[\zeta_s(y) \cdot \widehat{\psi}_{s-1}(y)] \cdot \langle w_l, \theta^\star \rangle^{s-1} \\
&= \mathbb{E}_{\mathbb{Q}}[\zeta_{s^\star}(y) \cdot \widehat{\psi}_{s^\star - 1}(y)] \cdot \frac{\sqrt{s^\star}}{L} \sum_{l=1}^{L} \langle w_l, \theta^\star \rangle^{s^\star - 1} + R, \quad\quad \text{(E.7)}
\end{aligned}$$

where all the remainder terms are collected by $R$, defined as

$$R = \sum_{s \geq s^\star} \frac{\sqrt{s+1}}{L} \sum_{l=1}^{L} \left( \mathbb{E}_{\mathbb{Q}}[\zeta_s(y) \cdot \widehat{\psi}_{s+1}(y)] \langle w_l, \theta^\star \rangle + \mathbb{E}_{\mathbb{Q}}[\zeta_{s+1}(y) \cdot \widehat{\psi}_s(y)] \right) \cdot \langle w_l, \theta^\star \rangle^s.$$

Below we will analyze the scale of each term in Eq. (E.7), and show that the remainder $R$ is negligible compared to the first term in Eq. (E.7) with high probability over the randomness of the injected noise $\xi_1, \ldots, \xi_L$.

**Analysis for the remainder term $R$ in Eq. (E.7).** To bound $|R|$, we apply the triangle inequality with the fact that $|\langle w_l, \theta^\star \rangle| \leq 1$ to get

$$|R| \leq \sum_{s \geq s^\star} \frac{\sqrt{s+1}}{L} \sum_{l=1}^{L} \mathbb{E}_{\mathbb{Q}}\Big[ |\zeta_s(y) \cdot \widehat{\psi}_{s+1}(y)| + |\zeta_{s+1}(y) \cdot \widehat{\psi}_s(y)| \Big] \cdot |\langle w_l, \theta^\star \rangle|^s.$$

Since $\mathbb{E}_{\mathbb{Q}}[\zeta_{s+1}(y)^2] \leq 1$ for all $s \geq 0$ by the property of the decomposition of the likelihood ratio, we have

$$\mathbb{E}_{\mathbb{Q}}[|\zeta_{s+1}(y) \cdot \widehat{\psi}_s(y)|] \leq \mathbb{E}_{\mathbb{Q}}[\zeta_{s+1}(y)^2]^{1/2} \cdot \mathbb{E}_{\mathbb{Q}}[\widehat{\psi}_s(y)^2]^{1/2}$$
$$\leq \mathbb{E}_{\mathbb{Q}}[\widehat{\psi}_s(y)^2]^{1/2}$$
$$\leq \sqrt{\sum_{s=0}^{\infty} \mathbb{E}[\widehat{\psi}_s(y)^2]} = O(1),$$

and similarly for $\mathbb{E}_{\mathbb{Q}}[|\zeta_s(y) \cdot \widehat{\psi}_{s+1}(y)|]$. It then suffices to bound $\sum_{s \geq s^\star} (\sqrt{s+1})/L \cdot \sum_{l=1}^{L} |\langle w_l, \theta^\star \rangle|^s$. Recall that we restrict ourselves to the following nice event

$$\mathcal{E}(\epsilon) : \left\{ |\langle \xi_l, \theta^\star \rangle| < \epsilon, \quad \max_{2 \leq i \leq d} |\langle \xi_l, v_i \rangle| < \epsilon, \quad \forall l \in [L] \right\},$$

where $\{v_1 = \theta^\star, v_2 = (\theta - \rho\theta^\star)/\sqrt{1-\rho^2}, v_3, \dots v_d\}$ is an orthonormal basis. Since we assume that $\gamma = o(1)$, it follows from Lemma E.1 that $|\langle w_l, \theta^\star \rangle| < 1/2$ for all $l \in [L]$ on $\mathcal{E}(\epsilon)$. Consequently, it holds on $\mathcal{E}(\epsilon)$ that

$$\sum_{s \geq s^\star} \frac{\sqrt{s+1}}{L} \sum_{l=1}^{L} |\langle w_l, \theta^\star \rangle|^s \leq \sum_{s \geq s^\star} \sqrt{s+1} \cdot \left( \frac{1}{2} \right)^{s-s^\star} \cdot \frac{1}{L} \sum_{l=1}^{L} |\langle w_l, \theta^\star \rangle|^{s^\star}$$
$$\lesssim \frac{1}{L} \sum_{l=1}^{L} |\langle w_l, \theta^\star \rangle|^{s^\star}, \tag{E.8}$$

where $\lesssim$ hides a constant that depends on $s^\star$. Now it reduces to upper bound the right hand side in Eq. (E.8). To proceed, we define

$$\widetilde{w}_l = \begin{cases} w_l & \text{if } \sup_i |\langle w_l, v_i \rangle| < \epsilon; \\ 0 & \text{otherwise.} \end{cases}$$

It can be easily verified that $\widetilde{w}_l = w_l$ for any $l \in [L]$ on $\mathcal{E}(\epsilon)$, and $\{\widetilde{w}_l\}_{l \in [L]}$ is a sequence of independent and bounded random vectors. By Lemma E.1, we have that

$$|\langle \widetilde{w}_l, \theta^\star \rangle| \leq 2\gamma \vee \epsilon.$$

To find its second moment, we have by definition that

$$\mathbb{E}_{\widetilde{w}_l}[\langle \widetilde{w}_l, \theta^\star \rangle^{2s^\star}] = \mathbb{E}_{w_l}\big[ \langle w_l, \theta^\star \rangle^{2s^\star} \cdot \mathbb{1}\big\{ \sup_i |\langle w_l, v_i \rangle| \leq \epsilon \big\} \big]$$
$$\leq \mathbb{E}_{w_l}[\langle w_l, \theta^\star \rangle^{2s^\star}]$$
$$\simeq (|\rho|\gamma + d^{-1/2})^{2s^\star},$$

where the last line holds by Lemma H.4. Therefore, we can apply the Bernstein's inequality (Lemma J.1) to the right hand side of Eq. (E.8) restricted to $\mathcal{E}(\epsilon)$. We deduce from it that there exists a event $\mathcal{E}_{1,1}$ with $\Pr(\mathcal{E}_{1,1}) \geq 1 - d^{-c}/(MT)$, and it holds on $\mathcal{E}_{1,1} \cap \mathcal{E}(\epsilon)$ that

$$\frac{1}{L} \sum_{l=1}^{L} |\langle w_l, \theta^\star \rangle|^{s^\star} = \frac{1}{L} \sum_{l=1}^{L} |\langle \widetilde{w}_l, \theta^\star \rangle|^{s^\star}$$
$$\lesssim \left( 1 + \sqrt{L^{-1} \log d} \right) \cdot (|\rho|\gamma + d^{-1/2})^{s^\star} + \frac{(\epsilon \vee \gamma)^{s^\star} \log d}{L}$$
$$\lesssim (|\rho|\gamma + d^{-1/2})^{s^\star} \tag{E.9}$$

Here we use the fact that $M$ and $T$ are at most polynomial in $d$ and the last line holds since we choose $L = \Omega\Big( \big( d^{1/2} \cdot (\epsilon \vee \gamma) \big)^{s^\star} \cdot \log d \vee \log d \Big)$.

**Analysis for the dominant term in Eq.** (E.7). We then consider the major term $L^{-1} \sum_{l=1}^{L} \langle w_l, \theta^\star \rangle^{s^\star - 1}$. By the definition of $\widetilde{w}_l$, we can approximate its expectation as follows:

$$
\begin{aligned}
\mathbb{E}_{\widetilde{w}_l}[\langle \widetilde{w}_l, \theta^\star \rangle^{s^\star - 1}] &= \mathbb{E}_{w_l}\left[ \langle w_l, \theta^\star \rangle^{s^\star - 1} \cdot \mathbb{1}\left\{ \sup_i |\langle w_l, v_i \rangle| \le \epsilon \right\} \right] \\
&\simeq \mathbb{E}_{w_l}[\langle w_l, \theta^\star \rangle^{s^\star - 1}] \pm \Pr\left( \sup_i |\langle w_l, v_i \rangle| > \epsilon \right) \\
&\simeq \mathbb{E}_{w_l}[\langle w_l, \theta^\star \rangle^{s^\star - 1}] \pm \Pr\left( \mathcal{E}(\epsilon)^c \right),
\end{aligned}
$$

where we use the fact that $|\langle w_l, \theta^\star \rangle|^{s^\star - 1} \le 1$ and the event $\{\sup_i |\langle w_l, v_i \rangle| > \epsilon\} \subset \mathcal{E}(\epsilon)^c$. Again, we have by Lemma H.4 that

$$
\mathbb{E}_{w_l}[\langle w_l, \theta^\star \rangle^{s^\star - 1}] \simeq \begin{cases} (|\rho|\gamma + d^{-1/2})^{s^\star - 1} & \text{if } s^\star \text{ is odd}; \\ \rho\gamma(|\rho|\gamma + d^{-1/2})^{s^\star - 2} & \text{if } s^\star \text{ is even}. \end{cases}
$$

Similarly, we have for the second moment that

$$
\begin{aligned}
\mathbb{E}_{\widetilde{w}_l}[\langle \widetilde{w}_l, \theta^\star \rangle^{2(s^\star - 1)}] &= \mathbb{E}_{w_l}\left[ \langle w_l, \theta^\star \rangle^{2(s^\star - 1)} \cdot \mathbb{1}\left\{ \sup_i |\langle w_l, v_i \rangle| \le \epsilon \right\} \right] \\
&\le \mathbb{E}_{w_l}[\langle w_l, \theta^\star \rangle^{2(s^\star - 1)}] \\
&\simeq (|\rho|\gamma + d^{-1/2})^{2(s^\star - 1)}.
\end{aligned}
$$

Given the boundedness on $\mathcal{E}(\epsilon)$ and the second moment characterization, the Bernstein's inequality (Lemma J.1) implies that there exists $\mathcal{E}_{1,2}$ with $\Pr(\mathcal{E}_{1,2}) \ge 1 - d^{-c}/(MT)$. Furthermore, it holds on $\mathcal{E}_{1,2} \cap \mathcal{E}(\epsilon)$ that

$$
\begin{aligned}
\frac{1}{L} \sum_{l=1}^{L} \langle w_l, \theta^\star \rangle^{s^\star - 1} &= \frac{1}{L} \sum_{l=1}^{L} \langle \widetilde{w}_l, \theta^\star \rangle^{s^\star - 1} \\
&= \mathbb{E}_{w_l}[\langle w_l, \theta^\star \rangle^{s^\star - 1}] + E,
\end{aligned}
$$

where the error term $E$ is absolutely bounded as

$$
\begin{aligned}
|E| &\lesssim (|\rho|\gamma + d^{-1/2})^{s^\star - 1} \cdot \sqrt{\frac{\log d}{L}} + \frac{(\epsilon \vee \gamma)^{s^\star - 1} \log d}{L} + \Pr\left( \mathcal{E}(\epsilon)^c \right) \\
&\lesssim (|\rho|\gamma + d^{-1/2})^{s^\star} + \Pr(\overline{\mathcal{E}}_m^{(t)}(\epsilon)) \\
&\lesssim (|\rho|\gamma + d^{-1/2})^{s^\star}.
\end{aligned}
$$

Here, the second line holds because $L = \Omega\left( \left( (\epsilon \vee \gamma)^{s^\star - 1} \cdot d^{s^\star/2} \right) \log d \vee d \log d \right)$ and the last line holds because $\Pr(\overline{\mathcal{E}}(\epsilon)) \le d^{-s^\star/2}$.

So far, we have obtained that on the event $\mathcal{E}(\epsilon) \cap \mathcal{E}_{1,1} \cap \mathcal{E}_{1,2}$, the following holds:

$$
\mathbb{E}_{\mathbb{P}_{\theta^\star}}[\langle g, \theta^\star \rangle] \simeq \mathbb{E}_{w_l}[\langle w_l, \theta^\star \rangle^{s^\star - 1}] + E + R,
$$

where $|E| + |R| \lesssim (|\rho|\gamma + d^{-1/2})^{s^\star}$ given our configuration of $L$ and $\Pr(\mathcal{E}(\epsilon)^c)$. On the other hand, provided that $|\rho|\gamma \gg d^{-1}$, we have that

$$
\begin{aligned}
(|\rho|\gamma + d^{-1/2})^{s^\star} &= (|\rho|\gamma + d^{-1/2})^{s^\star - 2} \cdot \left( (|\rho|\gamma)^2 + d^{-1} + 2 \cdot |\rho|\gamma \cdot d^{-1/2} \right) \\
&= (|\rho|\gamma + d^{-1/2})^{s^\star - 2} \cdot |\rho|\gamma \cdot (|\rho|\gamma + d^{-1} \cdot (|\rho|\gamma)^{-1} + 2d^{-1/2}) \\
&\ll (|\rho|\gamma + d^{-1/2})^{s^\star - 2} \cdot |\rho|\gamma.
\end{aligned}
$$

Therefore, $\mathbb{E}_{\mathbb{P}_{\theta^\star}}[\langle g, \theta^\star \rangle]$ is always the major term no matter whether $s^\star$ is even or odd, and we have that

$$
\mathbb{E}_{\mathbb{P}_{\theta^\star}}[\langle g, \theta^\star \rangle] \simeq \begin{cases} \gamma\rho \cdot (\gamma|\rho| + d^{-1/2})^{s^\star - 2} & \text{if } s^\star \text{ is even}; \\ (\gamma|\rho| + d^{-1/2})^{s^\star - 1} & \text{if } s^\star \text{ is odd}. \end{cases}
$$

We now turn to the norm of $\mathbb{E}_{\mathbb{P}_{\theta^\star}}[g]$. We have already shown the projection of $\mathbb{E}_{\mathbb{P}_{\theta^\star}}[g]$ onto $\theta^\star$. Next, define $P_{\theta^\star}^\perp = I - \theta^\star \theta^{\star\top}$ as the projection matrix onto the orthogonal complement of the space spanned by $\theta^\star$. Now, it follows from Eq. (H.6) that

$$\|P_{\theta^\star}^\perp \mathbb{E}_{\mathbb{P}_{\theta^\star}}[g]\|_2 \le \sum_{s \ge s^\star} |\mathbb{E}_{\mathbb{Q}}[\zeta_s(y) \cdot \widehat{\psi}_{s+1}(y)]| \cdot \left\| \frac{\sqrt{s+1}}{L} \sum_{l=1}^L \langle w_l, \theta^\star \rangle^s \cdot w_l \right\|_2$$

$$\lesssim \sum_{s \ge s^\star} \frac{\sqrt{s+1}}{L} \sum_{l=1}^L |\langle w_l, \theta^\star \rangle|^s.$$

Here, the first inequality holds by noting that the second term in Eq. (H.6) lies exactly along the direction of $\theta^\star$ and thus does not contribute to the norm of $P_{\theta^\star}^\perp \mathbb{E}_{\mathbb{P}_{\theta^\star}}[g]$, while for the first term in Eq. (H.6), we use the triangle inequality and the fact that $\|P_{\theta^\star}^\perp v\|_2 \le \|v\|_2$ for any $v \in \mathbb{R}^d$. In the second inequality, we also use the triangle inequality and the fact that $\|w_l\|_2 = 1$ for all $l \in [L]$. Here, the "$\lesssim$" hides a constant that depends on the boundedness of $\mathbb{E}_{\mathbb{Q}}[\zeta_s(y) \cdot \widehat{\psi}_{s+1}(y)]$ as we have shown in the previous analysis.

Note that the term $\sum_{s \ge s^\star} \frac{\sqrt{s+1}}{L} \sum_{l=1}^L |\langle w_l, \theta^\star \rangle|^s$ is already handled in Eq. (E.8) and (E.9) under the success of event $\mathcal{E}(\epsilon) \cap \mathcal{E}_{1,1}$, on which we have

$$\left| \|\mathbb{E}_{\mathbb{P}_{\theta^\star}}[g]\|_2 - |\langle \mathbb{E}_{\mathbb{P}_{\theta^\star}}[g], \theta^\star \rangle| \right| \lesssim \sum_{s \ge s^\star} \frac{\sqrt{s+1}}{L} \sum_{l=1}^L |\langle w_l, \theta^\star \rangle|^s \lesssim (|\rho|\gamma + d^{-1/2})^{s^\star}.$$

Setting $\mathcal{E}_1 = \mathcal{E}_{1,1} \cap \mathcal{E}_{1,2}$ gives the desired event.

$\square$

*Proof of Proposition E.4.* The polynomial-tail property allows us to control the fluctuation of the gradient estimator $g$ in each direction at the level that is determined by the sample size $n$ and the corresponding variance. To this end, we begin with calculating the variance of $g$ along each direction.

**Calculating the second moment.** Given $\theta$ and $\theta^\star$, recall that we consider the following $d$ orthonormal directions:

$$\theta^\star, v_2, v_3, \ldots, v_d,$$

where we set $v_2 = (\theta - \langle \theta, \theta^\star \rangle \theta^\star)/\|\theta - \langle \theta, \theta^\star \rangle \theta^\star\|_2$ and $v_i$ for $i \ge 3$ are orthogonal to $\theta^\star$ and $v_2$. Our goal is to show that $g$ has small variance on each of these directions.

As each sample $(z_i, y_i)$ is independently drawn from $\mathbb{P}_{\theta^\star}$, we just need to consider the variance of

$$g_1 = \frac{1}{L} \sum_{l=1}^L \left( \psi(y_1, \langle w_l, z_1 \rangle) \cdot z_1 - \widehat{\psi}_1(y_1) \cdot w_l \right)$$

in the direction of $v$ for $v \in \{\theta^\star, v_1, \ldots, v_{d-1}\}$ as

$$\mathrm{Var}_{\mathbb{P}_{\theta^\star}}[\langle g_1, v \rangle] = \mathbb{E}_{\mathbb{P}_{\theta^\star}}[\langle g_1, v \rangle^2] - \mathbb{E}_{\mathbb{P}_{\theta^\star}}[\langle g_1, v \rangle]^2$$

$$\le \mathbb{E}_{\mathbb{P}_{\theta^\star}}[\langle g_1, v \rangle^2].$$

From this we see that it suffices to bound the second moment of $\langle g_1, v \rangle$, which is given by

$$\mathbb{E}_{\mathbb{P}_{\theta^\star}}[\langle g_1, v \rangle^2] \lesssim \frac{1}{L^2} \sum_{l,l'=1}^L \mathbb{E}_{\mathbb{P}_{\theta^\star}} \left[ \psi(y, \langle w_l, z \rangle) \psi(y, \langle w_{l'}, z \rangle) \langle z, v \rangle^2 \right] + \frac{1}{L^2} \sum_{l,l'=1}^L \mathbb{E}_{\mathbb{P}_{\theta^\star}} \left[ \widehat{\psi}_1(y)^2 \langle w_l, v \rangle \langle w_{l'}, v \rangle \right]$$

$$= \frac{1}{L^2} \sum_{l \ne l'} \mathbb{E}_{\mathbb{Q}} \left[ \psi(y, \langle w_l, z \rangle) \psi(y, \langle w_{l'}, z \rangle) \langle z, v \rangle^2 \cdot \left( 1 + \sum_{s \ge s^\star} \zeta_s(y) h_s(\langle \theta^\star, z \rangle) \right) \right]$$

$$+ \frac{1}{L^2} \sum_{l=1}^L \mathbb{E}_{\mathbb{Q}} \left[ \psi(y, \langle w_l, z \rangle) \psi(y, \langle w_l, z \rangle) \langle z, v \rangle^2 \right] + \frac{1}{L^2} \sum_{l \ne l'} \mathbb{E}_{\mathbb{Q}}[\widehat{\psi}_1(y)^2] \langle w_l, v \rangle \langle w_{l'}, v \rangle$$

$$+ \frac{1}{L^2} \sum_{l=1}^L \mathbb{E}_{\mathbb{Q}}[\widehat{\psi}_1(y)^2] \langle w_l, v \rangle^2.$$

As $\psi(y,z)z$ is quadruple-integrable by Assumption 4.1, the above integral is well-defined. We split the summation into two parts: $l = l'$ and $l \neq l'$. For $l = l'$, we directly have an $O(L^{-1})$ bound for each direction $\theta^\star, v_2, \ldots, v_d$ thanks to the polynomial-like tail property of $\psi$ in Assumption 4.1.

For $l \neq l'$, Lemma E.1 implies that we have on the nice event $\widetilde{\mathcal{E}}(\widetilde{\epsilon})$ that

$$|\langle w_l, w_{l'} \rangle| \leq 4(\gamma^2 + 2\gamma\epsilon + \widetilde{\epsilon})$$
$$\leq 8(\gamma^2 + \widetilde{\epsilon}) := \epsilon_2.$$

Invoking Lemma H.3, for any $v \in \{\theta^\star, v_2, \ldots, v_d\}$, it holds on $\widetilde{\mathcal{E}}(\widetilde{\epsilon})$ that

$$
\mathbb{E}_\mathbb{Q}\left[\psi(y, \langle w_l, z \rangle)\psi(y, \langle w_{l'}, z \rangle)\langle z, v \rangle^2 \cdot \left(1 + \sum_{s \geq s^\star} \zeta_s(y) h_s(\langle \theta^\star, z \rangle)\right)\right]
$$
$$
\lesssim \epsilon_2^{s^\star - 1} \cdot \left(1 + \frac{\epsilon_1^2}{\epsilon_2} + \left(\frac{\epsilon_1^2}{\epsilon_2}\right)^{s^\star - 1} \cdot \epsilon + \mathbb{1}(v \perp \theta^\star) \cdot \left(\frac{\epsilon_1^2}{\epsilon_2}\right)^{s^\star - 2} \cdot \frac{\epsilon_0^2}{\epsilon_2} \cdot (\epsilon_1^2 + \epsilon_1 \cdot \mathbb{1}(s^\star \geq 4))\right)
$$
(E.10)

where we also define $\epsilon_1 := \max\{|\langle w_l, \theta^\star \rangle|, |\langle w_{l'}, \theta^\star \rangle|\}$, $\epsilon_0 := \max\{|\langle w_l, v \rangle|, |\langle w_{l'}, v \rangle|\}$. If the nice event $\mathcal{E}(\epsilon)$ also holds, on which the following holds for all $l \in [L]$:

$$|\langle \xi_l, \theta^\star \rangle| < \epsilon, \quad \max_{2 \leq i \leq d} |\langle \xi_l, v_i \rangle| < \epsilon,$$

then we have by Lemma E.1 that

$$|\langle w_l, \theta^\star \rangle| \lesssim \gamma|\rho| + \epsilon, \quad |\langle w_l, v_2 \rangle| \lesssim \sqrt{1 - \rho^2}\gamma + \epsilon, \quad |\langle w_l, v_i \rangle| \lesssim \epsilon, \quad \forall i \geq 3, \quad \forall l \in [L].$$

Consequently, we can set $\epsilon_1 \simeq \gamma|\rho| + \epsilon = o(1)$ and

$$
\epsilon_0 \simeq \begin{cases} \gamma|\rho| + \epsilon, & \text{if } v = \theta^\star, \\ \gamma\sqrt{1 - \rho^2} + \epsilon, & \text{if } v = v_2, \\ \epsilon, & \text{otherwise.} \end{cases}
$$

Therefore, we have the ratio

$$
\frac{\epsilon_1^2}{\epsilon_2} \simeq \frac{(\gamma|\rho| + \epsilon)^2}{4(\gamma^2 + \widetilde{\epsilon})} \simeq \frac{\gamma^2|\rho|^2 + \epsilon^2}{\gamma^2 + \widetilde{\epsilon}}, \quad \frac{\epsilon_0^2}{\epsilon_2} \simeq \begin{cases} \frac{(\gamma|\rho| + \epsilon)^2}{8(\gamma^2 + \widetilde{\epsilon})} \simeq \frac{\gamma^2|\rho|^2 + \epsilon^2}{\gamma^2 + \widetilde{\epsilon}}, & \text{if } v = \theta^\star, \\ \frac{(\gamma\sqrt{1 - \rho^2} + \epsilon)^2}{8(\gamma^2 + \widetilde{\epsilon})} \simeq \frac{\gamma^2(1 - \rho^2) + \epsilon^2}{\gamma^2 + \widetilde{\epsilon}}, & \text{if } v = v_2, \\ \frac{\epsilon^2}{4(\gamma^2 + \widetilde{\epsilon})} \simeq \frac{\epsilon^2}{\gamma^2 + \widetilde{\epsilon}}, & \text{otherwise.} \end{cases}
$$

Since $\epsilon^2 \leq \widetilde{\epsilon}$, we can conclude that $\epsilon_1^2/\epsilon_2 \lesssim 1$ and $\epsilon_0^2/\epsilon_2 \lesssim 1$. Hence, the right-hand side of Eq. (E.10) is bounded by $\epsilon_2^{s^\star - 1} \simeq (\gamma^2 + \widetilde{\epsilon})^{s^\star - 1}$ for all $v \in \{\theta^\star, v_2, \ldots, v_d\}$.

Similarly, let us consider the term $L^{-2} \cdot \sum_{l \neq l'} \mathbb{E}_\mathbb{Q}[\widehat{\psi}_1(y)^2]\langle w_l, v \rangle\langle w_{l'}, v \rangle$. On the good event $\mathcal{E}(\epsilon)$, we have

$$\frac{1}{L^2} \cdot \sum_{l \neq l'} \mathbb{E}_\mathbb{Q}[\widehat{\psi}_1(y)^2]\langle w_l, v \rangle\langle w_{l'}, v \rangle \lesssim \epsilon_0^2 \cdot \mathbb{1}\{s^\star \leq 2\}$$

$$\lesssim \epsilon_2 \mathbb{1}(s^\star \leq 2)$$
$$\lesssim (\gamma^2 + \widetilde{\epsilon})^{s^\star - 1} \cdot \mathbb{1}\{s^\star \leq 2\}.$$

The first inequality holds because $\widehat{\psi}_1(y) = 0$ whenever $s^\star \geq 2$ because of Assumption 4.1(b) and the second inequality holds due to the condition that $\epsilon^2 \leq \widetilde{\epsilon}$

Lastly for all the terms that take a single summation over $l \in [L]$, we have them bounded by $1/L$ as each term in the summation can be upper bounded by 1. Combining the results for $l = l'$ and $l \neq l'$, we have on the event $\mathcal{E}(\epsilon) \cap \widetilde{\mathcal{E}}(\widetilde{\epsilon})$ that

$$\text{Var}_{\mathbb{P}_{\theta^\star}}[\langle g_1, v \rangle] \leq \mathbb{E}_{\mathbb{P}_{\theta^\star}}[\langle g_1, v \rangle^2] \lesssim (\gamma^2 + \widetilde{\epsilon})^{s^\star - 1} + \frac{1}{L}, \quad \forall v \in \{\theta^\star, v_2, \ldots, v_d\}.$$

**Concentration.** The first thing is to control the variation of $g$ in the direction of $\theta^\star$, where we need to upper bound the $L^r(\mathbb{P}_{\theta^\star})$-norm of $\langle g_1, v \rangle$. To this end, we define $G : (\mathbb{R} \times \mathbb{R}) \times \mathbb{R} \to \mathbb{R}$ as

$$G(z, y, w) = |\psi(y, \langle w, z, \rangle) \cdot \langle z, v \rangle| + |\widehat{\psi}_1(y) \cdot \langle w, v \rangle|.$$

Also we define the empirical measure $\mathrm{d}\mu(w) = L^{-1}\sum_l \delta(w_l)$, then it holds by integral Minkowski's inequality that

$$\mathbb{E}_{\mathbb{P}_{\theta^\star}}\left[\langle g_1, v\rangle^r\right]^{1/r} \leq \left(\int \mathrm{d}\mathbb{P}_{\theta^\star}(y, z)\left(\int \mathrm{d}\mu(w) \cdot G(z, y, w)\right)^r\right)^{1/r}$$

$$\leq \int \mathrm{d}\mu(w)\left(\int \mathrm{d}\mathbb{P}_{\theta^\star}(y, z) \cdot G(z, y, w)^r\right)^{1/r}$$

$$\lesssim \frac{1}{L}\sum_l \mathbb{E}_{\mathbb{P}_{\theta^\star}}\left[|\psi(y, \langle w_l, z\rangle) \cdot \langle z, v\rangle|^r\right]^{1/r} + \frac{1}{L}\sum_l |\langle w_l, v\rangle|. \qquad \text{(E.11)}$$

For the second term, we have on $\mathcal{E}(\epsilon)$ that $|\langle w_l, v\rangle| \leq 2\gamma + 2\epsilon \leq 1$. Applying the Cauchy-Schwarz inequality, we have for the summand of the first term in Eq. (E.11) that

$$\mathbb{E}_{\mathbb{P}_{\theta^\star}}\left[|\psi(y, \langle w_l, z\rangle) \cdot \langle w_l, z\rangle|^r\right]^{1/r} \leq \mathbb{E}_{\mathbb{P}_{\theta^\star}}\left[|\psi(y, \langle w_l, z\rangle)|^{2r}\right]^{1/2r} \cdot \mathbb{E}_{\mathbb{P}_{\theta^\star}}\left[|\langle w_l, z\rangle|^{2r}\right]^{1/2r}.$$

Note that $\mathbb{E}_{\mathbb{P}_{\theta^\star}}\left[|\langle w_l, z\rangle|^{2r}\right]^{1/2r} \leq (2r-1)!!^{1/(2r)} \lesssim r^{1/2}$, it suffices to deal with $\mathbb{E}_{\mathbb{P}_{\theta^\star}}\left[|\langle w_l, z\rangle|^{2r}\right]^{1/2r}$. To proceed, we can decompose $\langle w_l, z\rangle$ to components that are correlated and independent with $y$ as

$$\langle w_l, z\rangle = \langle w_l - \langle w_l, \theta^\star\rangle\theta^\star, z\rangle + \langle w_l, \theta^\star\rangle\langle\theta^\star, z\rangle$$

$$= \sqrt{1 - \langle w_l, \theta^\star\rangle^2} \cdot x' + \langle w_l, \theta^\star\rangle x'$$

where $x = \langle \theta^\star, z\rangle \sim \mathcal{N}(0, 1)$ is independent to $x' = (1 - \langle w_l, \theta^\star\rangle^2)^{-1/2} \cdot \langle w_l - \langle w_l, \theta^\star\rangle\theta^\star, z\rangle \sim \mathcal{N}(0, 1)$. Therefore, we define a Gaussian noise operator as $\mathrm{U}_\rho\psi(y, x) = \mathbb{E}_{x'\sim\mathcal{N}(0,1)}[\psi(y, \rho x + \sqrt{1 - \rho^2}x')]$. And it holds that

$$\mathbb{E}_{\mathbb{P}_{\theta^\star}}\left[\psi(y, \langle w_l, z\rangle)^{2r}\right] = \mathbb{E}_{\mathbb{P}}\left[\mathrm{U}_{\langle\theta^\star, w_l\rangle}\psi(y, x)^{2r}\right]$$

$$= \mathbb{E}_{\mathbb{Q}}\left[\mathrm{U}_{\langle\theta^\star, w_l\rangle}\psi(y, x)^{2r} \cdot \frac{\mathbb{P}(x, y)}{\mathbb{Q}(x, y)}\right]$$

$$= \mathbb{E}_{\mathbb{Q}}\left[\psi(y, x)^{2r} \cdot \mathrm{U}_{\langle\theta^\star, w_l\rangle}\left(\frac{\mathbb{P}(x, y)}{\mathbb{Q}(x, y)}\right)\right]$$

$$\leq \left(\mathbb{E}_{\mathbb{Q}}[\psi(y, x)^{4r}] \cdot \mathbb{E}_{\mathbb{Q}}\left[\left(\mathrm{U}_{\langle\theta^\star, w_l\rangle}\left(\frac{\mathbb{P}(x, y)}{\mathbb{Q}(x, y)}\right)\right)^2\right]\right)^{1/2}, \qquad \text{(E.12)}$$

where the second line follows from the property of the Gaussian noise operator in (B.2). By assumption of the tail bound in Assumption 4.1, we have that $\mathbb{E}_{\mathbb{Q}}[\psi(y, x)^{4r}] \leq C_p(4r)^{4C_p r}$. For the second term, we have by the Parseval's identity that

$$\mathbb{E}_{\mathbb{Q}}\left[\left(\mathrm{U}_{\langle\theta^\star, w_l\rangle}\left(\frac{\mathbb{P}(x, y)}{\mathbb{Q}(x, y)}\right)\right)^2\right] = 1 + \sum_{s \geq s^\star}\langle\theta^\star, w_l\rangle^2 \cdot \mathbb{E}_{\mathbb{Q}}\left[\zeta_s(y)^2\right] \leq 2.$$

where we use the property that on the good event $\mathcal{E}(\epsilon)$ we have $|\langle w_l, \theta^\star\rangle| \leq \gamma|\rho| + \epsilon \leq \gamma + \epsilon < 1/2$ and also $\mathbb{E}_{\mathbb{Q}}[\zeta_s(y)^2] \leq 1$. And in conclusion, we get for the first term in Eq. (E.11) that

$$\frac{1}{L}\sum_l \mathbb{E}_{\mathbb{P}_{\theta^\star}}\left[|\psi(y, \langle w_l, z\rangle) \cdot \langle w_l, z\rangle|^r\right]^{1/r} \lesssim r^{C_p + 1/2}.$$

Combining everything, we have

$$\mathbb{E}_{\mathbb{P}_{\theta^\star}}\left[|\langle g_1, v\rangle|^r\right]^{1/r} \lesssim r^{C_p + 1/2}, \quad \forall v \in \{\theta^\star, v_2, \ldots, v_d\}.$$

Thus, by Lemma J.3, there exists a $\{(z_i, y_i)\}_{i\in[n]}$-measurable event $\mathcal{E}_{2,1}$ with $\Pr(\mathcal{E}_{2,1}) \geq 1 - d^{-c}/(MT)$ and it holds on $\mathcal{E}_2$ that

$$|\langle g, \theta^\star\rangle - \mathbb{E}_{\mathbb{P}_{\theta^\star}}[\langle g, \theta^\star\rangle]|$$

$$\lesssim \sqrt{\frac{\mathbb{E}_{\mathbb{P}_{\theta^\star}}[\langle g_1, \theta^\star\rangle^2] \cdot \log(d^c MT)}{n}} + \frac{\log(d^c MT) \cdot \log(d^c MTn)^{C_p + 1/2}}{n}$$

$$\lesssim \sqrt{\frac{((\gamma^2 + \widetilde{\epsilon})^{s^\star - 1} + L^{-1}) \cdot \log(d)}{n}} + \frac{\log(d)^{C_p + 3/2}}{n}, \quad \forall v \in \{\theta^\star, v_1, \ldots, v_{d-1}\}. \quad \text{(E.13)}$$

where we utilize the fact $T, M, n$ all have polynomial dependency on $d$. Moreover, since we assume that

$$n = \Omega\Big(\big((\gamma^2 + \widetilde{\epsilon})^{s^\star - 1} + L^{-1}\big)^{-1} \cdot \log(d)^{2C_p + 1}\Big),$$

we have that the first term in Eq. (E.13) dominates, which further implies that

$$|\langle g, \theta^\star \rangle - \mathbb{E}_{\mathbb{P}_{\theta^\star}}[\langle g, \theta^\star \rangle]| \lesssim \sqrt{\frac{((\gamma^2 + \widetilde{\epsilon})^{s^\star - 1} + L^{-1}) \cdot \log(d)}{n}}.$$

Meanwhile, for the $\ell_2$-norm of $g$, we have by the Jensen's inequality that for any $r \geq 1$,

$$\mathbb{E}_{\mathbb{P}_{\theta^\star}}[\|g_1\|_2^r]^{1/r} = \left(\mathbb{E}_{\mathbb{P}_{\theta^\star}}\left[\sum_{v \in \{\theta^\star, v_2, \ldots, v_d\}} \langle g_1, v \rangle^2\right]^{r/2}\right)^{1/r}$$

$$\leq \sqrt{d} \cdot \left(\frac{1}{d} \cdot \sum_{v \in \{\theta^\star, v_2, \ldots, v_d\}} \mathbb{E}_{\mathbb{P}_{\theta^\star}}[|\langle g_1, v \rangle|^r]\right)^{1/r} \lesssim \sqrt{d} \cdot (r)^{C_p + 1/2}.$$

This polynomial tail bound enables us to apply Lemma J.3 for the $\ell_2$-norm of $g$, which implies that there exists some event $\mathcal{E}_{2,2}$ with $\Pr(\mathcal{E}_{2,2}) \geq 1 - d^{-c}/(MT)$, and it holds on $\mathcal{E}_{2,2}$ that

$$\big\|g - \mathbb{E}_{\mathbb{P}_{\theta^\star}}[g]\big\|_2 \lesssim \sqrt{\frac{((\gamma^2 + \widetilde{\epsilon})^{s^\star - 1} + L^{-1}) \cdot d \cdot \log(d)}{n}}.$$

Setting $\mathcal{E}_2 = \mathcal{E}_{2,1} \cap \mathcal{E}_{2,2}$ gives the desired event. This concludes the proof of Proposition E.4. $\qquad\square$

## F  PROOF OF THE MAIN THEOREM FOR THE SPARSE PRIOR

### F.1  PROOF OUTLINE AND PRELIMINARIES

In this section, we provide a detailed proof for Theorem 5.1. We begin with some good events that we will work with.

**Signal concentration.** We begin with a good event on which the signal spreads almost evenly within its support. Define a series of event:

$$\mathcal{E}_{0,r} := \{\|\theta\|_r^r \leq C_r \cdot k^{1 - r/2}\};$$
$$\mathcal{E}_{0,\infty} := \{\|\theta^\star\|_\infty \leq C_\infty \cdot k^{-1/2} \log(k)^{1/2}\};$$
$$\mathcal{E}_{0,\sharp} := \Big\{ \sum_{j \in [d]} \mathbb{1}\big\{|\theta_j^\star| \geq \frac{1}{\sqrt{2k}}\big\} \geq \frac{k}{4}\Big\}.$$

The following lemma guarantees that the all the events above hold with high probability.

**Lemma F.1** (Good signal). *Suppose that $k$ is sufficiently large such that $k/\log(k) \geq 32c(r \vee 1)$, $k/\log^{r+2} k \geq \sqrt{2c + 2}$, then it holds that*

$$\Pr(\mathcal{E}_{0,r}) \wedge \Pr(\mathcal{E}_{0,\infty}) \geq 1 - O\big(k^{-c_{0,1}}\big);$$
$$\Pr(\mathcal{E}_{0,\sharp}) \geq 1 - O\big(\exp\{-c_{0,2}k\}\big),$$

*for some constants $c_{0,1}, c_{0,2} > 0$.*

*Proof of Lemma F.1.* See Appendix F.5. $\qquad\square$

For fixed $s^\star$, we collect all the indices $r$ such that the corresponding nice event $\mathcal{E}_{0,r}$ will be involved in the coming analysis. Define $S(s^\star) = \{s^\star - 1, s^\star - \mathbb{1}\{s^\star \text{ odd}\}, 2s^\star, 4s^\star\}$. And we will stick to the following high probability event

$$\mathcal{E}_0 := \mathcal{E}_{0,\infty} \cap \mathcal{E}_{0,\sharp} \cap (\cap_{r \in S(s^\star)} \mathcal{E}_{0,r}).$$

With Lemma F.1, we have that $\Pr(\mathcal{E}_0) \geq 1 - O(k^{-c_0})$ for some constant $c_0 > 0$.

**Preparation for characterizing one-step gradient.** Following the same manner as the proof for the non-sparse case, we first characterize the alignment of the gradient step (without adversarial error term $\mathrm{err}_{m,l,i}^{(t)}$). We begin with the definition of a minimal setup, that collects all the essential elements to form the one-step gradient. The following definition is the sparse analogue of Definition E.2. Apart from the method of generating the noise, in the sparse case, we will analyze the gradient in a coordinate-wise manner to adapt to the sparse structure.

**Definition F.2.** *Fix $k$-sparse vectors $\theta, \theta^\star \subset \mathbb{S}^{d-1}$ with $\phi = \mathsf{supp}(\theta)$ and $\phi^\star = \mathsf{supp}(\theta^\star)$. Let $\rho = \langle \theta, \theta^\star \rangle$. Suppose that a single batch of data $\{(z_i, y_i)\}_{i \in [n]}$ is i.i.d. generated from $\mathbb{P}_{\theta^\star}$. We fix the index $m$ as the current neuron. We first sample $\phi_{m,1}, \phi_{m,2}, \ldots \phi_{m,L} \overset{\text{i.i.d.}}{\sim} \mathsf{Unif}(\mathcal{S}_{k,m})$, i.e., uniform distribution over all $k$-sparse supports with $m$-th index always included. Given these random supports, we sample independent noises $\xi_{m,l} \sim \mathsf{Unif}(\mathbb{S}^{k-1}(\phi_{m,l}))$ for $l \in [L]$. Now, for each $l \in [L]$ and $\gamma = o(1)$, we define $w_{m,l} = (\gamma\theta + \xi_{m,l})/\|\gamma\theta + \xi_{m,l}\|_2$. Then our target is*

$$\bar{g}_m = \frac{1}{nL} \sum_{i=1}^{n} \sum_{l=1}^{L} \big( \psi(y_i, \langle w_{m,l}, z_i \rangle) \cdot z_i - \widehat{\psi}_1(y_i) \cdot w_{m,l} \big). \tag{F.1}$$

*The associated good event is defined as*

$$\mathcal{E}_m(\epsilon) = \{\sup_{l,j} |\langle \xi_{m,l}, e_j \rangle| \le \epsilon\}; \tag{F.2}$$

$$\widetilde{\mathcal{E}}_m = \Big\{ \max \big\{ \sup_{l \ne l'} |\phi_{m,l} \cap \phi_{m,l'}|, \sup_{l} |\phi_{m,l} \cap \phi^\star|, \sup_{l} |\phi_{m,l} \cap \phi| \big\} \le \log k \Big\}. \tag{F.3}$$

**Almost orthogonality.** Recall that our perturbation noise $\xi_{m,l}$ is sampled from $\mathsf{Unif}(\mathbb{S}^{k-1}(\phi_{m,l}))$, which is approximately isotropic. One can presume that each $\xi_{m,l}$ is evenly distributed among different coordinates. Additionally, we expect that $(\xi_{m,l}, \phi_{m,l})$ and $(\xi_{m,l'}, \phi_{m,l'})$ should have a negligible overlap. These two qualitative properties, which can help simplify the analysis, are captured by Eq. (F.2) and Eq. (F.3) in Definition F.2. The following lemma characterizes the property of the perturbated weights $w_{m,l}$ on the nice event $\mathcal{E}_m(\epsilon) \cap \widetilde{\mathcal{E}}$.

**Lemma F.3** (Polarized weight on nice event, sparse case)**.** *Consider the setting in Definition F.2 with $\gamma < 1/2$. Suppose that the nice event $\mathcal{E}_m(\epsilon) \cap \widetilde{\mathcal{E}}_m$ holds and $\|\theta^\star\|_\infty \le 1/\log k$, then we have that*

$$\sup_{l,j} |\langle w_{m,l}, e_j \rangle| \le 2(\gamma|\theta_j| + \epsilon);$$

$$\sup_{l} |\langle w_{m,l}, \theta^\star \rangle| \le 2(\gamma|\rho| + \epsilon).$$

*Additionally, we have that*

$$\sup_{l \ne l'} |\langle w_{m,l}, w_{m,l'} \rangle| \le 4(\gamma^2 + \epsilon^2 \log k).$$

*Proof of Lemma F.3.* See Appendix F.5. □

This lemma, serving as the counterpart of Lemma E.1, controls the behavior of the perturbated weight $w_{m,l}$ in its coordinates and alignment with $\theta^\star$. Additionally, they are approximately orthogonal with each other, which allows for good characterization to the second moment of the gradient.

One may notice that the definition of $\widetilde{\mathcal{E}}$ differs from the non-sparse case, where we explicitly bound the correlation between different $\xi_{m,l}$. This is the benefit of the sparse structure, as two randomly sampled $k$-sparse supports are naturally of low overlap.

Before delving into the component-wise analysis, we begin with a proposition that will be frequently used to calculate the average contribution of each term in the gradient. This proposition serves as the counterpart of Lemma H.4 in the sparse case.

**Proposition F.4.** *Suppose that the polarization level $\gamma = o(1)$ and the noise $(\xi_{m,l}, \phi_{m,l}) \sim \mathsf{Unif}(\mathbb{S}^{k-1}(\phi_{m,l})) \otimes \mathsf{Unif}(\mathcal{S}_{k,m})$. Let $\rho = \langle \theta, \theta^\star \rangle$ where $\theta$ is the polarized direction in $w_{m,l}$. Assume that the event $\mathcal{E}_{0,s-\mathbb{1}\{s \text{ odd}\}}$ holds. Then we have that*

$$\mathbb{E}_{w_{m,l}}[\langle w_{m,l}, \theta^\star \rangle^s] \simeq \begin{cases} \gamma\rho \cdot \big(\gamma|\rho| + k^{-1/2}|\theta_m^\star| + k^{-1}\delta_{s-1}\big)^{s-1} & \text{if } s \text{ odd}; \\ (\gamma|\rho| + k^{-1/2}|\theta_m^\star| + k^{-1}\delta_s)^s & \text{if } s \text{ even}, \end{cases}$$

*where $\delta_s = (k^2/d)^{1/s} = o(1)$ for any $s = O(1)$.*

*Proof of Proposition F.4.* See Appendix F.5. □

Recall that we define the good event over the signal as $\mathcal{E}_0 = \mathcal{E}_{0,\infty} \cap \mathcal{E}_{0,\sharp} \cap \bigcap_{r \in S(s^\star)} \mathcal{E}_{0,r}$, where $S(s^\star) = \{s^\star - 1, s^\star - \mathbb{1}\{s^\star \text{ odd}\}, 2s^\star, 4s^\star\}$. This definition facilitates our defered analysis where we need to control multiple moments of different orders and we collect all the necessary good events in $\mathcal{E}_0$ in the first place.

## F.2 Properties of the Gradient Step

In this section, we preview some properties regarding the gradient defined in Eq. (F.1). The following proposition deals with the first moment.

**Proposition F.5** (First-order moment of the gradient). *Suppose that $\theta^\star$ is fixed such that the nice event $\mathcal{E}_0$ holds. Under Definition F.2, we choose $\gamma \le \epsilon = o(1)$ such that $\Pr\left(\mathcal{E}_m(\epsilon)^c\right) \le O(k^{-s^\star})$ and*

$$L = \Omega\Big( \log(d) \cdot \big(k \vee (\epsilon^{s^\star - 1} \cdot k^{s^\star + 1})\big)\Big).$$

*Then there exists a $\{\xi_{m,l}\}_{l \in [L]}$-measurable event $\mathcal{E}_{m,1}$ with $\Pr(\mathcal{E}_{m,1}) \ge 1 - O(k^{-c_1})$ for some constant $c_1 > 0$, such that on $\mathcal{E}_{m,1} \cap \mathcal{E}_m(\epsilon) \cap \widetilde{\mathcal{E}}_m$, it holds for any $j \in [d]$ that*

$$\langle \mathbb{E}_{\mathbb{P}_{\theta^\star}}[\bar{g}_m], e_j \rangle = \mathbb{E}_{\mathbb{Q}}[\zeta_{s^\star}(y) \cdot \widehat{\psi}_{s^\star - 1}(y)] \cdot \sqrt{s^\star} \cdot \mathbb{E}_{\widetilde{w}_{m,l}}[\langle \widetilde{w}_{m,l}, \theta^\star \rangle^{s^\star - 1}]\theta_j^\star + R_{m,j},$$

*where the expectation*

$$\mathbb{E}_{\widetilde{w}_{m,l}}[\langle \widetilde{w}_{m,l}, \theta^\star \rangle^{s^\star - 1}] \simeq \begin{cases} \gamma\rho \cdot \big(\gamma|\rho| + k^{-1/2}|\theta_m^\star| + k^{-1}\delta_{s^\star - 2}\big)^{s^\star - 2} \cdot \theta_j^\star & \text{if } s^\star \text{ is even;} \\ \big(\gamma|\rho| + k^{-1/2}|\theta_m^\star| + k^{-1}\delta_{s^\star - 1}\big)^{s^\star - 1} \cdot \theta_j^\star & \text{if } s^\star \text{ is odd,} \end{cases}$$

*and $R_{m,j}$ is the remainder that can be bounded by*

$$|R_{m,j}| \lesssim \Big( \big(k^{-1} \vee (\gamma|\rho| + k^{-1/2}|\theta_m^\star|)\big) \cdot \big(\gamma|\rho| + k^{-1/2}|\theta_m^\star| + k^{-1}\delta_+\big)^{s^\star - 1} + k^{-s^\star}\Big) \cdot |\theta_j^\star|$$
$$+ (\gamma|\rho| + k^{-1/2}|\theta_m^\star| + k^{-1}\delta_+)^{s^\star} \cdot (\gamma|\theta_j| + k^{-1/2} \cdot (k/d)^{\mathbb{1}\{j \ne m\}/2}) + k^{-(s^\star + 1)}.$$

*Proof of Proposition F.5.* See Appendix F.4. □

The statement of this proposition clarifies the leading term explicitly, which enables us to track the leading term in the strong alignment more precisely. To complete this section, we provide a proposition that characterizes the fluctuation of the gradient, serving as the counterpart of Proposition E.4.

**Proposition F.6** (Fluctuation of mini-batch gradient). *Under the simplified setting Definition F.2 where $\psi$ follows Assumption 4.1. Additionally, suppose that the sample size*

$$n = \Omega\Big( \big((\gamma^2 + \epsilon^2 \log k)^{s^\star - 1} + L^{-1}\big)^{-1} \cdot \log(d)^{2C_p + 2}\Big),$$

*where $C_p$ is the order of the polynomial tail in Assumption 4.1(c). Then there exists a $\{(z_i, y_i)\}_{i \in [n]}$-measurable event $\mathcal{E}_{m,2}$ with $\Pr(\mathcal{E}_{m,2}) \ge 1 - O(d^{-(c+1)}/T)$, such that on $\mathcal{E}_{m,2} \cap \mathcal{E}_m(\epsilon) \cap \widetilde{\mathcal{E}}_m$, it holds that*

$$\big|\langle \bar{g}_m, e_j \rangle - \langle \mathbb{E}_{\mathbb{P}_{\theta^\star}}[\bar{g}_m], e_j \rangle\big| \le \sqrt{\frac{\big((\gamma^2 + \epsilon^2 \log k)^{s^\star - 1} + L^{-1}\big) \cdot \log(d)}{n}},$$

*for any $v \in \{e_1, e_2, \dots, e_d\}$*

*Proof of Proposition F.6.* See Appendix F.4. □

With these propositions, we have completed the preparation for the analysis of the gradient step and are ready to move on to the proof of the main theorem.

## F.3 Proof of the Main Theorem

*Proof of Theorem 5.1.*

**Preparations.** We first clarify the final good event that we will use throughout the proof. We first fix $\epsilon = k^{-1/2} \cdot \log k$, then it holds by Lemma J.6 that for each $m$ and $t$, we have

$$\Pr\left(\mathcal{E}_m^{(t)}(\epsilon)^c\right) \leq Ld \cdot O(\exp\{-ck\} + k^{-\log k/4}).$$

Since $L$ and $d$ are at most polynomials in $k$, we see that for sufficiently large $k$, it holds that $\Pr\left(\mathcal{E}_m^{(t)}(\epsilon)^c\right) \leq k^{-s^\star}$. Additionally, we see that $\gamma = k^{-1/2}$ is fixed and our parameter configuration

$$n = \Omega\left((k\log^3 k)^{s^\star} \cdot \log d\right), \quad L = \Omega(k^{(s^\star+3)/2} \cdot \log(k)^{s^\star-1})$$

are clearly compatible with the conditions in Proposition F.5 and Proposition F.6. At the $t$-th step, the mini-batch $\{(z_i^{(t)}, y_i^{(t)})\}_{i\in[n]}$, $\theta = \theta_m^{(t)}$ and the error-free gradient

$$\bar{g}_m^{(t)} = \frac{1}{nL}\sum_{l=1}^{L}\sum_{i=1}^{n}\left(\psi(y_i^{(t)}, \langle w_{m,l}^{(t)}, z_i^{(t)}\rangle) \cdot z_i^{(t)} - \widehat{\psi}_1(y_i^{(t)}) \cdot w_{m,l}^{(t)}\right) \tag{F.4}$$

together form an instance of Definition F.2, for which we can find the event $\mathcal{E}_{m,1}^{(t)}$ and $\mathcal{E}_{m,2}^{(t)}$, both with probability at least $1 - O(d^{-c-1}/T)$, such that on $\mathcal{E}_{m,1}^{(t)} \cap \mathcal{E}_{m,2}^{(t)}$ the results in Proposition F.5 and Proposition F.6 hold.

The gradient we use in the algorithm differs from Eq. (F.4) by the error term $\mathrm{err}_{m,l,i}^{(t)} \cdot z_i^{(t)}$. To control this difference, we define

$$\mathcal{E}_{m,3}^{(t)} = \left\{\sup_i \|z_i^{(t)}\| \leq \sqrt{d}\right\}.$$

Given our specification on $\sup_{m,l,i,t}\mathrm{err}_{m,l,i}^{(t)}$, it holds on $\mathcal{E}_{m,3}^{(t)}$ that for any $v \in \mathbb{S}^{d-1}$:

$$\begin{aligned}
\left|\|g_m^{(t)}\|_2 + \|\bar{g}_m^{(t)}\|_2\right| \vee |\langle g_m^{(t)}, v\rangle - \langle \bar{g}_m^{(t)}, v\rangle| &\leq \|g_m^{(t)} - \bar{g}_m^{(t)}\|_2 \\
&\leq \sup_i \|z_i^{(t)}\| \cdot \sup_l \|\mathrm{err}_{m,l,i}^{(t)}\|_2 \\
&\leq d^{-9s^\star}. \tag{F.5}
\end{aligned}$$

Our final event is fixed to be

$$\mathcal{E} = \mathcal{E}_0 \cap \bigcap_{m\in[d]}\bigcap_{t=1}^{T}\left(\mathcal{E}_m^{(t)}(\epsilon) \cap \widetilde{\mathcal{E}}_m \cap \mathcal{E}_{m,1}^{(t)} \cap \mathcal{E}_{m,2}^{(t)} \cap \mathcal{E}_{m,3}^{(t)}\right).$$

With union bound, we have that $\Pr(\mathcal{E}) \geq 1 - O(d^{-c})$ for some constant $c > 0$.

To avoid confusion, we denote for each $m$ that $\breve{g}_m^{(t)} = \mathbb{E}_{\mathbb{P}_{\theta^\star}}[\bar{g}_m^{(t)}]$. Later we will encounter some data dependent index $\widehat{m}$ and using $\breve{g}_{\widehat{m}}^{(t)}$ avoids the ambiguity of the expectation. With our choice of $n$ and $L$, Proposition F.6 guarantees that for any $m, t, j$, it holds that

$$\left|\bar{g}_{m,j}^{(t)} - \breve{g}_{m,j}^{(t)}\right| \lesssim k^{-(s^\star-1/2)} \cdot \log^{-3/2} k. \tag{F.6}$$

In the sequel, we will drop the superscript $t$ whenever there is no ambiguity. We will frequently involve $\widetilde{g}_m^{(t)} = P_{\mathsf{Top}_k(g_m^{(t)})}(g_m^{(t)})$.

**Weak alignment.** The proof towards the weak alignment in the initial step comprises three parts. In the first place, we will show that the index of the gradient we choose $\widehat{m}$ guarantees that $|\theta_{\widehat{m}}^\star| \gtrsim k^{-1/2}$. Thereby, the corresponding gradient exhibits good alignment towards the signal. Based on this, we can show that the support we choose $\mathsf{Top}_k(g_{\widehat{m}})$ is of considerable quality by successfully identifying

$\phi^{\star\star} = \{j : |\theta_j^\star| \geq 1/\sqrt{2k}\}$. Combining these elements, we can show that the gradient $\widetilde{g}_{\widehat{m}}$ is well-aligned with the signal $\theta^\star$.

We begin with analyzing the quality of $g_{\widehat{m}}$ where $\widehat{m} = \mathrm{argmax}_m \|\widetilde{g}_m\|_2$. With this objective in mind, we first work on deriving a signal-dependent upper bound for $\|\widetilde{g}_{\widehat{m}}\|$. Note that $\rho_m = \langle \theta_m, \theta^\star \rangle = |\theta_m^\star|$, where $\theta_m = e_m$ is the initial weight. Applying Proposition F.5, we have for any $\phi, |\phi| = k$ that

$$
\begin{aligned}
\sum_{j \in \phi} |\breve{g}_{\widehat{m},j}|^2 &\lesssim (k^{-1/2}|\theta_{\widehat{m}}^\star|)^{2\,\mathbb{1}\{s^\star \text{ even}\}} \cdot \left(k^{-1/2}|\theta_{\widehat{m}}^\star| + k^{-1}\delta_+\right)^{2(s^\star - 1 - \mathbb{1}\{s \text{ even}\})} \cdot \sum_{j \in \phi} \theta_j^{\star 2} \\
&\quad + \left(k^{-2} \vee (k^{-1/2}|\theta_{\widehat{m}}^\star|)^2 \cdot (k^{-1/2}|\theta_{\widehat{m}}^\star| + k^{-1}\delta_+)^{2(s^\star - 1)} + k^{-2s^\star}\right) \sum_{j \in \phi} \theta_j^{\star 2} \\
&\quad + (k^{-1/2}|\theta_{\widehat{m}}^\star| + k^{-1}\delta_+)^{2s^\star} \cdot k^{-1} \sum_{j \in \phi} \left(\theta_j^2 + \cdot(k/d)^{\mathbb{1}\{j \neq m\}}\right) + k^{-(2s^\star + 1)}
\end{aligned}
$$

Note that we have $\|\theta^\star\|_\infty \lesssim k^{-1/2} \cdot \log k$, it holds that

$$
k^{-2} \vee (k^{-1} \cdot |\theta_{\widehat{m}}^\star|^2) \cdot (k^{-1/2}|\theta_{\widehat{m}}^\star| + k^{-1}\delta_+)^{2(s^\star - 1)} \lesssim k^{-2s^\star} \cdot \log(k)^{2s^\star}.
$$

For any $\phi$ such that $|\phi| = k$, we have $\sum_{j \in \phi} \theta_j^2 \leq 1$ for any $\theta$ such that $\|\theta\|_0 = k$. Therefore, we can further upper bound this quantity by

$$
\begin{aligned}
\sup_{|\phi|=k} \sum_{j \in \phi} |\breve{g}_{\widehat{m},j}|^2 &\lesssim (k^{-1/2}|\theta_{\widehat{m}}^\star| + k^{-1}\delta_+)^{2s^\star - 2} + k^{-2s^\star} \cdot \log(k)^{2s^\star} \\
&\quad + (k^{-1/2}|\theta_{\widehat{m}}^\star| + k^{-1}\delta_+)^{2s^\star} \cdot k^{-1} \\
&\lesssim (k^{-1/2}|\theta_{\widehat{m}}^\star| + k^{-1}\delta_+)^{2s^\star - 2} \\
&\quad + k^{-2s^\star} \cdot \log^{s^\star/2} k + k^{-2s^\star} \cdot \log(k)^{2s^\star}.
\end{aligned} \tag{F.7}
$$

Now we combined Eq. (F.7), Eq. (F.6) and Eq. (F.5) to conclude that

$$
\begin{aligned}
\sup_{|\phi|=k} \sum_{j \in \phi} |g_{\widehat{m},j}|^2 &\lesssim k \cdot \sup_j |g_{\widehat{m},j} - \bar{g}_{\widehat{m},j}|^2 + k \cdot \sup_j |\bar{g}_{\widehat{m},j} - \breve{g}_{\widehat{m},j}|^2 + \sup_{|\phi|=k} \sum_{j \in \phi} |\breve{g}_{\widehat{m},j}|^2 \\
&\lesssim k^{-(s^\star - 1)} \cdot (|\theta_{\widehat{m}}^\star| + k^{-1/2}\delta_+)^{2s^\star - 2} + k^{-2s^\star} \cdot \log(k)^{2s^\star}.
\end{aligned}
$$

By definition of $\widetilde{g}_m$, we can further conclude that

$$
\begin{aligned}
\|\widetilde{g}_{\widehat{m}}\|_2^2 &= \sup_{|\phi|=k} \sum_{j \in \phi} |\langle g_{\widehat{m}}, e_j \rangle|^2 \\
&\lesssim k^{-(s^\star - 1)} \cdot |\theta_{\widehat{m}}^\star|^{2(s^\star - 1)} + o\left(k^{-2(s^\star - 1)}\right).
\end{aligned} \tag{F.8}
$$

On the other hand, for $m \in \phi^{\star\star}$, we have $(\gamma|\rho| + k^{-1/2}|\theta_m^\star| + k^{-1}\delta_+)^r \gtrsim k^{-r}$. Then it holds by Proposition F.5 that

$$
\sum_{j \in \phi^\star} |\breve{g}_{m,j}|^2 \gtrsim k^{-2(s^\star - 1)} - \widetilde{O}(k^{-2s^\star}),
$$

and by Eq. (F.6) and (F.5), we have that

$$
\|\widetilde{g}_m\|_2^2 \gtrsim k^{-2(s^\star - 1)} - \widetilde{O}(k^{-2s^\star}). \tag{F.9}
$$

Now, combinig Eq. (F.9) with Eq. (F.8) by the definition of $\widehat{m}$, we have that

$$
k^{-(s^\star - 1)}|\theta_{\widehat{m}}^\star|^{2(s^\star - 1)} + o\left(k^{-2(s^\star - 1)}\right) \gtrsim k^{-2(s^\star - 1)} - \widetilde{O}(k^{-2s^\star}).
$$

We conclude the first step from the last inequality that there exists a global constant $c_1 > 0$ such that for sufficiently large $k$:

$$
|\theta_{\widehat{m}}^\star| \geq \left(c_1 \cdot \left(1 - o(1)\right) \cdot k^{-2(s^\star - 1)} \cdot k^{s^\star - 1}\right)^{1/2(s^\star - 1)} \geq c_1' k^{-1/2}. \tag{F.10}
$$

Now we move on to the support identification. We have shown that $|\theta^\star_{\widehat{m}}| \gtrsim k^{-1/2}$. To establish $\phi^{\star\star} \subset \widehat{\phi} = \mathsf{Top}_k(g_{\widehat{m}})$, it is sufficient to demonstrate that

$$\sup_{j \notin \phi^\star} |g_{\widehat{m},j}| \leq \inf_{j \in \phi^\star} |g_{\widehat{m},j}|. \tag{F.11}$$

In the following, we will bound each side separately. Consider $j \notin \phi^\star$, we have by [Proposition F.5](#) that

$$|\breve{g}_{\widehat{m},j}| \lesssim \left(k^{-1/2}|\theta^\star_{\widehat{m}}| + k^{-1}\delta_+\right)^{s^\star} \cdot \left(k^{-1/2}|\theta_j| + (k/d)^{-1/2}\right) + k^{-(s^\star+1)}.$$

Combining this upper bound with Eq. ([F.10](#)) and Eq. ([F.6](#)) gives us that

$$\begin{aligned}
|g_{\widehat{m},j}| &\leq |g_{\widehat{m},j} - \bar{g}_{\widehat{m},j}| + |\bar{g}_{\widehat{m},j} - \breve{g}_{\widehat{m},j}| + |\breve{g}_{\widehat{m},j}| \\
&\lesssim (k^{-1/2} \cdot |\theta^\star_{\widehat{m}}| + k^{-1}\delta_+)^{s^\star} \cdot \left(k^{-1/2}|\theta_j| + (k/d)^{-1/2}\right) + k^{-(s^\star+1)} \\
&\quad + d^{-9s^\star} + k^{-(s^\star-1/2)} \cdot \log^{-3/2} k \\
&\lesssim k^{-(s^\star-1/2)} \cdot \log^{-3/2} k + \widetilde{O}(k^{-(s^\star+1)}). \tag{F.12}
\end{aligned}$$

On the other hand, for $j \in \phi^{\star\star}$, we have that

$$\begin{aligned}
|\breve{g}_{\widehat{m},j}| &\gtrsim |k^{-1/2}\theta^\star_{\widehat{m}} + k^{-1}\delta_+|^{s^\star-1} \cdot |\theta^\star_j| \\
&\gtrsim k^{-(s^\star-1/2)}
\end{aligned}$$

Similarly, it holds that

$$\begin{aligned}
|g_{\widehat{m},j}| &\gtrsim |\breve{g}_{\widehat{m},j}| - |g_{\widehat{m},j} - \bar{g}_{\widehat{m},j}| - |\bar{g}_{\widehat{m},j} - \breve{g}_{\widehat{m},j}| \\
&\gtrsim k^{-(s^\star-1/2)} - o(k^{-(s^\star-1/2)}). \tag{F.13}
\end{aligned}$$

Comparing Eq. ([F.12](#)) and Eq. ([F.13](#)), we successfully validate Eq. ([F.11](#)) holds, and consequently $\phi^{\star\star} \subset \widehat{\phi}$.

We complete our proof of weak alignment by analyzing the inner product between $\widetilde{g}_{\widehat{m}}$ and $\theta^\star$ and the norm of $\widetilde{g}_{\widehat{m}}$ respectively. Since $|\theta^\star_{\widehat{m}}| \gtrsim k^{-1/2}$, we have that

$$(\gamma|\rho_{\widehat{m}}| + k^{-1/2} \cdot |\theta^\star_{\widehat{m}}| + k^{-1}\delta_+)^{s^\star-1} \wedge (\gamma|\rho_{\widehat{m}}| + k^{-1/2} \cdot |\theta^\star_{\widehat{m}}| + k^{-1}\delta_+)^{s^\star-2} \cdot |\gamma\rho_m| \gtrsim k^{-(s^\star-1)/2} \cdot |\theta^\star_{\widehat{m}}|^{s^\star-1}$$

For the inner product, we apply [Proposition F.5](#) for $\widehat{m}$ and each $j \in \phi^{\star\star}$ and get that

$$\begin{aligned}
g_{\widehat{m},j} \cdot \theta^\star_j &\simeq \theta^{\star 2}_j \cdot k^{-(s^\star-1)/2} \cdot |\theta^\star_{\widehat{m}}|^{s^\star-1} \cdot \mathrm{sign}(\theta^\star_{\widehat{m}}) \\
&\quad - \left(|R_{\widehat{m},j}| + |g_{\widehat{m},j} - \bar{g}_{\widehat{m},j}| + |\bar{g}_{\widehat{m},j} - \breve{g}_{\widehat{m},j}|\right) \cdot |\theta^\star_j|. \tag{F.14}
\end{aligned}$$

Since $\phi^{\star\star} \subset \widehat{\phi}_1$, we can lower bound the summation of the leading term as

$$\begin{aligned}
\sum_{j \in \widehat{\phi}_1} \theta^{\star 2}_j \cdot k^{-(s^\star-1)/2} \cdot |\theta^\star_{\widehat{m}}|^{s^\star-1} &\gtrsim \sum_{j \in \widehat{\phi}^{\star\star}} \theta^{\star 2}_j \cdot k^{-(s^\star-1)/2} \cdot |\theta^\star_{\widehat{m}}|^{s^\star-1} \\
&\gtrsim k^{-(s^\star-1)/2} \cdot |\theta^\star_{\widehat{m}}|^{s^\star-1} \cdot k^{-1} \cdot |\phi^{\star\star}| \\
&\geq k^{-(s^\star-1)/2} \cdot |\theta^\star_{\widehat{m}}|^{s^\star-1}, \tag{F.15}
\end{aligned}$$

where the last line holds by the definition of $\mathcal{E}_{0,\sharp} \subset \mathcal{E}_0$. For the term associated with $R_{\widehat{m},j}$, we have by the characterization in [Proposition F.5](#) that

$$\begin{aligned}
\sum_{j \in \widehat{\phi}_1} |R_{m,j}| \cdot |\theta^\star_j| &\lesssim (k^{-s^\star} \cdot \log(k)^{s^\star} + k^{-s^\star}) \cdot \sum_{j \in \phi^\star} |\theta^\star_j|^2 \\
&\quad + k^{-s^\star} \cdot \log(k)^{s^\star} \cdot k^{-1/2} \cdot \left(\sum_{j \in \widehat{\phi}_1} |\theta_j| \cdot |\theta^\star_j| + |\theta^\star_j| \cdot (k/d)^{\mathbb{1}\{j \neq m\}}\right) \\
&\quad + k^{-(s^\star+1)} \cdot \sum_{j \in \widehat{\phi}_1} |\theta^\star_j|.
\end{aligned}$$

Applying the Cauchy-Schwarz inequality, we have that

$$\sum_{j \in \hat{\phi}_1} |\theta_j^\star| \cdot \left(|\theta_j^\star| + (k/d)^{\mathbb{1}\{j \neq m\}}\right) \leq \left(\sum_{j \in [d]} |\theta_j^\star|^2\right)^{1/2} \cdot \left(\left(\sum_{j \in [d]} |\theta_j|^2\right)^{1/2} + (1 + k^3/d^2)^{1/2}\right)$$

$$= O(1). \tag{F.16}$$

And therefore

$$\sum_{j \in \hat{\phi}_1} |R_{m,j}| \cdot |\theta_j^\star| \lesssim k^{-s^\star} \cdot \log(k)^{s^\star} + k^{-s^\star} + \widetilde{O}(k^{-(s^\star+1/2)}). \tag{F.17}$$

For the rest of the error terms, we have by Eq. (F.6) and Eq. (F.5) that

$$\sum_{j \in \hat{\phi}_1} \left(|g_{\widehat{m},j} - \bar{g}_{\widehat{m},j}| + |\bar{g}_{\widehat{m},j} - \breve{g}_{\widehat{m},j}|\right) \cdot |\theta_j^\star| \lesssim \sum_{j \in \phi^\star} |\theta_j^\star| \cdot k^{-(s^\star-1/2)} \cdot \log(k)^{-3/2}$$

$$\leq k^{-(s^\star-1)} \cdot \log(k)^{-3/2}. \tag{F.18}$$

Combining Eq. (F.14), Eq. (F.15), Eq. (F.17) and Eq. (F.18), we have that

$$|\langle \widetilde{g}_{\widehat{m}}, \theta^\star \rangle| \gtrsim k^{-(s^\star-1)/2} \cdot |\theta_{\widehat{m}}^\star|^{s^\star-1} - o(k^{-(s^\star-1)}).$$

For the norm, we already have in Eq. (F.8) that

$$\|\widetilde{g}_{\widehat{m}}\|_2 \lesssim k^{-(s^\star-1)/2} \cdot |\theta_{\widehat{m}}^\star|^{(s^\star-1)} + o(k^{-(s^\star-1)}).$$

Combining last two inequalities, we concludes that

$$\frac{|\langle P_{\hat{\phi}_1} g_{\widehat{m}}, \theta^\star \rangle|}{\|P_{\hat{\phi}_1} g_{\widehat{m}}\|} \gtrsim \frac{|\theta_{\widehat{m}}^\star|^{s^\star-1} - o(k^{-(s^\star-1)/2})}{|\theta_{\widehat{m}}^\star|^{s^\star-1} + o(k^{-(s^\star-1)/2})} = \Theta(1),$$

given that $|\theta_{\widehat{m}}^\star| \geq c_1' k^{-1/2}$ for some constant $c_1' > 0$. This concludes the proof of weak alignment.

**Strong Alignment.** Starting from the second step, we have by Algorithm 2 that all the neurons share the same weight parameter. Let $\theta$ be the weight parameter in any step after the first step and we suppose that $\rho = \langle \theta, \theta^\star \rangle = \Theta(1)$. From Proposition F.5, we see that now the choice of the gradient should not change the quality of the gradient significantly, as $\gamma|\rho| \gg k^{-1/2}|\theta_m^\star|$ for any $m \in [d]$. Therefore, we start by analyzing the alignment increment using any $g_m$. This alteration does not affect the alignment increment, but substantially simplifies the analysis.

We additionally define a support $\phi^\dagger = \{j \in [d] : |\theta_j^\star| \geq k^{-1}\}$. In the following, we fix an arbitrary $m \in [d]$. We begin with analyzing the magnitude of $g_{m,j}$ for $j \in \phi^\dagger \setminus \{m\}$. Since $|\rho| = \Omega(1)$, it holds that

$$(\gamma|\rho| + k^{-1/2} \cdot |\theta_m^\star| + k^{-1}\delta_{s^\star-1})^{s^\star-1} \simeq (\gamma|\rho| + k^{-1/2} \cdot |\theta_m^\star| + k^{-1}\delta_{s^\star-2})^{s^\star-2} \cdot (\gamma|\rho|) \simeq k^{-(s^\star-1)/2}.$$

With triangle inequality, Proposition F.5 indicates that, for $j \in \phi^\dagger$:

$$|g_{m,j}| \geq |\breve{g}_{m,j}| - |\bar{g}_{m,j} - \breve{g}_{m,j}| - |\bar{g}_{m,j} - g_{m,j}|$$

$$\geq k^{-(s^\star-1)/2} \cdot |\theta_j^\star| - k^{-s^\star/2} \cdot |\theta_j^\star|$$

$$- k^{-(s^\star+1)/2} \cdot (|\theta_j| + (k/d)^{\mathbb{1}\{j \neq m\}/2}) - \widetilde{O}(k^{-(s^\star-1/2)}).$$

Similarly, we have for $j \notin \phi^\star$ that

$$|g_{m,j}| \leq |\breve{g}_{m,j}| + |\bar{g}_{m,j} - \breve{g}_{m,j}| + |\bar{g}_{m,j} - g_{m,j}|$$

$$\lesssim k^{-(s^\star+1)/2} \cdot (|\theta_j| + (k/d)^{\mathbb{1}\{j \neq m\}/2}) + \widetilde{O}(k^{-(s^\star-1/2)}).$$

Comparing last two inequalities, we have that $|\theta_j| > k^{-1}$ implies that $|g_{m,j}| \geq \max_{j \notin \phi^\star} |g_{m,j}|$ for sufficiently large $k$, and therefore

$$\min_{j \in \phi^\dagger} |g_{m,j}| > \max_{j \notin \phi^\star} |g_{m,j}|,$$

which means that $\phi^\dagger \subset \widehat{\phi}_m = \mathsf{Top}_k(\theta_m)$. Thereby, we have

$$\sum_{j \in \widehat{\phi}_m} |\theta_j^\star|^2 \geq 1 - \sum_{j \notin \phi^\dagger} |\theta_j^\star|^2 \geq 1 - k^{-1}.$$

We then move on to the alignment analysis. Retreat to Eq. (F.23), we define

$$\beta_m(\rho, \{\xi_{m,l}\}_{l \in [L]}, \theta^\star, \theta) = \frac{\mathbb{E}_{\mathbb{Q}}[\zeta_{s^\star}(y) \cdot \widehat{\psi}_{s^\star - 1}(y)] \cdot \sqrt{s^\star} \cdot \mathbb{E}_{\widetilde{w}_{m,l}}[\langle w_m, l, \theta^\star \rangle^{s^\star - 1}]}{\mathrm{sign}(\rho)^{\mathbb{1}\{s^\star \text{ even}\}} \cdot (\gamma|\rho|)^{s^\star - 1}};$$

$$r_{m,j}(\rho, \{\xi_{m,l}\}_{l \in [L]}, \theta^\star, \theta) = \mathbb{E}_{\mathbb{P}_{\theta^\star}}[\langle \bar{g}_m, e_j \rangle] - \theta_j^\star \cdot (\gamma|\rho|)^{s^\star - 1} \cdot \mathrm{sign}(\rho)^{\mathbb{1}\{s^\star \text{ even}\}} \cdot \beta_m. \quad \text{(F.19)}$$

Then it follows from Proposition F.5 that whenever $|\rho| = \Omega(1)$, we have that $\beta_m > 0$ and

$$\beta_m \vee \beta_m^{-1} < B$$

$$|r_{m,j}| \leq r_{m,j}^a + r_{m,j}^b,$$

where

$$r_{m,j}^a \leq C_a k^{-(s^\star + 1)/2} \cdot \left( |\theta_j| + (k/d)^{\mathbb{1}\{j \neq m\}/2} \right);$$

$$r_{m,j}^b \leq C_b \cdot |\theta_j^\star| \cdot (\gamma|\rho|)^{s^\star} \quad \text{(F.20)}$$

with some global positive constant $B, C_a$ and $C_0$, whenever the designated parameters are compatible with the definition of our nice event. With this representation, we can deduce by Eq. (F.6) and Eq. (F.5) that

$$|\langle \widetilde{g}_m, \theta^\star \rangle| = \Big| \sum_{j \in \widehat{\phi}_m} \langle \widecheck{g}_m, e_j \rangle \cdot \theta_j^\star \Big| + \sum_{j \in \widehat{\phi}_m} \left( |\langle g_m, e_j \rangle - \langle \bar{g}_m, e_j \rangle| + |\langle \bar{g}_m, e_j \rangle - \langle \widecheck{g}_m, e_j \rangle| \right) \cdot |\theta_j^\star|$$

$$\geq \Big| \sum_{j \in \widehat{\phi}_m} \langle \widecheck{g}_m, e_j \rangle \cdot \theta_j^\star \Big| - \sum_{j \in \phi^\star} |\theta_j^\star| \cdot O(k^{-(s^\star - 1/2)} \cdot \log^{-3/2} k)$$

$$\geq \beta_m \cdot (\gamma|\rho|)^{s^\star - 1} \cdot \sum_{j \in \widehat{\phi}_m} |\theta_j^\star|^2 - \sum_{j \in \phi^\star} |r_{m,j}| \cdot |\theta_j^\star| - O(k^{-(s^\star - 1)} \cdot \log^{-3/2} k)$$

$$\geq \left( \beta_m \cdot (\gamma|\rho|)^{s^\star - 1} - C_b \cdot (\gamma|\rho|)^{s^\star} \right) \cdot \sum_{j \in \widehat{\phi}_m} |\theta_j^\star|^2 - 2C_a \cdot k^{-(s^\star + 1)/2} - O(k^{-(s^\star - 1)} \cdot \log^{-3/2} k).$$

$$\text{(F.21)}$$

where the last inequality holds by Eq. (F.16). To complete the analysis, we need an upper bound for $\|\widetilde{g}_m\|_2$. Note that by the triangle inequality and Eq. (F.19), we have that

$$\|\widetilde{g}_m\| \leq \|P_{\widehat{\phi}_m} \widecheck{g}_m\| + \|P_{\widehat{\phi}_m} \bar{g}_m - P_{\widehat{\phi}_m} \widecheck{g}_m\| + \|P_{\widehat{\phi}_m} \bar{g}_m - P_{\widehat{\phi}_m} g_m\|$$

$$\leq \|\beta_m (\gamma|\rho|)^{s^\star - 1} \cdot P_{\widehat{\phi}_m} \theta^\star\| + \|P_{\widehat{\phi}_m} r_m^a\| + \|P_{\widehat{\phi}_m} r_m^b\|$$

$$+ O(k^{-(s^\star - 1)} \cdot \log^{-3/2} k),$$

where $r_m^a = (r_{m,1}^a, \ldots, r_{m,d}^a) \in \mathbb{R}^d$ and $r_m^b = (r_{m,1}^b, \ldots, r_{m,d}^b) \in \mathbb{R}^d$. To proceed, note that by Eq. (F.20) and Eq. (F.6), we have that

$$\|P_{\widehat{\phi}} r_m^a\| \leq C_a k^{-(s^\star + 1)/2} \cdot (\|\theta\|_2 + \sqrt{1 + k^2/d})$$

$$\leq 3C_a k^{-(s^\star + 1)/2};$$

$$\|P_{\widehat{\phi}} r_m^b\| \leq C_b \cdot (\gamma|\rho|)^{s^\star} \cdot \Big( \sum_{j \in \widehat{\phi}_m} |\theta_j^\star|^2 \Big)^{1/2}.$$

Putting these upper bounds together, we have that

$$\|P_{\widehat{\phi}} g_m\|_2 \leq \left( \beta_m \cdot (\gamma|\rho|)^{s^\star - 1} + C_b (\gamma|\rho|)^{s^\star} \right) \cdot \Big( \sum_{j \in \widehat{\phi}_m} |\theta_j^\star|^2 \Big)^{1/2}$$

$$+ 3C_a k^{-(s^\star + 1)/2} + O(k^{-(s^\star - 1)} \cdot \log^{-3/2} k). \quad \text{(F.22)}$$

Note that for any $a_1 \wedge a_2 > b > 0$, it holds that

$$\frac{a_1 - b}{a_2 + b} = \frac{(a_1 - b) \cdot (a_2 - b)}{a_2^2 - b^2} \geq (a_1/a_2 - b/a_2) \cdot (1 - b/a_2).$$

Setting $\Delta = \cdot k^{-1} + k^{-(s^\star - 1)/2} \cdot \log^{-3/2} k \vee k^{-1} = o(1)$, we get by combining Eq. (F.21) and Eq. (F.22) that

$$\frac{\langle \widetilde{g}_m, \theta^\star \rangle}{\|\widetilde{g}_m\|} \geq \frac{(\beta_m - C_b \cdot \gamma |\rho|) \cdot \left( \sum_{j \in \hat{\phi}_m} \theta_j^{\star 2} \right) - 3 C_a \cdot k^{-1} - O(k^{-(s^\star - 1)/2} \cdot \log^{-3/2} k)}{(\beta_m + C_b \cdot \gamma |\rho|) \cdot \left( \sum_{j \in \hat{\phi}_m} \theta_j^{\star 2} \right)^{1/2} + 3 C_a \cdot k^{-1} + O(k^{-(s^\star - 1)/2} \cdot \log^{-3/2} k)}$$

$$\geq \left( 1 - O(\Delta) \right)^{-1} \cdot \left( \frac{1 - C_b \beta_m^{-1} \gamma |\rho|}{1 + C_b \beta_m^{-1} \gamma |\rho|} \cdot \left( \sum_{j \in \hat{\phi}_m} \theta_j^{\star 2} \right)^{1/2} - O(\Delta) \right)$$

$$\geq 1 - C \cdot k^{-1} - O(\Delta)$$

where the last line holds because $(1 - Ck^{-1})^r \geq 1 - rC \cdot k^{-1}$ for any $C, r > 0$ and sufficiently large $k$. Note that $k^{-1} = O(\Delta)$, we see that

$$\frac{\langle \widetilde{g}_m, \theta^\star \rangle}{\|\widetilde{g}_m\|} \geq 1 - O(\Delta).$$

This concludes the proof of Theorem 5.1.

$\square$

## F.4    PROOF OF THE KEY RESULTS

*Proof of Proposition F.5.* First, by Lemma H.2, we have for each $j \in [d]$ that

$$\langle \mathbb{E}_{\mathbb{P}_{\theta^\star}}[\bar{g}_m], e_j \rangle = \sum_{s \geq s^\star} \mathbb{E}_{\mathbb{Q}}[\zeta_s(y) \cdot \widehat{\psi}_{s+1}(y)] \cdot \frac{\sqrt{s+1}}{L} \sum_{l=1}^{L} \langle w_{m,l}, \theta^\star \rangle^s \cdot \langle w_{m,l}, e_j \rangle$$

$$+ \sum_{s \geq s^\star} \mathbb{E}_{\mathbb{Q}}[\zeta_s(y) \cdot \widehat{\psi}_{s-1}(y)] \cdot \frac{\sqrt{s}}{L} \cdot \sum_{l=1}^{L} \langle w_{m,l}, \theta^\star \rangle^{s-1} \cdot \langle \theta^\star, e_j \rangle.$$

$$= \mathbb{E}_{\mathbb{Q}}[\zeta_{s^\star}(y) \cdot \widehat{\psi}_{s^\star - 1}(y)] \cdot \frac{\sqrt{s^\star}}{L} \sum_{l=1}^{L} \langle w_{m,l}, \theta^\star \rangle^{s^\star - 1} \cdot \langle \theta^\star, e_j \rangle$$

$$+ R_1 + R_2, \tag{F.23}$$

where the remainders $R_1, R_2$ are defined as

$$R_1 = \sum_{s \geq s^\star} \mathbb{E}_{\mathbb{Q}}[\zeta_{s+1}(y) \cdot \widehat{\psi}_s(y)] \cdot \frac{\sqrt{s+1}}{L} \sum_{l=1}^{L} \langle w_{m,l}, \theta^\star \rangle^s \cdot \langle \theta^\star, e_j \rangle;$$

$$R_2 = \sum_{s \geq s^\star} \mathbb{E}_{\mathbb{Q}}[\zeta_s(y) \cdot \widehat{\psi}_{s+1}(y)] \cdot \frac{\sqrt{s+1}}{L} \cdot \sum_{l=1}^{L} \langle w_{m,l}, \theta^\star \rangle^s \cdot \langle w_{m,l}, e_j \rangle.$$

We also denote the leading signal term as

$$S = \mathbb{E}_{\mathbb{Q}}[\zeta_{s^\star}(y) \cdot \widehat{\psi}_{s^\star - 1}(y)] \cdot \frac{\sqrt{s^\star}}{L} \sum_{l=1}^{L} \langle w_{m,l}, \theta^\star \rangle^{s^\star - 1} \cdot \langle \theta^\star, e_j \rangle.$$

By definition $R_1$ collects the higher order term that aligns with the signal and the $R_2$ collects all the terms in the expected gradient that are parallel to $w_{m,l}$. In comparison to the non-sparse case, here we are analyzing the gradient coordinate-wisely. Therefore, $R_1$ and $R_2$ need to be controlled separately.

**Analysis for the dominant term $S$ in Eq.** (F.23). We first define

$$\widetilde{w}_{m,l} = w_{m,l} \cdot \mathbb{1}\{|\sup_j \langle \xi_{m,l}, e_j \rangle| \le \epsilon\}.$$

By definition of $\widetilde{w}_{m,l}$, we have that $\widetilde{w}_{m,l}, l \in [L]$ are independent to each other and $\widetilde{w}_{m,l} = w_{m,l}$ on event $\mathcal{E}_m(\epsilon)$. We can approximate the expectation of the $\langle w_{m,l}, \theta^\star \rangle$ as

$$\mathbb{E}_{\widetilde{w}_{m,l}}[\langle \widetilde{w}_{m,l}, \theta^\star \rangle^{s^\star - 1}] = \mathbb{E}_{w_{m,l}}\left[\langle w_{m,l}, \theta^\star \rangle^{s^\star - 1} \cdot \mathbb{1}\left\{\sup_j |\langle \xi_{m,l}, e_j \rangle| \le \epsilon\right\}\right]$$

$$\simeq \mathbb{E}_{w_{m,l}}[\langle w_{m,l}, \theta^\star \rangle^{s^\star - 1}] \pm \Pr\left(\sup_j |\langle \xi_{m,l}, e_j \rangle| > \epsilon\right)$$

$$\simeq \mathbb{E}_{w_{m,l}}[\langle w_{m,l}, \theta^\star \rangle^{s^\star - 1}] \pm \Pr(\mathcal{E}_m(\epsilon)^c).$$

Here the last line holds because $\mathcal{E}_m(\epsilon)^c = \cup_l \{\sup_j |\langle \xi_{m,l}, e_j \rangle| > \epsilon\}$. For the first term, it holds by Proposition F.4, we have that on $\mathcal{E}_{0,s^\star - 1 - \mathbb{1}\{s^\star \text{ even}\}}$

$$\mathbb{E}_{w_{m,l}}[\langle w_{m,l}, \theta^\star \rangle^{s^\star - 1}] \simeq \begin{cases} \gamma\rho \cdot \left(\gamma|\rho| + k^{-1/2}|\theta_m^\star| + k^{-1}\delta_{s^\star - 2}\right)^{s^\star - 2} & \text{if } s^\star \text{ even;} \\ \left(\gamma|\rho| + k^{-1/2}|\theta_m^\star| + k^{-1}\delta_{s^\star - 1}\right)^{s^\star - 1} & \text{if } s^\star \text{ odd.} \end{cases}$$

For the second moment that is involved in the Bernstein's inequality, we have that on $\mathcal{E}_{0,2s^\star - 2}$

$$\mathbb{E}[\langle \widetilde{w}_{m,l}, \theta^\star \rangle^{2s^\star - 2}] = \mathbb{E}[\langle w_{m,l}, \theta^\star \rangle^{2s^\star - 2} \mathbb{1}\{\sup_j |\langle w_{m,l}, e_j \rangle| \le \epsilon\}]$$

$$\le \mathbb{E}[\langle w_{m,l}, \theta^\star \rangle^{2s^\star - 2}]$$

$$\simeq (\gamma|\rho| + k^{-1/2}|\theta_m^\star| + k^{-1}\delta_{2s^\star - 2})^{2s^\star - 2}$$

To proceed, we have by Bernstein's inequality (Lemma J.1) that there exists an event $\mathcal{E}_{m,11}$ with $\Pr(\mathcal{E}_{m,11}) \ge 1 - O(d^{-c_{b,11}})$. And it holds on $\mathcal{E}_{m,11} \cap \mathcal{E}_m(\epsilon)$ that

$$\frac{1}{L}\sum_l \langle w_{m,l}, \theta^\star \rangle^{s^\star - 1} = \frac{1}{L}\sum_l \langle \widetilde{w}_{m,l}, \theta^\star \rangle^{s^\star - 1}$$

$$\simeq \mathbb{E}_{w_{m,l}}[\langle w_{m,l}, \theta^\star \rangle^{s^\star - 1}] + E,$$

where the error term $E$ can be bounded by

$$|E| \le \left(\gamma|\rho| + k^{-1/2}|\theta_m^\star| + k^{-1}\delta_{2s^\star - 2}\right)^{s^\star - 1} \cdot \sqrt{\frac{\log(d)}{L}} + \frac{\epsilon^{s^\star - 1}\log(d)}{L} + \Pr\left(\mathcal{E}_m^c(\epsilon)\right).$$

Moreover, the assumption that

$$L \gtrsim \log d \cdot \left(k^2 \vee \left(\epsilon^{s^\star - 1} \cdot k^{s^\star}\right)\right); \qquad \Pr(\mathcal{E}_m(\epsilon)^c) \le k^{-s^\star};$$

allows us to simplify the upper bound for $E$, since

$$|E| \lesssim k^{-1} \cdot \left(\gamma|\rho| + k^{-1/2}|\theta_m^\star| + k^{-1}\delta_{2s^\star - 2}\right)^{s^\star - 1} + k^{-s^\star}. \tag{F.24}$$

In conclusion, we have on $\mathcal{E}_{m,11} \cap \mathcal{E}_m(\epsilon)$ that

$$S \simeq (\mathbb{E}_{w_{m,l}}[\langle w_{m,l}, \theta^\star \rangle^{s^\star - 1}] + E) \cdot \langle \theta^\star, e_j \rangle.$$

We remain this form for further simplification.

**Analysis for the first remainder $R_1$ in Eq.** (F.23). For any $s, s'$, it holds by the property of likelihood ratio decomposition that

$$\mathbb{E}_\mathbb{Q}[|\zeta_s(y) \cdot \widehat{\psi}_{s'}(y)|] \le \mathbb{E}_\mathbb{Q}[\zeta_s(y)^2]^{1/2} \cdot \mathbb{E}_\mathbb{Q}[\widehat{\psi}_{s'}(y)^2]^{1/2}$$

$$\le \sqrt{\sum_{s' \ge 0} \mathbb{E}_\mathbb{Q}[\widehat{\psi}_{s'}(y)^2]},$$

and the last quantity is a constant that is independent to $s, s'$. To bound the summation for $s \geq s^\star$, we have on by Lemma F.3 that $|\langle w_{m,l}, \theta^\star \rangle| \leq \gamma|\rho| + \epsilon < 1/2$ on

$$\sum_{s \geq s^\star} \frac{\sqrt{s+1}}{L} \sum_{l=1}^{L} |\langle w_{m,l}, \theta^\star \rangle|^s \lesssim \sum_{s \geq s^\star} \sqrt{s+1} \cdot \left(\frac{1}{2}\right)^{s-s^\star} \cdot \frac{1}{L} \sum_{l=1}^{L} |\langle w_{m,l}, \theta^\star \rangle|^{s^\star}$$

$$\lesssim \frac{1}{L} \sum_{l=1}^{L} |\langle w_{m,l}, \theta^\star \rangle|^{s^\star}. \tag{F.25}$$

Now it reduces to bound the right-hand side of Eq. (F.25). Note that on $\mathcal{E}_m(\epsilon)$, $\widetilde{w}_{m,l} = w_{m,l}$. We can first track the first and second moment of $\langle w_{m,l}, \theta^\star \rangle$ as

$$\mathbb{E}_{\widetilde{w}_{m,l}}[|\langle \widetilde{w}_{m,l}, \theta^\star \rangle|^{s^\star}] \leq \mathbb{E}_{w_{m,l}}[|\langle w_{m,l}, \theta^\star \rangle|^{s^\star}]$$

$$\leq \mathbb{E}_{w_{m,l}}[\langle w_{m,l}, \theta^\star \rangle^{2s^\star}]^{1/2}.$$

To bound the last quantity, we see that given $\mathcal{E}_{0,2s^\star}$, Proposition F.4 reads

$$\mathbb{E}_{w_{m,l}}[\langle w_{m,l}, \theta^\star \rangle^{2s^\star}] \leq (\gamma|\rho| + k^{-1/2}|\theta_m^\star| + k^{-1}\delta_+)^{2s^\star}.$$

By Bernstein's inequality, there exists a event $\mathcal{E}_{m,12}$ with $\Pr(\mathcal{E}_{m,12}) \geq 1 - O(d^{-c_{b,12}})$ such that on $\mathcal{E}_{m,12} \cap \mathcal{E}_m(\epsilon)$, it holds that

$$\frac{1}{L} \sum_{l=1}^{L} |\langle w_{m,l}, \theta^\star \rangle|^{s^\star} \lesssim \left(1 + \sqrt{\frac{\log d}{L}}\right) \cdot (\gamma|\rho| + k^{-1/2}|\theta_m^\star| + k^{-1}\delta_+)^{s^\star} + \frac{\epsilon^{s^\star} \log(d)}{L},$$

Given that $L \gtrsim \log(d) \cdot (k \vee (\epsilon^{s^\star} \cdot k^{s^\star}))$, it further holds that

$$\frac{1}{L} \sum_{l=1}^{L} |\langle w_{m,l}, \theta^\star \rangle|^{s^\star} \lesssim (\gamma|\rho| + k^{-1/2}|\theta_m^\star| + k^{-1}\delta_+)^{s^\star} + k^{-s^\star}.$$

In conclusion, it holds on $\mathcal{E}_{m,12} \cap \mathcal{E}_m(\epsilon)$ that

$$R_1 \lesssim \left((\gamma|\rho| + k^{-1/2}|\theta_m^\star| + k^{-1}\delta_+)^{s^\star-1} + k^{-s^\star}\right) \cdot |\langle \theta^\star, e_j \rangle|.$$

**Analysis for the second remainder $R_2$ in Eq. (F.23).** Similar to Eq. (F.25), we can first upper bound $R_2$ as

$$|R_2| \lesssim \frac{1}{L} \sum_{l=1}^{L} |\langle w_{m,l}, \theta^\star \rangle|^{s^\star} \cdot |\langle w_{m,l}, e_j \rangle|$$

$$= \frac{1}{L} \sum_{l=1}^{L} |\langle \widetilde{w}_{m,l}, \theta^\star \rangle|^{s^\star} \cdot |\langle \widetilde{w}_{m,l}, e_j \rangle|,$$

where the last line holds by the definition of $\mathcal{E}_m(\epsilon)$. We decouple the product with the Cauchy-Schwarz inequality as follows:

$$\mathbb{E}_{\widetilde{w}_{m,l}}[|\langle \widetilde{w}_{m,l}, \theta^\star \rangle|^{s^\star} \cdot |\langle \widetilde{w}_{m,l}, e_j \rangle|] \leq \mathbb{E}_{w_{m,l}}[|\langle w_{m,l}, \theta^\star \rangle|^{s^\star} \cdot |\langle w_{m,l}, e_j \rangle|]$$

$$\leq \mathbb{E}_{w_{m,l}}[|\langle w_{m,l}, \theta^\star \rangle|^{2s^\star}]^{1/2} \cdot \mathbb{E}_{w_{m,l}}[|\langle w_{m,l}, e_j \rangle|^2]^{1/2}$$

The first term in the upper bound can be tackled with Proposition F.4. For the second term, we have that

$$\mathbb{E}_{w_{m,l}}[\langle w_{m,l}, e_j \rangle^2] \lesssim \mathbb{E}[(\langle \xi_{m,l}, e_j \rangle + \gamma \cdot \langle \theta, e_j \rangle)^2]$$

$$\lesssim \mathbb{E}[\langle \xi_{m,l}, e_j \rangle^2] + \gamma^2 \theta_j^2$$

$$= \gamma^2 \theta_j^2 + \mathbb{E}[\mathbb{1}\{j \in \phi_{m,l}\} \cdot \xi_{m,l,j}^2]$$

$$\lesssim \gamma^2 \theta_j^2 + k^{-1} \cdot (k/d)^{\mathbb{1}\{j \neq m\}}.$$

Here, the first line holds because $\|\gamma\theta + \xi\|_2 \geq 1/2$. The last line holds by applying Lemma I.5 and that $\mathbb{P}(j \in \phi_{m,l}) \leq k/d$ for $j \neq m$. Note that each term in the summation of is bounded by $\epsilon \log(k)$ up to a constant on $\mathcal{E}_m(\epsilon)$. We have by Bernstein's inequality (Lemma J.1) that, there exists an event $\mathcal{E}_{m,13}$ with $\Pr(\mathcal{E}_{m,13}) \geq 1 - O(d^{-c_{b,13}})$. And it holds on $\mathcal{E}_{m,13} \cap \mathcal{E}_m(\epsilon)$ that

$$|R_2| \lesssim \left(1 + \sqrt{\frac{\log(d)}{L}}\right) \cdot \left(\gamma|\rho| + k^{-1/2}|\theta_m^\star| + k^{-1}\delta_+\right)^{s^\star} \cdot \left(\gamma|\theta_j| + k^{-1/2}(k/d)^{\mathbb{1}\{j \neq m\}/2}\right)$$
$$+ \frac{\epsilon^{s^\star+1}\log(d)}{L}.$$

Given that

$$L \gtrsim \log(d) \cdot \left(k \vee (\epsilon^{s^\star+1} \cdot k^{s^\star+1})\right),$$

we conclude that it holds on $\mathcal{E}_{m,13} \cap \mathcal{E}_m(\epsilon)$ that

$$|R_2| \lesssim \left(\gamma|\rho| + k^{-1/2}|\theta_m^\star| + k^{-1}\delta_+\right)^{s^\star} \cdot \left(\gamma|\theta_j| + k^{-1/2}(k/d)^{\mathbb{1}\{j \neq m\}/2}\right) + k^{-(s^\star+1)}.$$

**Summary of first-order moment.** We now merge previous results to summarize the results for the first-order moment. Note that it is sufficient to set

$$L = \Omega\left(\log(d) \cdot \left(k \vee \epsilon^{s^\star-1}(k \cdot \log k)^{s^\star+1}\right)\right)$$

Define the final event as $\mathcal{E}_{m,1} = \mathcal{E}_{m,11} \cap \mathcal{E}_{m,12} \cap \mathcal{E}_{m,13}$, which is $\{w_{m,l}\}_{l \in [L]}$ measurable. By previous analysis, it holds on this event that

$$S \simeq \theta_j^\star \cdot (\gamma\rho)^{\mathbb{1}\{s^\star \text{ even}\}} \cdot \left(\gamma|\rho| + k^{-1/2}|\theta_m^\star| + k^{-1}\delta_{s^\star-1-\mathbb{1}\{s^\star \text{ even}\}}\right)^{s^\star-1-\mathbb{1}\{s^\star \text{ even}\}} + \theta_j^\star \cdot E;$$
$$R_1 \lesssim \left(\left(\gamma|\rho| + k^{-1/2}|\theta_m^\star| + k^{-1}\delta_+\right)^{s^\star} + k^{-s^\star}\right) \cdot |\theta_j^\star|;$$
$$R_2 \lesssim \left(\gamma|\rho| + k^{-1/2}|\theta_m^\star| + k^{-1}\delta_+\right)^{s^\star} \cdot (\gamma|\theta_j| + k^{-1/2} \cdot (k/d)^{\mathbb{1}\{j \neq m\}/2}) + k^{-(s^\star+1)}.$$

Following the error term $E$ in Eq. (F.24), we define $R = R_1 + R_2 + E$, which be bounded by d

$$|R| \lesssim \left(k^{-1} \vee (\gamma|\rho| + k^{-1/2}|\theta_m^\star|) \cdot (\gamma|\rho| + k^{-1/2}|\theta_m^\star| + k^{-1}\delta_+)^{s^\star-1} + k^{-s^\star}\right) \cdot |\theta_j^\star|$$
$$+ \left(\gamma|\rho| + k^{-1/2}|\theta_m^\star| + k^{-1}\delta_+\right)^{s^\star} \cdot (\gamma|\theta_j| + k^{-1/2} \cdot (k/d)^{\mathbb{1}\{j \neq m\}/2}) + k^{-(s^\star+1)}. \tag{F.26}$$

And we summarize the first moment on $\mathcal{E}_{m,1} \cap \mathcal{E}_m(\epsilon)$ as

$$\mathbb{E}_{\mathbb{P}_{\theta^\star}}[\langle \bar{g}_m, e_j\rangle] \simeq \theta_j^\star \cdot (\gamma\rho)^{\mathbb{1}\{s \text{ even}\}} \cdot \left(\gamma|\rho| + k^{-1/2}|\theta_m^\star| + k^{-1}\delta_{s^\star-1-\mathbb{1}\{s \text{ even}\}}\right)^{s^\star-1-\mathbb{1}\{s \text{ even}\}} + R,$$

where $R$ is upper bounded in Eq. (F.26). $\qquad \square$

*Proof of Proposition F.6.* Similar to the proof of Proposition E.4, the proof of this proposition comprises two parts. To begin with, we calculate the variance of each coordinate of $\bar{g}_m$.

**Second moment calculation.** It suffices to consider the variance of the first sample. To this end, we define

$$\bar{g}_{m,1} = \frac{1}{L} \sum_{l=1}^{L} \left(\psi(y_1, \langle w_{m,l}, z_1\rangle) \cdot z_1 - \widehat{\psi}_1(y_1) \cdot w_{m,l}\right).$$

For any $v \in \{e_1, e_2, \ldots, e_d\}$, it holds by the definition of $\bar{g}_{m,1}$ that

$$
\mathbb{E}_{\mathbb{P}_{\theta^\star}}[\langle \bar{g}_{m,1}, v \rangle^2] \lesssim \frac{1}{L^2} \sum_{l,l'=1}^{L} \mathbb{E}_{\mathbb{P}_{\theta^\star}} \left[ \psi(y, \langle w_l, z \rangle) \psi(y, \langle w_{l'}, z \rangle) \langle z, v \rangle^2 \right] + \frac{1}{L^2} \sum_{l,l'=1}^{L} \mathbb{E}_{\mathbb{P}_{\theta^\star}} \left[ \widehat{\psi}_1(y)^2 \langle w_l, v \rangle \langle w_{l'}, v \rangle \right]
$$

$$
= \frac{1}{L^2} \sum_{l \neq l'} \mathbb{E}_{\mathbb{Q}} \left[ \psi(y, \langle w_l, z \rangle) \psi(y, \langle w_{l'}, z \rangle) \langle z, v \rangle^2 \cdot \left( 1 + \sum_{s \geq s^\star} \zeta_s(y) h_s(\langle \theta^\star, z \rangle) \right) \right]
$$

$$
+ \frac{1}{L^2} \sum_{l=1}^{L} \mathbb{E}_{\mathbb{Q}} \left[ \psi(y, \langle w_l, z \rangle) \psi(y, \langle w_l, z \rangle) \langle z, v \rangle^2 \right] + \frac{1}{L^2} \sum_{l \neq l'} \mathbb{E}_{\mathbb{Q}}[\widehat{\psi}_1(y)^2] \langle w_l, v \rangle \langle w_{l'}, v \rangle
$$

$$
+ \frac{1}{L^2} \sum_{l=1}^{L} \mathbb{E}_{\mathbb{Q}}[\widehat{\psi}_1(y)^2] \langle w_l, v \rangle^2. \tag{F.27}
$$

In the same manner as the non-sparse case, we can derive a $O(1/L)$ upper bound for the second and the last summation, which traverse through all $l = l'$. Recalling Eq. Lemma F.3, we already have that

$$
\sup_{l,j} |\langle w_{m,l}, e_j \rangle| \lesssim \gamma + \epsilon;
$$

$$
\sup_{l} |\langle w_{m,l}, \theta^\star \rangle| \lesssim \gamma |\rho| + \epsilon.
$$

$$
\sup_{l \neq l'} |\langle w_{m,l}, w_{m,l'} \rangle| \lesssim \gamma^2 + \epsilon^2 \log(k).
$$

To incorporate with the notations in Lemma H.3, we denote the upper bounds of the $\langle w_{m,l}, e_j \rangle$, $\langle w_{m,l}, \theta^\star \rangle$ and $\langle w_{m,l}, w_{m,l'} \rangle$ ($l \neq l'$) as

$$
\epsilon_0 = \gamma + \epsilon; \qquad \epsilon_1 = \gamma |\rho| + \epsilon; \qquad \epsilon_2 = \gamma^2 + \epsilon^2 \log(k),
$$

respectively. By the virtue of Lemma H.3, the desired expectation is behaving nicely if the ratio $(\epsilon_0^2 \vee \epsilon_1^2)/\epsilon_2$ is a constant term. To validate this fact, we note that

$$
\frac{\epsilon_0^2}{\epsilon_2} \simeq \frac{\gamma^2 + \epsilon^2}{\gamma^2 + \epsilon^2 \log(k)}, \qquad \frac{\epsilon_1^2}{\epsilon_2} \simeq \frac{\gamma^2 \rho^2 + \epsilon^2}{\gamma^2 + \epsilon^2 \log(k)}.
$$

Since $\epsilon \ll 1 \ll \log(k)^{1/2}$, we conclude that $\epsilon_0^2 \vee \epsilon_1^2/\epsilon_2 \lesssim 1$ for sufficiently large $k$. Therefore, we have by Lemma H.3 that

$$
\frac{1}{L^2} \sum_{l \neq l'} \mathbb{E}_{\mathbb{Q}} \left[ \psi(y_1, \langle w_{m,l}, z_1 \rangle) \cdot \psi(y_1, \langle w_{m,l'}, z_1 \rangle) \cdot \langle z_1, v \rangle^2 \cdot \left( 1 + \sum_{s=s^\star}^{\infty} \zeta_s(y) h_s(\langle \theta^\star, z \rangle) \right) \right] \lesssim \epsilon_2^{s^\star - 1}.
$$

On the other hand, we have for the third term in Eq. (F.27) that

$$
\frac{1}{L^2} \sum_{l \neq l'} \mathbb{E}_{\mathbb{Q}}[\widehat{\psi}_1(y)^2] \langle w_{m,l}, v \rangle \cdot \langle w_{m,l'}, v \rangle \lesssim \sup_{l} |\langle w_{m,l'}, v \rangle|^2 \cdot \mathbb{1}\{s^\star \leq 2\}
$$

$$
\lesssim \epsilon_0^2 \cdot \mathbb{1}\{s^\star \leq 2\}
$$

$$
\lesssim \epsilon_2 \cdot \mathbb{1}\{s^\star \leq 2\},
$$

where the first line holds by Lemma F.3.

In summary, we have on the event $\mathcal{E}_m(\epsilon) \cap \widetilde{\mathcal{E}}_m$ that

$$
\sup_{v \in \{e_1, e_2, \ldots, e_d\}} n \operatorname{Var}_{\mathbb{P}_{\theta^\star}}[\langle g_m, v \rangle] \lesssim \epsilon_2^{s^\star - 1} + \frac{1}{L} = \left( \gamma^2 + \epsilon^2 \log(k) \right)^{s^\star - 1} + \frac{1}{L}.
$$

**Concentration .** We now turn to validate the condition of Lemma J.3. For any $v \in \{e_1, e_2, \ldots, e_d\}$, we set $G(z, y, w) = |\psi(y, \langle w, z \rangle) \cdot \langle z, v \rangle| + |\widehat{\psi}_1(y) \cdot \langle w, v \rangle|$ with the domain measure defined as $d\mathbb{P}_{\theta^\star}(z_1, y_1) \times d\mu(w)$, where $d\mu(w) = L^{-1} \sum_l \delta_{w_{m,l}}$, the integral Minkowski's inequality implies

that

$$\mathbb{E}_{\mathbb{P}_{\theta^\star}}[|\langle g_{m,1}, v\rangle|^r]^{1/r} = \Big(\int d\mathbb{P}_{\theta^\star}(y,z)\Big(\int d\mu(w)|G(z,y,w)|\Big)^r\Big)^{1/r}$$

$$\leq \int d\mu(w)\Big(\int d\mathbb{P}_{\theta^\star}(y,z)|G(z,y,w)|^r\Big)^{1/r}$$

$$= \frac{1}{L}\sum_{l=1}^{L}\mathbb{E}_{\mathbb{P}_{\theta^\star}}[|\psi(y_i, \langle w_{m,l}, z_i\rangle)\cdot\langle z_i, v\rangle|^r]^{1/r} + \frac{1}{L}\sum_{l}|\langle w_{m,l}, v\rangle|. \quad \text{(F.28)}$$

To proceed, we leverage Cauchy-Schwarz inequality to decouple the average of the product in the first term, which reads

$$\frac{1}{L}\sum_{l=1}^{L}\mathbb{E}_{\mathbb{P}_{\theta^\star}}[|\psi(y_i, \langle w_{m,l}, z_i\rangle)\cdot\langle z_i, v\rangle|^r]^{1/r} \leq \frac{1}{L}\sum_{l=1}^{L}\mathbb{E}_{\mathbb{P}_{\theta^\star}}[|\psi(y_i, \langle w_{m,l}, z_i\rangle)|^{2r}]^{1/2r}\cdot\mathbb{E}_{\mathbb{P}_{\theta^\star}}[|\langle z_i, v\rangle|^{2r}]^{1/2r}.$$

Similar to the proof of Proposition E.4, we have that

$$\mathbb{E}_{\mathbb{P}_{\theta^\star}}[\psi(y, \langle w_{m,l}z,\rangle)^{2r}] \leq \mathbb{E}_{\mathbb{Q}}[U_{\langle\theta^\star, w_{m,l}\rangle}\Big(\frac{\mathbb{P}(x,y)}{\mathbb{Q}(x,y)}\Big)^2]^{1/2}\cdot\mathbb{E}_{\mathbb{Q}}[\psi(y,x)^{4r}]^{1/2} \lesssim r^{C_p 4r},$$

where the first inequality exactly repeats Eq. (E.12) and the second inequality holds by Assumption 4.1(c). On the other hand, we have that $\mathbb{E}_{\mathbb{P}_{\theta^\star}}[\langle z_i, v\rangle^{2r}]^{1/2r} \leq r^{1/2}$. Since the second term in Eq. (F.28) is bounded by $O(1)$, we conclude that

$$\mathbb{E}_{\mathbb{P}_{\theta^\star}}[|\langle g_{m,1}, v\rangle|^r]^{1/r} \lesssim r^{C_p + 1/2}.$$

Thus, Lemma J.3 implies that there exists a $\{(z_i, y_i)\}_{i\in[n]}$-measurable event $\mathcal{E}_{m,2}$ with probability at least $1 - O(d^{-c-1}/T)$, on which for any $v \in \{e_1, e_2, \ldots, e_d\}$, it holds that

$$\big|\langle g_m, v\rangle - \mathbb{E}_{\mathbb{P}_{\theta^\star}}[\langle g_m, v\rangle]\big| \lesssim \sqrt{\frac{\mathbb{E}_{\mathbb{P}_{\theta^\star}}[\langle g_{m,1}, v\rangle^2]\cdot\log(d^{c+1}T)}{n}} + \frac{\log(d^{c+1}T)\cdot\log(d^{c+1}Tn)^{C_p + 1/2}}{n}$$

$$\lesssim \sqrt{\frac{\big((\gamma^2 + \epsilon^2\log(k))^{s^\star - 1} + L^{-1}\big)\cdot\log(d)}{n}} + \frac{\log(d)^{C_p + 3/2}}{n},$$

given that $T, n$ are at most of polynomial rate in $d$. Since we assume that

$$n = \Omega\Big(\big((\gamma^2 + \epsilon^2\log(k))^{s^\star - 1} + L^{-1}\big)^{-1}\cdot\log(d)^{2C_p + 2}\Big),$$

the above inequality can be further simplified as

$$\big|\langle g_m, v\rangle - \mathbb{E}_{\mathbb{P}_{\theta^\star}}[\langle g_m, v\rangle]\big| \lesssim \sqrt{\frac{\big((\gamma^2 + \epsilon^2\log(k))^{s^\star - 1} + L^{-1}\big)\cdot\log(d)}{n}}.$$

Additionally, $\mathcal{E}_{m,2}$ is the desired event. This concludes the proof of Proposition F.6. $\qquad\square$

## F.5 Proofs for Technical Results in the Sparse Case

*Proof of Lemma F.1.* With slightly abuse of notation, we assume that $\theta^\star \sim \text{Unif}(\mathbb{S}^{k-1})$. We first consider the event $\mathcal{E}_{0,\infty} = \{\|\theta^\star\|_\infty \leq C\cdot k^{-1/2}\log(k)^{1/2}\}$. From the proof of Lemma J.6, we see that

$$\mathbb{P}(\|\theta^\star\|_\infty \geq t) \leq 2k\cdot\mathbb{P}(\theta_1^\star \geq t) \leq 2k\exp(-k/16) + 2k\exp(-t^2 k/4).$$

Take $t = C\cdot k^{-1/2}\log(k)^{1/2}$, we have that the failure probability is upper bounded by $2k\exp(-k/16) + 2k^{1-C^2/4}$.

For the $r$-norm, we leverage the property that $\theta^\star \overset{d.}{=} Z/\|Z\|_2$ where $Z \sim \mathcal{N}(0, I_k)$. Now, $\|Z\|_2^2 = \sum_{i\leq k} Z_i^2$, where $Z_i^2 - \mathbb{E}[Z_i^2] \geq -1$. Applying one-sided Bernstein's inequality with failure probability $k^{-c_0}$, we have that

$$\|Z\|_2^2 \leq k + \sqrt{2c_0 k\log(k)} + c_0\log(k)/3.$$

On the other hand, note that we have for $c_1 > 2$ that

$$\mathbb{P}(\max_{i \leq k} |Z_i| > \sqrt{2c_1 \log k}) \leq k \cdot \mathbb{P}(|Z_1| > \sqrt{2c_1 \log k}) \leq k \exp\{-c \log k\} = k^{1-c_1},$$

To apply Bernstein's inequality, we note that

$$\mathbb{E}[|Z_i|^r \cdot \mathbb{1}\{|Z_1| \leq \sqrt{2c_1 \log k}\}] \leq \mathbb{E}[|Z_i|^{2r}]^{1/2};$$
$$\mathbb{E}[|Z_i|^{2r} \cdot \mathbb{1}\{|Z_1| \leq \sqrt{2c_1 \log k}\}] \leq \mathbb{E}[|Z_i|^{2r}],$$

where $\mathbb{E}[Z_i^{2r}] = (2r-1)!!$. Therefore, it holds by truncated Bernstein's inequality that

$$\mathbb{P}\Big(\|Z\|_r^r > (k + \sqrt{2c_2 k \log(k)}) \cdot \mathbb{E}[|Z_1|^{2r}]^{1/2} + \big(\sqrt{2C \log(k)}\big)^r \cdot c_2 \log(k)/3\Big) \leq k^{1-c_1} + k^{-c_2}$$

Combining the two bounds, we conclude that with probability $1 - O(k^{1-c_0 \vee c_1 \vee c_2})$, it holds that

$$\|\theta^\star\|_r^r \overset{d.}{=} \frac{\|Z\|_r^r}{\|Z\|_2^r} \lesssim \frac{k + \sqrt{k \log k} + (\log k)^{1+r/2}}{(k + \sqrt{k \log k} + \log k)^{r/2}} \lesssim k^{1-r/2},$$

with probability at least $1 - O(k^{-c})$ for some constant $c > 0$.

We now move on to consider the event $\mathcal{E}_{0,\sharp} = \big\{ \sum_{i \leq k} \mathbb{1}\{|\theta_i^\star| \geq 1/\sqrt{2k}\} \geq k/4 \big\}$. First, it holds by the Hoeffding's inequality that

$$\mathbb{P}\Big(\Big| \sum_{i \leq k} \mathbb{1}\{Z_i^2 \geq 1/2\} - kp \Big| \leq \frac{kp}{2}\Big) \geq 1 - 2\exp(-2p^2 k),$$

where $p = \mathbb{P}(Z_1^2 \geq 3/4) > 0.5$. Denote above event as $\mathcal{A}_1$. On the other hand, we have by the Bernstein's inequality that

$$\mathbb{P}\Big(\Big| k^{-1} \sum_{i \leq k} Z_i^2 - 1 \Big| \leq 1/2\Big) \geq 1 - 2\exp\{-k/32\}.$$

Denote above event as $\mathcal{A}_2$. Then on the event $\mathcal{A}_1 \cap \mathcal{A}_2$, we have that

$$\sum_{i \leq k} \mathbb{1}\Big\{ \frac{Z_1}{\|Z\|} > \frac{1}{\sqrt{2k}}\Big\} = \sum_{i \leq k} \mathbb{1}\Big\{ Z_i^2 > \frac{1}{2k} \sum_{i \leq k} Z_i^2 \Big\}$$
$$\overset{\mathcal{A}_2}{\geq} \sum_{i \leq k} \mathbb{1}\{ Z_i^2 > \frac{3}{4}\}$$
$$\overset{\mathcal{A}_1}{\geq} \frac{kp}{2} > k/4.$$

In conclusion we have that $\mathbb{P}(|\{i : |\theta_i^\star| > 1/\sqrt{2k}\}| > k/4) \geq \mathbb{P}(\mathcal{A}_1 \cap \mathcal{A}_2) \geq 1 - \exp\{-c_3 k\}$ for some constant $c_3 > 0$.

$\square$

*Proof of Lemma F.3.* Clearly, it holds that

$$\|\gamma\theta + \xi_{m,l}\|_2 \geq \|\xi_{m,l}\|_2 - \gamma \cdot \|\xi_{m,l}\|_2 \geq 1/2.$$

Bu substituting this lower bound for the denominator, we have for any $j, l$ that

$$|\langle w_{m,l}, e_j \rangle| \leq 2(\gamma|\theta_j| + |\langle \xi_{m,l}, e_j \rangle|)$$
$$\leq 2(\gamma|\theta_j| + \epsilon).$$

The last line holds by the definition of $\mathcal{E}_m(\epsilon)$. On the other hand, we have for any $l$ that

$$|\langle w_{m,l}, \theta^\star \rangle| \leq 2(\gamma|\rho| + \sum_{j \in [d]} \xi_{m,l,j} \cdot \theta_j^\star \cdot \mathbb{1}\{j \in \phi^\star \cap \phi_{m,l}\})$$
$$\leq 2\gamma|\rho| + 2\big( \sum_j \xi_{m,l,j}^2 \cdot \mathbb{1}\{j \in \phi^\star \cap \phi_{m,l}\}\big)^{1/2} \cdot \big( \sum_j \theta_j^{\star 2} \cdot \mathbb{1}\{j \in \phi^\star \cap \phi_{m,l}\}\big)^{1/2}$$
$$\leq 2\gamma|\rho| + 2\sup_j |\xi_{m,l,j}| \cdot \|\theta^\star\|_\infty \cdot |\phi^\star \cap \phi_{m,l}|$$

To proceed, note that on the event $\widetilde{\mathcal{E}}_m \cap \mathcal{E}_m(\epsilon)$, it holds that $|\phi^\star \cap \phi_{m,l}| \le \log k$ and that $\sup_j |\xi_{m,l,j}| \le \epsilon$. Since we assume that $\|\theta^\star\|_\infty \le 1/\log k$, it holds that

$$|\langle w_{m,l}, \theta^\star \rangle| \le 2(\gamma|\rho| + \epsilon).$$

Now we turn to consider the correlation between $w_{m,l}$ and $w_{m,l'}$.

$$
\begin{aligned}
|\langle w_{m,l}, w_{m,l'} \rangle| \le 2\Big(&\gamma^2 + \sum_j |\xi_{m,l,j}| \cdot |\xi_{m,l',j}| \cdot \mathbb{1}\{j \in \phi_{m,l} \cap \phi_{m,l'}\} \\
&+ \gamma \sum_j |\theta_j| \cdot |\xi_{m,l,j}| \cdot \mathbb{1}\{j \in \phi_{m,l} \cap \mathsf{supp}(\theta)\} \\
&+ \gamma \sum_j |\theta_j| \cdot |\xi_{m,l',j}| \cdot \mathbb{1}\{j \in \phi_{m,l'} \cap \mathsf{supp}(\theta)\}\Big).
\end{aligned}
$$

For the second term, we have with the definition of $\mathcal{E}_m(\epsilon)$ that

$$
\sum_j |\xi_{m,l,j}| \cdot |\xi_{m,l',j}| \cdot \mathbb{1}\{j \in \phi_{m,l} \cap \phi_{m,l'}\} \le \max_{j,l} |\xi_{m,l,j}|^2 \cdot |\phi_{m,l} \cap \phi_{m,l'}|
$$

$$
\le \epsilon^2 \log k.
$$

For the third term, applying the Cauchy-Schwarz inequality, we have that

$$
\gamma \sum_j |\theta_j| \cdot |\xi_{m,l',j}| \cdot \mathbb{1}\{j \in \phi_{m,l} \cap \phi_{m,l'}\} \le \gamma \|\theta\|_2 \cdot \epsilon \sqrt{\log k} = \le \gamma^2 + \epsilon^2 \log k.
$$

Putting them together, we have that

$$|\langle w_{m,l}, w_{m,l'} \rangle| \le 4(\gamma^2 + \epsilon^2 \log k).$$

This concludes the proof of Lemma F.3. $\qquad\qquad\square$

*Proof of Proposition F.4.* For conciseness, we momentarily drop the subscript $m, l$ in the following analysis. Conditioning on fixed $\phi$, we have that

$$
\begin{aligned}
\mathbb{E}_w[\langle w, \theta^\star \rangle^s] &= \mathbb{E}_w\big[\|\gamma\theta + \xi\|_2^{-s} \cdot \big(\gamma\langle\theta, \theta^\star\rangle + \langle\xi, \theta^\star\rangle\big)^s\big] \\
&= \mathbb{E}_\phi\big[\mathbb{E}_w\big[\|\gamma\theta + \xi\|_2^{-s} \cdot \big(\gamma\rho + \langle\xi, P_\phi\theta^\star\rangle\big)^s \,\big|\, \phi\big]\big]. \tag{F.29}
\end{aligned}
$$

Given the polarization level $\gamma = o(1)$, we see that $\|\gamma\theta + \xi\|_2^{s+1} \simeq 1 \pm o(1)$, and it suffices to evaluate $\mathbb{E}_w\big[\big(\gamma\rho + \langle\xi, P_\phi\theta^\star\rangle\big)^s \,\big|\, \phi\big]$. Without loss of generality, we assume that $1 \in \phi$ and we can translate $P_\phi\theta^\star$ into the first coordinate by the isotropy of $\xi$ over $\mathbb{S}^{k-1}(\phi)$. To this end, we can characterize the first term as follows:

$$
\begin{aligned}
\mathbb{E}\big[\big(\gamma\rho + \langle\xi, P_\phi\theta^\star\rangle\big)^s \,\big|\, \phi\big] &= \mathbb{E}\big[\big(\gamma\rho + \langle\xi, \|P_\phi\theta^\star\|_2 \cdot e_1\rangle\big)^s \,\big|\, \phi\big] \\
&= \sum_{r=0}^s \binom{2\lfloor s/2 \rfloor}{r}(\gamma\rho)^{s-r} \cdot \|P_\phi\theta^\star\|_2^r \cdot \mathbb{E}\big[\xi_1^r \,\big|\, \phi\big] \cdot \mathbb{1}\{r \text{ even}\} \\
&\overset{(i)}{\simeq} \sum_{r=0}^{\lfloor s/2 \rfloor} \binom{2\lfloor s/2 \rfloor}{2r}(\gamma\rho)^{2\lfloor s/2 \rfloor - 2r} \cdot \|P_\phi\theta^\star\|_2^{2r} \cdot k^{-r} \cdot (\gamma\rho)^{\mathbb{1}\{s \text{ odd}\}} \\
&= (\gamma\rho)^{\mathbb{1}\{s \text{ odd}\}} \cdot \big((\gamma\rho + k^{-1/2}\|P_\phi\theta^\star\|_2)^{2\lfloor s/2 \rfloor} + (\gamma\rho - k^{-1/2}\|P_\phi\theta^\star\|_2)^{2\lfloor s/2 \rfloor}\big)/2 \\
&\simeq (\gamma\rho)^{\mathbb{1}\{s \text{ odd}\}} \cdot (\gamma|\rho| + k^{-1/2}\|P_\phi\theta^\star\|_2)^{2\lfloor s/2 \rfloor}. \tag{F.30}
\end{aligned}
$$

Here, (i) holds by applying Lemma I.5 and $\simeq$ denotes the equality that is up to a $s$-dependent multiplicative constant.

Putting together Eq. (F.29) and (F.30), we conclude that

$$
\mathbb{E}_w[\langle w, \theta^\star \rangle^s \,|\, \phi] \simeq (\gamma\rho)^{\mathbb{1}\{s \text{ odd}\}}(\gamma|\rho| + k^{-1/2}\|P_\phi\theta^\star\|_2)^{s - \mathbb{1}\{s \text{ odd}\}}. \tag{F.31}
$$

In the sequel, we consider averaging over $\phi$. From Eq. (F.31), we see that it suffices to consider $\mathbb{E}_\phi[(\gamma|\rho| + k^{-1/2}\|P_\phi\theta^\star\|_2)^r]$ for some $r \geq 2$. We alter the notation to facilitate some deferred calculation. Consider $\mathbf{m} \subset [d]$ with constant size $|\mathbf{m}| = O(1)$ that does not scale with $k$ or $d$. Now define $\phi_\mathbf{m} \sim \mathrm{Unif}\{\mathcal{S}_{k,\mathbf{m}}\}$, where $\mathcal{S}_{k,\mathbf{m}} = \{S \subset [d] : |S| = k, \mathbf{m} \subset S\}$. It is easily seen that this definition covers previous definition of $\mathcal{S}_{k,m}$ by setting $\mathbf{m} = \{m\}$. We characterize the magnitude of $\mathbb{E}_{\phi_\mathbf{m}}[\|P_{\phi_\mathbf{m}}\theta^\star\|_2^r]$ from both sides as follows. For the lower bound, we have that

$$
\begin{aligned}
\mathbb{E}_{\phi_\mathbf{m}}[\|P_{\phi_\mathbf{m}}\theta^\star\|_2^r] &= \mathbb{E}_{\phi_\mathbf{m}}\left[\Big(\|\theta_\mathbf{m}^\star\|_2^2 + \sum_{j\notin\mathbf{m}}|\theta_j^\star|^2\, \mathbb{1}\{j \in \phi_\mathbf{m}\}\Big)^{r/2}\right] \\
&\geq \mathbb{E}_{\phi_\mathbf{m}}\left[\|\theta_\mathbf{m}^\star\|_r^r + \sum_{j\notin\mathbf{m}}|\theta_j^\star|^r\, \mathbb{1}\{j \in \phi_\mathbf{m}\}\right] \\
&\overset{(i)}{\simeq} (1 - k/d)\cdot\|\theta_\mathbf{m}^\star\|_r^r + \frac{k}{d}\cdot\|\theta^\star\|_r^r \\
&\overset{(ii)}{\gtrsim} \|\theta_\mathbf{m}^\star\|_r^r + \frac{k}{d}\cdot k^{1-r/2}\cdot\|\theta^\star\|_2^{r/2} \\
&= \|\theta_\mathbf{m}^\star\|_r^r + \frac{k^2}{d}\cdot k^{-r/2}.
\end{aligned}
$$

Here (i) holds by the fact that $\mathbb{E}[\mathbb{1}\{j \in \phi_\mathbf{m}\}] \simeq k/d$ for $j \notin \mathbf{m}$, and (ii) is a consequence of Jensen's inequality. For the upper bound, we have that

$$
\begin{aligned}
\mathbb{E}_{\phi_\mathbf{m}}[\|P_\phi\theta^\star\|_2^r] &= \mathbb{E}_{\phi_\mathbf{m}}\left[\Big(\|\theta_\mathbf{m}^\star\|_2^2 + \sum_{j\notin\mathbf{m}}|\theta_j^\star|^2\cdot\mathbb{1}\{j \in \phi_\mathbf{m}\}\Big)^{r/2}\right] \\
&\lesssim \mathbb{E}_{\phi_\mathbf{m}}\left[\|\theta_\mathbf{m}^\star\|_r^r + \Big(\underbrace{\sum_{j\notin\mathbf{m}}|\theta_j^\star|^2\cdot\mathbb{1}\{j \in \phi_\mathbf{m}\}}_{|(\phi_\mathbf{m}\cap\phi^\star)\setminus\{m\}|\text{ nonzero summands}}\Big)^{r/2}\right] \\
&\overset{\text{Jensen}}{\lesssim} \|\theta_\mathbf{m}^\star\|_r^r + \mathbb{E}_{\phi_\mathbf{m}}\left[|(\phi_\mathbf{m}\cap\phi^\star)\setminus\mathbf{m}|^{r/2-1}\cdot\Big(\sum_{j\notin\mathbf{m}}|\theta_j^\star|^r\cdot\mathbb{1}\{j \in \phi_\mathbf{m}\}\Big)\right]. \quad \text{(F.32)}
\end{aligned}
$$

Next, we apply Cauchy-Schwarz inequality as follows:

$$
\begin{aligned}
\text{(F.32)} &= \|\theta_\mathbf{m}^\star\|_r^r + \mathbb{E}_{\phi_\mathbf{m}}\left[\sum_{j\notin\mathbf{m}}|\theta_j^\star|^r\cdot\mathbb{1}\{j \in \phi_\mathbf{m}\}^2\cdot|(\phi_\mathbf{m}\cap\phi^\star)\setminus\mathbf{m}|^{r/2-1}\right] \\
&\leq \|\theta_m^\star\|_r^r + \mathbb{E}_{\phi_\mathbf{m}}\left[\sum_{j\notin\mathbf{m}}|\theta_j^\star|^{2r}\cdot\mathbb{1}\{j \in \phi_\mathbf{m}\}\right]^{1/2}\cdot\mathbb{E}_{\phi_\mathbf{m}}\left[\sum_{j\notin\mathbf{m}}\mathbb{1}\{j \in \phi_\mathbf{m}\}\cdot|(\phi_\mathbf{m}\cap\phi^\star)\setminus\mathbf{m}|^{r-2}\right]^{1/2} \\
&= \|\theta_\mathbf{m}^\star\|_r^r + \Big(\frac{k}{d}\cdot\sum_{j\notin\mathbf{m}}|\theta_j^\star|^{2r}\Big)^{1/2}\cdot\mathbb{E}_{\phi_\mathbf{m}}\left[|(\phi_\mathbf{m}\cap\phi^\star)\setminus\mathbf{m}|^{r-1}\right]^{1/2} \\
&\overset{(i)}{\lesssim} \|\theta_\mathbf{m}^\star\|_r^r + \Big(\frac{k}{d}\cdot k^{1-r}\Big)^{1/2}\cdot\Big(\frac{k^2}{d}\Big)^{(r-1)/2} \\
&= \|\theta_\mathbf{m}^\star\|_r^r + \frac{k^2}{d}\cdot k^{-r/2},
\end{aligned}
$$

where (i) holds by $\mathcal{E}_{0,2r}$ and Lemma J.7. In conclusion, we have that $\mathbb{E}_{\phi_\mathbf{m}}[\|P_\phi\theta^\star\|_2^r] \simeq \|\theta_\mathbf{m}^\star\|_r^r + k^{-r/2}\cdot\delta$ given that $k = o(\sqrt{d})$. Combining this result with Eq. (F.31), we obtain that

$$
\mathbb{E}_w[\langle w, \theta^\star\rangle^s] \simeq (\gamma\rho)^{\mathbb{1}\{s\text{ odd}\}}(\gamma|\rho| + k^{-1/2}|\theta_m^\star| + k^{-1}\delta_{1/(s-\mathbb{1}\{s\text{ odd}\})})^{s-\mathbb{1}\{s\text{ odd}\}}
$$

where $\delta = k^2/d = o(1)$ and $\delta_r = \delta^{1/r}$. $\qquad\square$

# G  STATISTICAL QUERY LOWER BOUND FOR SPARSE SIGNAL RECOVERY

In this section, we provide a $k^{s^\star}$ sample complexity lower bound for the single index model with $k$-sparse signal when querying a VSTAT oracle. The statistical query (SQ) framework was developed in Feldman et al. (2017) and for completeness, we present essential definition and results here.

**Definition G.1** (VSTAT Oracle). *Let $D^\star$ be the input distribution over domain $\mathcal{X}$. For a sample size parameter $n > 0$, $\mathrm{VSTAT}(D^\star, n)$ oracle is the oracle that for any query function $h : \mathcal{X} \to [0, 1]$, returns a value $v \in [p - \tau, p + \tau]$, where $p = \mathbb{E}_{x \sim D^\star}[h(x)]$ and $\tau = \max\{t^{-1}, \sqrt{p(1-p)/n}\}$.*

To define a key concept *statistical query dimension*, we first introduce the following notation.

**Definition G.2** (Relative Pairwise Correlation). *Given two distributions $D_1, D_2 \in \Delta(\mathcal{X})$ and a reference distribution $D \in \Delta(\mathcal{X})$,*

$$\chi_D(D_1, D_2) = \mathbb{E}_{x \sim D}\left[\frac{D_1(x)}{D(x)} \cdot \frac{D_2(x)}{D(x)}\right] - 1.$$

**Definition G.3** (Statistical Dimension). *For $\bar{\gamma} > 0$, $\eta \in (0, 1)$, domain $\mathcal{X}$, a set of distributions $\mathcal{D}$ over $\mathcal{X}$, the **statistical dimension** $\mathrm{SDA}(\mathcal{D}, \bar{\gamma}, \eta)$ of $\mathcal{D}$ with average correlation $\bar{\gamma}$ and solution set bound $\eta$ is defined as the largest value $m'$ such that there exists a reference distribution $D \in \Delta(\mathcal{X})$ and a finite set of distributions $\mathcal{D}_D \subseteq \mathcal{D}$ which can depend on the reference $D$ with the following property: for any solution $D^\star \in \mathcal{D}$,*

*(i) $|\mathcal{D}_D \setminus \{D^\star\}| \geq (1 - \eta)|\mathcal{D}_D|$;*

*(ii) for any subset $\mathcal{D}'_D \subseteq \mathcal{D}_D \setminus \{D^\star\}$ such that $|\mathcal{D}'_D| \geq |\mathcal{D}_D \setminus \{D^\star\}|/m'$,*

$$\frac{1}{|\mathcal{D}'_D|^2} \sum_{D_i, D_j \in \mathcal{D}'_D} \chi_D(D_i, D_j) \leq \bar{\gamma}.$$

The above definition of the statistical dimension is a speical case of the original Definition 3.1 in Feldman et al. (2017) where we consider a search problem of *exact recovery* of the ground truth $D^\star$.

**Definition G.4** (($\gamma, \beta$)-correlated Distributions). *We say that a set of $m$ distributions $\mathcal{D} = \{D_1, \ldots, D_m\}$ over $\mathcal{X}$ is $(\gamma, \beta)$-correlated relative to a reference distribution $D \in \Delta(\mathcal{X})$ if:*

$$\chi_D(D_i, D_j) \leq \begin{cases} \beta & \text{for } i = j \in [m] \\ \gamma & \text{for } i \neq j \in [m]. \end{cases}$$

The following lemma borrowed from Lemma 3.10 of Feldman et al. (2017) provides a lower bound on the statistical dimension in terms of the $(\gamma, \beta)$-correlation property of the set of candidate distributions.

**Lemma G.5.** *Given a set of candidate distributions $\mathcal{D}$ that are $(\gamma, \beta)$-correlated with respect to a reference distribution $D$, then for any $\gamma' > 0$ and $\eta > |\mathcal{D}|^{-1}$,*

$$\mathrm{SDA}(\mathcal{D}, \gamma + \gamma', \eta) \geq \frac{(|\mathcal{D}| - 1)\gamma'}{\beta - \gamma}.$$

The main result in the SQ framework is the following statement that relates the number of queries required to the statistical dimension, which is borrowed from Theorem 3.2 of Feldman et al. (2017).

**Lemma G.6.** *Let $\mathcal{X}$ be a domain and $\mathcal{D}$ be a set of candidate distributions over $\mathcal{X}$. For any $\bar{\gamma} > 0$ and $\eta \in (0, 1)$, Any randomized SQ algorithm that solves the problem of finding the input distribution $D^\star \in \mathcal{D}$ with probability at least $\alpha > \eta$ requires at least $(\alpha - \eta)/(1 - \eta) \cdot \mathrm{SDA}(\mathcal{D}, \bar{\gamma}, \eta)$ calls to the $\mathrm{VSTAT}(D^\star, (3\bar{\gamma})^{-1})$ oracle.*

Our strategy for proving the lower bound is to first construct a set of candidate distributions $\mathcal{D}$ that are $(\omega(k^{-1}), \beta)$-correlated with respect to reference distribution $\mathbb{Q}$ with $\beta = D_{\chi^2}(\mathbb{P}_{\theta^\star} \| \mathbb{Q})$ and $|\mathcal{D}|$ exponentially large. Then by Lemma G.5 and Lemma G.6, we can derive the desired hardness result. It remains to construct the set of candidate distributions $\mathcal{D}$ that are $(\omega(k^{-1}), \beta)$-correlated with respect to $\mathbb{Q}$. To this end, we introduce the following result on the packing number of $k$-sparse vectors.

**Lemma G.7** (Packing Number for $k$-Sparse Vectors). *Define $\rho(u, v) = |\langle u, v \rangle|$. Let packing number $\mathcal{M}_\rho(d, k, t)$ be the maximal cardinality of the set of $k$-sparse vectors in $\mathbb{S}^{d-1}$ such that $\rho(u, v) < t$ for any $u \neq v$ in the set. We have for any $t \in (1/k, 1)$ that*

$$\mathcal{M}_\rho(d, k, t) \geq \frac{1}{2} \cdot \exp\left(\frac{\min\{(d-k)t^2, 3kt\}}{8}\right).$$

With all these ingredients in place, we are ready to prove the main theorem.

*Proof of Theorem 5.4.* Let us pick parameter $\kappa_d \in ((\log d)^2, k/4)$ that scales with $d$ and set

$$t \geq \max\left\{ \sqrt{\frac{\kappa_d}{d-k}}, \frac{\kappa_d}{3k} \right\} \in (1/k, 1/2). \tag{G.1}$$

Note that $t \in (1/k, 1/2)$ is able to hold by our choice of $\kappa_d$ and condition that $\omega((\log d)^2) \leq k \leq d/2$. In this vein, we can pick $\mathcal{D}$ to be the *maximal* set of distributions $\mathbb{P}_\theta$ for some $k$-sparse vectors $\theta \in \mathbb{S}^{d-1}$ satisfying $\rho(\theta, \theta') < t$ for any $\theta \neq \theta'$ in the set. It follows from plugging (G.1) into Lemma G.7 that $|\mathcal{D}| \geq \exp(\kappa_d/8)/2$, which is super polynomially large in $d$ for our choice of $\kappa_d$.

Next, we configure the remaining parameters in Lemma G.5 and Lemma G.6. We choose the reference distribution to be $\mathbb{Q}$, in which the covariate $z$ is independent of the output $y$. For $\beta$, we note that

$$\chi_\mathbb{Q}(\mathbb{P}_\theta, \mathbb{P}_\theta) = D_{\chi^2}(\mathbb{P}_\theta \,\|\, \mathbb{Q}) = O(1),$$

which is a constant independent of $\theta$ due to the rotational invariance of the likelihood ratio with respect to $\theta$. Thus, we define this quantity as $B$ can just set $\beta = D_{\chi^2}(\mathbb{P}_{\theta^\star} \,\|\, \mathbb{Q}) = B$. For $\gamma$, we note that for any two $\mathbb{P}_\theta, \mathbb{P}_{\theta'}$ in $\mathcal{D}$ for $\theta \neq \theta'$,

$$|\chi_\mathbb{Q}(\mathbb{P}_\theta, \mathbb{P}_{\theta'})| = \left| \mathbb{E}_{x\sim\mathbb{Q}}\left[ \frac{\mathbb{P}_\theta(x)}{\mathbb{Q}(x)} \cdot \frac{\mathbb{P}_{\theta'}(x)}{\mathbb{Q}(x)} \right] - 1 \right|$$

$$= \left| \mathbb{E}_{x\sim\mathbb{Q}}\left[ \left( 1 + \sum_{s \geq s^\star} \zeta_s(y) h_s(\langle\theta, z\rangle) \right) \cdot \left( 1 + \sum_{s' \geq s^\star} \zeta_{s'}(y) h_{s'}(\langle\theta', z\rangle) \right) \right] - 1 \right|$$

$$= \sum_{s \geq s^\star} \mathbb{E}_\mathbb{Q}[\zeta_s(y)^2] \cdot |\langle\theta, \theta'\rangle|^s \leq \sum_{s \geq s^\star} \mathbb{E}_\mathbb{Q}[\zeta_s(y)^2] \cdot t^s \leq \mathbb{E}_\mathbb{Q}[\zeta_{s^\star}(y)^2] \cdot t^{s^\star} + \frac{t^{s^\star+1}}{1-t}.$$

Here, the third equality follows from the fact that only when $s = s'$, the cross term $\mathbb{E}_\mathbb{Q}[h_s(\langle\theta, z\rangle) h_s(\langle\theta', z\rangle)]$ is non-zero. In particular, by the property of the Gaussian noise operator introduced in (B.3), we have that $\mathbb{E}_\mathbb{Q}[h_s(\langle\theta, z\rangle) h_s(\langle\theta', z\rangle)] = \langle\theta, \theta'\rangle^s < t^s$. For the last inequality above, we simply use the fact that $\mathbb{E}_\mathbb{Q}[\zeta_s(y)^2] \leq 1$ for any $s$ (Damian et al., 2024) and $t < 1$. Now, we conclude that

$$|\chi_\mathbb{Q}(\mathbb{P}_\theta, \mathbb{P}_{\theta'})| \leq \left( \mathbb{E}_\mathbb{Q}[\zeta_{s^\star}(y)^2] + \frac{t}{1-t} \right) \cdot t^{s^\star} \leq \left( \mathbb{E}_\mathbb{Q}[\zeta_{s^\star}(y)^2] + 1 \right) \cdot t^{s^\star}.$$

We thus set $\gamma' = \gamma = \left( \mathbb{E}_\mathbb{Q}[\zeta_{s^\star}(y)^2] + 1 \right) \cdot t^{s^\star} = \Theta(t^{s^\star})$. Finally, we set $\eta = 1/3$ and $\alpha = 2/3$. Then all the conditions in both Lemma G.5 and Lemma G.6 are satisfied and we have

$$\mathrm{SDA}(\mathcal{D}, 2\gamma, 1/3) \geq \frac{(|\mathcal{D}|-1)\gamma}{\beta - \gamma} \geq \frac{|\mathcal{D}|\gamma}{2\beta} \geq \frac{\gamma \exp(\kappa_d/8)}{4\beta}.$$

Lastly, recall that we have $|\langle\theta, \theta'\rangle| \leq t$ for any $\theta \neq \theta'$ in $\mathcal{D}$, which means that in order to achieve alignment at least $2t$ with the true signal $\theta^\star$, we need to *exactly* identify the distribution $\mathbb{P}_{\theta^\star}$. Consequently, by Lemma G.6, we have that any randomized SQ algorithm that solves the problem of achieving alignment $2t$ with probability at least $2/3$ requires at least $\gamma \exp(\kappa_d/8)/(8B)$ calls to the $\mathrm{VSTAT}(\mathbb{P}_{\theta^\star}, (6\gamma)^{-1})$ oracle.

**Simplification of the lower bound.** To simplify the lower bound, let us take $\kappa_d = (\log d)^c/2$ for some constant $c > 2$. Thus, the alignment $2t$ is upper bounded by

$$2t \leq \begin{cases} \widetilde{\omega}(k^{-1}) & \text{if } k < \sqrt{d} \\ \widetilde{\omega}(d^{-1/2}) & \text{if } k \geq \sqrt{d} \end{cases},$$

where $\widetilde{\omega}(\cdot)$ hides some poly-logarithmic factors. The number of queries is still super polynomially large in $d$. Following from (G.1), we can safely set

$$t = \begin{cases} (\log d)^c/k & \text{if } (\log d)^2 < k < \sqrt{d(\log d)^c} \\ \sqrt{(\log d)^c/d} & \text{if } \sqrt{d(\log d)^c} \leq k \leq d/2 \end{cases},$$

Hence, the number of sample

$$(6\gamma)^{-1} = \frac{t^{-s^\star}}{6} \simeq \begin{cases} \frac{k^{s^\star}}{(\log d)^{cs^\star}} & \text{if } (\log d)^2 < k < \sqrt{d(\log d)^c} \\ \frac{d^{s^\star/2}}{6(\log d)^{cs^\star/2}} & \text{if } \sqrt{d(\log d)^c} \leq k \leq d/2 \end{cases}.$$

Hence, we have established the desired lower bound on the sample complexity. $\square$

**Remark G.8** (Difference between CSQ and SQ lower bound). *The following is a comparison between our SQ lower bound and the CSQ lower bound in* Vural & Erdogdu (2024):

- **Difference**: *CSQ is constrained to operate on a specific set of functions, whereas SQ is not. In this sense, CSQ creates a large family of functions with small average correlation, while SQ constructs a large covering net over the support of $\theta$ as a hard instance, allowing for arbitrary query forms.*

- **Similarity**: *Both methods rely on Bernstein concentration inequalities on the correlation level for two random sparse vectors. This leads to the critical correlation threshold $\max\{1/\sqrt{d}, 1/k\}$, where $d$ is the dimensionality and $k$ is the sparsity level.*

## G.1    PROOF OF TECHNICAL LEMMAS FOR SQ LOWER BOUND

*Proof of Lemma G.7.* We use the probability method to prove the existence of a set of $k$-sparse vectors in $\mathbb{S}^{d-1}$ with the desired property. We i.i.d. sample $m$ vectors $\omega^{(1)}, \ldots, \omega^{(m)}$ from the following distribution:

$$\omega: \quad \phi \sim \text{Unif}(\mathcal{S}_k), \quad \omega_j = \begin{cases} \frac{1}{\sqrt{k}}, & \text{w.p. } \frac{1}{2} \text{ if } j \in \phi \\ -\frac{1}{\sqrt{k}}, & \text{w.p. } \frac{1}{2} \text{ if } j \in \phi, \quad j \in [d]. \\ 0, & j \notin \phi. \end{cases}$$

where we recall that $\mathcal{S}_k$ is the set of all size-$k$ subsets in $[d]$. Since each $\omega^{(i)}$ is i.i.d. sampled, we can equivalently view $\langle \omega^{(i)}, \omega^{(j)} \rangle$ for $i \neq j$ as a random variable sampled from the following distribution:

$$\langle \omega^{(i)}, \omega^{(j)} \rangle \stackrel{d}{=} \frac{R_X}{k}, \quad \text{where} \quad R_X = r_1 + \ldots, r_X, \quad X \sim \text{Hypergeometric}(d, k, k), \quad \text{(G.2)}$$

where $r_1, r_2, \ldots$ are i.i.d. Rademacher random variables. Let us consider random variable $W$ distributed as

$$W \stackrel{d}{=} \frac{R_Y}{k}, \quad \text{where} \quad R_Y = r_1 + \ldots + r_Y, \quad Y \sim \text{Binomial}\left(k, \frac{k}{d-k}\right). \quad \text{(G.3)}$$

We will invoke the following fact on the tail probability regarding the above two random variables.

**Proposition G.9.** *For $R_X$ and $R_Y$ defined in* (G.2) *and* (G.3)*, respectively, we have that $\mathbb{P}(R_X \geq t) \leq 2\mathbb{P}(R_Y \geq t)$ for any $t > 1$.*

The proof of the proposition is deferred to the end of the proof. Thus, it suffices to study the tail probability of $W$. Note that $W \stackrel{d}{=} \sum_{j=1}^k w_j$ where $w_j$ are i.i.d. sampled from

$$w_j = \begin{cases} \frac{1}{k}, & \text{w.p. } \frac{k}{2(d-k)} \\ -\frac{1}{k}, & \text{w.p. } \frac{k}{2(d-k)} \\ 0, & \text{w.p. } 1 - \frac{k}{d-k} \end{cases}, \quad j \in [k].$$

where $\mathbb{E}[w_j] = 0$ and $\mathbb{E}[w_j^2] = (k(d-k))^{-1}$. Hence, we can apply the Bernstein inequality to obtain that for any $t > 1/k$,

$$\mathbb{P}(\langle \omega^{(i)}, \omega^{(j)} \rangle \geq t) \leq 2\mathbb{P}(W \geq t) \leq 2\exp\left(-\frac{k(t/k)^2/2}{(k(d-k))^{-1} + t/(3k^2)}\right)$$

$$= 2\exp\left(-\frac{k^2 t^2}{2k^2/(d-k) + 2kt/3}\right) \leq 2\exp\left(-\min\left\{\frac{(d-k)t^2}{4}, \frac{3kt}{4}\right\}\right).$$

Suppose we randomly sample $m$ i.i.d. $\omega^{(i)}$ from the same distribution. Then the probability that all such pair $|\langle \omega^{(i)}, \omega^{(j)} \rangle| < t$ for $t > 1/k$ is lower bounded by

$$\mathbb{P}\big(|\langle \omega^{(i)}, \omega^{(j)} \rangle| < t, \forall i \neq j\big) \geq 1 - m^2 \cdot 2\mathbb{P}(\langle \omega^{(i)}, \omega^{(j)} \rangle \geq t)$$

$$\geq 1 - 4m^2 \cdot \exp\left(-\min\left\{\frac{(d-k)t^2}{4}, \frac{3kt}{4}\right\}\right).$$

Ensuring that the probability is nonzero will give us a valid construction of the set $\mathcal{D}$. Therefore, there must exist a $\mathcal{D}$ satisfying $|\langle \omega^{(i)}, \omega^{(j)} \rangle| < t$ for any $i \neq j$ and with size

$$|\mathcal{D}| \geq \frac{1}{2} \cdot \exp\left(\frac{\min\left\{(d-k)t^2, 3kt\right\}}{8}\right).$$

Hence, we complete the proof. $\qquad\square$

Next, we aim to present the proof of Proposition G.9. To proceed, let us introduce the definition of stochastic dominance.

**Definition G.10** (Stochastic Dominance). *For any real-valued random variable $X$ and $Y$, we say that $X$ is* stochastically dominated *by $Y$, denoted by $X \overset{\text{s.t.}}{\leq} Y$, if $\mathbb{P}(X \geq t) \leq \mathbb{P}(Y \geq t)$ for every $t$.*

The following result is from Theorem A, Chapter 2 of Szekli (2012).

**Proposition G.11.** *We have $X \overset{\text{s.t.}}{\leq} Y$ if and only if there exists a coupling $(\widehat{X}, \widehat{Y})$ with $\mathrm{law}(\widehat{X}) = \mathrm{law}(X)$ and $\mathrm{law}(\widehat{Y}) = \mathrm{law}(Y)$ such that $\widehat{X} \leq \widehat{Y}$ almost surely.*

**Proposition G.12** (Theorem 1.1, Klenke & Mattner (2010)). $\mathrm{Hypergeometric}(d, k, k) \overset{\text{s.t.}}{\leq} \mathrm{Binomial}(k, k/(d-k))$.

Another way to think of the problem is that $\mathrm{Hypergeometric}(d, k, k)$ corresponds to the number of times a black ball is drawn when sampling for $k$ times from an urn with $d - k$ white ball and $k$ black ball without replacement, while $\mathrm{Binomial}(k, k/(d-k))$ corresponds to sampling in the same urn but with replacement. We claim the following fact on the tail probability of sum of Rademacher random variables.

**Proposition G.13** (Sum of Rademacher Random Variables). *Let $r_1, r_2, \ldots$ be i.i.d. Rademacher random variables. Let $R_l = r_1 + \ldots + r_l$ for $l = 1, 2, \ldots$. Let $p_l(\cdot)$ be the probability mass function of $B_l$. Then the following holds for any $l = 1, 2, \ldots$:*

1. *$p_l$ is symmetric and supported on the set of odd integers if $l$ is odd, and supported on the set of even integers if $l$ is even.*

2. *For $i \in \mathrm{supp}(p_l)$ and $i \geq 0$, $p_l(i)$ is a non-increasing function of $i$.*

3. *$\mathbb{P}(R_l \geq t) \leq \mathbb{P}(R_{l+2} \geq t)$ for any $t > 1$.*

4. *$\mathbb{P}(R_l \geq t) \leq 2\mathbb{P}(R_{l+1} \geq t)$ for any $t > 1$.*

5. *$\mathbb{P}(R_l \geq t) \leq 2\mathbb{P}(R_{l+l'} \geq t)$ for any $l \geq 1$ and $l' \geq 1$.*

*Proof of Proposition G.13.* The first claim is immediate from the symmetry of the Rademacher random variables and the fact that the sum of an odd number of Rademacher random variables is odd, while the sum of an even number of Rademacher random variables is even. For the second claim, we note that

$$p_l(i) = 2^{-l} \cdot \binom{l}{(i+l)/2}, \quad i \in \mathrm{supp}(p_l),$$

which is a non-increasing function for $i \geq 0$. For the third claim, we let $t^\star = 2\lceil t/2 \rceil$ if $l$ is even and $t^\star = 2\lceil (t-1)/2 \rceil + 1$ if $l$ is odd. In other words, $t^\star = \min\{\tau \in \mathrm{supp}(p_l) : \tau \geq t\}$. Then we have

that

$$\begin{aligned}
\mathbb{P}(R_{l+2} \geq t) &= \mathbb{P}(R_l \geq t^\star + 2) + \mathbb{P}(R_l = t^\star) \cdot \mathbb{P}(r_{l+1} + r_{l+2} \geq 0) \\
&\quad + \mathbb{P}(R_l = t^\star - 2) \cdot \mathbb{P}(r_{l+1} + r_{l+2} = 2) \\
&= \mathbb{P}(R_l \geq t^\star) + (\mathbb{P}(R_l = t^\star - 2) - \mathbb{P}(R_l = t^\star)) \cdot \mathbb{P}(r_{l+1} + r_{l+2} = 2) \\
&\geq \mathbb{P}(R_l \geq t^\star) = \mathbb{P}(R_l \geq t).
\end{aligned}$$

where in the first equality we use the fact that $r_{l+1} + r_{l+2}$ is supported on $\{-2, 0, 2\}$ and in the second equality we use the symmetric property of the distribution of $r_{l+1} + r_{l+2}$. The last inequality follows from the monotonicity of the probability mass function of $R_l$ for $t^\star - 2 \geq 0$ when $t > 1$. For the forth claim, we similarly have that

$$\mathbb{P}(R_{l+1} \geq t) \geq \mathbb{P}(R_l \geq t^\star) - \mathbb{P}(R_l = t^\star) \cdot \mathbb{P}(r_{l+1} = -1) \geq \frac{1}{2}\mathbb{P}(R_l \geq t^\star) = \frac{1}{2}\mathbb{P}(R_l \geq t).$$

The last claim follows from a combination of the third and forth claims where

$$\mathbb{P}(R_{l+l'} \geq t) \geq \frac{1}{2}\mathbb{P}(R_{l+2\lfloor l'/2 \rfloor} \geq t) \geq \frac{1}{2}\mathbb{P}(R_{l+2\lfloor l'/2 \rfloor - 2} \geq t) \geq \ldots \geq \frac{1}{2}\mathbb{P}(R_l \geq t).$$

Hence, the proof is complete. $\qquad\square$

Next, we proceed to the proof of Proposition G.9.

*Proof of Proposition G.9.* By Proposition G.12 and Proposition G.11, there exists a coupling $\widehat{X}, \widehat{Y}$ with $\mathrm{law}(\widehat{X}) = \mathrm{law}(X)$ and $\mathrm{law}(\widehat{Y}) = \mathrm{law}(Y)$ such that $\widehat{X} \leq \widehat{Y}$ almost surely where $X \sim$ Hypergeometric$(d, k, k)$ and $Y \sim$ Binomial$(k, k/(d-k))$.

Consider i.i.d. Rademacher random variables $r_1, r_2, \ldots, r_k$. Let $R_l = r_1 + \ldots + r_l$ for $l = 1, 2, \ldots$. Since $R_{\widehat{X}} = r_1 + \ldots + r_{\widehat{X}} \,|\, \widehat{X} \stackrel{d}{=} 2\text{Binomial}(\widehat{X}, 1/2) - L$ and $R_{\widehat{Y}} = r_1 + \ldots + r_{\widehat{Y}} \,|\, \widehat{Y} \stackrel{d}{=} 2\text{Binomial}(\widehat{Y}, 1/2) - L$ for the coupling $(\widehat{X}, \widehat{Y})$ with $\widehat{X} \leq \widehat{Y}$, we consider the conditional random variable

$$r_{\widehat{X}+i} \,|\, (R_{\widehat{X}+i-1}, \widehat{X}, \widehat{Y}) = r_{\widehat{X}+i} = \begin{cases} 1, & \text{w.p. } 1/2 \\ -1, & \text{w.p. } 1/2 \end{cases}, \quad i = 1, 2, \ldots, \widehat{Y} - \widehat{X}$$

The equality holds by the i.i.d. property of these Rademacher random variables. From the distributional perspective, the distribution of $R_{\widehat{Y}}$ is obtained by conducting convolution with the Rademacher distribution for $\widehat{Y} - \widehat{X}$ times on the distribution of $R_{\widehat{X}}$. Invoking Proposition G.13, we directly conclude that $\mathbb{P}(R_{\widehat{Y}} \geq t \,|\, \widehat{X}, \widehat{Y}) \geq \mathbb{P}(R_{\widehat{X}} \geq t \,|\, \widehat{X}, \widehat{Y})/2$ for any $t > 1$ and $\widehat{Y} \geq \widehat{X}$. As $\widehat{Y} \geq \widehat{X}$ holds almost surely, by the law of total probability, we arrive at the conclusion that $\mathbb{P}(R_{\widehat{Y}} \geq t) \geq \mathbb{P}(R_{\widehat{X}} \geq t)/2$ for any $t > 1$. $\qquad\square$

# H SUPPORTING LEMMAS ON MOMENT CALCULATIONS

**Lemma H.1** (First moment). *Under Assumption 4.1, for any $s \geq 0$, it holds for any $y \in \mathbb{R}$ and $w, \theta \in \mathbb{S}^{d-1}$ that*

$$\mathbb{E}_{z \sim \mathcal{N}_d}\left[\psi(y, \langle w, z \rangle) z \cdot h_s(\langle \theta, z \rangle)\right] = \sqrt{s+1} \cdot \widehat{\psi}_{s+1}(y) \cdot \langle w, \theta \rangle^s w + \sqrt{s} \cdot \widehat{\psi}_{s-1}(y) \cdot \langle w, \theta \rangle^{s-1} \theta,$$

*in the $L^2$ sense over the marginal distribution of $y$ under $\mathbb{Q}$.*

*Proof of Lemma H.1.* For convenience, we denote $\rho := \langle w, \theta^\star \rangle$. We claim the following identities:

$$\mathbb{E}_{z \sim \mathcal{N}_d}[\psi(y, w^\top z) z \cdot h_s(\theta^{\star\top} z)]$$
$$= \mathbb{E}_{z \sim \mathcal{N}_d}\left[\psi(y, w^\top z) \cdot \theta^{\star\top} z \cdot h_s(\theta^{\star\top} z)\right] \cdot \theta^\star + \mathbb{E}_{z \sim \mathcal{N}_d}\left[\psi(y, w^\top z) \cdot h_s(\theta^{\star\top} z) \cdot P_{\theta^\star}^\perp z\right] \quad \text{(H.1)}$$
$$= \underbrace{\mathbb{E}_{z \sim \mathcal{N}_d}\left[\psi(y, w^\top z) \cdot \theta^{\star\top} z \cdot h_s(\theta^{\star\top} z)\right]}_{(\mathrm{I})} \cdot \frac{\theta^\star - \rho w}{1 - \rho^2} + \underbrace{\mathbb{E}_{z \sim \mathcal{N}_d}\left[\psi(y, w^\top z) \cdot w^\top z \cdot h_s(\theta^{\star\top} z)\right]}_{(\mathrm{II})} \cdot \frac{w - \rho \theta^\star}{1 - \rho^2}.$$

Here, in the first identity, we project $z$ in the direction of $\theta^\star$ and the orthogonal complement of $\theta^\star$, where $P_{\theta^\star}^\perp = I - \theta^\star \theta^{\star \top}$ is the projection operator onto the orthogonal complement of $\theta^\star$. To see how the second identity holds, we first look at the second term $\mathbb{E}_{z \sim \mathcal{N}_d} \left[ \psi(y, w^\top z) \cdot h_s(\theta^{\star \top} z) \cdot P_{\theta^\star}^\perp z \right]$. For each direction $v$ orthogonal to both $\theta^\star$ and $w$, we have

$$\mathbb{E}_{z \sim \mathcal{N}_d} \left[ \psi(y, w^\top z) \cdot h_s(\theta^{\star \top} z) \cdot \langle P_{\theta^\star}^\perp z, v \rangle \right] = \mathbb{E}_{z \sim \mathcal{N}_d, x \sim \mathcal{N}} \left[ \psi(y, w^\top z) \cdot h_s(\theta^{\star \top} z) \cdot x \right] = 0.$$

Also, by projection $P_{\theta^\star}^\perp z$ is always orthogonal to $\theta^\star$. Thus, the only direction left for consideration is $v = (w - \rho\theta^\star)/\sqrt{1 - \rho^2}$, for which we have

$$\mathbb{E}_{z \sim \mathcal{N}_d} \left[ \psi(y, w^\top z) \cdot h_s(\theta^{\star \top} z) \cdot \langle P_{\theta^\star}^\perp z, v \rangle \right] \cdot v$$

$$= \mathbb{E}_{z \sim \mathcal{N}_d} \left[ \psi(y, w^\top z) \cdot h_s(\theta^{\star \top} z) \cdot \frac{w^\top z - \rho\theta^{\star \top} z}{\sqrt{1 - \rho^2}} \right] \cdot \frac{w - \rho\theta^\star}{\sqrt{1 - \rho^2}}$$

$$= \mathbb{E}_{z \sim \mathcal{N}_d} \left[ \psi(y, w^\top z) \cdot h_s(\theta^{\star \top} z) \cdot (w^\top z - \rho\theta^{\star \top} z) \right] \cdot \frac{w - \rho\theta^\star}{1 - \rho^2}. \tag{H.2}$$

Plugging Eq. (H.2) into the second term of line 2 in Eq. (H.1), we thus have the last identity in Eq. (H.1). Next, we analyze terms (I) and (II) in Eq. (H.1). For our convenience, we define $\mathrm{U}_\rho$ as the Gaussian noise operator such that

$$\mathrm{U}_\rho \psi(y, x) = \mathbb{E}_{x' \sim \mathcal{N}} \left[ \psi(y, \rho x + \sqrt{1 - \rho^2} x') \right].$$

For term (I), we have by the definition of $\mathrm{U}_\rho$ that

$$(\mathrm{I}) = \mathbb{E}_{x \sim \mathcal{N}} \left[ \mathrm{U}_\rho \psi(y, x) \cdot x \cdot h_s(x) \right]$$

$$= \mathbb{E}_{z \sim \mathcal{N}_d} \left[ \sqrt{s+1} \cdot \mathrm{U}_\rho \psi(y, x) \cdot h_{s+1}(x) + \sqrt{s} \cdot \mathrm{U}_\rho \psi(y, x) \cdot h_{s-1}(x) \cdot \right]$$

$$\overset{L^2(\mathbb{Q})}{=} \sqrt{s+1} \cdot \widehat{\psi}_{s+1}(y) \cdot \rho^{s+1} + \sqrt{s} \cdot \widehat{\psi}_{s-1}(y) \cdot \rho^{s-1}. \tag{H.3}$$

where the second line follows from the recurrence relation of the Hermite polynomials in Eq. (B.1), and the last line follows from the property of the Gaussian noise operator in Eq. (B.3). Similarly for term (II), we have

$$(\mathrm{II}) = \mathbb{E}_{x \sim \mathcal{N}} \left[ \mathrm{U}_\rho \left( \psi(y, x) x \right) \cdot h_s(x) \right]$$

$$\overset{L^2(\mathbb{Q})}{=} \rho^s \cdot \mathbb{E}_{x \sim \mathcal{N}} \left[ \psi(y, x) \cdot x \cdot h_s(x) \right]$$

$$\overset{L^2(\mathbb{Q})}{=} \rho^s \cdot \left( \sqrt{s+1} \cdot \widehat{\psi}_{s+1}(y) + \sqrt{s} \cdot \widehat{\psi}_{s-1}(y) \right), \tag{H.4}$$

where in the last line we borrow the calculation in Eq. (H.3) by letting $\rho = 1$. Plugging Eq. (H.3) and (H.4) into Eq. (H.1), we hence have

$$(\text{H.1}) \overset{L^2(\mathbb{Q})}{=} \left( \sqrt{s+1} \cdot \widehat{\psi}_{s+1}(y) \cdot \rho^{s+1} + \sqrt{s} \cdot \widehat{\psi}_{s-1}(y) \cdot \rho^{s-1} \right) \cdot \frac{\theta^\star - \rho w}{1 - \rho^2}$$

$$+ \rho^s \cdot \left( \sqrt{s+1} \cdot \widehat{\psi}_{s+1}(y) + \sqrt{s} \cdot \widehat{\psi}_{s-1}(y) \right) \cdot \frac{w - \rho\theta^\star}{1 - \rho^2}$$

$$= \sqrt{s+1} \cdot \widehat{\psi}_{s+1}(y) \cdot \rho^s w + \sqrt{s} \cdot \widehat{\psi}_{s-1}(y) \cdot \rho^{s-1} \theta^\star,$$

which completes the proof. $\qquad\square$

An implication of the previous lemma is that

$$\mathbb{E}_{\mathbb{Q}}[h_{s^\star}(\langle \theta^\star, z \rangle) \cdot \sigma'(\langle z, \theta \rangle) \cdot \langle z, \theta^\star \rangle] = s \cdot \widehat{\sigma}^{(s^\star)} \cdot \langle \theta^\star, \theta \rangle^{s^\star - 1} + \sqrt{(s+1)(s+2)} \cdot \widehat{\sigma}^{(s^\star + 2)} \cdot \langle \theta^\star, \theta \rangle^{s^\star + 1},$$

where we take $\widehat{\sigma}^{(s)}$ as the $s$-th normalized Hermite coefficient of $\sigma$. Here, we take $\psi(y, x)$ as $\sigma'(x)$ and thus $\widehat{\psi}_s(y) = \sqrt{s+1} \cdot \widehat{\sigma}^{(s+1)}$.

**Lemma H.2** (Decomposition of first order moment). *Suppose that $\psi$ follows Assumption 4.1 and*

$$g = \frac{1}{nL} \sum_{i=1}^{n} \sum_{l=1}^{L} \left( \psi(y_i, \langle w_l, z_i \rangle) \cdot z_i - \widehat{\psi}_1(y_i) \cdot w_l \right),$$

*where $(z_i, y_i) \overset{\text{i.i.d.}}{\sim} \mathbb{P}_{\theta^\star}$ and $\{w_l\}_{l \leq L}$ is fixed. Then it holds that*

$$\mathbb{E}_{\mathbb{P}_{\theta^\star}}[g] = \sum_{s \geq s^\star} \mathbb{E}_{\mathbb{Q}}[\zeta_s(y) \cdot \widehat{\psi}_{s-1}(y)] \cdot \frac{\sqrt{s}}{L} \sum_{l=1}^{L} \langle w_l, \theta^\star \rangle^{s-1} \cdot \theta^\star$$

$$+ \sum_{s \geq s^\star} \mathbb{E}_{\mathbb{Q}}[\zeta_s(y) \cdot \widehat{\psi}_{s+1}(y)] \cdot \frac{\sqrt{s+1}}{L} \sum_{l=1}^{L} \langle w_l, \theta^\star \rangle^s \cdot w_l.$$

*Proof of Lemma H.2.* Applying a change of measure from $\mathbb{P}_{\theta^\star}$ to $\mathbb{Q}$ and invoking Eq. (2.2), we get

$$\mathbb{E}_{\mathbb{P}_{\theta^\star}}[g] = \frac{1}{L} \sum_{l=1}^{L} \mathbb{E}_{\mathbb{P}_{\theta^\star}} \left[ \psi(y, \langle w_l, z \rangle) \cdot z - \widehat{\psi}_1(y) \cdot w_l \right]$$

$$= \frac{1}{L} \sum_{l=1}^{L} \mathbb{E}_{\mathbb{Q}} \left[ \psi(y, \langle w_l, z \rangle) z \cdot \left( 1 + \sum_{s \geq s^\star} \zeta_s(y) h_s(\langle \theta^\star, z \rangle) \right) - \widehat{\psi}_1(y) w_l \right], \quad \text{(H.5)}$$

Note that for $s = 0$, we have for the first term in the summation of Eq. (H.5) that

$$L^{-1} \sum_{l=1}^{L} \mathbb{E}_{\mathbb{Q}}[\psi(y, \langle w_l, z \rangle) z] = \mathbb{E}_{\mathbb{Q}}[\widehat{\psi}_1(y)] \cdot \frac{1}{L} \sum_{l=1}^{L} w_l,$$

which is cancelled out by the debiasing term in the algorithm. Applying the result of Lemma H.1 to the remaining terms in Eq. (H.5) yields

$$\mathbb{E}_{\mathbb{P}_{\theta^\star}}[g] = \sum_{s \geq s^\star} \mathbb{E}_{\mathbb{Q}}[\zeta_s(y) \cdot \widehat{\psi}_{s+1}(y)] \cdot \frac{\sqrt{s+1}}{L} \sum_{l=1}^{L} \langle w_l, \theta^\star \rangle^s \cdot w_l$$

$$+ \sum_{s \geq s^\star} \mathbb{E}_{\mathbb{Q}}[\zeta_s(y) \cdot \widehat{\psi}_{s-1}(y)] \cdot \frac{\sqrt{s}}{L} \sum_{l=1}^{L} \langle w_l, \theta^\star \rangle^{s-1} \cdot \theta^\star, \quad \text{(H.6)}$$

where the $\mathbb{E}_{\mathbb{Q}}[\widehat{\psi}_1(y)] \cdot \frac{1}{L} \sum_{l=1}^{L} w_l$ term from Lemma H.1 with $s = 0$ is cancelled out by the debiasing term in the algorithm. $\qquad\square$

**Lemma H.3** (Second moment on nice event). *Suppose $\psi : \mathbb{R} \times \mathbb{R} \to \mathbb{R}$ satisfies the quadruple-integrable and high-pass assumptions in Assumption 4.1. Let $s^\star$ be the generative exponent defined in Definition 2.1. Suppose $\mathbb{E}_{\mathbb{Q}}[\zeta_s(y)^2] \leq C$ for some universal $C = O(1)$ and for all $s \geq s^\star$. For any $w, w', \theta^\star, v \in \mathbb{S}^{d-1}$ where either $v = \theta^\star$ or $\langle v, \theta^\star \rangle = 0$ in the non-sparse case, and either $v = e_j$ for $j \in \text{supp}(\theta^\star)$ or $v = e_j$ for $j \notin \text{supp}(\theta^\star)$ in the sparse case, suppose that*

$$\max\{|\langle v, w \rangle|, |\langle v, w' \rangle|\} \leq \epsilon_0, \quad \max\{|\langle \theta^\star, w \rangle|, |\langle \theta^\star, w' \rangle|\} \leq \epsilon, \quad |\langle w, w' \rangle| \leq \epsilon_1$$

*for some $\epsilon, \epsilon_0, \epsilon_1$ such that $4es^\star \epsilon < 1/2$. Then, we have for $s^\star \geq 2$ that*

$$\mathbb{E}_{\mathbb{Q}} \left[ \psi(y, \langle w, z \rangle) \psi(y, \langle w', z \rangle) \langle v, z \rangle^2 \cdot \left( 1 + \sum_{s = s^\star}^{\infty} \zeta_s(y) h_s(\langle \theta^\star, z \rangle) \right) \right]$$

$$\lesssim \epsilon_1^{s^\star - 1} \cdot \left( 1 + \frac{\epsilon^2}{\epsilon_1} + \left( \frac{\epsilon^2}{\epsilon_1} \right)^{s^\star - 1} \cdot \epsilon + \mathbb{1}(v \perp \theta^\star) \cdot \left( \frac{\epsilon^2}{\epsilon_1} \right)^{s^\star - 2} \cdot \frac{\epsilon_0^2}{\epsilon_1} \cdot (\epsilon^2 + \epsilon \cdot \mathbb{1}(s^\star \geq 4)) \right),$$

*and for $s^\star = 1$, the bound is $O(1)$. Here, $\lesssim$ hides constants that only depend on $s^\star$, $\mathbb{E}_Q[\psi(x, y)^4]$ and $C$.*

*Proof.* Using the results from Proposition I.1, we have that

$$h_s(\langle\theta^\star, z\rangle)\langle v, z\rangle^2 = \sqrt{(s+2)(s+1)} \cdot h_{s+2}(z)[(\theta^\star)^{\otimes s} \otimes v^{\otimes 2}] + h_s(z)[(\theta^\star)^{\otimes s}] \tag{H.7}$$
$$+ 2s \cdot h_s(z)[(\theta^\star)^{\otimes s-1} \otimes v]^\top \cdot \langle\theta^\star, v\rangle + \sqrt{s(s-1)} \cdot h_{s-2}(z)[(\theta^\star)^{\otimes s-2}] \cdot \langle\theta^\star, v\rangle^2.$$

Thus, we only need to focus on these degree terms in $\psi(y, \langle w, z\rangle)\psi(y, \langle w', z\rangle)$. Our goal is to compute the following quantity, which we denoted by $F$:

$$F = \mathbb{E}_\mathbb{Q}\left[\psi(y, \langle w, z\rangle)\psi(y, \langle w', z\rangle)\langle v, z\rangle^2 \cdot \left(1 + \sum_{s=s^\star}^\infty \zeta_s(y)h_s(\langle\theta^\star, z\rangle)\right)\right]$$
$$= \mathbb{E}_\mathbb{Q}\left[\psi(y, \langle w, z\rangle)\psi(y, \langle w', z\rangle)\langle v, z\rangle^2\right]$$
$$+ \sum_{s=s^\star}^\infty \left|\mathbb{E}_\mathbb{Q}\left[\zeta_s(y)\psi(y, \langle w, z\rangle)\psi(y, \langle w', z\rangle)\langle v, z\rangle^2 h_s(\langle\theta^\star, z\rangle)\right]\right|. \tag{H.8}$$

Here, for the term corresponding to $s = 0$ in Eq. (H.8), we plug in Eq. (H.7) and have by Lemma I.3 that

$$\left|\mathbb{E}_\mathbb{Q}\left[\psi(y, \langle w, z\rangle)\psi(y, \langle w', z\rangle)\langle v, z\rangle^2\right]\right|$$
$$= \left|\mathbb{E}_\mathbb{Q}\left[\psi(y, \langle w, z\rangle)\psi(y, \langle w', z\rangle)(\sqrt{2} \cdot h_2(z)[v^{\otimes 2}] + 1)\right]\right|$$
$$\lesssim \epsilon_1^{(s^\star-2)\vee 0} \cdot \epsilon_0^{2\wedge c_0} \cdot \epsilon^{(2-c_0)\vee 0} + \epsilon_1^{s^\star-1} \lesssim \epsilon_1^{(s^\star-2)\vee 0} \cdot \epsilon_0^2 + \epsilon_1^{s^\star-1} \lesssim \mathbb{1}(s^\star = 1) + \epsilon_1^{s^\star-2}\epsilon_0^2 + \epsilon_1^{s^\star-1}.$$

Here, to use Lemma I.3, for $h_2(z)[v^{\otimes 2}]$ we take test tensor $T_2 = v^{\otimes 2}$ and set $c_0 = 2$. The last line also holds by using the Cauchy-Schwarz inequality for $\mathbb{E}_\mathbb{Q}[|\zeta_s(y)| \cdot \psi(y, x)^2] \leq \mathbb{E}_\mathbb{Q}[|\zeta_s(y)|^2]^{1/2} \cdot \mathbb{E}_\mathbb{Q}[\psi(y, x)^4]^{1/2} \leq \mathbb{E}_\mathbb{Q}[\psi(y, x)^4]^{1/2} = O(1)$. As for the case $s^\star = 1$, we already have a constant outside, and noting that the second moment is at most $O(1)$ due to the quadruple-integrable assumption, it suffices to consider in the following $s^\star \geq 2$. For the second part of Eq. (H.8), we can split the expectation according to Eq. (H.7). For the first term in Eq. (H.7) which corresponds to $\sqrt{(s+2)(s+1)} \cdot h_{s+2}(z)[(\theta^\star)^{\otimes s} \otimes v^{\otimes 2}]$, we take test tensor $T_s = v^{\otimes 2} \otimes (\theta^\star)^{\otimes(s-2)}$ with $c_0 = 2, s_0 = s^\star + 2$ and have by Proposition I.4 that

$$\left|\sum_{s=s^\star}^\infty \sqrt{(s+2)(s+1)} \cdot \mathbb{E}_\mathbb{Q}\left[\zeta_s(y)\psi(y, \langle w, z\rangle)\psi(y, \langle w', z\rangle)h_{s+2}(z)[(\theta^\star)^{\otimes s} \otimes v^{\otimes 2}]\right]\right|$$
$$\lesssim \mathbb{1}(s_0 \leq c_0) \cdot \left(\epsilon_1^{s^\star-1-\lfloor s_0/2\rfloor} \cdot \epsilon_0^{s_0} + \epsilon_1^{s^\star-1-\lfloor c_0/2\rfloor} \cdot \epsilon_0^{c_0}\right) + \epsilon_0^{c_0} \cdot \epsilon^{(2s^\star)\vee s_0-c_0}$$
$$+ \mathbb{1}(s_0 \leq 2(s^\star-1)) \cdot \left(\epsilon_1^{s^\star-1-\lfloor(c_0+1)/2\rfloor} \cdot \epsilon_0^{c_0} \cdot \epsilon + \epsilon_0^{c_0} \cdot \epsilon^{2(s^\star-1)+1-c_0}\right)$$
$$\lesssim \epsilon_0^2 \cdot \epsilon^{2s^\star-2} + \mathbb{1}(s^\star \geq 4) \cdot \left(\epsilon_1^{s^\star-2} \cdot \epsilon_0^2 \cdot \epsilon + \epsilon_0^2 \cdot \epsilon^{2s^\star-3}\right).$$

For the second term $h_s(z)[(\theta^\star)^{\otimes s}]$, we take test tensor $T_s = (\theta^\star)^{\otimes s}$ with $c_0 = 0, s_0 = s^\star$ and have by Proposition I.4 that

$$\left|\sum_{s=s^\star}^\infty \mathbb{E}_\mathbb{Q}\left[\zeta_s(y)\psi(y, \langle w, z\rangle)\psi(y, \langle w', z\rangle)h_s(z)[(\theta^\star)^{\otimes s}]\right]\right|$$
$$\lesssim \epsilon_0^{c_0} \cdot \epsilon^{(2s^\star)\vee s_0-c_0} + \left(\epsilon_1^{s^\star-1-\lfloor(c_0+1)/2\rfloor} \cdot \epsilon_0^{c_0} \cdot \epsilon + \epsilon_0^{c_0} \cdot \epsilon^{2(s^\star-1)+1-c_0}\right)$$
$$\lesssim \epsilon^{2s^\star} + \epsilon_1^{s^\star-1} \cdot \epsilon + \epsilon^{2s^\star-1} \lesssim \epsilon_1^{s^\star-1} \cdot \epsilon + \epsilon^{2s^\star-1}.$$

For the third term $2s \cdot h_s(z)[(\theta^\star)^{\otimes s-1} \otimes v]^\top \cdot \langle\theta^\star, v\rangle$, we take test tensor $T_s = v \otimes (\theta^\star)^{\otimes s-1}$ with $c_0 = 1, s_0 = s^\star$ and have by Proposition I.4 that

$$\left|\sum_{s=s^\star}^\infty 2s \cdot \mathbb{E}_\mathbb{Q}\left[\zeta_s(y)\psi(y, \langle w, z\rangle)\psi(y, \langle w', z\rangle)h_s(z)[(\theta^\star)^{\otimes s-1} \otimes v]^\top\right]\right|$$
$$\lesssim \epsilon_0^{c_0} \cdot \epsilon^{(2s^\star)\vee s_0-c_0} + \left(\epsilon_1^{s^\star-1-\lfloor(c_0+1)/2\rfloor} \cdot \epsilon_0^{c_0} \cdot \epsilon + \epsilon_0^{c_0} \cdot \epsilon^{2(s^\star-1)+1-c_0}\right)$$
$$\lesssim \epsilon_0 \cdot \epsilon^{2s^\star-1} + \epsilon_1^{s^\star-2} \cdot \epsilon_0 \cdot \epsilon + \epsilon_0 \cdot \epsilon^{2s^\star-2} \lesssim \epsilon_1^{s^\star-2} \cdot \epsilon_0 \cdot \epsilon + \epsilon_0 \cdot \epsilon^{2s^\star-2}.$$

For the last term $\sqrt{s(s-1)} \cdot \boldsymbol{h}_{s-2}(z)[(\theta^\star)^{\otimes s-2}] \cdot \langle \theta^\star, v \rangle^2$, we take test tensor $T_s = (\theta^\star)^{\otimes s}$ with $c_0 = 0, s_0 = s^\star - 2$ and have by Proposition I.4 that

$$\left| \sum_{s=s^\star}^\infty \sqrt{s(s-1)} \cdot \mathbb{E}_{\mathbb{Q}} \left[ \zeta_s(y)\psi(y, \langle w, z \rangle)\psi(y, \langle w', z \rangle)\boldsymbol{h}_{s-2}(z)[(\theta^\star)^{\otimes s-2}] \cdot \langle \theta^\star, v \rangle^2 \right] \right|$$

$$\lesssim \mathbb{1}(s^\star = 2) \cdot \left( \epsilon_1^{s^\star-1-\lfloor s_0/2 \rfloor} \cdot \epsilon_0^{s_0} + \epsilon_1^{s^\star-1-\lfloor c_0/2 \rfloor} \cdot \epsilon_0^{c_0} \right) + \epsilon_0^{c_0} \cdot \epsilon^{(2s^\star)\vee s_0 - c_0}$$

$$+ \left( \epsilon_1^{s^\star-1-\lfloor (c_0+1)/2 \rfloor} \cdot \epsilon_0^{c_0} \cdot \epsilon + \epsilon_0^{c_0} \cdot \epsilon^{2(s^\star-1)+1-c_0} \right)$$

$$\lesssim \mathbb{1}(s^\star = 2)\epsilon_1 + \epsilon^{2s^\star} + \epsilon_1^{s^\star-1} \cdot \epsilon + \epsilon^{2s^\star-1} \lesssim \mathbb{1}(s^\star = 2)\epsilon_1 + \epsilon_1^{s^\star-1} \cdot \epsilon + \epsilon^{2s^\star-1}.$$

Summing up the above terms, we have that for $s^\star \geq 2$,

$$F \lesssim \left( \epsilon_1^{s^\star-2}\epsilon_0^2 + \epsilon_1^{s^\star-1} \right) + \left( \epsilon_0^2 \cdot \epsilon^{2s^\star-2} + \mathbb{1}(s^\star \geq 4) \cdot \left( \epsilon_1^{s^\star-2} \cdot \epsilon_0^2 \cdot \epsilon + \epsilon_0^2 \cdot \epsilon^{2s^\star-3} \right) \right)$$

$$+ \left( \epsilon_1^{s^\star-1} \cdot \epsilon + \epsilon^{2s^\star-1} \right) + \left( \epsilon_1^{s^\star-2} \cdot \epsilon_0 \cdot \epsilon + \epsilon_0 \cdot \epsilon^{2s^\star-2} + \mathbb{1}(s^\star = 2)\epsilon_1 + \epsilon_1^{s^\star-1} \cdot \epsilon + \epsilon^{2s^\star-1} \right) \mathbb{1}(v = \theta^\star)$$

$$\lesssim \epsilon_1^{s^\star-2}\epsilon_0^2 + \epsilon_1^{s^\star-1} + \epsilon_0^2 \cdot \epsilon^{2s^\star-2} + \mathbb{1}(s^\star \geq 4) \cdot \epsilon_0^2 \cdot \epsilon^{2s^\star-3} + \epsilon^{2s^\star-1}$$

$$+ \left( \epsilon_1^{s^\star-2} \cdot \epsilon_0 \cdot \epsilon + \epsilon_0 \cdot \epsilon^{2s^\star-2} \right) \mathbb{1}(v = \theta^\star).$$

If $v = \theta^\star$, then we additionally have $\epsilon_0 = \epsilon$, which simplifies the above bound to

$$F \big|_{s^\star \geq 2, v = \theta^\star} \lesssim \epsilon_1^{s^\star-2}\epsilon^2 + \epsilon_1^{s^\star-1} + \epsilon^{2s^\star-1} = \epsilon_1^{s^\star-1} \cdot \left( 1 + \frac{\epsilon^2}{\epsilon_1} + \left( \frac{\epsilon^2}{\epsilon_1} \right)^{s^\star-1} \cdot \epsilon \right).$$

For $v \perp \theta^\star$, we have that

$$F \big|_{s^\star \geq 2, v \perp \theta^\star} \lesssim \epsilon_1^{s^\star-2}\epsilon_0^2 + \epsilon_1^{s^\star-1} + \epsilon_0^2 \cdot \epsilon^{2s^\star-2} + \mathbb{1}(s^\star \geq 4) \cdot \epsilon_0^2 \cdot \epsilon^{2s^\star-3} + \epsilon^{2s^\star-1}$$

$$\lesssim \epsilon_1^{s^\star-1} \cdot \left( 1 + \frac{\epsilon^2}{\epsilon_1} + \left( \frac{\epsilon^2}{\epsilon_1} \right)^{s^\star-1} \cdot \epsilon + \left( \frac{\epsilon^2}{\epsilon_1} \right)^{s^\star-2} \cdot \frac{\epsilon_0^2}{\epsilon_1} \cdot (\epsilon^2 + \epsilon \cdot \mathbb{1}(s^\star \geq 4)) \right).$$

Where for $s^\star = 1$, we have $F|_{s^\star=1} \lesssim 1$. Hence, we complete the proof. $\qquad\square$

**Lemma H.4.** *For polarization level $\gamma = o(1)$, take the polarized random vector*

$$w = \frac{\gamma e_1 + \xi}{\|\gamma e_1 + \xi\|_2}, \quad \text{where} \quad \xi \sim \text{Unif}(\mathbb{S}^{d-1}),$$

*where $e_1 = (1, 0, \ldots, 0)^\top$ is the first standard basis vector in $\mathbb{R}^d$. Let $\theta^\star = (\rho, \sqrt{1-\rho^2}, 0, \ldots, 0) \in \mathbb{S}^{d-1}$ be a fixed direction. Then, we have that*

$$\mathbb{E}[\langle \theta^\star, w \rangle^s] \simeq \begin{cases} \left( |\rho|(\gamma + d^{-1/2}) + \sqrt{1-\rho^2}d^{-1/2} \right)^s & \text{if } s \text{ is even} \\ \rho\gamma \left( |\rho|(\gamma + d^{-1/2}) + \sqrt{1-\rho^2}d^{-1/2} \right)^{s-1} & \text{if } s \text{ is odd}. \end{cases}$$

*Proof of Lemma H.4.* For $w$, we have by Lemma I.9 that the first moment is given by

$$\mathbb{E}[w] = C(e_1, \gamma) \cdot \gamma \cdot e_1 \simeq \gamma e_1,$$

and the second moment is controlled by

$$\mathbb{E}[ww^\top] = \begin{bmatrix} C(2e_1, \gamma) \cdot (\gamma + d^{-1/2})^2 & 0 & \cdots & 0 \\ 0 & & & \\ \vdots & & C(2e_2, \gamma)d^{-1} \cdot I_{d-1} & \\ 0 & & & \end{bmatrix} \asymp \begin{bmatrix} (\gamma + d^{-1/2})^2 & 0 & \cdots & 0 \\ 0 & & & \\ \vdots & & d^{-1} \cdot I_{d-1} & \\ 0 & & & \end{bmatrix}.$$

For $\langle \theta^\star, w \rangle^s \cdot w$, we look at coordinate $\tau$ of $w$, and the first moment is given by

$$\mathbb{E}\left[ \langle \theta^\star, w \rangle^s \cdot w_\tau \right] = \mathbb{E}\left[ (\rho w_1 + \sqrt{1-\rho^2}w_2)^s w_\tau \right].$$

Note that if $\tau \neq 1, 2$, then the expectation is zero. For more generality, let us take $r_1, r_2 \in \mathbb{N}$. We study the following expectation:

$$
\mathbb{E}\left[\langle \theta^\star, w \rangle^s \cdot w_1^{r_1} w_2^{r_2}\right]
$$

$$
= \sum_{j=0}^{s} \binom{s}{j} \rho^j \sqrt{1-\rho^2}^{s-j} \mathbb{E}\left[w_1^{j+r_1} w_2^{s-j+r_2}\right] \cdot \mathbb{1}(s-j+r_2 \text{ even})
$$

$$
= \begin{cases}
\displaystyle\sum_{j=0}^{\lfloor s/2 \rfloor} \binom{s}{2j} \rho^{2j} \sqrt{1-\rho^2}^{s-2j} \mathbb{E}\left[w_1^{2j+r_1} w_2^{s-2j+r_2}\right] & \text{if } s+r_2 \text{ is even,} \\[2ex]
\displaystyle\sum_{j=0}^{\lfloor (s-1)/2 \rfloor} \binom{s}{2j+1} \rho^{2j+1} \sqrt{1-\rho^2}^{s-2j-1} \mathbb{E}\left[w_1^{2j+1+r_1} w_2^{s-2j-1+r_2}\right] & \text{if } s+r_2 \text{ is odd.}
\end{cases}
$$

Here, the first equality holds by noting that each term in the sum is zero if the degree on $w_2$ is odd due to the symmetry in the distribution of $w_2$. Next, we invoke Lemma I.9 for the moment as

$$
\mathbb{E}\left[\langle \theta^\star, w \rangle^s \cdot w_1^{r_1} w_2^{r_2}\right]
$$

$$
\simeq \begin{cases}
\text{if } s+r_2 \text{ is even:} \\
\displaystyle\sum_{j=0}^{\lfloor s/2 \rfloor} \binom{s}{2j} \rho^{2j} \sqrt{1-\rho^2}^{s-2j} \gamma^{\mathbb{1}(r_1 \text{ odd})} \left(\gamma + d^{-1/2}\right)^{2j+2\lfloor r_1/2 \rfloor} \left(d^{-1/2}\right)^{s-2j+r_2} \\
\text{if } s+r_2 \text{ is odd:} \\
\displaystyle\sum_{j=0}^{\lfloor (s-1)/2 \rfloor} \binom{s}{2j+1} \rho^{2j+1} \sqrt{1-\rho^2}^{s-2j-1} \gamma^{\mathbb{1}(r_1 \text{ even})} \left(\gamma + d^{-1/2}\right)^{2j+2\lfloor \frac{r_1+1}{2} \rfloor} \left(d^{-1/2}\right)^{s-2j-1+r_2}
\end{cases}
$$

$$
\simeq \begin{cases}
\text{if } s+r_2 \text{ is even:} \\
\sqrt{1-\rho^2}^{\mathbb{1}(s \text{ odd})} \gamma^{\mathbb{1}(r_1 \text{ odd})} \left(\gamma + \frac{1}{\sqrt{d}}\right)^{2\lfloor \frac{r_1}{2} \rfloor} \left(\frac{1}{\sqrt{d}}\right)^{r_2+\mathbb{1}(s \text{ odd})} \left(|\rho|(\gamma + d^{-1/2}) + \sqrt{1-\rho^2} d^{-1/2}\right)^{2\lfloor \frac{s}{2} \rfloor} \\
\text{if } s+r_2 \text{ is odd:} \\
\rho \sqrt{1-\rho^2}^{\mathbb{1}(s \text{ even})} \gamma^{\mathbb{1}(r_1 \text{ even})} \left(\gamma + \frac{1}{\sqrt{d}}\right)^{2\lfloor \frac{r_1+1}{2} \rfloor} \left(\frac{1}{\sqrt{d}}\right)^{r_2+\mathbb{1}(s \text{ even})} \left(|\rho|(\gamma + d^{-1/2}) + \sqrt{1-\rho^2} d^{-1/2}\right)^{2\lfloor \frac{s-1}{2} \rfloor}
\end{cases}
$$

Here, the symbol $\simeq$ conceals some constant factors that are governed by upper and lower bounds dependent on $s$ only. Using the above calculation, we have We can specialize the above results to the case $r_1 = r_2 = 0$ and obtain

$$
\mathbb{E}[\langle \theta^\star, w \rangle^s] \simeq \begin{cases}
\left(|\rho|(\gamma + d^{-1/2}) + \sqrt{1-\rho^2} d^{-1/2}\right)^s & \text{if } s \text{ is even} \\[1ex]
\rho \gamma \left(|\rho|(\gamma + d^{-1/2}) + \sqrt{1-\rho^2} d^{-1/2}\right)^{s-1} & \text{if } s \text{ is odd,}
\end{cases}
$$

which completes the proof. $\qquad\square$

# I TECHNICAL RESULTS

## I.1 TECHNICAL RESULTS FOR HERMITE TENSOR

**Proposition I.1.** *Let $s \in \mathbb{N}_0$. For any $z \in \mathbb{R}^d$, we have*

$$
z_i z_j \boldsymbol{h}_s(z) = \mathrm{Sym}\big(\sqrt{(s+2)(s+1)} \cdot \boldsymbol{h}_{s+2}(z)[e_i \otimes e_j] + \delta_{ij} \boldsymbol{h}_s(z) + s \cdot \boldsymbol{h}_s(z)[e_j] \otimes e_i
$$
$$
+ s \cdot \boldsymbol{h}_s(z)[e_i] \otimes e_j + \sqrt{s(s-1)} \cdot \boldsymbol{h}_{s-2}(z) \otimes e_i \otimes e_j\big),
$$

*where we define $\boldsymbol{h}_{-1}(z)$ and $\boldsymbol{h}_{-2}(z)$ to be all zero tensors of any conformable shape.*

*Proof of Proposition I.1.* Note that each element of $h_s(\theta^{\star\top} z) z z^\top$ must lie in the polynomial space with degree at most $s+2$, i.e., $\mathbb{R}_{s+2}[z]$. We take a test function $F : \mathbb{R}^d \to \mathbb{R}$ such that $R \in \mathbb{R}_{s+2}[z]$. Thus, we can write down the inner product of $F$ and $h_s(\theta^{\star\top} z) z_i z_j$ for $i, j \in [d]$ as

$$
\mathbb{E}_{z \sim \mathcal{N}_d}[F(z) h_s(\theta^{\star\top} z) z_i z_j] = \mathbb{E}_{z \sim \mathcal{N}_d}[F(z) z_i z_j \cdot \boldsymbol{h}_s(z)[(\theta^\star)^{\otimes s}]],
$$

where in the equation, we use (B.4) to rewrite the Hermite polynomial in terms of the Hermite tensor. It suffices to understand the tensor $F(z)z_i z_j \boldsymbol{h}_s(z)$. Note that $F$ is differentiable to any order, and the tensor obtained by differentiating $F$ to any order is square-integrable with respect to the standard normal distribution. By the Stein's lemma for Hermite tensor (B.5), we have

$$\sqrt{s!} \cdot \mathbb{E}_{z \sim \mathcal{N}_d}[F(z)z_i z_j \boldsymbol{h}_s(z)] = \mathbb{E}_{z \sim \mathcal{N}_d}[\nabla^s (F(z)z_i z_j)]$$

$$= \mathrm{Sym}\left(\mathbb{E}_{z \sim \mathcal{N}_d}[\nabla^s F(z)z_i z_j] + s\mathbb{E}_{z \sim \mathcal{N}_d}[\nabla^{s-1} F(z)\nabla(z_i z_j)] + \frac{s(s-1)}{2}\mathbb{E}_{z \sim \mathcal{N}_d}[\nabla^{s-2} F(z)\nabla^2(z_i z_j)]\right)$$

$$= \mathrm{Sym}\big(\mathbb{E}_{z \sim \mathcal{N}_d}[\nabla^s F(z)\delta_{ij}] + \mathbb{E}_{z \sim \mathcal{N}_d}[\nabla^{s+2} F(z)[e_i \otimes e_j]] + s\mathbb{E}_{z \sim \mathcal{N}_d}[\nabla^s F(z)[e_j] \otimes e_i]$$

$$+ s\mathbb{E}_{z \sim \mathcal{N}_d}[\nabla^s F(z)[e_i] \otimes e_j] + s(s-1)\mathbb{E}_{z \sim \mathcal{N}_d}[\nabla^{s-2} F(z) \otimes e_i \otimes e_j]\big).$$
(I.1)

Here, the "Sym" operation symmetrizes the tensor in the parentheses. The last equality holds by the following calculations. For $\mathbb{E}_{z \sim \mathcal{N}_d}[\nabla^s F(z)z_i z_j]$, we have

$$\mathbb{E}_{z \sim \mathcal{N}_d}[\nabla^s F(z)z_i z_j] = \mathrm{Sym}\left(\mathbb{E}_{z \sim \mathcal{N}_d}[\nabla^s F(z)\delta_{ij}] + \mathbb{E}_{z \sim \mathcal{N}_d}[\nabla^{s+1} F(z)[e_j]z_i]\right)$$

$$= \mathrm{Sym}\left(\mathbb{E}_{z \sim \mathcal{N}_d}[\nabla^s F(z)\delta_{ij}] + \mathbb{E}_{z \sim \mathcal{N}_d}[\nabla^{s+2} F(z)[e_i \otimes e_j]]\right),$$

where we use the Stein's lemma for both equalities. For $\mathbb{E}_{z \sim \mathcal{N}_d}[\nabla^{s-2} F(z)\nabla^2(z_i z_j)]$, we have

$$\mathrm{Sym}\left(\mathbb{E}_{z \sim \mathcal{N}_d}[\nabla^{s-1} F(z)\nabla(z_i z_j)]\right) = \mathrm{Sym}\left(\mathbb{E}_{z \sim \mathcal{N}_d}[z_j \nabla^{s-1} F(z) \otimes e_i + z_i \nabla^{s-1} F(z) \otimes e_j]\right)$$

$$= \mathrm{Sym}\left(\mathbb{E}_{z \sim \mathcal{N}_d}[\nabla^s F(z)[e_j] \otimes e_i] + \mathbb{E}_{z \sim \mathcal{N}_d}[\nabla^s F(z)[e_i] \otimes e_j]\right).$$

Now, for each derivative of $F$ in (I.1), we have by the Stein's lemma stated in (B.5) that $\mathbb{E}_{z \sim \mathcal{N}_d}[\nabla^s F(z)] = \sqrt{s!} \cdot \mathbb{E}_{z \sim \mathcal{N}_d}[F(z)\boldsymbol{h}_s(z)]$, which gives us

$$\mathbb{E}_{z \sim \mathcal{N}_d}[F(z)z_i z_j \boldsymbol{h}_s(z)] = \mathbb{E}_{z \sim \mathcal{N}_d}\Big[F(z) \cdot \mathrm{Sym}\big(\delta_{ij}\boldsymbol{h}_s(z) + \sqrt{(s+2)(s+1)} \cdot \boldsymbol{h}_{s+2}(z)[e_i \otimes e_j]$$

$$+ s \cdot \boldsymbol{h}_s(z)[e_j] \otimes e_i + s \cdot \boldsymbol{h}_s(z)[e_i] \otimes e_j + \sqrt{s(s-1)} \cdot \boldsymbol{h}_{s-2}(z) \otimes e_i \otimes e_j\big)\Big].$$

Since $F \in \mathbb{R}_{s+2}[z]$ is arbitrary, we conclude that

$$z_i z_j \boldsymbol{h}_s(z) = \mathrm{Sym}\big(\sqrt{(s+2)(s+1)} \cdot \boldsymbol{h}_{s+2}(z)[e_i \otimes e_j] + \delta_{ij}\boldsymbol{h}_s(z) + s \cdot \boldsymbol{h}_s(z)[e_j] \otimes e_i$$

$$+ s \cdot \boldsymbol{h}_s(z)[e_i] \otimes e_j + \sqrt{s(s-1)} \cdot \boldsymbol{h}_{s-2}(z) \otimes e_i \otimes e_j\big).$$

The proof is completed by further taking the tensor inner product operation with respect to $(\theta^\star)^{\otimes s}$ on both side. □

**Proposition I.2.** *Let $w, w' \in \mathbb{S}^{d-1}$ and $s \in \mathbb{N}_0$. We have*
$$\mathbb{E}\left[h_i(\langle w, z\rangle)h_j(\langle w', z\rangle) \cdot \boldsymbol{h}_s(z)\right]$$

$$= \sum_{\tau=0}^s \mathbb{1}(j = i + s - 2\tau, i \geq \tau) \cdot \binom{s}{\tau} \sqrt{\frac{i!j!}{s!((i-\tau)!)^2}} \cdot \langle w, w'\rangle^{i-\tau} \cdot \mathrm{Sym}\big(w^{\otimes \tau} \otimes w'^{\otimes s-\tau}\big).$$

*The above term is equal to*
$$\binom{s}{(i-j+s)/2} \cdot \sqrt{\frac{i!j!}{s!}} \cdot \frac{\langle w, w'\rangle^{(i+j-s)/2}}{((i+j-s)/2)!} \cdot \mathrm{Sym}\big(w^{\otimes (i-j+s)/2} \otimes w'^{\otimes (j-i+s)/2}\big).$$

*if $|i - j| \leq s \leq i + j$ and $s \equiv i - j \mod 2$. Otherwise the expectation gives the zero tensor.*

*Proof of Proposition I.2.* By the Stein's lemma for the Hermite tensor (B.5), we have

$$\mathbb{E}\left[h_i(\langle w, z\rangle)h_j(\langle w', z\rangle) \cdot \boldsymbol{h}_s(z)\right] = \frac{1}{\sqrt{s!}} \cdot \mathbb{E}\left[\nabla^s(h_i(\langle w, z\rangle)h_j(\langle w', z\rangle))\right]$$

$$= \frac{1}{\sqrt{s!}} \cdot \sum_{\tau=0}^s \binom{s}{\tau} \mathbb{E}\left[\mathrm{Sym}\left(\nabla^\tau h_i(\langle w, z\rangle) \otimes \nabla^{s-\tau} h_j(\langle w', z\rangle)\right)\right]$$

$$= \frac{1}{\sqrt{s!}} \cdot \sum_{\tau=0}^s \binom{s}{\tau} \sqrt{\frac{i!j!}{(i-\tau)!(j-s+\tau)!}} \mathbb{E}\left[h_{i-\tau}(\langle w, z\rangle) \otimes h_{j-s+\tau}(\langle w', z\rangle) \cdot \mathrm{Sym}(w^{\otimes \tau} \otimes w'^{\otimes s-\tau})\right]$$

$$= \sum_{\tau=0}^s \mathbb{1}(j = i + s - 2\tau, i \geq \tau) \cdot \frac{\binom{s}{\tau}}{(i-\tau)!} \sqrt{\frac{i!j!}{s!}} \langle w, w'\rangle^{i-\tau} \mathrm{Sym}\big(w^{\otimes \tau} \otimes w'^{\otimes s-\tau}\big).$$

the condition can be translated into $|i - j| \leq s \leq i + j$ and $s \equiv i - j \mod 2$. Then, we can take $\tau = (i - j + s)/2$, $s - \tau = (j - i + s)/2$, $i - \tau = (i + j - s)/2$ to obtain the desired result. $\square$

**Lemma I.3.** *Let $\psi : \mathbb{R} \to \mathbb{R}$ such that $\psi^2 \in L^2(\mathcal{N})$. Suppose that $\psi$ is high-pass in the sense that $\widehat{\psi}_i = 0$ for any $i < s^\star - 1$ for some $s^\star \in \mathbb{N}_0$. For $w, w' \in \mathbb{S}^{d-1}$, take a series of test tensor $\{T_s = v_1 \otimes v_2 \otimes \ldots \otimes v_s \in (\mathbb{R}^d)^{\otimes s}\}_{s=0}^{\infty}$ such that $\sup_{i > c_0}\{|\langle w, v_i \rangle| \vee |\langle w', v_i \rangle|\} \leq \epsilon$ for some $\epsilon \in (0, 1/2)$ and integer $c_0 \in \mathbb{N}_0$. Let*

$$\epsilon_0 := \max_{1 \leq i \leq c_0} \{|\langle w, v_i \rangle| \vee |\langle w', v_i \rangle|\}.$$

*Suppose that $|\langle w, w' \rangle| \leq \epsilon_1$. Then we have for any $s \in \mathbb{N}_0$ that*

$$\left| \mathbb{E}_{z \sim \mathcal{N}_d}\left[ \psi(\langle w, z \rangle)\psi(\langle w', z \rangle) \cdot \boldsymbol{h}_s(z)[T_s] \right] \right| \leq 4\|\psi\|_2^2 (4es^\star)^{s/2} \sqrt{s + s^\star} \cdot \epsilon_1^{(s^\star - 1 - \lfloor s/2 \rfloor) \vee 0} \cdot \epsilon_0^{s \wedge c_0} \cdot \epsilon^{(s - c_0) \vee 0},$$

*where $\|\psi\|_2^2 = \mathbb{E}_{x \sim \mathcal{N}}[\psi^2(x)]$.*

*Proof of Lemma I.3.* As $\psi(x)\psi(x')$ is also square-integrable, we are able to extract the $s$-th tensor coefficient of the Hermite expansion of $\psi(x)\psi(x')$ as

$$\mathbb{E}_{z \sim \mathcal{N}_d}\left[ \psi(\langle w, z \rangle)\psi(\langle w', z \rangle) \cdot \boldsymbol{h}_s(z) \right]$$

$$= \mathbb{E}_{z \sim \mathcal{N}_d}\left[ \sum_{i=0}^{\infty} \sum_{j=0}^{\infty} \widehat{\psi}_i \widehat{\psi}_j h_i(\langle w, z \rangle) h_j(\langle w', z \rangle) \cdot \boldsymbol{h}_s(z) \right]$$

$$= \sum_{i=0}^{\infty} \sum_{j=0}^{\infty} \sum_{\tau=0}^{s} \mathbb{1}(j = i + s - 2\tau, i \geq \tau) \cdot \binom{s}{\tau} \sqrt{\frac{i!j!}{s!((i - \tau)!)^2}} \widehat{\psi}_i \widehat{\psi}_j \cdot \langle w, w' \rangle^{i - \tau} \cdot \mathrm{Sym}\left( w^{\otimes \tau} \otimes w'^{\otimes s - \tau} \right)$$

$$= \sum_{i=0}^{\infty} \sum_{\tau=0}^{s} \mathbb{1}(i \geq \tau) \binom{s}{\tau} \sqrt{\frac{1}{s!}} \sqrt{\frac{i!(i + s - 2\tau)!}{((i - \tau)!)^2}} \widehat{\psi}_i \widehat{\psi}_{i+s-2\tau} \cdot \langle w, w' \rangle^{i - \tau} \cdot \mathrm{Sym}\left( w^{\otimes \tau} \otimes w'^{\otimes (s - \tau)} \right),$$

where the last second identity follows from Proposition I.2, and in the last line we restrict the condition $j = i + s - 2\tau$. Note that the double sums are interchangeable only if the series converges for each $\tau$. However, by our condition that $\langle w, w' \rangle \leq 1 - \epsilon$ for some $\epsilon > 0$, then for each $\tau$, we have for any test tensor $T = v_1 \otimes v_2 \otimes \ldots \otimes v_s$ with $\|v_i\|_2 = 1$ that

$$\left| \sum_{i=\tau}^{\infty} \sqrt{\frac{i!(i + s - 2\tau)!}{((i - \tau)!)^2}} \widehat{\psi}_i \widehat{\psi}_{i+s-2\tau} \cdot \langle w, w' \rangle^{i - \tau} \cdot \mathrm{Sym}\left( w^{\otimes \tau} \otimes w'^{\otimes (s - \tau)} \right)[T] \right|$$

$$\leq \sum_{i=\tau}^{\infty} (i + s)^{s/2} \cdot \left| \widehat{\psi}_i \widehat{\psi}_{i+s-2\tau} \right| \cdot (1 - \epsilon)^{i - \tau} < \infty,$$

where we note that $1 - \epsilon$ will dominate the polynomial growth of $(i + s)^{s/2}$, and also using the fact that $\widehat{\psi}_i \widehat{\psi}_{i+s-2\tau}$ is uniformly bounded by $\sum_{i=0}^{\infty} \widehat{\psi}_i^2 < \infty$. Now, we interchange the double sum and apply the high-pass assumption and have that

$$\mathbb{E}_{z \sim \mathcal{N}_d}\left[ \psi(\langle w, z \rangle)\psi(\langle w', z \rangle) \cdot \boldsymbol{h}_s(z)[T] \right]$$

$$= \sum_{\tau=0}^{s} \binom{s}{\tau} \sqrt{\frac{1}{s!}} \underbrace{\sum_{i=i_0}^{\infty} \sqrt{\frac{i!(i + s - 2\tau)!}{((i - \tau)!)^2}} \widehat{\psi}_i \widehat{\psi}_{i+s-2\tau} \cdot \langle w, w' \rangle^{i - \tau}}_{(\mathrm{I})} \cdot \mathrm{Sym}\left( w^{\otimes \tau} \otimes w'^{\otimes (s - \tau)} \right)[T].$$

where we define

$$i_0 = \max\left\{ (s^\star - 1), (s^\star - 1 + 2\tau - s), \tau \right\}$$

Note that the tensor product part is independent of $i$. Hence, we pull out term (I) and have

$$(\mathrm{I}) = \sqrt{\frac{i_0!(i_0 + s - 2\tau)!}{((i_0 - \tau)!)^2}} \widehat{\psi}_{i_0} \widehat{\psi}_{i_0 + s - 2\tau} \langle w, w' \rangle^{i_0 - \tau}$$

$$\pm \|\psi\|_2^2 \cdot \langle w, w' \rangle^{i_0 - \tau + 1} \left( \sum_{j=0}^{\infty} (j + s + s^\star)(j + s + s^\star - 1) \cdots (j + s^\star + \lceil (s + 1)/2 \rceil) |\langle w, w' \rangle|^j \right).$$

Here, the first term is given by splitting out the term with $i = i_0$ from the summation, and the second term is for $i \geq i_0 + 1$. For the second term, we have the following argument:

$$
\begin{aligned}
\max\{i_0, i_0 + s - 2\tau\} &= (s^\star - 1) \vee (s^\star - 1 + 2\tau - s) \vee \tau \vee (s^\star - 1 + s - 2\tau) \vee (s^\star - 1) \vee (s - \tau) \\
&= (s^\star - 1 + 2\tau - s) \vee (s^\star - 1 + s - 2\tau) \vee \tau \vee (s - \tau) \\
&= (s^\star - 1 + |2\tau - s|) \vee (|\tau - s/2| + s/2) \leq s^\star + s - 1.
\end{aligned}
$$

Hence, we then have that for any $i \geq i_0 + 1$ with $j = i - (i_0 + 1)$ that

$$
\max\{i, i + s - 2\tau\} \leq s^\star + s + j.
$$

Therefore, for any $i \geq i_0 + 1$, we have

$$
\sqrt{\frac{i!(i + s - 2\tau)!}{((i - \tau)!)^2}} = \sqrt{i(i-1)\cdots(i - \tau + 1)} \cdot \sqrt{(i + s - 2\tau)(i + s - 2\tau - 1)\cdots(i - \tau + 1)}
$$

$$
\leq \sqrt{(j + s^\star + s)(j + s^\star + s - 1)\cdots(j + s^\star + 1)}
$$

$$
\leq (j + s^\star + s)(j + s^\star + s - 1)\cdots(j + s^\star + \lceil(s+1)/2\rceil).
$$

To characterize the second term, we use Proposition J.5 where we have conditions $|\langle w, w'\rangle| < 1/2$ and $s^\star + s \geq 2\lfloor(s+1)/2\rfloor - 1$ satisfied, which gives us

$$
\sum_{j=0}^\infty (j + s + s^\star)(j + s + s^\star - 1)\cdots(j + s^\star + \lceil(s+1)/2\rceil)|\langle w, w'\rangle|^j
$$

$$
\leq 2(s + s^\star)\cdots(s^\star + \lceil(s+1)/2\rceil) \cdot \frac{1}{1 - |\langle w, w'\rangle|}
$$

$$
\leq 4(s + s^\star)\cdots(s^\star + \lceil(s+1)/2\rceil).
$$

Combining these results, we have for (I) that

$$
\text{(I)} = \sqrt{\frac{i_0!(i_0 + s - 2\tau)!}{((i_0 - \tau)!)^2}} \widehat{\psi}_{i_0} \widehat{\psi}_{i_0 + s - 2\tau} \langle w, w'\rangle^{i_0 - \tau} \pm \|\psi\|_2^2 \cdot 4(s + s^\star)\cdots(s^\star + \lceil(s+1)/2\rceil) \cdot \langle w, w'\rangle^{i_0 - \tau + 1}
$$

$$
= C(s^\star, s, \tau, \langle w, w'\rangle) \cdot \langle w, w'\rangle^{i_0 - \tau},
$$

where we define $C(s^\star, s, \tau, \langle w, w'\rangle) = \text{(I)}/\langle w, w'\rangle^{i_0 - \tau}$ as the coefficient, which is given by

$$
C(s^\star, s, \tau, \langle w, w'\rangle) = \sqrt{\frac{i_0!(i_0 + s - 2\tau)!}{((i_0 - \tau)!)^2}} \widehat{\psi}_{i_0} \widehat{\psi}_{i_0 + s - 2\tau} \pm \|\psi\|_2^2 \cdot 4(s + s^\star)\cdots(s^\star + \lceil(s+1)/2\rceil) \cdot |\langle w, w'\rangle|,
$$

and also enjoys the following upper bound

$$
|C(s^\star, s, \tau, \langle w, w'\rangle)| \leq \|\psi\|_2^2 \cdot 4(s + s^\star - 1)\cdots(s^\star - 1 + \lceil(s+1)/2\rceil). \tag{I.2}
$$

Here, the upper bound can be obtained by noting that the previous upper bound for terms $i \geq i_0 + 1$ can be also applied to $i \geq i_0$.

**Case $s \geq 1$.** Let us plug in test tensor $T_s$ into the expression, which gives us

$$
|\mathbb{E}_{z \sim \mathcal{N}_d}[\psi(\langle w, z\rangle)\psi(\langle w', z\rangle) \cdot \boldsymbol{h}_s(z)[T_s]]|
$$

$$
\leq \sum_{\tau=0}^s \binom{s}{\tau} \sqrt{\frac{1}{s!}} |C(s^\star, s, \tau, \langle w, w'\rangle)| \cdot |\langle w, w'\rangle|^{i_0 - \tau} \cdot \left|\text{Sym}(w^{\otimes \tau} \otimes w'^{\otimes(s-\tau)})[T_s]\right|
$$

$$
\leq \sum_{\tau=0}^s \binom{s}{\tau} \sqrt{\frac{1}{s!}} |C(s^\star, s, \tau, \langle w, w'\rangle)| \cdot |\langle w, w'\rangle|^{i_0 - \tau} \cdot \left(\frac{1}{s!} \sum_{\pi \in \Pi_s} \prod_{i=1}^\tau |\langle w, v_{\pi(i)}\rangle| \prod_{j=\tau+1}^s |\langle w', v_{\pi(j)}\rangle|\right),
$$

where $\Pi_s$ denotes the set of all permutations of $s$ elements. We further have this term bounded by

$$
|\mathbb{E}_{z \sim \mathcal{N}_d}[\psi(\langle w, z\rangle)\psi(\langle w', z\rangle) \cdot \boldsymbol{h}_s(z)[T_s]]| \tag{I.3}
$$

$$
\leq \sum_{\tau=0}^s \binom{s}{\tau} \sqrt{\frac{1}{s!}} \max_{0 \leq \tau \leq s} |C(s^\star, s, \tau, \langle w, w'\rangle)| \cdot |\langle w, w'\rangle|^{i_0 - \tau} \cdot \epsilon_0^{s \wedge c_0} \cdot \epsilon^{(s - c_0) \vee 0}
$$

$$
\leq \frac{2^s}{\sqrt{s!}} \max_{0 \leq \tau \leq s} |C(s^\star, s, \tau, \langle w, w'\rangle)| \cdot |\langle w, w'\rangle|^{(s^\star - 1 - \lfloor s/2 \rfloor) \vee 0} \cdot \epsilon_0^{s \wedge c_0} \cdot \epsilon^{(s - c_0) \vee 0}.
$$

where the last inequality follows from the following fact

$$i_0 - \tau = (s^\star - 1 - \tau) \vee (s^\star - 1 + \tau - s) \vee 0 = (s^\star - 1 - s/2 + |\tau - s/2|) \vee 0$$
$$\geq (s^\star - 1 - s/2 + \mathbb{1}(s \text{ odd})/2) \vee 0 = (s^\star - 1 - \lfloor s/2 \rfloor) \vee 0.$$

Plugging (I.2) into (I.3), and by noting that $|\langle w, v_i \rangle| \vee |\langle w', v_i \rangle| \leq \epsilon$ for any $v_i \in \{\theta^\star, v\}$ and $|\langle w, w' \rangle| \leq \epsilon$, we have that

$$|\mathbb{E}_{z \sim \mathcal{N}_d} [\psi(\langle w, z \rangle)\psi(\langle w', z \rangle) \cdot \boldsymbol{h}_s(z)[T_s]]|$$
$$\leq \frac{2^s}{\sqrt{s!}} \|\psi\|_2^2 \cdot 4(s + s^\star - 1) \cdots (s^\star - 1 + \lceil (s+1)/2 \rceil) \cdot \epsilon_1^{(s^\star - 1 - \lfloor s/2 \rfloor) \vee 0} \cdot \epsilon_0^{s \wedge c_0} \cdot \epsilon^{(s - c_0) \vee 0}$$
$$\leq \left( \frac{4e(s + s^\star - 1)}{s} \right)^{s/2} \cdot \sqrt{s + s^\star - 1} \cdot 4\|\psi\|_2^2 \cdot \epsilon_1^{(s^\star - 1 - \lfloor s/2 \rfloor) \vee 0} \cdot \epsilon_0^{s \wedge c_0} \cdot \epsilon^{(s - c_0) \vee 0}$$
$$\leq (4es^\star)^{s/2} \cdot \sqrt{s + s^\star} \cdot 4\|\psi\|_2^2 \cdot \epsilon_1^{(s^\star - 1 - \lfloor s/2 \rfloor) \vee 0} \cdot \epsilon_0^{s \wedge c_0} \cdot \epsilon^{(s - c_0) \vee 0}.$$

Here, the second inequality follows from the Stirling's approximation, and the last inequality holds because $s \geq 1$.

**Case $s = 0$.** For the case $s = 0$, we have that

$$|\mathbb{E}_{z \sim \mathcal{N}_d} [\psi(\langle w, z \rangle)\psi(\langle w', z \rangle)]| = \left| C(s^\star, 0, 0, \langle w, w' \rangle) \cdot \langle w, w' \rangle^{s^\star - 1} \right| \leq 4\|\psi\|_2^2 \epsilon_1^{s^\star - 1},$$

which can also be upper bounded by the quantity derived for the case $s \geq 1$. Hence, the proof is completed. $\qquad \square$

**Proposition I.4.** *Let $\psi(\cdot, \cdot)^2$ satisfies the quadratic integrability condition and high-pass condition in Assumption 4.1 with $s^\star$ being the generative exponent. Suppose the remaining definitions ($c_0$, $\{T_s\}_{s=0}^\infty$, $\epsilon$, $\epsilon_0$, $\epsilon_1$) and conditions are the same as in Lemma I.3. Suppose $c_0 \in \{0, 1, 2\}$ and $s_0 \in \mathbb{N}_0$. Take function series $\{\zeta_s(\cdot)\}_{s=0}^\infty$ satisfying $\mathbb{E}_\mathbb{Q}[\zeta_s(y)^2] < C, \forall s \in \mathbb{N}_0$ for some universal $C = O(1)$. Suppose $4e\epsilon^2 < 1/2$. If $s^\star = 1$, then we have that*

$$\sum_{s \geq s_0} (s+2) \cdot |\mathbb{E}_\mathbb{Q} [\zeta_s(y)\psi(y, \langle w, z \rangle)\psi(y, \langle w', z \rangle)\boldsymbol{h}_s(z)[T_s]]| \lesssim \epsilon_0^{s_0} + \epsilon_0^{c_0} \epsilon^{(s_0 - c_0) \vee 0}.$$

*If $s^\star \geq 2$, then we have that*

$$\sum_{s \geq s_0} (s+2) \cdot |\mathbb{E}_\mathbb{Q} [\zeta_s(y)\psi(y, \langle w, z \rangle)\psi(y, \langle w', z \rangle)\boldsymbol{h}_s(z)[T_s]]|$$
$$\lesssim \mathbb{1}(s_0 \leq c_0) \cdot \left( \epsilon_1^{s^\star - 1 - \lfloor s_0/2 \rfloor} \cdot \epsilon_0^{s_0} + \epsilon_1^{s^\star - 1 - \lfloor c_0/2 \rfloor} \cdot \epsilon_0^{c_0} \right) + \epsilon_0^{c_0} \cdot \epsilon^{(2s^\star) \vee s_0 - c_0}$$
$$+ \mathbb{1}(s_0 \leq 2(s^\star - 1)) \cdot \left( \epsilon_1^{s^\star - 1 - \lfloor (c_0+1)/2 \rfloor} \cdot \epsilon_0^{c_0} \cdot \epsilon + \epsilon_0^{c_0} \cdot \epsilon^{2(s^\star - 1) + 1 - c_0} \right).$$

*Here, $\lesssim$ hides constants that depend on $s_0, c_0, C, \mathbb{E}_\mathbb{Q}[\psi(y, x)^4]$.*

*Proof of Proposition I.4.* Let $F$ denote the target quantity. Invoking Lemma I.3 for each degree $s$, we have that

$$F \leq \sum_{s \geq s_0} \sqrt{s(s+1)} \cdot \mathbb{E}_{y \sim \mathbb{Q}} |\mathbb{E}_{z \sim \mathcal{N}_d} [\zeta_s(y)\psi(y, \langle w, z \rangle)\psi(y, \langle w', z \rangle)\boldsymbol{h}_s(z)[T_s]]|$$
$$\leq \sum_{s \geq s_0} \sqrt{s(s+1)} \cdot \mathbb{E}_{y \sim \mathbb{Q}} \left[ 4|\zeta_s(y)|\mathbb{E}_{x \sim \mathcal{N}}[\psi(y, x)^2] \right] (4es^\star)^{s/2} \sqrt{s + s^\star} \epsilon_1^{(s^\star - 1 - \lfloor s/2 \rfloor) \vee 0} \epsilon_0^{s \wedge c_0} \epsilon^{(s - c_0) \vee 0}$$
$$\leq \sum_{s \geq s_0} \sqrt{s(s+1)} \cdot 4\sqrt{\mathbb{E}_\mathbb{Q}[\zeta_s(y)^2]\mathbb{E}_\mathbb{Q}[\psi(y, x)^4]} (4es^\star)^{s/2} \sqrt{s + s^\star} \epsilon_1^{(s^\star - 1 - \lfloor s/2 \rfloor) \vee 0} \epsilon_0^{s \wedge c_0} \epsilon^{(s - c_0) \vee 0},$$

where the last inequality follows from the Cauchy-Schwarz inequality. Noting that by our assumptions, $\mathbb{E}_\mathbb{Q}[\zeta_s(y)^2]\mathbb{E}_\mathbb{Q}[\psi(y, x)^4] = O(1)$ uniformly over $s$, which gives us

$$F \lesssim \sum_{s \geq s_0} \sqrt{s(s+1)(s+s^\star)} (4es^\star)^{s/2} \epsilon_1^{(s^\star - 1 - \lfloor s/2 \rfloor) \vee 0} \epsilon_0^{s \wedge c_0} \epsilon^{(s - c_0) \vee 0}.$$

For the case $s^\star = 1$, we have the above term controlled by

$$F\big|_{s^\star=1} \lesssim \sum_{s \geq s_0} \sqrt{s(s+1)^2}\,(4e)^{s/2}\,\epsilon_0^{s \wedge c_0}\,\epsilon^{(s-c_0)\vee 0}$$

$$\lesssim \mathbb{1}(c_0 > s_0) \cdot \sup_{s_0 \leq s \leq c_0-1} \epsilon_0^s + \sum_{s \geq c_0 \vee s_0} \sqrt{s(s+1)^2}\,(4e)^{s/2}\,\epsilon_0^{c_0}\,\epsilon^{s-c_0}$$

$$\lesssim \mathbb{1}(c_0 > s_0) \cdot \epsilon_0^{s_0} + \epsilon_0^{c_0}\,\epsilon^{(s_0-c_0)\vee 0} \sum_{s \geq c_0 \vee s_0} \sqrt{s(s+1)^2}\,\big(4e\epsilon^2\big)^{(s-c_0 \vee s_0)/2}$$

$$\lesssim \mathbb{1}(s_0 < c_0) \cdot \epsilon_0^{s_0} + \epsilon_0^{c_0}\,\epsilon^{(s_0-c_0)\vee 0} \lesssim \epsilon_0^{c_0 \wedge s_0}\,\epsilon^{(s_0-c_0)\vee 0},$$

where $\lesssim$ only hides constants that depend on $s_0, c_0$. The last second inequality holds by noting that $4e\epsilon^2 < 1/2$.

For the case $s^\star \geq 2$, we note that $c_0 \leq 2 \leq 2(s^\star - 1)$, and we have

$$F\big|_{s^\star \geq 2} \lesssim \sum_{s \geq s_0} \sqrt{s(s+1)(s+s^\star)}\,(4es^\star)^{s/2}\,\epsilon_1^{(s^\star-1-\lfloor s/2 \rfloor)\vee 0}\,\epsilon_0^{s \wedge c_0}\,\epsilon^{(s-c_0)\vee 0}$$

$$\lesssim \mathbb{1}(s_0 \leq c_0) \cdot \max_{s_0 \leq s \leq c_0} \epsilon_1^{s^\star-1-\lfloor s/2 \rfloor} \cdot \epsilon_0^s$$

$$+ \mathbb{1}(s_0 \leq 2(s^\star-1)) \cdot \max_{c_0+1 \leq s \leq 2(s^\star-1)+1} \epsilon_1^{s^\star-1-\lfloor s/2 \rfloor} \cdot \epsilon_0^{c_0} \cdot \epsilon^{s-c_0}$$

$$+ \sum_{s \geq (2(s^\star-1)+2)\vee s_0} \sqrt{s(s+1)(s+s^\star)}\,(4es^\star)^{s/2}\,\epsilon_0^{c_0}\,\epsilon^{s-c_0}$$

$$\lesssim \mathbb{1}(s_0 \leq c_0) \cdot \Big(\epsilon_1^{s^\star-1-\lfloor s_0/2 \rfloor} \cdot \epsilon_0^{s_0} + \epsilon_1^{s^\star-1-\lfloor c_0/2 \rfloor} \cdot \epsilon_0^{c_0}\Big) + \epsilon_0^{c_0} \cdot \epsilon^{(2(s^\star-1)+2)\vee s_0-c_0}$$

$$+ \mathbb{1}(s_0 \leq 2(s^\star-1)) \cdot \Big(\epsilon_1^{s^\star-1-\lfloor (c_0+1)/2 \rfloor} \cdot \epsilon_0^{c_0} \cdot \epsilon + \epsilon_0^{c_0} \cdot \epsilon^{2(s^\star-1)+1-c_0}\Big).$$

Thus, we complete the proof. $\qquad\square$

### I.2 Technical Results for Uniform Distribution on the Sphere

**Lemma I.5** (Moment of polynomial on a sphere, adapted from Folland (2001)). *Let* $\xi = (\xi_1, \xi_2, \ldots, \xi_d) \sim Unif(\mathbb{S}^{d-1})$ *and* $s_1, s_2, \ldots, s_d \in \mathbb{N}_0$. *Let* $s = \sum_{i=1}^d s_i$. *Then we have*

$$\mathbb{E}_\xi\left[\prod_{i=1}^d \xi_i^{s_i}\right] = \begin{cases} 0 & \text{if some } s_i \text{ is odd,} \\ \prod_{i=1}^d \dfrac{\Gamma((s_i+1)/2)}{\Gamma(1/2)} \cdot \dfrac{\Gamma(d/2)}{\Gamma((s+d)/2)} & \text{if all } s_i \text{ are even,} \end{cases}$$

To evaluate this moment, we provide the following bound.

**Fact I.6.** *Take even degrees* $s_1, s_2, \ldots, s_d \in \mathbb{N}_0$ *and* $s = \sum_{i=1}^d s_i$. *Then we have*

$$\left(\frac{1}{d}\right)^{s/2} \leq \prod_{i=1}^d \frac{\Gamma((s_i+1)/2)}{\Gamma(1/2)} \cdot \frac{\Gamma(d/2)}{\Gamma((s+d)/2)} \leq \left(\frac{s}{d}\right)^{s/2},$$

*where we define* $(0)^0 = 1$.

*Proof of Fact I.6.* Note that we can rewrite the product as

$$\prod_{i=1}^d \frac{\Gamma((s_i+1)/2)}{\Gamma(1/2)} \cdot \frac{\Gamma(d/2)}{\Gamma((s+d)/2)} = \frac{\prod_{i=1}^d (s_i-1)!!}{(s+d-2) \cdot (s+d-4) \cdots d}.$$

Using the fact that $a/b \geq (a-1)/(b-1)$ for $2 \leq a \leq b$, we can recursively apply this inequality to the factorial until it gives us the desired lower bound of $(1/d)^{s/2}$. For the upper bound, we can lower bound the denominator by $d^{s/2}$ and upper bound the numerator by $s^{s/2}$. We thus have the desired result. $\qquad\square$

Note that the second moment $\mathbb{E}[\xi_i^2] \asymp d^{-1}$, thus Fact I.6 can be viewed as some kind of *reverse Holder's inequality* where we use lower moment to control higher moment. Notably, the reverse inequality gives a *dimension-free* bound for the moments of the polynomial on the sphere. The following proposition formalizes this intuition.

**Proposition I.7.** *Suppose $\xi = (\xi_1, \xi_2, \ldots, \xi_d) \sim Unif(\mathbb{S}^{d-1})$. Let $f : \mathbb{R}^d \to \mathbb{R}$ be a function such that $\mathbb{E}[(f(\xi) - 1)^2] \le \varepsilon^2$. Take nonnegative degree $s = (s_1, s_2, \ldots, s_d)$ with $\|s\|_1 = s$, and suppose each $s_i$ is even for $i \in [d]$. We then have*

$$(1 - (2s)^{s/2}\varepsilon) \cdot \left(\frac{1}{d}\right)^{s/2} \le \mathbb{E}\left[f(\xi) \cdot \prod_{i=1}^d \xi_i^{s_i}\right] \le (1 + 2^{s/2}\varepsilon) \cdot \left(\frac{s}{d}\right)^{s/2}.$$

*Proof of Proposition I.7.* By the Cauchy-Schwarz inequality, we have for any $j = 0, 1, \ldots, s_1/2$,

$$\left|\mathbb{E}\left[(f(\xi) - 1) \cdot \prod_{i=1}^d \xi_i^{s_i}\right]\right| \le \sqrt{\mathbb{E}[(f(\xi) - 1)^2]} \cdot \sqrt{\mathbb{E}\left[\prod_{i=1}^d \xi_i^{2s_i}\right]} \le \varepsilon \cdot \left(\frac{2s}{d}\right)^{s/2},$$

where we use Lemma I.5 and the upper bound in Fact I.6 for the second inequality. Additionally, note that each $s_i$ is even, we use the same argument to have

$$\left(\frac{1}{d}\right)^{s/2} \le \mathbb{E}\left[\prod_{i=1}^d \xi_i^{s_i}\right] \le \left(\frac{s}{d}\right)^{s/2}.$$

Combining these two inequalities, we conclude the proof. □

**Proposition I.8.** *Let $\xi = (\xi_1, \xi_2, \ldots, \xi_d) \sim Unif(\mathbb{S}^{d-1})$, $\gamma > 0$ be a fixed parameter, and $s = (s_1, s_2, \ldots, s_d)$ be a nonnegative integer vector with $\|s\|_1 = s$. It then holds that*

$$G(s) := \mathbb{E}\left[(\xi_1 + \gamma)^{s_1} \cdot \prod_{i=2}^d \xi_i^{s_i}\right] = \mathbb{E}\left[\sum_{j=0}^{\lfloor s_1/2 \rfloor} \binom{s_1}{2j} \gamma^{s_1 - 2j} \cdot \xi_1^{2j} \cdot \prod_{i=2}^d \xi_i^{s_i}\right]$$

$$= \begin{cases} 0, & \text{if some } s_i \text{ is odd for } i = 2, 3, \ldots, d, \\ C(s, \gamma) \cdot \left(\gamma + \frac{1}{\sqrt{d}}\right)^{2\lfloor s_1/2 \rfloor} \cdot \gamma^{\mathbb{1}(s_1 \text{ odd})} \cdot \left(\frac{1}{\sqrt{d}}\right)^{s - s_1}, & \text{otherwise,} \end{cases}$$

*where $1/5 \le C(s, \gamma) \le s^{s/2}$.*

*Proof of Proposition I.8.* Note that

$$G(s) = \mathbb{E}\left[(\xi_1 + \gamma)^{s_1} \cdot \prod_{i=2}^d \xi_i^{s_i}\right] = \mathbb{E}\left[\sum_{j=0}^{s_1} \binom{s_1}{j} \xi_1^j \gamma^{s_1 - j} \cdot \prod_{i=2}^d \xi_i^{s_i}\right].$$

Note that if there exists any odd degree $s_i$ for $i \ge 2$, then $G(s) = 0$. For $s_2, s_3, \ldots, s_d$ being even, we have

$$G(s) = \mathbb{E}\left[\sum_{j=0}^{\lfloor s_1/2 \rfloor} \binom{s_1}{2j} \gamma^{s_1 - 2j} \cdot \xi_1^{2j} \cdot \prod_{i=2}^d \xi_i^{s_i}\right]$$

$$= \sum_{j=0}^{\lfloor s_1/2 \rfloor} \binom{s_1}{2j} \gamma^{s_1 - 2j} \cdot \frac{\Gamma(j + 1/2)}{\Gamma(1/2)} \cdot \prod_{i=2}^d \frac{\Gamma((s_i + 1)/2)}{\Gamma(1/2)} \cdot \frac{\Gamma(d/2)}{\Gamma((s - s_1 + 2j + d)/2)}$$

$$= \sum_{j=0}^{\lfloor s_1/2 \rfloor} \binom{s_1}{2j} \gamma^{s_1 - 2j} \cdot \underbrace{\frac{(2j - 1)!! \cdot \prod_{i=2}^d (s_i - 1)!!}{(s - s_1 + 2j + d - 2) \cdot (s - s_1 + 2j + d - 4) \cdots d}}_{(I)},$$

where for the second identity, we use Lemma I.5 to compute the moments of the polynomial on the sphere. Now, we compute the lower and upper bounds of $G(s)$.

**Lower Bound.** For the lower bound, we have

$$G(s) \geq \sum_{j=0}^{\lfloor s_1/2 \rfloor} \binom{s_1}{2j} \gamma^{s_1-2j} \cdot d^{-j-(s_2+s_3+\cdots+s_d)/2}$$

$$\geq \frac{1}{5} \cdot 5 \sum_{j=0}^{\lfloor s_1/2 \rfloor} \binom{2\lfloor s_1/2 \rfloor}{2j} \cdot \gamma^{2\lfloor s_1/2 \rfloor - 2j} \left(\frac{1}{\sqrt{d}}\right)^{2j} \cdot d^{-(s_2+s_3+\cdots+s_d)/2} \cdot \gamma^{\mathbb{1}(s_1 \text{ odd})}$$

$$\geq \frac{1}{5} \cdot \sum_{j=0}^{2\lfloor s_1/2 \rfloor} \binom{2\lfloor s_1/2 \rfloor}{j} \cdot \gamma^{2\lfloor s_1/2 \rfloor - j} \left(\frac{1}{\sqrt{d}}\right)^{j} \cdot d^{-(s_2+s_3+\cdots+s_d)/2} \cdot \gamma^{\mathbb{1}(s_1 \text{ odd})}.$$

Here in the first inequality, we use the fact that $a/b \geq (a-1)/(b-1)$ for $2 \leq a \leq b$ and apply it recursively to the factorial (I) until it gives us $d^{-j-(s_2+s_3+\cdots+s_d)/2}$. For the second inequality, we first rearrange the terms in the summation and lower bound the binomial coefficients by changing $s_1$ to $2\lfloor s_1/2 \rfloor$. For the last inequality, let us define

$$A_j = \binom{2\lfloor s_1/2 \rfloor}{j} \cdot \gamma^{2\lfloor s_1/2 \rfloor - j} \left(\frac{1}{\sqrt{d}}\right)^{j}.$$

We invoke Lemma J.4 and have that for each odd $j$, we have that

$$A_j \leq 2(A_{j-1} + A_{j+1}), \quad j = 1, 3, \ldots, 2\lfloor s_1/2 \rfloor - 1.$$

Therefore, the summation of all the odd terms is upper bounded by $4$ times the summation of all the even terms, which gives us the last inequality. Therefore, the lower bound of $G(s)$ is

$$G(s) \geq \frac{1}{5} \cdot \left(\gamma + \frac{1}{\sqrt{d}}\right)^{2\lfloor s_1/2 \rfloor} \cdot \gamma^{\mathbb{1}(s_1 \text{ odd})} \cdot \left(\frac{1}{\sqrt{d}}\right)^{(s-s_1)}.$$

**Upper Bound.** For the upper bound, we have

$$G(s) \leq \sum_{j=0}^{\lfloor s_1/2 \rfloor} \binom{s_1}{2j} \gamma^{s_1-2j} \cdot d^{-j-(s_2+s_3+\cdots+s_d)/2} \cdot s^{s/2}$$

$$\leq \sum_{j=0}^{2\lfloor s_1/2 \rfloor} \binom{2\lfloor s_1/2 \rfloor}{j} \cdot \gamma^{2\lfloor s_1/2 \rfloor - j} \left(\frac{1}{\sqrt{d}}\right)^{j} \cdot d^{-(s_2+s_3+\cdots+s_d)/2} \cdot \gamma^{\mathbb{1}(s_1 \text{ odd})} \cdot s^{s/2}$$

$$= s^{s/2} \cdot \left(\gamma + \frac{1}{\sqrt{d}}\right)^{2\lfloor s_1/2 \rfloor} \cdot \gamma^{\mathbb{1}(s_1 \text{ odd})} \cdot \left(\frac{1}{\sqrt{d}}\right)^{-(s-s_1)},$$

where in the first line, we lower bound the denominator in the factorial (I) by $d^{j+(s_2+s_3+\cdots+s_d)/2}$ and upper bound the numerator by $s^{s/2}$. For the second inequality, we use the nonnegativity of each terms and append the terms with $j$ being odd to the summation.

Combining the lower and upper bounds, we have that

$$G(s) = \begin{cases} 0, & \text{if some } s_i \text{ is odd for } i = 2, 3, \ldots, d, \\ C(s, \gamma) \cdot \left(\gamma + \frac{1}{\sqrt{d}}\right)^{2\lfloor s_1/2 \rfloor} \cdot \gamma^{\mathbb{1}(s_1 \text{ odd})} \cdot \left(\frac{1}{\sqrt{d}}\right)^{-(s-s_1)}, & \text{otherwise,} \end{cases}$$

for $1/5 \leq C(s, \gamma) \leq s^{s/2}$. Hence, the proof is complete. $\square$

## I.3 TECHNICAL RESULTS ON POLARIZED RANDOM VECTORS

**Lemma I.9** (Moments of weakly polarized random vector). *Suppose* $\xi = (\xi_1, \xi_2, \ldots, \xi_d) \sim Unif(\mathbb{S}^{d-1})$ *and define the polarized vector* $w$ *as*

$$w = \frac{\xi + \gamma e_1}{\|\xi + \gamma e_1\|_2},$$

*where $\gamma = o(1) > 0$ is a parameter that describes the polarization strength, and $e_1 = (1, 0, \ldots, 0)$ is the first standard basis vector. Take nonnegative integer degree $\boldsymbol{s} = (s_1, s_2, \ldots, s_d)$ with $\|\boldsymbol{s}\|_1 = s = O(1) < (2\sqrt{e}\gamma)^{-1} - 2$. Then we have*

$$\mathbb{E}_w\left[\prod_{i=1}^d w_i^{s_i}\right] = \begin{cases} 0, & \text{if some } s_i \text{ is odd for } i = 2, 3, \ldots, d, \\ C(\boldsymbol{s}, \gamma) \cdot \left(\gamma + \frac{1}{\sqrt{d}}\right)^{s_1} \cdot \left(\frac{1}{\sqrt{d}}\right)^{s - s_1}, & \text{if } s_1 \text{ is even, } s_2, \ldots, s_d \text{ are even,} \\ C(\boldsymbol{s}, \gamma) \cdot \gamma \cdot \left(\gamma + \frac{1}{\sqrt{d}}\right)^{s_1 - 1} \cdot \left(\frac{1}{\sqrt{d}}\right)^{s - s_1}, & \text{if } s_1 \text{ is odd, } s_2, \ldots, s_d \text{ are even,} \end{cases}$$

*where $1/5 - O(\gamma) \le C(\boldsymbol{s}, \gamma) \le s^{(s+1)/2} + O(\gamma)$.*

*Proof of Lemma I.9.* Let $r = \|\xi + \gamma e_1\|_2$. We reformulate the moment as

$$\mathbb{E}\left[\prod_{i=1}^d w_i^{s_i}\right] = \mathbb{E}\left[\frac{(\xi_1 + \gamma)^{s_1}}{r^s} \cdot \prod_{i=2}^d \xi_i^{s_i}\right].$$

By symmetry, we have the moment equal zero if some $s_i$ is odd for $i = 2, 3, \ldots, d$. Hence, we only need to consider $s_2, s_3, \ldots, s_d$ being even. In the following, we study the case for $s_1$ being even and odd separately.

**Case 1: $s_1$ is even.** We first look at the simpler case where $s_1$ is even. We note by weak polarization in the sense of $\|\xi\|_2 \gg \|\gamma e_1\|_2$, we can approximate $1/r^s$ as 1 with approximation error

$$\left|\mathbb{E}\left[\prod_{i=1}^d w_i^{s_i}\right] - \mathbb{E}\left[(\xi_1 + \gamma)^{s_i} \cdot \prod_{i=2}^d \xi_i^{s_i}\right]\right| \le \sqrt{\mathbb{E}\left[(1 - r^{-s})^2\right]} \cdot \sqrt{\mathbb{E}\left[(\xi_1 + \gamma)^{2s_1} \cdot \prod_{i=2}^d \xi_i^{2s_i}\right]}$$

$$\le es\gamma \cdot \sqrt{\mathbb{E}\left[(\xi_1 + \gamma)^{2s_1} \cdot \prod_{i=2}^d \xi_i^{2s_i}\right]}, \qquad (I.4)$$

where we use the Cauchy-Schwarz inequality in the first line and for the second line, we use the fact

$$|1 - r^{-s}| \le \frac{1}{(1-\gamma)^s} - 1 = \left(1 + \frac{\gamma}{1-\gamma}\right)^s - 1 \le \exp\left(\frac{s\gamma}{1-\gamma}\right) - 1 \le es\gamma \qquad (I.5)$$

for $s < (2\sqrt{e}\gamma)^{-1}$. Define

$$G(\boldsymbol{s}) := \mathbb{E}\left[(\xi_1 + \gamma)^{s_i} \cdot \prod_{i=2}^d \xi_i^{s_i}\right].$$

By Proposition I.8, we have that for even $s_1, s_2, \ldots, s_d$,

$$G(\boldsymbol{s}) = C'(\boldsymbol{s}, \gamma) \cdot \left(\gamma + \frac{1}{\sqrt{d}}\right)^{s_1} \cdot \left(\frac{1}{\sqrt{d}}\right)^{s - s_1},$$

with $1/5 \le C'(\boldsymbol{s}, \gamma) \le s^{s/2}$. Plugging the form of $G(\boldsymbol{s})$ into (I.4), we have that

$$\mathbb{E}_w\left[\prod_{i=1}^d w_i^{s_i}\right] = M(\boldsymbol{s}) \pm es\gamma \cdot \sqrt{M(2\boldsymbol{s})}$$

$$= \left(C'(\boldsymbol{s}, \gamma) \pm es\gamma\sqrt{C'(2\boldsymbol{s}, \gamma)}\right) \cdot \left(\gamma + \frac{1}{\sqrt{d}}\right)^{s_1} \cdot \left(\frac{1}{\sqrt{d}}\right)^{s - s_1}$$

$$= \left(C'(\boldsymbol{s}, \gamma) \pm es\gamma(2s)^{s/2}\right) \cdot \left(\gamma + \frac{1}{\sqrt{d}}\right)^{s_1} \cdot \left(\frac{1}{\sqrt{d}}\right)^{s - s_1}.$$

Here, $C'(\boldsymbol{s}, \gamma) - es\gamma(2s)^{s/2} \ge 1/5 - es\gamma(2s)^{s/2}$ and $C'(\boldsymbol{s}, \gamma) + es\gamma(2s)^{s/2} \le s^{s/2} + es\gamma(2s)^{s/2}$.

**Case 2: $s_1$ is odd.** We now consider the more complicated case where $s_1$ is odd. In the following proof, we will frequently invoke the following proposition, whose proof is deferred to Proposition I.7 of Appendix H.

**Proposition I.7** (Restated). *Suppose $\xi = (\xi_1, \xi_2, \ldots, \xi_d) \sim \text{Unif}(\mathbb{S}^{d-1})$. Let $f : \mathbb{R}^d \to \mathbb{R}$ be a function such that $\mathbb{E}[(f(\xi) - 1)^2] \leq \varepsilon^2$. Take nonnegative degree $\boldsymbol{s} = (s_1, s_2, \ldots, s_d)$ with $\|\boldsymbol{s}\|_1 = s$, and suppose each $s_i$ is even for $i \in [d]$. We then have*

$$(1 - (2s)^{s/2}\varepsilon) \cdot \left(\frac{1}{d}\right)^{s/2} \leq \mathbb{E}\left[f(\xi) \cdot \prod_{i=1}^d \xi_i^{s_i}\right] \leq (1 + 2^{s/2}\varepsilon) \cdot \left(\frac{s}{d}\right)^{s/2}.$$

We first rewrite the moment as

$$\mathbb{E}\left[\prod_{i=1}^d w_i^{s_i}\right] = \mathbb{E}\left[\frac{(\xi_1 + \gamma)^{s_1}}{r^s} \cdot \prod_{i=2}^d \xi_i^{s_i}\right] \tag{I.6}$$

$$= \underbrace{\sum_{j=0}^{(s_1-1)/2} \binom{s_1}{2j}\gamma^{s_1-2j} \cdot \mathbb{E}\left[\frac{\xi_1^{2j}}{r^s} \cdot \prod_{i=2}^d \xi_i^{s_i}\right]}_{\text{(I) even terms}} + \underbrace{\sum_{j=0}^{(s_1-1)/2} \binom{s_1}{2j+1}\gamma^{s_1-2j-1} \cdot \mathbb{E}\left[\frac{\xi_1^{2j+1}}{r^s} \cdot \prod_{i=2}^d \xi_i^{s_i}\right]}_{\text{(II) odd terms}}.$$

Let us look at the odd terms of (I.6). Let $r_+ = \|\xi + \gamma e_1\|_2$ and $r_- = \|-\xi + \gamma e_1\|_2$. By symmetry, we have for the expectation within the odd terms that

$$\mathbb{E}\left[\frac{\xi_1^{2j+1}}{r^s} \cdot \prod_{i=2}^d \xi_i^{s_i}\right] = \frac{1}{2} \cdot \mathbb{E}\left[\left(\frac{1}{r_+^s} - \frac{1}{r_-^s}\right) \cdot \xi_1^{2j+1} \cdot \prod_{i=2}^d \xi_i^{s_i}\right]$$

$$= \frac{1}{2} \cdot \mathbb{E}\left[\frac{(r_-^2 - r_+^2)\left(\sum_{l=0}^{s-1} r_-^l r_+^{s-1-l}\right)}{r_+^s \cdot r_-^s \cdot (r_+ + r_-)} \cdot \xi_1^{2j+1} \cdot \prod_{i=2}^d \xi_i^{s_i}\right]$$

$$= -\gamma \cdot \mathbb{E}\left[\frac{2 \cdot \sum_{l=0}^{s-1} r_-^l r_+^{s-1-l}}{r_+^s \cdot r_-^s \cdot (r_+ + r_-)} \cdot \xi_1^{2j+2} \cdot \prod_{i=2}^d \xi_i^{s_i}\right].$$

Here, the last identity holds by noting that $r_-^2 - r_+^2 = -4\gamma\xi_1$. Note that

$$\sup_{\xi \in \mathbb{S}^{d-1}} \left|\frac{2 \cdot \sum_{l=0}^{s-1} r_-^l r_+^{s-1-l}}{r_+^s \cdot r_-^s \cdot (r_+ + r_-)} - 1\right| \leq \left(\frac{1}{1-\gamma}\right)^{s+2} - 1 \leq e\gamma(s+2).$$

Hence, we have for the odd terms (II) that

$$-\text{(II)} = \mathbb{E}\left[\frac{2 \cdot \sum_{l=0}^{s-1} r_-^l r_+^{s-1-l}}{r_+^s \cdot r_-^s \cdot (r_+ + r_-)} \cdot \sum_{j=0}^{(s_1-1)/2} \binom{s_1}{2j+1}\gamma^{s_1-2j} \cdot \xi_1^{2j+2} \cdot \prod_{i=2}^d \xi_i^{s_i}\right]$$

$$\leq \sum_{j=0}^{(s_1-1)/2} \binom{s_1}{2j+1}\gamma^{s_1-2j} \cdot (1 + 2^{(s+2)/2}e\gamma(s+2)) \cdot \left(\frac{s+2}{d}\right)^{(s-s_1+2j+2)/2}$$

$$\leq s_1 \cdot (1 + 2^{(s+2)/2}e\gamma(s+2)) \cdot \left(\frac{s+2}{d}\right)^{(s-s_1+2)/2} \cdot \gamma \cdot \sum_{j=0}^{s_1-1} \binom{s_1-1}{j} \cdot \gamma^{s_1-1-j} \cdot \left(\sqrt{\frac{s+2}{d}}\right)^j,$$

where the first inequality holds by Proposition I.7 and the second inequality holds by $\binom{s_1}{2j+1}/\binom{s_1-1}{2j} = s_1/(2j+1) \leq s_1$ and also appending the odd terms to the summation. Thus, we have

$$-\text{(II)} \leq s_1 \cdot (1 + 2^{(s+2)/2}e\gamma(s+2)) \cdot \gamma \cdot \left(\gamma + \sqrt{\frac{s+2}{d}}\right)^{s_1-1} \cdot \left(\sqrt{\frac{s+2}{d}}\right)^{s-s_1+2}.$$

Next, we study the even terms (I) in (I.6). Using Proposition I.7 with the uniform bound in (I.5), we have (I) upper bounded by

$$
\begin{aligned}
\text{(I)} &= \sum_{j=0}^{(s_1-1)/2} \binom{s_1}{2j} \gamma^{s_1-2j} \cdot \mathbb{E}\left[ \frac{\xi_1^{2j}}{r^s} \cdot \prod_{i=2}^{d} \xi_i^{s_i} \right] \\
&\leq (1 + 2^{s/2} es\gamma) \cdot \sum_{j=0}^{(s_1-1)/2} \binom{s_1}{2j} \gamma^{s_1-2j} \cdot \left( \sqrt{\frac{s}{d}} \right)^{s-s_1+2j} \\
&\leq (1 + 2^{s/2} es\gamma) \cdot s_1 \cdot \sum_{j=0}^{s_1-1} \binom{s_1-1}{j} \cdot \gamma^{s_1-1-j} \cdot \left( \sqrt{\frac{s}{d}} \right)^{j} \cdot \left( \sqrt{\frac{s}{d}} \right)^{s-s_1} \cdot \gamma \\
&= s_1 \cdot (1 + 2^{s/2} es\gamma) \cdot \gamma \cdot \left( \gamma + \sqrt{\frac{s}{d}} \right)^{s_1-1} \cdot \left( \sqrt{\frac{s}{d}} \right)^{s-s_1}.
\end{aligned}
$$

Similarly, we have the lower bound for (I) as

$$
\begin{aligned}
\text{(I)} &= \sum_{j=0}^{(s_1-1)/2} \binom{s_1}{2j} \gamma^{s_1-2j} \cdot \mathbb{E}\left[ \frac{\xi_1^{2j}}{r^s} \cdot \prod_{i=2}^{d} \xi_i^{s_i} \right] \\
&\geq (1 - (2s)^{s/2} es\gamma) \cdot \sum_{j=0}^{(s_1-1)/2} \binom{s_1}{2j} \gamma^{s_1-2j} \cdot \left( \sqrt{\frac{1}{d}} \right)^{s-s_1+2j} \\
&\geq \frac{1}{5}(1 - (2s)^{s/2} es\gamma) \cdot \sum_{j=0}^{s_1-1} \binom{s_1-1}{j} \gamma^{s_1-1-j} \cdot \left( \sqrt{\frac{1}{d}} \right)^{j} \cdot \left( \sqrt{\frac{1}{d}} \right)^{s-s_1} \cdot \gamma \\
&= \frac{1}{5}(1 - (2s)^{s/2} es\gamma) \cdot \gamma \cdot \left( \gamma + \sqrt{\frac{1}{d}} \right)^{s_1-1} \cdot \left( \sqrt{\frac{1}{d}} \right)^{s-s_1}.
\end{aligned}
$$

Combining these upper and lower bounds for (I) together with the upper bound for (II), we have

$$
\mathbb{E}\left[ \prod_{i=1}^{d} w_i^{s_i} \right] \geq \left( \frac{1}{5}(1 - (2s)^{s/2} es\gamma) - \frac{s_1 \cdot (1 + 2^{(s+2)/2} e\gamma(s+2)) \cdot (s+2)^{(s+2)/2}}{d} \right)
$$
$$
\cdot \gamma \cdot \left( \gamma + \sqrt{\frac{1}{d}} \right)^{s_1-1} \cdot \left( \sqrt{\frac{1}{d}} \right)^{s-s_1},
$$

and

$$
\mathbb{E}\left[ \prod_{i=1}^{d} w_i^{s_i} \right] \leq (1 + 2^{s/2} es\gamma) \cdot s^{(s+1)/2} \cdot \gamma \cdot \left( \gamma + \sqrt{\frac{1}{d}} \right)^{s_1-1} \cdot \left( \sqrt{\frac{1}{d}} \right)^{s-s_1}.
$$

Hence, we complete our proof. $\qquad \square$

## J  AUXILIARY LEMMAS

**Lemma J.1** (Bernstein's inequality). *Let $X_1, \ldots, X_n$ be independent random variables with $|X_i - \mathbb{E}[X_i]| \leq C$ for all $i \in [n]$. Then for any $t > 0$, it holds that*

$$
\mathbb{P}\left( \left| \frac{1}{n} \sum_{i=1}^{n} X_i - \frac{1}{n} \sum_{i=1}^{n} \mathbb{E}X_i \right| \geq t \right) \leq 2\exp\left( -\frac{nt^2/2}{n^{-1} \cdot \sum_{i=1}^{n} \mathrm{Var}[X_i] + Ct/3} \right),
$$

*or equivalently, for any $\delta \in (0, 1)$,*

$$
\mathbb{P}\left( \left| \frac{1}{n} \sum_{i=1}^{n} X_i - \mathbb{E}X_i \right| \leq \sqrt{\frac{2 \cdot n^{-1} \sum_{i=1}^{n} \mathrm{Var}[X_i] \cdot \log \delta^{-1}}{n}} + \frac{C \log \delta^{-1}}{3n} \right) \geq 1 - \delta.
$$

For the vector case, by a union bound over all the coordinates, we have the following corollary.

**Corollary J.2** (Vector version of Bernstein's inequality). *Let $X_1, \ldots, X_n$ be independent random vectors in $\mathbb{R}^d$ with $\|X_i - \mathbb{E}[X_i]\|_\infty \leq C$ for all $i \in [n]$. Then for any $\delta \in (0,1)$, it holds with probability at least $1 - \delta$ that*

$$\left\| \frac{1}{n} \sum_{i=1}^n X_i \right\|_2 \lesssim \left\| \mathbb{E}\left[ \frac{1}{n} \sum_{i=1}^n X_i \right] \right\|_2 + \sqrt{\frac{n^{-1} \sum_{i=1}^n \operatorname{Tr}(\operatorname{Cov}[X_i]) \cdot \log(d\delta^{-1})}{n}} + \frac{\sqrt{d} C \log(d\delta^{-1})}{n}.$$

**Lemma J.3** (Lemma I.3. in Damian et al. (2024)). *Let $X_1, \ldots, X_n \in \mathbb{R}^d$ be independent mean-zero random vectors such that for all $p \geq 2$, $\mathbb{E}[\|X_i\|^p]^{1/p} \leq Cp^{k/2}$ for some constants $k, C \geq 0$ and vector norm $\|\cdot\|$. Define $\sigma^2 := n^{-1} \sum_{i=1}^n \mathbb{E}[\|X_i\|^2]$ and $Y := n^{-1} \cdot \sum_{i=1}^n X_i$. Then with probability at least $1 - 2\delta$,*

$$\|Y\| \lesssim \sigma \cdot \sqrt{\frac{\log(1/\delta)}{n}} + \frac{C \log(1/\delta) \log(n/\delta)^{k/2}}{n}.$$

*where $\lesssim$ only hides constant that depends on $k$.*

**Lemma J.4** (Ratio bound of binomial expansion). *For real numbers $a, b > 0$ and integer $s \geq 2$, define*

$$A_j := \binom{s}{j} \cdot a^{s-j} \cdot b^j, \quad \text{for } j = 0, 1, \ldots, s.$$

*Then for all $j = 1, 2, \ldots, s - 1$, it holds that $A_j \leq 2(A_{j-1} + A_{j+1})$.*

*Proof of Lemma J.4.* By the definition of $A_j$,

$$\min\left\{ \frac{A_j}{A_{j-1}}, \frac{A_j}{A_{j+1}} \right\} = \min\left\{ \frac{b}{a} \cdot \frac{s-j+1}{j}, \frac{a}{b} \cdot \frac{j+1}{s-j} \right\} \leq \sqrt{\frac{b}{a} \cdot \frac{s-j+1}{j} \cdot \frac{a}{b} \cdot \frac{j+1}{s-j}}$$

$$\leq \sqrt{\left(1 + \frac{1}{s-j}\right)\left(1 + \frac{1}{j}\right)} \leq 2.$$

Hence, the proof is complete by the nonnegativity of $A_j$. $\qquad\square$

**Proposition J.5.** *For $\epsilon \in [0, 1/2)$ and $s, r \in \mathbb{N}_0$ with $s \geq 2r - 1$, it holds that*

$$\sum_{j=0}^\infty (j+s)(j+s-1)\cdots(j+s-r+1) \cdot \epsilon^j \leq 2s \cdot (s-1) \cdots (s-r+1) \cdot \frac{1}{1-\epsilon}.$$

*Proof of Proposition J.5.* Denote $F(x) = \sum_{j=0}^\infty (j+s)(j+s-1)\cdots(j+s-r+1)x^j$ for $x \in (0, 1)$. The desired quantity on the left-hand side of the inequality is simply $F(\epsilon)$. It can be verified using the expansion of $1/(1-x) = \sum_{j=0}^\infty x^j$ for $x \in (0, 1)$ that

$$F(x) = \frac{\mathrm{d}^r}{\mathrm{d}x^r}\left(\frac{x^s}{1-x}\right) \cdot x^{-(s-r)}.$$

Expanding this expression, we have

$$F(x) = \sum_{\tau=0}^r \binom{r}{\tau} \frac{\mathrm{d}^\tau}{\mathrm{d}x^\tau}(x^s) \cdot \frac{\mathrm{d}^{r-\tau}}{\mathrm{d}x^{r-\tau}}\left(\frac{1}{1-x}\right) \cdot x^{-(s-r)}$$

$$= \sum_{\tau=0}^r \binom{r}{\tau} s(s-1)\cdots(s-\tau+1) \cdot x^{s-\tau} \cdot \frac{(-1)^{r-\tau}(r-\tau)!}{(1-x)^{r-\tau+1}} \cdot x^{-(s-r)}$$

$$= \sum_{\tau=0}^r (r \cdot (r-1) \cdots (\tau+1)) \cdot (s \cdot (s-1) \cdots (s-\tau+1)) \cdot (-1)^{r-\tau} \cdot \frac{x^{r-\tau}}{(1-x)^{r-\tau+1}}.$$

Write the above summation as $F(x) = \sum_{\tau=0}^{r} F_\tau(x)$, where $F_\tau(x)$ is the $\tau$-th term in the summation. Then for each $\tau = 0, \ldots, r-1$, when $x \in (0,1)$, we have

$$\frac{F_\tau(x)}{F_{\tau+1}(x)} = -\frac{s-\tau}{\tau+1} \cdot \frac{1-x}{x} < -1.$$

Note that $F_r(x)$ is positive, and for each $k = 1, \ldots, \lfloor r/2 \rfloor$, $F_{r-2k+1}(x) < -F_{r-2k}(x) < 0$. Since $F(x)$ is positive, it is then upper bounded by $F_\tau(x) + |F_0(x)|$, i.e.,

$$F(x) \le s \cdot (s-1) \cdots (s-r+1) \cdot \frac{1}{1-x} + r! \cdot \frac{x^r}{(1-x)^{r+1}} \le 2s \cdot (s-1) \cdots (s-r+1) \cdot \frac{1}{1-x}.$$

The proof is complete by setting $x = \epsilon$. $\qquad\square$

**Lemma J.6** (Gaussian-like tail bound for spherical coordinate). *Suppose $\xi \sim \text{Unif}(\mathbb{S}^{d-1})$. Then the first coordinate of $\xi$, denoted by $\xi_1$, satisfies that for any $t \ge 0$,*

$$\Pr(\xi_1 \ge \sqrt{Cd^{-1}\log d}) \le \exp(-d/16) + d^{-C/4},$$

*where $C > 0$ is a constant.*

*Proof of Lemma J.6.* Consider $z \sim \mathcal{N}(0, I_d)$, and it holds that $\xi_1 \overset{d}{=} z_1/\|z\|_2$, where $z_1$ is the first coordinate of $z$. Note that $\|z\|_2^2 \sim \chi_d^2$, and by a standard tail bound for the $\chi_d^2$ distribution, we have

$$\Pr(\|z\|_2^2 \le d - 2\sqrt{dx}) \le \exp(-x), \quad \text{for any } x \ge 0.$$

By taking $x = d/16$, we get $\Pr(\|z\|_2^2 \le d/2) \le \exp(-d/16)$. Thus, applying a union bound,

$$\Pr(\xi_1 \ge t) = \Pr\left(\frac{z_1}{\|z\|_2} \ge t\right) \le \Pr\left(\|z\|_2^2 \le d/2\right) + \Pr(z_1 \ge t\sqrt{d/2})$$

$$\le \exp(-d/16) + \exp(-t^2 d/4).$$

The proof is complete by taking $t = \sqrt{Cd^{-1}\log d}$. $\qquad\square$

**Lemma J.7** (Hypergeometric behavior). *Consider random variable $X \sim \text{Hypergeometric}(d, k, k)$ with probability mass*

$$\Pr(X = x) = \frac{\binom{k}{x}\binom{d-k}{k-x}}{\binom{d}{k}}, \quad \text{for } x = 1, 2, \ldots k.$$

*Suppose $k = o(\sqrt{d})$, then for any constant $s > 0$, it holds that $\mathbb{E}[X^s] \simeq k^2/d$. In addition, the following tail bound holds:*

$$\Pr(X \ge \log k) \lesssim (k^2/d)^{\log k}.$$

*Proof of Lemma J.7.* We first notice that for $x \ge k^2/d$, since $k = o(\sqrt{d})$,

$$\frac{\Pr(X = x+1)}{\Pr(X = x)} = \frac{(k-x)^2}{(x+1)(d-2k-x+1)} = \left(1 - \frac{k}{d}\right)^2 \bigg/ \left(\frac{d}{k^2} + o\left(\frac{d}{k^2}\right)\right) \lesssim \frac{k^2}{d}. \quad \text{(J.1)}$$

This immediately implies the tail bound:

$$\Pr(X \ge \log k) \le \sum_{j=\lceil \log k \rceil}^{k} \Pr(X = j) \lesssim \left(\frac{k^2}{d}\right)^{\log k}.$$

Next, for the moment $\mathbb{E}[X^s]$, we first study the magnitude of $\mathbb{P}(X = 0)$, which, by Stirling's approximation, is given by

$$\Pr(X = 0) = \frac{((d-k)!)^2}{d! \cdot (d-2k)!} \simeq \frac{(d-k)^{2(d-k)+1} \cdot e^{-2(d-k)}}{d^{d+1/2} \cdot (d-2k)^{(d-2k)+1/2} \cdot e^{-2(d-k)}}$$

$$= \frac{(1 - 2k/d + k^2/d^2)^{d-k+1/2}}{(1 - 2k/d)^{d-2k+1/2}}$$

$$= \left(1 + \frac{1}{(1-2k/d) \cdot d^2/k^2}\right)^{d-k+1/2} \left(1 - \frac{2k}{d}\right)^{k}.$$

Since $k = o(\sqrt{d})$, we have $(1 - 2k/d)^k \simeq 1$. Further applying the fact that $(1 + 1/m)^m = \Theta(1)$,

$$\Pr(X = 0) \simeq \left(1 + \frac{1}{(1 - 2k/d) \cdot d^2/k^2}\right)^{(1-2k/d) \cdot d^2/k^2 \cdot k^2/d} = \Theta(1).$$

As a consequence, using the first equality in (J.1), we can lower bound the expectation of $X^s$ by

$$\mathbb{E}[X^s] \geq \Pr(X = 1) = \Pr(X = 0) \cdot \frac{k^2}{d - 2k + 1} \gtrsim \frac{k^2}{d}.$$

For the upper bound, we again use the first equality in (J.1) to get

$$\Pr(X = x + 1) \leq \frac{k^2}{(x+1)(d - 2k - x + 1)} \cdot \Pr(X = x) \lesssim \frac{\Pr(X = x)}{x + 1} \cdot \frac{k^2}{d}.$$

Recursive application of this inequality yields that $\mathbb{P}(X = x) \lesssim \mathbb{P}(X = 0) \cdot (k^2/d)^x/x!$, and thus

$$\mathbb{E}[X^s] = \sum_{x=1}^{k} x^s \cdot \Pr(X = x) \leq \Pr(X = 0) \cdot \sum_{x=1}^{k} \frac{x^s}{x!}\left(\frac{k^2}{d}\right)^x \lesssim \frac{k^2}{d}.$$

Therefore, we conclude that $\mathbb{E}[X^s] \simeq k^2/d$. This completes the proof. $\qquad\square$

