# OpenReview forum: "Can Neural Networks Achieve Optimal Computational-statistical Tradeoff? An Analysis on Single-Index Model"
_ICLR.cc/2025/Conference — ICLR 2025 Oral_

### Official Review · Reviewer_TYz2 · 2024-10-26

**Soundness:** 3
**Presentation:** 3
**Contribution:** 3
**Rating:** 8
**Confidence:** 3

**Summary:**

The author consider the problem of recovering the unknown direction in Gaussian single-index models. They present a class of online algorithms for that match the Statistical Query lower bound for this problem. As specific realizations of this algorithm, they consider the class where their iteration matches a randomized gradient-type update scheme. They also present results in the case of sparse models.

**Strengths:**

The class of algorithms they present seem to capture other state-of-the-art online SGD schemes, such as "batched reuse" SGD (for generative exponent $\leq  2$) and "landscape smoothing" (for generative exponent $\geq 2$). The key difference in the latter case, is the use a tailored randomized objective that allows them to improve the rates down to the SQ lower bound, as opposed to the CSQ bound, which is a substantial improvement.

**Weaknesses:**

*Content*:  The algorithm for generative exponent > 2 seems quite tailored to the problem. You need to know the generative exponent of the, nominally unknown, link function. See questions below.


*Presentation:*  My main concern with this paper is the presentation. While globally, the paper is reasonably well-written, it could do with a thorough round of editing. I had trouble interpreting essential aspects of this paper.  I have tried to made precise suggestions in "Questions" below.

*Correctness:* The paper seems plausibly correct, though the sketch presented in Appendix D appears to have a flaw? Unfortunately given the timescale of refereeing, I could not find a fix to this on my own. I'm sure the authors will find a quick fix. See "Questions" below. [I will retract this comment, upon clarification from the authors during the discussion period.]

**Questions:**

As the authors will see, my rating is largely due to the presentation. Please consider making some of the following edits and/or answering my questions and I will gladly raise my score.

**Major comments:**

*Alg 1:* I am having trouble understanding the definition of the main algorithm, Alg 1. What is $err^t$? It never seems to be defined in the paper (main body or appendix).

*Eq D.2* Was this sketch intended only for the case of $s_* \geq 2$? If $s_* = 1$ then the $\tilde{O}$ term in the norm bound in Eq D.2 seems to be the dominant term and divergent.

*Gradient based:* It is not to me why the authors refer to their algorithm as gradient based. On the one hand, their algorithm as stated is more general, it is an online algorithm given some function $\psi$ which, in certain cases, is a gradient.

However this does not seem to correspond to an optimization algorithm for risk based inference on a loss of the form $L(y - f(z,\theta))$.
How can you get a small loss since the algorithm is only updating the hidden layer? Also, why can't we just take $a = 1$ in this algorithm?

It seems that the answer is that something essential is baked-in to Assumption 4.1(b).  E.g., if $y$ can be both positive and negative but we take $a,\sigma$ to both be positive we run in to issues; This will presumably violate (b)? Am I missing something?

Also, perhaps your point is that you *can* view it as an optimization scheme where youre finding the best fit in this model with constant second layer, and we don't really care about minimizing the loss because our goal is really to infer $\theta_*$?


**Minor questions/comments + typos:**

*unknown link* Contributions item 2, you state that the link function is unknown.  Is the assumption that the generative exponent mild? Can you access this given just the data/without knowing the link? What happens if it is misspecified?

*Definition of model* In Eq 2.1 you define the model, but your first example, immediately after seems to not fit in to this framework due to overloaded notation. Perhaps choose different letters for the link *function*  and link *probability*?

*Ln 83:* "thus there is no...gap..." I do not see how this is concluded. what you stated only implies that the algorithmic and information theoretic thresholds are the same order of magnitude. They can still be different, no?

*Ln 176:* The notation $\mathcal{N}(z^\perp;0,I)$ should be defined.

*L^p* The notation $\stackrel{L^2(P)}{=}$ is a bit excessive. Perhaps just write = ?

*Assumption 4.1* Please condense this. For example, instead of "Both $\psi(x,y)^2$ and $\psi(x,y)^2x^2$ are square integrable..." maybe just "$\psi(x,y),\psi(x,y)\cdot x \in L^4(Q)$? Also, the remainder of Item (a) is more of a downstream consequence. Perhaps move this observation outside of the assumption environment.

*Ln 254* "Fourier coefficient" Perhaps "Hermite coefficient"?
*Ln 302* Netwok ---> Network ?

*Ln1280* “follows from [a] Bernstien type...”

---

### Official Review · Reviewer_mmjr · 2024-11-03

**Soundness:** 3
**Presentation:** 3
**Contribution:** 3
**Rating:** 6
**Confidence:** 4

**Summary:**

This paper studies the learning of Gaussian single index models with an SQ complexity bound using a gradient-based technique. The main contributions are twofold: 1) They demonstrate that a carefully tuned two-layer neural network, trained with a gradient-based algorithm, can learn a non-polynomial link function within an SQ complexity framework, extending prior results that focused on polynomial link functions. Notably, their analysis is agnostic to the specific form of the link function. 2) They further extend these results to sparse single index models, achieving the optimal SQ bound in the sparse setting.

**Strengths:**

* The paper is reasonably well-written and provides ample details on its technical contributions.
* It extends the existing results on learning with SQ bounds from polynomial link functions to more general non-polynomial link functions.
* The proof techniques are involved, involving careful arguments to address challenges such as avoiding zero correlation and enhancing the signal-to-noise ratio without prior knowledge of the link function.

**Weaknesses:**

* Although the paper considers neural networks, the proof techniques require numerous modifications that make the proposed algorithm somewhat unnatural. For example, the use of Hermite polynomials as the activation function is required to apply the weight perturbation technique, and the specific label transformation depends on the cumulative distribution function (CDF) of the labels.

* Notably, the second assumption for label transformation and Assumption C.2 are not discussed in the main text. I recommend that the authors include a "Limitations" section to explicitly outline the constraints of their proof techniques for future reference.

**Questions:**

* Is there any potential to use losses other than the squared loss to avoid label transformation, similar to [1]?

* In the case of sparse priors, prior work [2] also employs a similar approach to that used in this paper, involving index-sensitive perturbation followed by gradient sparsification and support identification via a top-k operator, achieving the CSQ-optimal bound for sparse priors. Could the authors elaborate on how their technique differs from this previous approach and discuss the challenges encountered in extending these CSQ results to the SQ setting?

[1] Joshi, N., Misiakiewicz, T., & Srebro, N. (2024). On the Complexity of Learning Sparse Functions with Statistical and Gradient Queries. ArXiv, abs/2407.05622.

[2] Vural, N., & Erdogdu, M.A. (2024). Pruning is Optimal for Learning Sparse Features in High-Dimensions. Annual Conference Computational Learning Theory.

---

### Official Review · Reviewer_A2wU · 2024-11-03

**Soundness:** 3
**Presentation:** 3
**Contribution:** 3
**Rating:** 8
**Confidence:** 4

**Summary:**

This paper studies the problem of learning single-index models with neural networks. Under a general gradient oracle, the authors present an algorithm that can achieve the optimal computational-statistical tradeoff by obtaining a sample complexity of $\tilde{O}(d^{s^*/2} \lor d)$, and contains the batch-reuse approach of prior work as an example for learning polynomials where $s^* \leq 2$. Furthermore, the authors construct a random loss for each single-index model that implements this general gradient oracle for all $s^*$ when using the $s^*$ degree Hermite polynomial as the activation. Finally, the authors consider the case where the single-index model has additional sparsity, proving an SQ sample complexity lower bound of $n = \tilde{\Omega}(k^{s^*})$ for $k = o(\sqrt{d})$, and presenting an algorithm that matches this lower bound.

**Strengths:**

The paper has solid technical contributions and presents an alternative to the tensor power iteration algorithm of Damian et al., 2024 for learning single-index models with an algorithm that instead relies on a training procedure for neurons. In particular, having a framework that encompasses both batch-reusing and loss modification approaches of prior works is valuable. Further investigating the sparse case presents a nice example of additional hidden structure that can benefit sample complexity.

**Weaknesses:**

* The original motivation as stated in the abstract is to see whether neural networks trained with gradient-based methods can obtain the optimal computational-statistical tradeoff. But the algorithm presented, while interesting from a technical perspective, has little resemblance to the training of neural networks in practice. In that sense, the problem of having a more standard gradient-based algorithm that achieves this tradeoff is still open.

* I believe the writing can be improved in terms of clarity and citing relevant works, with some examples provided below.

**Questions:**

* There are some ambiguities in the discussion starting from Line 258:
    * The paragraph before introduces a general class of gradients $\psi(y,x)z$, but here we switch back to activations. What kind of loss function are we using here? The calculation seems to be for squared loss, but my impression was that we have to change the loss to go beyond correlational queries in this online setting.
    * The identity in Line 267 might require further clarification. For example, I think $E_Q[y^2]$ should be replaced with $E_Q[\zeta_s(y)y^2]$. Maybe we can apply Cauchy-Schwartz from here to use the assumption on $\zeta_s$, but this would lead to the fourth moment of $y$ appearing. Perhaps the authors could add more details on the derivation to make it clearer.
     * My impression was that the only benefit of weight perturbation is to give a smaller denominator in the SNR, however, that doesn't match the following argument. If I try to replicate the SNR calculation of the paragraph starting at Line 220 for this paragraph, the numerator in the SNR is still $d^{-(s^*-1)}$ while the new denominator coming from weight perturbation is $d^{-(s^*-3)/2}$. This leads to an SNR of order $d^{-(s^*+1)/2}$. We argued we need $\sqrt{n\mathrm{SNR}} \gg d^{-1/2}$, therefore $n$ should be of the order $d^{(s^*-1)/2}$. Are there other effects happening when we perturb the weights?

* Line 320 mentions that $L = \tilde{\Omega}(n/\sqrt{d})$ is sufficient. Given $n = d^{s^*/2}\lor d$, this leads to $L = d^{(s^*-1)/2}\lor d^{1/2}$ (which seems to be used in Remark 4.3 as well), which is different from $L$ in Theorem 4.2.

* The loss defined at the end of Page 22 is not exactly a loss function, for example it seems that it can decrease while $y$ is moving away from zero due to the derivative being negative. If this interpretation is correct, perhaps it could be highlighted in the main text that the goal of this loss is not to measure the difference with ground-truth labels, but to construct suitable gradient oracles.

* Can Assumption C.2 be verified for distributions with high generative exponent?

* Theorem 4.2 only states a lower bound on the learning rate. To me it seems like the ideal scenario would be to let the learning rate go to infinity, and directly update the model weights with the gradients as in Algorithm 2. Is there a reason we would want to have a finite learning rate in Algorithm 1?

* Relevant prior works:
    * Theorem 5.4 resembles Theorem 3.1 of [VE24], which presents the CSQ lower bound under sparsity, where generative exponent is replaced with information exponent. I think having a discussion on the similarities and differences of the two statements and techniques would be interesting.
    * The authors cite [Bach17] as an example of neural networks achieving $O(d)$ sample complexity for learning single-index models. From [Bach17, Table 1], it seems the sample complexity of learning single-index models is $O(d^2)$. In [Bach17, Table 2], the sample complexity will be $d$ with a sign activation and $d^2$ with the ReLU activation. There is a discussion on the sample complexity obtained from [Bach17] in [MHWE24], where the authors also show that standard gradient-based algorithms can achieve the optimal $O(d)$ sample complexity, albeit with a number of queries going beyond the SQ lower bound. This might be interesting to point out while having the discussion on information-theoretically optimal sample complexity with neural networks.
    * Another example of additional structure reducing sample complexity is structure in the input covariance, which is known to help in the CSQ setting [BESWW23, MHWSE23]. Developing algorithms that benefit this structure to achieve a smaller sample complexity in the SQ setting can be an interesting direction for future work.
---
References:

[VE24] N. M. Vural and M. A. Erdogdu. "Pruning is Optimal for Learning Sparse Features in High-Dimensions". COLT 2024.

[Bach17] F. Bach. "Breaking the Curse of Dimensionality with Convex Neural Networks". JMLR 2017.

[MHWE24] A. Mousavi-Hosseini, D. Wu, M. A. Erdogdu. "Learning Multi-Index Models with Neural Networks via Mean-Field Langevin Dynamics". arXiv 2024.

[BESWW23] J. Ba, M. A. Erdogdu, T. Suzuki, Z. Wang, D. Wu. "Learning in the presence of low-dimensional structure: a spiked random matrix perspective". NeurIPS 2023.

[MHWSE23] A. Mousavi-Hosseini, D. Wu, T. Suzuki, M. A. Erdogdu. "Gradient-Based Feature Learning under Structured Data". NeurIPS 2023.

---

### Official Review · Reviewer_NQEc · 2024-11-04

**Soundness:** 3
**Presentation:** 4
**Contribution:** 3
**Rating:** 8
**Confidence:** 3

**Summary:**

This paper investigates the sample complexity of learning Gaussian single-index models with a two-layer neural network. The authors introduce a gradient-based training method that learns the target with $d^{s^*/2}$ samples, where $s^*$ is the generative exponent, hence matching the SQ lower bound up to poly-log factors and improving on previous works. The authors present also an improved sample complexity bound in the case of sparse teacher prior, a setting not addressed much in previous literature.

**Strengths:**

- Gaussian single index models have received a substantial interest recently, and this paper provides a detailed unifying framework, supported by theoretical proofs, which is of interest to the community.
- The framework includes several previous works, which allow to compare this work to previous result.
- The sparse teacher setting is novel and of interest.

**Weaknesses:**

- The algorithm requires a gradient oracle, which is a bit unusual and requires choosing the loss and the activation ad hoc.

**Questions:**

- Is there anything you can say for non-high-pass activations? More generally, I would like to understand whether these results generalises to some extents to natural gradient oracles.

---

### Meta-Review · Area_Chair_1ir6 · 2024-12-17

**Metareview:**

This paper gave an algorithm for neural network style learning of single-index model. The result shows that with some modifications in training, a gradient based algorithm can train a 2-layer neural network in d^{s*/2} (s*>1) samples, where s* is the generative exponent and this matches the SQ-lowerbound. Recently there has been a long line of work to understand the sample complexity for learning single/multi-index models using neural network ideas, and this work cleanly settles the problem for single-index models. The reviews mostly agree that the paper makes interesting contribution to the understanding of learning single-index models. The slight concerns is that the authors should make it clear that the algorithm is (necessarily) different from just running gradient descent on a neural network architecture.

**Additional Comments On Reviewer Discussion:**

There were some clarifying questions and suggestions raised by the reviewers. The authors addressed most of the concerns and the reviewers are satisfied.

---

### Decision · Program_Chairs · 2025-01-22

Accept (Oral)